# Geographic variability in freshwater methane hydrogen isotope ratios and its implications for global isotopic source signatures

Peter M.J. Douglas[1], Emerald Stratigopoulos[1], Jenny Park[1], Dawson Phan[1]

[1]Earth and Planetary Sciences, McGill University, Montreal, H3A 0E8, Canada

*Correspondence to*: Peter M. J. Douglas (peter.douglas@mcgill.ca)

**Abstract.** There is growing interest in developing spatially resolved methane ($CH_4$) isotopic source signatures to aid in geographic source attribution of $CH_4$ emissions. $CH_4$ hydrogen isotope measurements ($\delta^2H$-$CH_4$) have the potential to be a powerful tool for geographic differentiation of $CH_4$ emissions from freshwater environments, as well as other microbial sources. This is because microbial $\delta^2H$-$CH_4$ values are partially dependent on the $\delta^2H$ of environmental water ($\delta^2H$-$H_2O$), which exhibits large and well-characterized spatial variability globally. We have refined the existing global relationship between $\delta D$-$CH4$ - $\delta D$-$H2O$ by compiling a more extensive global dataset of $\delta^2H$-$CH_4$ from freshwater environments, including wetlands, inland waters, and rice paddies, comprising a total of 129 different sites, and compared these with measurements and estimates of $\delta^2H$-$H_2O$, as well as $\delta^{13}C$-$CH_4$ and $\delta^{13}C$-$CO_2$ measurements. We found that estimates of $\delta^2H$-$H_2O$ explain approximately 42% of the observed variation in $\delta^2H$-$CH_4$, with a flatter slope than observed in previous studies. The inferred global $\delta^2H$-$CH_4$ vs $\delta^2H$-$H_2O$ regression relationship is not sensitive to using either modelled precipitation $\delta^2H$ or measured $\delta^2H$-$H_2O$ as the predictor variable. The slope of the global freshwater relationship between $\delta^2H$-$CH_4$ and $\delta^2H$-$H_2O$ is similar to observations from incubation experiments, but is different from pure culture experiments, and is consistent with previous suggestions that variation in the $\delta^2H$ of acetate controlled by environmental $\delta^2H$-$H_2O$ is important in determining variation in $\delta^2H$-$CH_4$. The relationship between $\delta^2H$-$CH_4$ and $\delta^2H$-$H_2O$ leads to significant differences in the distribution of freshwater $\delta^2H$-$CH_4$ between the northern high latitudes (60-90 ºN), relative to other global regions. We estimate a flux-weighted global freshwater $\delta^2H$-$CH_4$ of -310±15‰, which is higher than most previous estimates. Comparison of the residual variability in $\delta^2H$-$CH_4$ with $\delta^{13}C$ measurements of both $CH_4$ and $CO_2$ does not support a dominant role for either differential isotopic fractionation related to methanogenesis pathways or methane oxidation in controlling variation in $\delta^2H$-$CH_4$, but instead suggests that residual $\delta^2H$-$CH_4$ variation is the result of complex interactions between these and other biogeochemical variables. We observe significantly higher distribution of $\delta^2H$-$CH_4$ values, corrected for $\delta^2H$-$H_2O$, in inland waters relative to wetlands, and suggest this difference is caused by more prevalent $CH_4$ oxidation in inland waters. We used the expanded freshwater $CH_4$ isotopic dataset to calculate a bottom-up estimate of global $CH_4$ $\delta^2H$ and $\delta^{13}C$ sources that includes spatially resolved isotopic signatures for freshwater $CH_4$ sources. The bottom-up global source $\delta^2H$-$CH_4$ estimate is higher than a previous estimate using a similar approach, as a result of the more enriched global freshwater $\delta^2H$-$CH_4$ signature. However, it is in agreement with top-down

estimates of global source $\delta^2$H-CH$_4$ based on atmospheric measurements and estimated atmospheric sink fractionations.
In contrast our bottom-up global source $\delta^{13}$C-CH$_4$ estimate is lower than top-down estimates, partly as a result of a lack
of $\delta^{13}$C-CH$_4$ data from C$_4$ plant dominated ecosystems. In general, we find there is a particular need for more data to
constrain isotopic signatures for low-latitude microbial CH$_4$ sources.
**1 Introduction**
Methane (CH$_4$) is an important greenhouse gas that accounts for approximately 25% of current anthropogenic global
warming, but we do not have a complete understanding of the current relative or absolute fluxes of different CH$_4$ sources
to the atmosphere (Schwietzke et al., 2016;Saunois et al., 2019), nor is there consensus on the causes of recent decadal-
scale changes in the rate of increase in atmospheric CH$_4$ (Kai et al., 2011;Pison et al., 2013;Rice et al., 2016;Schaefer et
al., 2016;Worden et al., 2017;Thompson et al., 2018;Turner et al., 2019). Freshwater ecosystems are an integral
component of the global CH$_4$ budget. They are one of the largest sources of atmospheric CH$_4$ and are unequivocally the
largest natural, or non-anthropogenic, source (Bastviken et al., 2011;Saunois et al., 2019). At the same time the
geographic distribution of freshwater CH$_4$ emissions, changes in the strength of this source through time, and the relative
importance of wetland versus inland water CH$_4$ emissions all remain highly uncertain (Pison et al., 2013;Schaefer et al.,
2016;Ganesan et al., 2018;Saunois et al., 2019;Turner et al., 2019). Gaining a better understanding of freshwater CH$_4$
emissions on a global scale is of great importance for understanding potential future climate feedbacks related to CH$_4$
emissions from these ecosystems (Bastviken et al., 2011;Koven et al., 2011;Yvon-Durocher et al., 2014;Zhang et al.,
2017). It is also necessary in order to better constrain the quantity and rate of change of other CH$_4$ emissions sources,
including anthropogenic sources from fossil fuels, agriculture, and waste (Kai et al., 2011;Pison et al., 2013;Schaefer et
al., 2016).
Isotopic tracers, particularly $\delta^{13}$C, have proven to be very useful in constraining global CH$_4$ sources and sinks
(Kai et al., 2011;Nisbet et al., 2016;Rice et al., 2016;Schaefer et al., 2016;Schwietzke et al., 2016;Nisbet et al., 2019).
However, $\delta^{13}$C source signatures cannot fully differentiate CH$_4$ sources, leaving residual ambiguity in source
apportionment (Schaefer et al., 2016;Schwietzke et al., 2016;Worden et al., 2017;Turner et al., 2019). Applying
additional isotopic tracers to atmospheric CH$_4$ monitoring has the potential to greatly improve our understanding of CH$_4$
sources and sinks (Saunois et al., 2019;Turner et al., 2019). Recently developed laser-based methods, including cavity
ringdown spectroscopy, quantum cascade laser absorption spectroscopy, and tunable infrared laser direct absorption
spectroscopy (Chen et al., 2016; Röckmann et al., 2016; Yacovitch et al., 2020) could greatly enhance the practicality of
atmospheric $\delta^2$H-CH$_4$ measurements at greater spatial and temporal resolution, similar to recent developments for $\delta^{13}$C-
CH$_4$ measurements (Zazzeri et al., 2015; Miles et al., 2018). $\delta^2$H-CH$_4$ measurements have proven useful in
understanding past CH$_4$ sources in ice-core records (Whiticar and Schaefer, 2007; Mischler et al., 2009; Bock et al.,
2010; Bock et al., 2017), but have seen only limited use in modern atmospheric CH$_4$ budgets (Kai et al., 2011; Rice et al.,
2016), in part because of loosely constrained source terms, as well as relatively sparse atmospheric measurements.
Atmospheric inversion models have shown that increased spatial and temporal resolution of $\delta^2$H-CH$_4$ measurements
could provide substantial improvements in precision for global and regional methane budgets (Rigby et al., 2012).
$\delta^2$H-CH$_4$ measurements could prove especially useful in understanding freshwater CH$_4$ emissions. Freshwater
$\delta^2$H-CH$_4$ is thought to be highly dependent on $\delta^2$H-H$_2$O (Waldron et al., 1999a;Whiticar, 1999;Chanton et al., 2006).
Since $\delta^2$H-H$_2$O exhibits large geographic variation as a function of temperature and fractional precipitation (Rozanski et
al., 1993; Bowen and Revenaugh, 2003), $\delta^2$H-CH$_4$ measurements have the potential to differentiate freshwater CH$_4$
sources by latitude. This approach has been applied in some ice core studies (Whiticar and Schaefer, 2007;Bock et al.,
2010), but geographic source signals remain poorly constrained, in part because of small datasets and because of
incompletely understood relationships between $\delta^2$H-H$_2$O and $\delta^2$H-CH$_4$. In contrast, recent studies of modern atmospheric
$\delta^2$H-CH$_4$ have typically not accounted for geographic variation in freshwater CH$_4$ sources (Kai et al., 2011;Rice et al.,
2016). Relatedly, other studies have found an important role for variation in $\delta^2$H-H$_2$O in controlling $\delta^2$H-CH$_4$ from
biomass burning (Umezawa et al., 2011) and from plants irradiated by UV light (Vigano et al., 2010), as well as the $\delta^2$H
of H$_2$ produced by wood combustion (Röckmann et al., 2016).
In addition to variance caused by $\delta^2$H-H$_2$O, a number of additional biogeochemical variables have been
proposed to influence $\delta^2$H-CH$_4$ in freshwater environments. These include differences in the predominant biochemical
pathway of methanogenesis (Whiticar et al., 1986; Whiticar, 1999; Chanton et al., 2006), the extent of methane oxidation
(Happell et al., 1994; Waldron et al., 1999a; Whiticar, 1999; Cadieux et al., 2016), isotopic fractionation resulting from
diffusive gas transport (Waldron et al., 1999a; Chanton, 2005), and differences in the thermodynamic favorability or
enzymatic reversibility of methanogenesis (Valentine et al., 2004b; Stolper et al., 2015; Douglas et al., 2016). These
influences on $\delta^2$H-CH$_4$ have the potential to complicate geographic signals, but also provide the potential to differentiate
ecosystem sources if specific ecosystems are characterized by differing rates and pathways of methanogenesis, rates of
CH$_4$ oxidation, or gas transport processes. A recent study proposed that freshwater $\delta^{13}$C-CH$_4$ could be differentiated
geographically based on ecosystem differences in the prevalence of different methanogenic pathways and in the
predominance of C$_4$ plants, in addition to the geographic distribution of wetland ecosystems (Ganesan et al., 2018). $\delta^2$H-
CH$_4$ measurements have the potential to complement this approach by providing an additional isotopic parameter for
differentiating ecosystem and geographic CH$_4$ source signatures.
In order to use $\delta^2$H-CH$_4$ as an indicator of freshwater ecosystem contributions to global and regional CH$_4$
emissions budgets, a clearer understanding of freshwater $\delta^2$H source signals, and how they vary by geographic location,
ecosystem type, and other variables is needed. In order to address this need we have assembled and analyzed a dataset of
897 $\delta^2$H-CH$_4$ measurements from 129 individual ecosystems, or sites, derived from 40 publications (Schoell, 1983;
Woltemate et al., 1984; Burke Jr and Sackett, 1986; Whiticar et al., 1986; Burke Jr et al., 1988; Burke Jr, 1992; Burke Jr
et al., 1992 ;Lansdown et al., 1992; Lansdown, 1992; Martens et al., 1992; Wassmann et al., 1992; Happell et al., 1993;
Levin et al., 1993; Happell et al., 1994; Wahlen, 1994; Bergamaschi, 1997; Chanton et al., 1997; Hornibrook et al., 1997;
Tyler et al., 1997; Zimov et al., 1997; Bellisario et al., 1999; Popp et al., 1999; Waldron et al., 1999b; Chasar et al., 2000;
Marik et al., 2002; Nakagawa et al., 2002b; Nakagawa et al., 2002a; Chanton et al., 2006; Walter et al., 2006; Walter et
al., 2008; Alstad and Whiticar, 2011; Brosius et al., 2012; Sakagami et al., 2012; Bouchard et al., 2015; Stolper et al.,
2015; Wang et al., 2015; Cadieux et al., 2016; Douglas et al., 2016; Thompson et al., 2016; Lecher et al., 2017). We have
advanced existing datasets of freshwater $\delta^2$H-CH$_4$ (Whiticar et al., 1986; Waldron et al., 1999a; Sherwood et al., 2017) in
the following key attributes:1) compiling a significantly larger dataset than was previously available; 2) compiling paired
$\delta^{13}$C-CH4 data for all sites, $\delta^{13}$C-CO$_2$ data for 50% of sites, and $\delta^2$H-H$_2$O data for 47% of sites; 3) compiling geographic
coordinates for all sites, providing the ability to perform spatial analyses and compare with gridded datasets of
precipitation isotopic composition; and 4) classifying all sites by ecosystem and sample type (dissolved vs. gas samples),
allowing for a clearer differentiation of how these variables influence $\delta^2$H-CH$_4$.
Using this data set we applied statistical analyses to address key questions surrounding the global distribution of
freshwater $\delta^2$H-CH$_4$, the variables that control this distribution, and its implications for atmospheric $\delta^2$H-CH$_4$.
Specifically, we investigated the nature of the global dependence of $\delta^2$H-CH$_4$ on $\delta^2$H-H$_2$O, and whether this relationship
results in significant differences in freshwater $\delta^2$H-CH$_4$ by latitude. We also assessed whether variability in $\delta^{13}$C-CH$_4$,
$\delta^{13}$C-CO$_2$, and $\alpha_C$, was correlated with $\delta^2$H-CH$_4$, and whether there are significant differences in $\delta^2$H-CH$_4$ between
different ecosystem and sample types. Finally, we used our dataset, combined with other isotopic datasets (Sherwood et
al., 2017) and flux estimates (Saunois et al., 2020), to estimate the global $\delta^2$H-CH$_4$ and $\delta^{13}$C-CH$_4$ of global emissions
sources, and compared this with previous estimates based on atmospheric measurements or isotopic datasets (Whiticar
and Schaefer, 2007;Rice et al., 2016;Sherwood et al., 2017),.
**2 Methods**
**2.1 Isotope Nomenclature**
The isotope notation used in this study is briefly introduced here. Hydrogen and carbon isotope ratios are primarily
discussed as delta values, using the generalized formula (Coplen, 2011):
$$\delta = \frac{\left(R_{sample} - R_{standard}\right)}{R_{standard}} \tag{1}$$
where R is the ratio of the heavy isotope to the light isotope, and the standard is Vienna Standard Mean Ocean Water
(VSMOW) for $\delta^2$H and Vienna Pee Dee Belemnite (VPDB) for $\delta^{13}$C. $\delta$ values are expressed in per mil (‰) notation.
We also refer to the isotopic fractionation factor between two phases, or $\alpha$, which is defined as:
$$\alpha_{a-b} = \frac{R_a}{R_b} = \frac{\delta_a + 1}{\delta_b + 1}$$ (2)
Specifically, we discuss the carbon isotope fractionation factor between $CO_2$ and $CH_4$ ($\alpha_C$) and the hydrogen isotope
fractionation factor between $H_2O$ and $CH_4$ ($\alpha_H$).

**2.2 Dataset Compilation**

**2.2.1 Literature Survey**

To identify datasets we used a set of search terms (methane OR $CH_4$ AND freshwater OR wetland OR peatland OR
swamp OR marsh OR lake OR pond OR 'inland water' AND 'hydrogen isotope' OR 'δD' OR 'δ²H') in Google Scholar
to find published papers that discussed this measurement. We also identified original publications using previously
compiled datasets (Waldron et al., 1999a;Sherwood et al., 2017). Data for 90% of sites were from peer-reviewed
publications. Data from 13 sites were from a Ph.D. dissertation (Lansdown, 1992).

**2.2.2 Dataset structure**

Most samples were associated with geographic coordinates in data tables or text documentation, or with specific
geographic locations such as the name of a town or city. In a few cases we identified approximate geographic locations
based on text descriptions of sampling sites, with the aid of Google Earth software. Sampling sites were defined as
individual water bodies or wetlands as identified in the relevant study. In some cases where a number of small ponds
were sampled from the same location, we grouped ponds of a given type as a single site (Bouchard et al., 2015). We
divided sampling sites into six ecosystem categories: 1) lakes and ponds (hereafter lakes), 2) rivers and floodplains
(hereafter rivers), 3) bogs, 4) fens, 5) swamps and marshes, and 6) rice paddies. Most data (7 of 8 sites) in the rivers
category are from floodplain lake or delta environments. Swamps and marshes were combined as one category because
of a small number of sites, and because there is no clear indication of biogeochemical differences between these
ecosystems. To make these categorizations we relied on site descriptions in the data source publications. We also
analyzed data in two larger environment types, inland waters (lakes and rivers) and wetlands (bogs, fens, swamps and
marshes, and rice paddies), which correspond to two flux categories (freshwaters and natural wetlands) documented by
Saunois et al. (2020). While rice paddies are an anthropogenic ecosystem, they are wetlands where microbial
methanogenesis occurs under generally similar conditions to natural wetlands, and therefore we included them as
wetlands in our analysis. In some cases the type of wetland was not specified. We did not differentiate between
ombrotrophic and minerotrophic peatlands since most publications did not specify this difference, although it has been
inferred to be important for $\delta^{13}C$-$CH_4$ distributions (Hornibrook, 2009). For studies of bogs and fens that sampled by soil
depth we have only included sample measurements from the upper 50 cm. This is based on the observation of large-scale

isotopic variability with soil depth in these ecosystems (Hornibrook et al., 1997;Waldron et al., 1999b), and the observation that shallow peat is typically the dominant source of atmospheric emissions (Waldron et al., 1999b;Bowes and Hornibrook, 2006;Shoemaker et al., 2012), which is our primary focus in this study. Other wetland ecosystems were not sampled by soil depth.

We also categorized samples by the form in which $CH_4$ was sampled, differentiating between dissolved $CH_4$ and $CH_4$ emitted through diffusive fluxes, which we group as dissolved $CH_4$, and gas-phase samples, including bubbles sampled either by disturbing sediments or by collecting natural ebullition fluxes. In some cases the sampling method or type of sample was not specified, or samples were a mix of both categories, which we did not attempt to differentiate.

Where possible (78% of sites), $\delta^2$H-$CH_4$ and $\delta^{13}$C-$CH_4$ values, as well as $\delta^{13}$C-$CO_2$ and $\delta^2$H-$H_2O$, were gathered from data files or published tables. In a number of publications, representing 22% of sites, data were only available graphically. For these studies we used Webplot Digitizer (https://automeris.io/WebPlotDigitizer/) software to extract data for these parameters. Previous studies have shown that user errors from Webplot Digitizer are typically small, with 90% of user generated data within 1% of the actual value (Drevon et al., 2017). Based on this, we estimate a typical error for $\delta^2$H-$CH_4$ data of less than 3‰. Studies where data were derived from graphs are identified in Supplementary Table S1 (Douglas et al., 2020).

### 2.2.3 Estimates of $\delta^2$H-$H_2O$ and its effects on $\delta^2$H-$CH_4$

To estimate $\delta^2$H-$H_2O$ for sites where it was not measured we relied on estimates of the isotopic composition of precipitation ($\delta^2H_p$), derived the Online Isotopes in Precipitation Calculator v.3.1 (OIPC3.1; www.waterisotopes.org; Bowen and Wilkinson, 2002; Bowen and Revenaugh, 2003; Bowen et al., 2005). Inputs for $\delta^2H_p$ estimates are latitude, longitude, and elevation. We estimated elevation for each site surface elevation at the site's geographic coordinates reported by Google Earth. We tabulated estimates of both annual precipitation-amount weighted $\delta^2H_p$, and growing season precipitation-amount weighted $\delta^2H_p$, where the growing season is defined as months with a mean temperature greater than 0 ºC. We then analysed whether annual or growing season $\delta^2H_p$ is a better estimate of environmental $\delta^2$H-$H_2O$ for both wetlands and inland waters by comparing these values with measured $\delta^2$H-$H_2O$ for sites with measurements (See Sect. 3.2). Based on this analysis, we then identified a 'best-estimate' $\delta^2$H-$H_2O$ value for each site, using an approach similar to that of Waldron et al. (1999a). Namely, we apply measured $\delta^2$H-$H_2O$ where available, and estimates based on the regression analyses detailed in Section 3.2 for sites without measurements.

To account for the effects of $\delta^2$H-$H_2O$ on $\delta^2$H-$CH_4$, we introduce the term $\delta^2$H-$CH_{4,W0}$, which is the estimated $\delta^2$H-$CH_4$ of a sample if it had formed in an environment where $\delta^2$H-$H_2O$ = 0‰. This is defined by the equation:

$$\delta^2\text{H-CH}_{4,\text{W0}} = \delta^2\text{H-CH}_4 - \left( b \times \delta^2\text{H}_2\text{O} \right) \tag{3}$$

where $\delta^2$H-$H_2$O is the 'best-estimate' value for each site described above, $b$ is the slope of the regression relationship of
$\delta^2$H-$H_2$O vs. $\delta^2$H-$CH_4$ for the entire dataset, as reported in Sect. 3.3. We also performed the same calculation separately
for the subset of sites with measured $\delta^2$H-$H_2$O. We analyze $\delta^2$H-$CH_{4,W0}$ instead of $\alpha_H$ because, as discussed in Sect. 3.3.1,
the global relationship between $\delta^2$H$_p$ vs. $\delta^2$H-$CH_4$ does not correspond to a constant value of $\alpha_H$, and therefore deviations
from the global empirical relationship are more clearly expressed as a residual as opposed to a fractionation factor.

## 2.3 Statistical analyses

For all statistical analyses we use site-level mean isotopic values. This avoids biasing our analyses towards sites with a
large number of measurements, since there are large differences in the number of samples analyzed per site ($n$ ranges
from 66 to 1). To calculate $\alpha_C$ we used average $\delta^{13}$C-$CH_4$ and $\delta^{13}$C-$CO_2$ at a given site. This approach entails some
additional uncertainty in this variable, but was necessary because at many sites these measurements were not made on the
same samples.
We perform a set of linear regression analyses $\delta^2$H-$CH_4$ against other isotopic variables, in addition to latitude.
All statistical analyses were performed in Matlab. We considered $p < 0.05$ to be the threshold for identifying significant
regression relationships. We chose to perform unweighted regression, as opposed to weighted regression based on the
standard error of sample measurements, for two reasons. First, a small number of sites with a large number of
measurements, and therefore small standard error, had a disproportionate effect on weighted regression results. Second,
in environmental research unweighted regression is frequently less biased than weighted regression (Fletcher and Dixon,
2012). Based on a test proposed by Fletcher and Dixon (2012) unweighted regression is appropriate for this dataset. We
used analysis of covariance to test for significant differences ($p < 0.05$) between regression relationships.
To compare isotopic data ($\delta^2$H-$CH_4$ and $\delta^{13}$C-$CH_4$) between groups (i.e. latitudinal bands, ecosystem types,
sample types) we used non-parametric statistical tests to test whether the groups were from different distributions. We
used non-parametric tests because some sample groups were not normally distributed, as determined by a Shapiro-Wilk
test (Shapiro and Wilk, 1965). For comparing differences between the distributions of two groups we used the Mann-
Whitney U-test (Mann and Whitney, 1947), whereas when comparing differences between the distributions of more than
two groups we used the Kruskal-Wallis H-test (Kruskal and Wallis, 1952), combined with Dunn's test to compare
specific sample group pairs (Dunn, 1964). We considered $p < 0.05$ to be the threshold for identifying groups with
significantly different distributions.
When comparing $\delta^{13}$C-$CH_4$ by latitude and ecosystem we combined the data from this study with additional data
from Sherwood et al. (2017) (32 additional sites) where $\delta^2$H-$CH_4$ was not measured to make our dataset as representative
as possible. To our knowledge this combined dataset is the largest available compiled dataset of freshwater $\delta^{13}$C-$CH_4$,
although there are many more $\delta^{13}$C-$CH_4$ measurements that have not yet been aggregated. We did not include these

additional data when analysing differences by sample type, as sample type was not specified in the dataset of Sherwood et al. (2017).

## 2.4 Estimation of global atmospheric CH$_4$ δ$^2$H and δ$^{13}$C source values

To better understand how latitudinal differences in wetland isotopic source signatures influence atmospheric δ$^2$H-CH$_4$ and δ$^{13}$C-CH$_4$, we calculated a 'bottom-up' mixing model of δ$^2$H-CH$_4$ and δ$^{13}$C-CH$_4$. For this calculation we ascribed all CH$_4$ sources a flux (derived from Saunois et al., 2020; see details below) and a δ$^2$H and δ$^{13}$C value, and calculated the global atmospheric source value using an isotopic mixing model. Because of non-linearity when calculating mixtures using δ$^2$H values, we performed the mixing equation using isotopic ratios (see Sect. 2.1). The mixing equation is as follows:

$$R_{mix} = f_1 R_1 + f_2 R_2 + ... + f_n R_n \tag{4}$$

where $f_n$ is the fractional flux for each source term (i.e. the ratio of the source flux to total flux), and R$_n$ is the isotope ratio for each source term.

Values for the flux, δ$^2$H, and δ$^{13}$C applied for each source term are shown in Table 1. We used bottom-up source fluxes from Saunois et al. (2020) for the period 2008-2017. For categories other than wetlands, inland waters, and rice paddies, we used global fluxes and isotope values, since geographically resolved isotopic source signature estimates are not available. For these sources we used δ$^2$H values published by Sherwood et al. (2017), using the mean value for each source term. For wetlands, inland waters, and rice paddies, we used geographically resolved (60-90 ºN; 30-60 ºN, 90º S-30ºN) fluxes derived from Saunois et al. (2019) for the period 2008-2017, and mean δ$^2$H-CH$_4$ for these latitudinal bands from this study.

To calculate mean δ$^{13}$C-CH$_4$ from wetlands, inland waters, and rice paddies for different latitudinal bands we combined the data from this study along with additional data from Sherwood et al. (2017) (32 additional sites) to make our estimated source signatures as representative as possible. To our knowledge this combined dataset is the largest available compiled dataset of freshwater δ$^{13}$C-CH$_4$ (See Sect. 2.3). Sites dominated by C$_4$ plants are notably underrepresented in this combined dataset. In addition, the biomass burning dataset of Sherwood e al. (2017) contains very few data from C$_4$ plant combustion. We performed a separate estimate of global source δ$^{13}$C-CH$_4$ that attempted to correct for these likely biases by making two adjustments: 1) using the estimated low-latitude wetland δ$^{13}$C-CH$_4$ signature of Ganesan et al., (2018) (-56.7‰), which takes into account the predicted spatial distribution of C$_4$ plant dominated wetlands; and 2) using the biomass burning δ$^{13}$C-CH$_4$ signature of Schwietzke et al., (2016) (-22.3‰), which is weighted by the predicted contribution from C$_4$ plant combustion. We did not attempt to take into account δ$^{13}$C-CH$_4$ from ruminants feeding on C$_4$ plants. For these C$_4$ plant corrections we applied the same uncertainties that are reported in Table 1.

**Table 1: Estimates of source-specific fluxes, $\delta^2$H-CH4, and $\delta^{13}$C-CH4, and their uncertainties, used in mixing models and Monte Carlo analyses**

| Category | Flux (Tg/Yr) | Uncertainty | $\delta^2$H signature (‰, VSMOW) | Uncertainty | $\delta^{13}$C signature (‰, VPDB) | Uncertainty |
|---|---|---|---|---|---|---|
| Wetlands (<30N) | 115 | 37.5 | -301 | 15 | -64.4 | 1.9 |
| Wetlands (30-60N) | 25 | 16.5 | -324 | 14 | -61.8 | 2.6 |
| Wetlands (>60N) | 9 | 8.0 | -374 | 10 | -62.7 | 3.0 |
| Inland Waters (<30N) | 80 | 39.4 | -301 | 12 | -57.1 | 3.0 |
| Inland Waters (30-60N) | 64 | 31.9 | -308 | 18 | -62.0 | 3.8 |
| Inland Waters (>60N) | 16 | 7.5 | -347 | 9 | -65.0 | 1.8 |
| Geological (onshore)[a] | 38 | 13.0 | -189 | 44 | -43.8 | 10.0 |
| Wild animals[b] | 2 | 2.0 | -316 | 28 | -65.4 | 3.5 |
| Termites[c] | 9 | 6.0 | -343 | 50 | -63.4 | 3.5 |
| Permafrost soils (direct)[d] | 1 | 0.5 | -374 | 15 | -64.4 | 1.7 |
| Geological (offshore)[a] | 7 | 7.0 | -189 | 44 | -43.8 | 10.0 |
| Biogenic open and coastal[e] | 6 | 3.0 | -200 | 50 | -80.0 | 20.0 |
| Enteric fermentation and manure | 111 | 5.0 | -308 | 28 | -65.4 | 3.5 |
| Landfills and waste | 65 | 4.5 | -297 | 6 | -56.0 | 4.9 |
| Rice cultivation (<30N) | 19 | 1.2 | -324 | 8 | -55.0 | 6.5 |
| Rice cultivation (30-60N) | 12 | 0.5 | -325 | 8 | -62.3 | 2.1 |
| Coal mining | 42 | 15.5 | -232 | 5 | -49.5 | 1.0 |
| Oil and gas | 79 | 13.0 | -189 | 2 | -43.8 | 0.5 |
| Industry[f] | 3 | 3.0 | -189 | 2 | -43.8 | 0.5 |
| Transport[f] | 4 | 4.0 | -189 | 2 | -43.8 | 0.5 |
| Biomass burning[g] | 17 | 6.0 | -211 | 15 | -26.2 | 2.0 |
| Biofuel burning[g] | 12 | 2.0 | -211 | 15 | -26.2 | 2.0 |

a-No specific isotopic measurements in the database (Sherwood et al., 2017). We applied the mean isotopic values for oil and gas, and applied the standard deviation of for oil and gas as the uncertainty

b-No specific isotopic measurements in database (Sherwood et al., 2017). We used the isotopic values and uncertainties from livestock

c-Only one $\delta^2$H measurement in database (Sherwood et al., 2017). We applied 50‰ as a conservative uncertainty estimate.

d- No specific isotopic measurement in database (Sherwood et al., 2017). We used the isotopic values and uncertainties for high-latitude wetlands

e- No specific isotopic measurements in database (Sherwood et al., 2017). We applied approximate isotopic values based on Whiticar, (1999), and conservatively large uncertainty estimates.

f-No specific isotopic measurements in database (Sherwood et al., 2017). We used the isotopic values and uncertainties for oil and gas.

g-We applied all isotopic measurements of biomass burning to both the biomass burning and biofuel burning categories. We did not correct for the relative proportion of $C_3$ and $C_4$ plant combustion sources (See Sect. 2.4)


Since fluxes from *other natural sources* are not differentiated for the period 2008-2017, we calculated the
proportional contribution of each category of other natural sources for the period 2000-2009 (Saunois et al., 2020), and
applied this to the total flux from other natural sources for 2008-2017. Inland waters and rice paddies do not have
geographically resolved fluxes reported in Saunois et al. (2020). Therefore, we calculated the proportion of *other natural*
*sources* attributed to inland waters from 2000-2009 (71%), and applied this proportion to the geographically resolved
fluxes of *other natural sources*. Similarly, we calculated the proportion of *agricultural and waste sources* attributed to
rice agriculture from 2008-2017 (15%), and applied this to the geographically resolved fluxes of *agricultural and waste*
*fluxes*.
To estimate uncertainty in the modelled total source $\delta^2H$ and $\delta^{13}C$ values we conducted Monte Carlo analyses
(Thompson et al., 1992). We first estimated the uncertainty for each flux, $\delta^2H$, and $\delta^{13}C$ term. Flux uncertainties were
defined as one half of the range of estimates provided by Saunois et al., (2020). For sources where fluxes were calculated
as a proportion of a larger flux, we applied the same proportional calculation to uncertainty estimates. In cases where one
half of the range of reported studies was larger than the flux estimate, we set the uncertainty to be equal to the flux
estimate to avoid negative fluxes in the mixing model. Isotopic source signal uncertainties were defined as the 95%
confidence interval of the mean value for a given source category. For some sources there is insufficient data to calculate
a 95% confidence interval, and we applied a conservative estimate of uncertainty for these sources, as detailed in Table 1.
We then recalculated the $\delta^2H$ and $\delta^{13}C$ mixing models 10,000 times, each time sampling inputs from the uncertainty
distribution for each variable. We assumed all uncertainties were normally distributed. We interpret the 2-sigma standard
deviation of the resulting Monte Carlo distributions as an estimate of the uncertainty of our total atmospheric $CH_4$ source
isotopic values. To examine how the Monte Carlo analyses were specifically influenced by uncertainty in isotopic source
signatures vs. flux estimates, we conducted sensitivity tests where we set the uncertainty in either isotopic source
signatures or flux estimates to zero. We also used the mixing model and Monte Carlo method to estimate the mean flux-
weighted freshwater $\delta^2H-CH_4$ and $\delta^{13}C-CH_4$, using only the inputs for freshwater environments (Wetlands, Inland
Waters, and Rice Cultivation) from Table 1 (See Sect. 3.5)
**3 Results and Discussion**
**3.1 Dataset distribution**
The dataset is primarily concentrated in the northern hemisphere (Fig. 1A), but is distributed across a wide range
of latitudes between 3 ºS to 73 ºN (Fig. 1C). The majority of sampled sites are from North America (Fig. 1B), but there
are numerous sites from Eurasia. A much smaller number of sites are from South America and Africa. We define three
latitudinal bands for describing geographic trends: low latitudes (3 ºS to 30 ºN); mid-latitudes (30 ºto 60 ºN); and high-
latitudes; (60º to 90º N). This definition was used primarily because it corresponds with a commonly applied geographic
classification of CH$_4$ fluxes (Saunois et al., 2020).

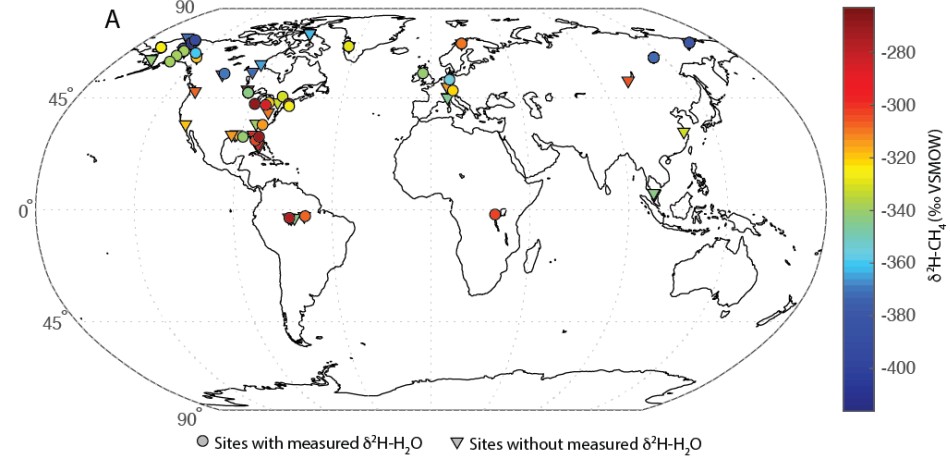

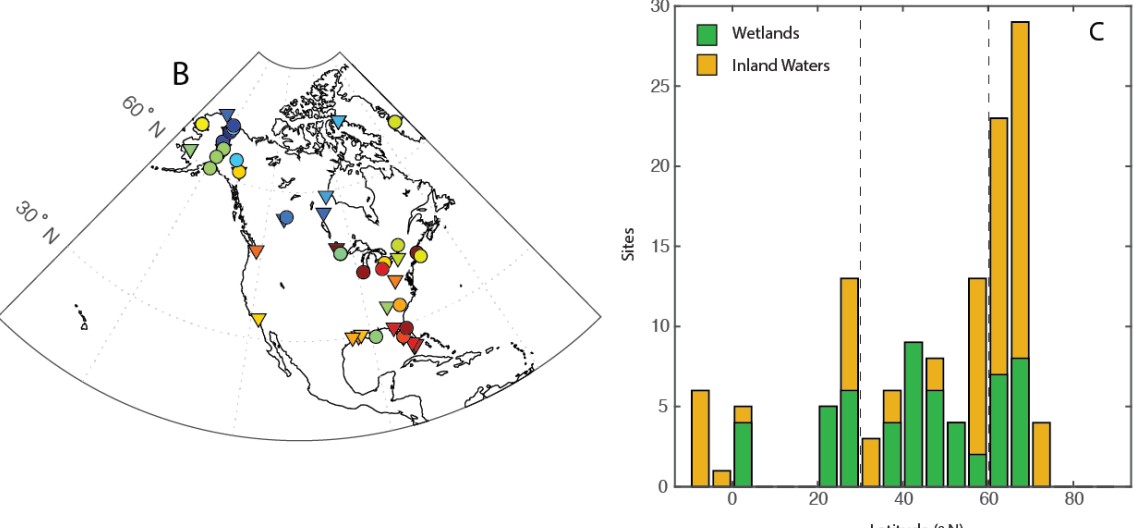


**Figure 1: Distribution of sites shown; A) on a global map, with site mean CH$_4$-δ$^2$H values indicated in relation to a color bar.**
**Sites with and without measured δ$^2$H-H$_2$O are differentiated; B) on a map of North America; and C) as a histogram of sites by**
**latitude, differentiated between wetlands and inland waters. Dashed lines in (C) indicate divisions between low-latitude, mid-**
**latitude, and high-latitude sites.**
74 of 129 sites are classified as inland waters, primarily lakes (n = 66), with a smaller number from rivers (n =
8). To our knowledge, all of the inland water sites are natural ecosystems and do not include reservoirs. 55 sites are
classified as wetlands, including 16 bogs, 14 swamps and marshes, 12 fens, and 8 rice paddies. For the majority of sites
(n = 84) gas samples were measured, whereas studies at 36 sites measured dissolved CH$_4$ or diffusive fluxes.

### 3.2 Use of $\delta^2H_p$ as an estimator for freshwater $\delta^2H$-$H_2O$

As discussed in Sect. 2.2.3, we compared modelled annual and growing season $\delta^2H_p$ with measured $\delta^2H$-$H_2O$ to determine which is a better estimator for sites where $\delta^2H$-$H_2O$ is not measured. We performed this analysis separately for wetland and inland water environments because these broad environmental categories have distinct hydrological characteristics. For all comparisons we found strong correlations, with $R^2$ values between 0.82 to 0.88 (Fig. 2). For wetlands, regression using annual $\delta^2H_p$ produces a slightly better fit, and also produces a slope within error of 1 (Fig 2A), suggesting that variation in annual $\delta^2H_p$ scales proportionately with variation in measured $\delta^2H$-$H_2O$. However, the intercept of this relationship was significantly greater than 0 (19±9 ‰). We interpret this intercept as indicating that evaporative isotopic enrichment is generally important in controlling $\delta^2H$-$H_2O$ in wetlands. A slope slightly greater than 1 is also consistent with evaporative enrichment, since greater evaporation rates would be expected in low-latitude environments with higher $\delta^2H$-$H_2O$. These results are consistent with detailed studies of wetland isotope hydrology that indicate a major contribution from groundwater, with highly dampened seasonal variability relative to precipitation, but also indicate evaporative enrichment of water isotopes in shallow soil water (Sprenger et al., 2017; David et al., 2018)}.

For inland waters, regression with growing season $\delta^2H_p$ produces a relationship that is within error of the 1:1 line (Fig. 2C), in contrast to annual $\delta^2H_p$, which produces a flatter slope (Fig. 2D). We infer that seasonal differences in $\delta^2H_p$ are important in determining $\delta^2H$-$H_2O$ in the inland water environments analyzed, especially at high latitudes, implying that these environments generally have water residence times on subannual timescales. This finding is generally consistent with evidence for seasonal variation in lake water isotopic compositions that is dependent on lake water residence times (Tyler et al., 2007;Jonsson et al., 2009). Lake water residence times vary widely, primarily as a function of lake size, but isotopic data implies that small lakes have water residence times of less than a year (Brooks et al., 2014), resulting in seasonal isotopic variability (Jonsson et al., 2009). Isotopic enrichment of lake water is highly variable, but is typically minor in humid and high-latitude regions (Jonsson et al., 2009;Brooks et al., 2014), which characterizes most of our study sites.

Based on these results we, combine measured and estimated $\delta^2H$-$H_2O$ to determine a 'best-estimate' value for each site, an approach similar to that of Waldron et al. (1999a). For sites with measured $\delta^2H$-$H_2O$ values we use the measured value. For inland water sites without measured $\delta^2H$-$H_2O$ we use modeled growing season $\delta^2H_p$ since the regression of this against measured $\delta^2H$-$H_2O$ is indistinguishable from the 1:1 line (Fig. 2D). For wetland sites without measured $\delta^2H$-$H_2O$ we estimate $\delta^2H$-$H_2O$ using the regression relationship with annual precipitation $\delta^2H$-$H_2O$ shown in Fig. 2A. The root mean square errors (RMSE) of these relationships (16‰ for wetlands, 22‰ for inland waters) provide an estimate of the uncertainty associated with estimating $\delta^2H$-$H_2O$ using $\delta^2H_p$. Given the uncertainty associated with estimating $\delta^2H$-$H_2O$ using $\delta^2H_p$, for all analyses presented below that depend on $\delta^2H$-$H_2O$ values we also analyse the dataset only including sites with measured $\delta^2H$-$H_2O$.

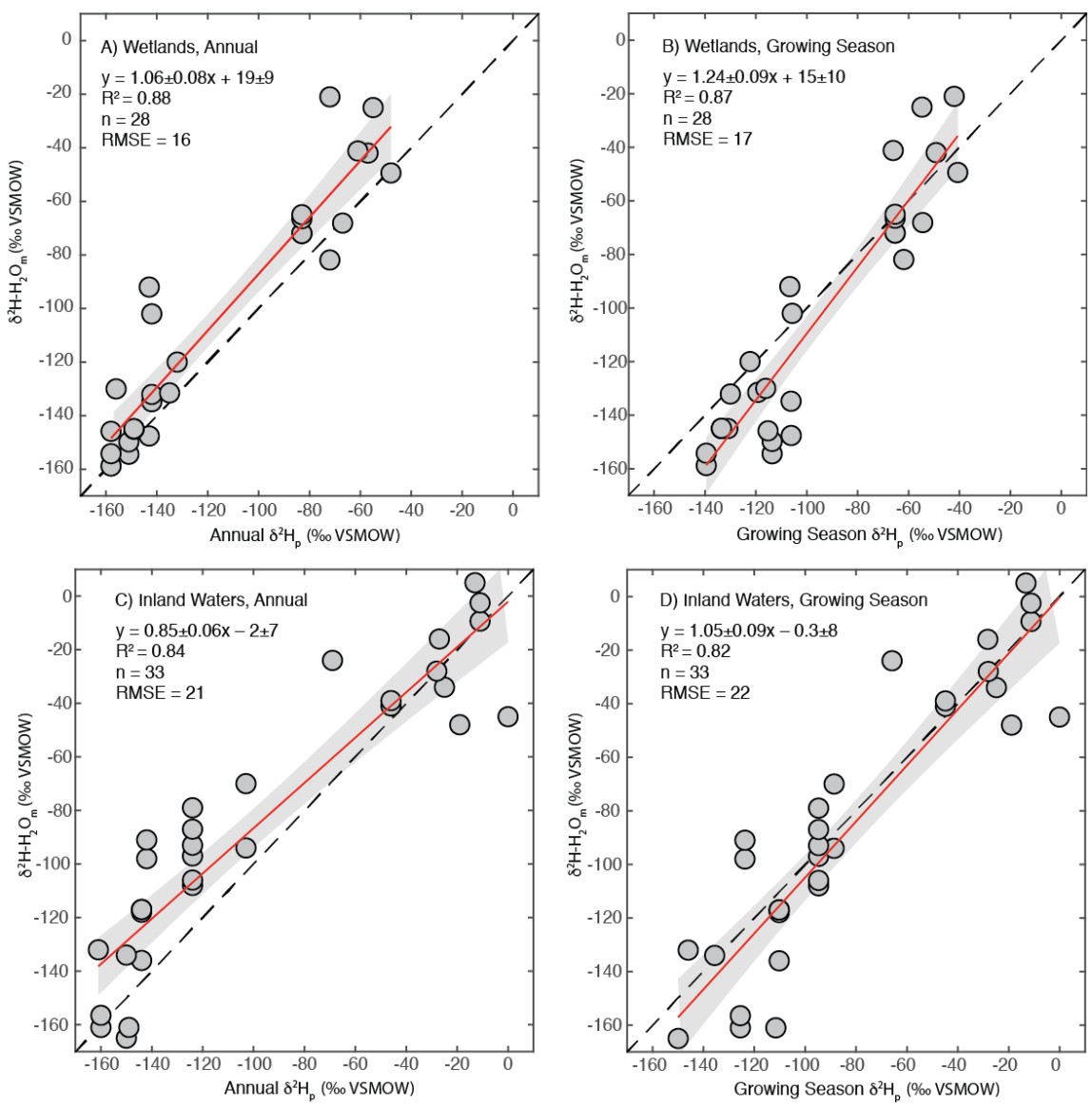

321

**Figure 2: Scatter plots of annual or growing season $\delta^2H_p$ vs. measured $\delta^2H$-$H_2O$ for wetland (A,B) and inland water (C,D) sites. The red lines indicates the best fit, with a 95% confidence interval (gray envelopes), and the dashed black lines are the 1:1 relationship.**

### 3.3 Relationship between $\delta^2H$-$H_2O$ and $\delta^2H$-$CH_4$

We carried out regression analyses of $\delta^2H$-$H_2O$ vs. $\delta^2H$-$CH_4$, both using 'best-estimate' $\delta^2H$-$H_2O$ as described in sect. 3.2 (Fig. 3A), and only including sites with measured $\delta^2H$-$H_2O$ (Fig. 3B). In addition we analysed the relationship for all sites using annual (Fig. 3C) and growing season (Fig. 3D) $\delta^2H_p$. Identifying the relationship between modelled $\delta^2H_p$ and $\delta^2H$-$CH_4$ is of value because this could be used to create gridded global predictions of $\delta^2H$-$CH_4$ based on gridded datasets

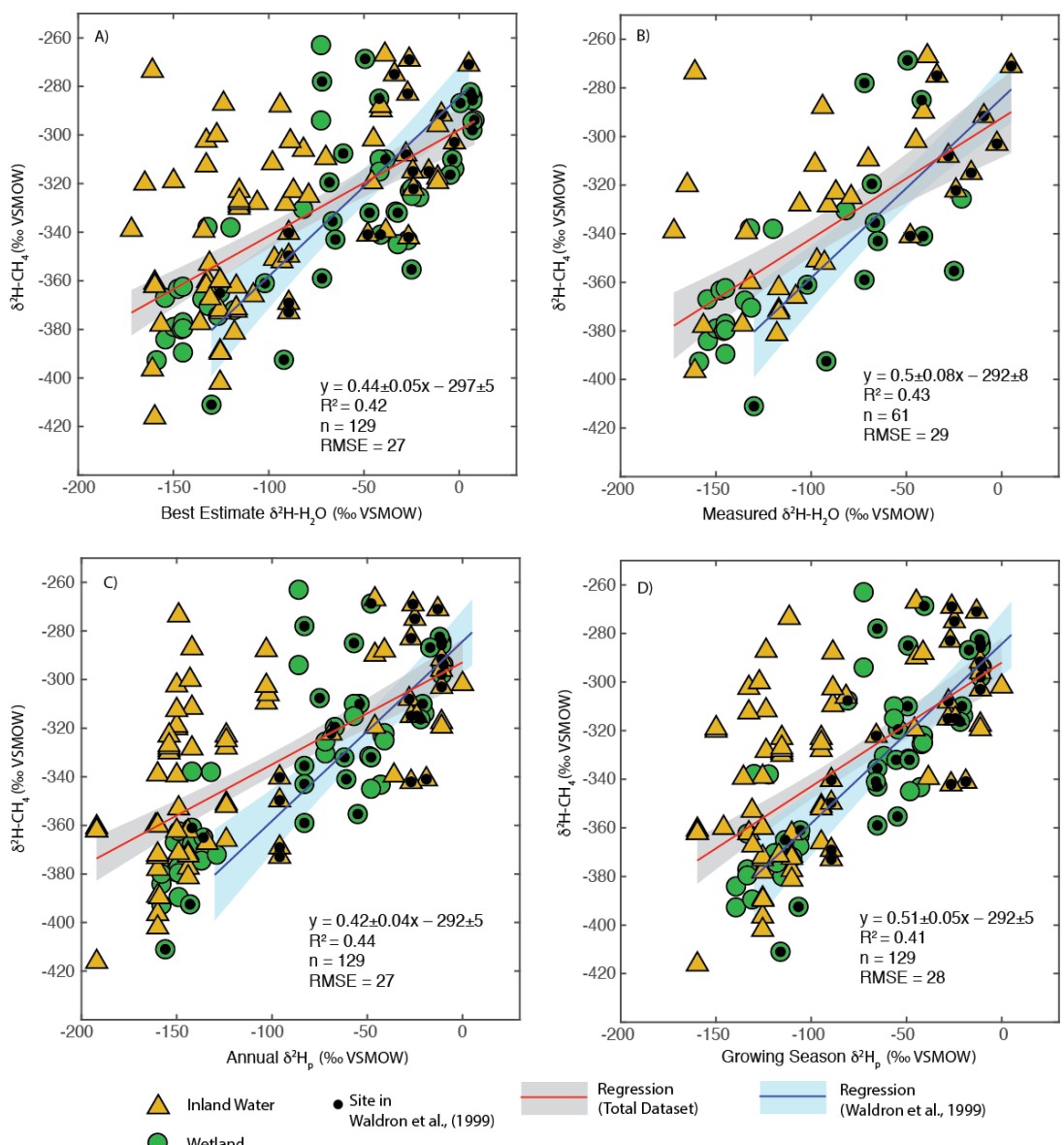


**Figure 3: Scatter plots of $\delta^2$H-CH$_4$ vs. (A) best-estimate $\delta^2$H-H$_2$O; (B) measured $\delta^2$H-H$_2$O; (C) annual $\delta^2$H$_p$; and (D) growing**
**season $\delta^2$H$_p$. Sites that were included in the analysis of Waldron et al. (1999a) are indicated. The regression relationship for the**
**total dataset in each plot is shown by the red line, with its 95% confidence interval (grey envelope). The regression relationship**
**and confidence interval for the dataset of Waldron et al., (1999a) is shown in blue.  Uncertainties for reported regression**
**relationships are standard errors.**

of $\delta^2$H$_p$ (Bowen and Revenaugh, 2003), as well as to predict the distribution of $\delta^2$H-CH$_4$ under past and future global
climates using isotope enabled Earth system models (Zhu et al., 2017).

$\delta^2$H-CH$_4$ is significantly positively correlated with $\delta^2$H-H$_2$O when using all four methods of estimating $\delta^2$H-

H$_2$O (Fig. 3, Supplemental Table 2). This is the case when analysing all sites together, as well as when analysing
wetlands and inland waters separately (Supplemental Table 2, Fig. 4). There is no significant difference in regression
relationships, based on analysis of covariance, when $\delta^2$H-CH$_4$ is regressed against best-estimate $\delta^2$H-H$_2$O, measured $\delta^2$H-
H$_2$O, or modelled $\delta^2$H$_p$, nor is there a major difference in R$^2$ values or RMSE (Supplemental Table S2). Wetland sites
consistently have a steeper regression slope than inland water sites (Supplemental Table S2), but this difference is not
significant. Regression with wetland sites also consistently results in a higher R$^2$ values and lower RMSE.

Given the similar results when regressing with estimated or measured $\delta^2$H-H$_2$O, we infer that using either the

'best-estimate' $\delta^2$H-H$_2$O or modelled $\delta^2$H$_p$, instead of measured $\delta^2$H-H$_2$O, to predict $\delta^2$H-CH$_4$ does not result in
substantial additional error. This implies that isotope-enabled Earth Systems models (ESMs) could be used to predict the
distribution of freshwater $\delta^2$H-CH$_4$ under past and future climates based on modeled $\delta^2$H$_p$, although the substantial
scatter in Figures 3C and D should be taken into account. The southern hemisphere is highly underrepresented in
available $\delta^2$H-CH$_4$ data. However, the mechanisms linking $\delta^2$H-CH$_4$ with H$_2$O-$\delta^2$H should not differ in the southern
hemisphere, and we argue that the relationships observed in this study are suitable to predict southern hemisphere
freshwater $\delta^2$H-CH$_4$. The choice of predicting $\delta^2$H-CH$_4$ using growing-season vs. annual precipitation $\delta^2$H$_p$ could be
important, with steeper slopes overall when regressing against growing season $\delta^2$H$_p$. Based on our analysis in sect. 3.2,
we suggest that annual $\delta^2$H$_p$ may be more appropriate for estimating wetland $\delta^2$H-CH$_4$, while growing season $\delta^2$H$_p$ may
be more appropriate for estimating inland water $\delta^2$H-CH$_4$. Future research will combine gridded datasets of wetland
distribution (Ganesan et al., 2018), modeled annual $\delta^2$H$_p$ (Bowen and Revenaugh, 2003), and the regression relationships
from this study to predict spatially-resolved wetland $\delta^2$H-CH$_4$ at a global scale.

Overall, our results are consistent with those of Waldron et al., (1999a), and confirm the finding of that study

that $\delta^2$H-H$_2$O is the predominant predictor of global variation in $\delta^2$H-CH$_4$. All of the regression slopes produced using
our dataset are flatter than the regression relationship found by Waldron et al. (1999a) using a smaller dataset (0.68±0.1),
although the slopes are not significantly different based on analysis of covariance. Based on this result we infer that the
true global relationship is likely flatter than that estimated by Waldron et al. (1999a), but more data will be needed to
further constrain this relationship. The difference between the regression relationships reported here and that of Waldron
et al. (1999a) is largely a result of a much greater number of samples from the high latitudes (Fig. 1C), where $\delta^2$H-H$_2$O
values are typically lower. The small number of high-latitude sites sampled by Waldron et al. (1999a) are skewed
towards the low end of the high-latitude $\delta^2$H-CH$_4$ data from this study (Fig. 3). A similarly flatter slope (0.54±0.05) was
found by Chanton et al. (2006) when combining a dataset of $\delta^2$H-CH$_4$ from Alaskan wetlands, which are included in this
study, with the dataset of Waldron et al. (1999a). As discussed below in sect. 3.3.1, our regression relationship slopes are
very similar to that of the 'in-vitro' line of Waldron et al. (1999a). Based on the range of R$^2$ values shown in Figure 3, we
estimate that $\delta^2H$-$H_2O$ explains approximately 42% of variability in $\delta^2H$-$CH_4$, implying substantial residual variability,
with greater residual variability inland water sites than in wetlands (Supplemental Table 2).

Given that $\delta^2H$-$H_2O$ is strongly influenced by latitude, although it is also influenced by other geographic and

climatic variables, we examined whether $\delta^2H$-$CH_4$ is also significantly correlated with latitude. There is indeed a
significant, negative relationship between latitude and $\delta^2H$-$CH_4$, indicating an approximate decrease of 0.9‰/° latitude
(Fig. 4). The slope is significantly flatter than that for latitude vs. $\delta^2H$-$H_2O$ in this dataset (-2 ‰/° latitude), which is
consistent with the inferred slope for $\delta^2H$-$H_2O$ vs. $\delta^2H$-$CH_4$ (0.44 to 0.5). There is greater scatter in $\delta^2H$-$CH_4$ at higher
latitudes, especially for inland waters, but it is unclear if this is simply a result of a larger sample set or of differences in
the underlying processes controlling $\delta^2H$-$CH_4$. We discuss latitudinal differences in $\delta^2H$-$CH_4$ in further detail in Sect. 3.5

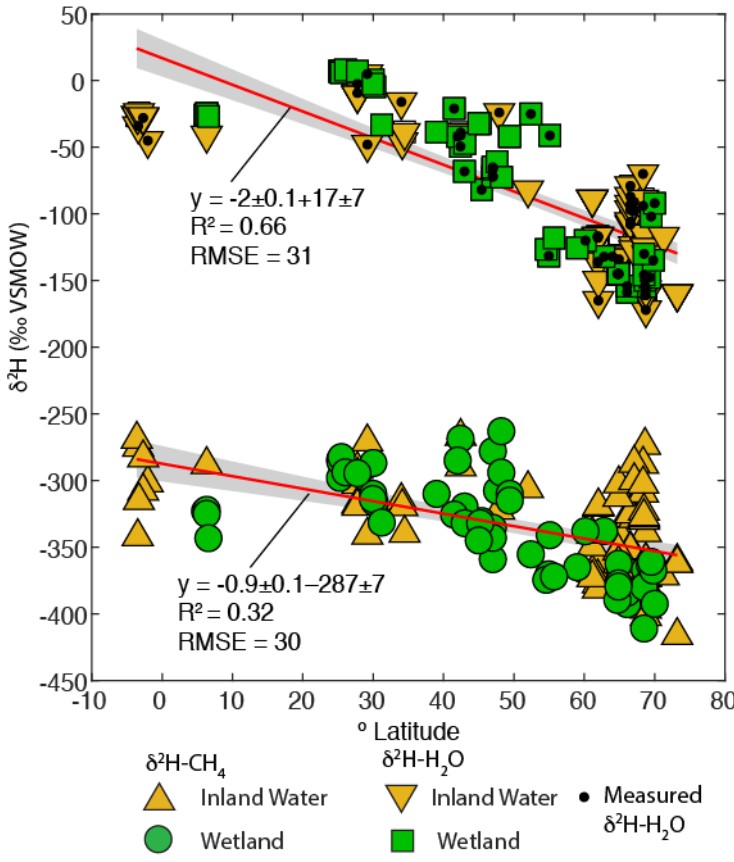


**Figure 4: Scatter plots of $\delta^2H$-$CH_4$ and best estimate $\delta^2H$-$H_2O$, vs. latitude (° N). Sites with measured $\delta^2H$-$H_2O$ are indicated.**
**Envelopes indicate 95% confidence intervals for regression lines.**
**3.3.1 Comparison of $\delta^2H$-$H_2O$ vs $\delta^2H$-$CH_4$ relationships between environmental and experimental studies**

To further understand the processes controlling the observed $\delta^2H$-$H_2O$ vs. $\delta^2H$-$CH_4$ relationships we compared

them to results from pure culture and incubation experiments across a wide range of $\delta^2H$-$H_2O$ values (Fig. 5), focusing

on regression against best-estimate $\delta^2$H-H$_2$O. The regression slopes for both wetlands and inland waters (0.5 and 0.42) are within error of the 'in-vitro' relationship compiled by Waldron et al. (1999a) (0.44), based on laboratory incubations from three separate studies (Schoell, 1980;Sugimoto and Wada, 1995;Waldron et al., 1998). The intercept for the wetland and inland water regressions is higher than that for the 'in-vitro' relationship, although only the difference with inland waters is significant. In contrast, the regression slope for pure-culture acetoclastic methanogenesis experiments is much flatter (0.18 to 0.2) (Valentine et al., 2004b;Gruen et al., 2018), consistent with the prediction that one hydrogen atom is exchanged between water and the acetate methyl group during CH$_4$ formation (Pine and Barker, 1956;Whiticar, 1999). The large difference in intercept between the two acetate pure culture datasets is likely a function of differences in the $\delta^2$H of acetate, but could also be influenced by differences in kinetic isotope effects (Valentine et al., 2004b).

Pure culture hydrogenotrophic methanogenesis experiments (Gruen et al., 2018) yield a regression slope that is consistent with a constant $\alpha_H$ value, although $\alpha_H$ clearly varies depending on experimental or environmental conditions (Valentine et al., 2004b;Stolper et al., 2015;Douglas et al., 2016)}. The wetland, inland water, and 'in-vitro' regression relationships are not consistent with a constant value of $\alpha_H$ (Fig. 5). Our comparison supports previous inferences that the in-vitro line of Waldron et al., (1999a) provides a good estimate of the slope of environmental $\delta^2$H-H$_2$O vs. $\delta^2$H-CH$_4$ relationships. This slope is likely controlled by the relative proportion of acetoclastic and hydrogenotrophic methanogenesis, the net kinetic isotope effect associated with these two methanogenic pathways, and variance in $\delta^2$H of acetate (Waldron et al., 1998;Waldron et al., 1999a;Valentine et al., 2004a), but the relative importance of these variables remains uncertain.

In particular, the $\delta^2$H of acetate methyl hydrogen is probably influenced by environmental $\delta^2$H-H$_2$O, and therefore likely varies geographically as a function of $\delta^2$H$_p$, as originally hypothesized by Waldron et al. (1999a). To our knowledge there are no measurements of acetate or acetate-methyl $\delta^2$H from natural environments with which to test this hypothesis. In general, variability in the $\delta^2$H of environmental organic molecules in lake sediments and wetlands, including fatty acids and cellulose, is largely controlled by $\delta^2$H-H$_2$O (Huang et al., 2002;Sachse et al., 2012;Mora and Zanazzi, 2017), albeit with widely varying fractionation factors. The $\delta^2$H of methoxyl groups in plants has also been shown to vary as a function of $\delta^2$H-H$_2$O (Vigano et al., 2010). Furthermore, culture experiments with acetogenic bacteria imply that there is rapid isotopic exchange between H$_2$ and H$_2$O during chemoautotrophic acetogenesis (Valentine et al., 2004a), implying that the $\delta^2$H of chemoautotrophic acetate is also partially controlled by environmental $\delta^2$H-H$_2$O. Incubation experiments, such as those included in the 'in-vitro line' (Schoell, 1980;Sugimoto and Wada, 1995;Waldron et al., 1998), probably contain acetate-$\delta^2$H that varies as a function of ambient $\delta^2$H-H$_2$O, given that the acetate in these incubation experiments was actively produced by fermentation and/or acetogenesis during the course of the experiment. This differs from pure cultures of methanogens, where acetate is provided in the culture medium and therefore would not vary in its $\delta^2$H value (Valentine et al., 2004b;Gruen et al., 2018).

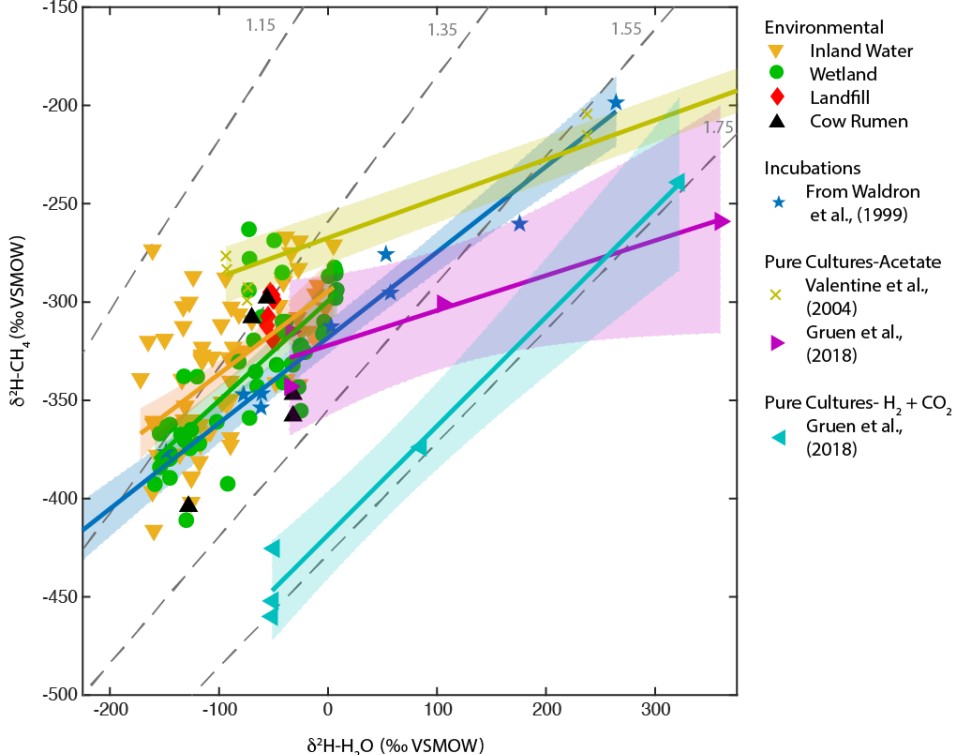

**Figure 5: Scatter plots of $\delta^2$H-CH$_4$ vs. $\delta^2$H-H$_2$O for wetlands, inland waters, landfills, and cow rumen, compared with incubation and pure-culture experiments. Regression lines and confidence intervals corresponding to each dataset (except landfills and cow rumen) are shown. Dashed gray lines indicate constant values of $\alpha_H$. Regression line statistics are listed in Supplemental Table S2. Plotted $\delta^2$H-H$_2$O values are 'best-estimate' values for wetlands and inland waters, measured values for culture experiments, and a combination of measured values and annual $\delta^2$H$_p$ for landfills and cow rumen (See supplemental Table S3 for more details).**

**3.4 Relationship of $\delta^2$H-CH$_4$ with $\delta^{13}$C-CH$_4$, $\delta^{13}$C-CO$_2$, and $\alpha_C$**

As shown in Fig. 3, there is a large amount of residual variability in $\delta^2$H-CH$_4$ that is not explained by $\delta^2$H-H$_2$O. Several biogeochemical variables have been proposed to influence freshwater $\delta^2$H-CH$_4$ independently of $\delta^2$H-H$_2$O, including the predominant biochemical pathway of methanogenesis (Whiticar et al., 1986;Whiticar, 1999;Chanton et al., 2006), the extent of methane oxidation (Happell et al., 1994;Waldron et al., 1999a;Whiticar, 1999;Cadieux et al., 2016), isotopic fractionation resulting from diffusive gas transport (Waldron et al., 1999a;Chanton, 2005), and differences in the thermodynamic favorability or reversibility of methanogenesis (Valentine et al., 2004b;Stolper et al., 2015;Douglas et al., 2016). These variables are also predicted to cause differences in $\delta^{13}$C-CH$_4$, $\delta^{13}$C-CO$_2$, and $\alpha_C$. Therefore, we analysed co-variation between $\delta^2$H-CH$_{4,W0}$ (see definition in Sect. 2.2.3) and $\delta^{13}$C-CH$_4$, $\delta^{13}$C-CO$_2$, and $\alpha_C$ to see if it could partially explain the residual variability in $\delta^2$H-CH$_4$ (Fig. 6).

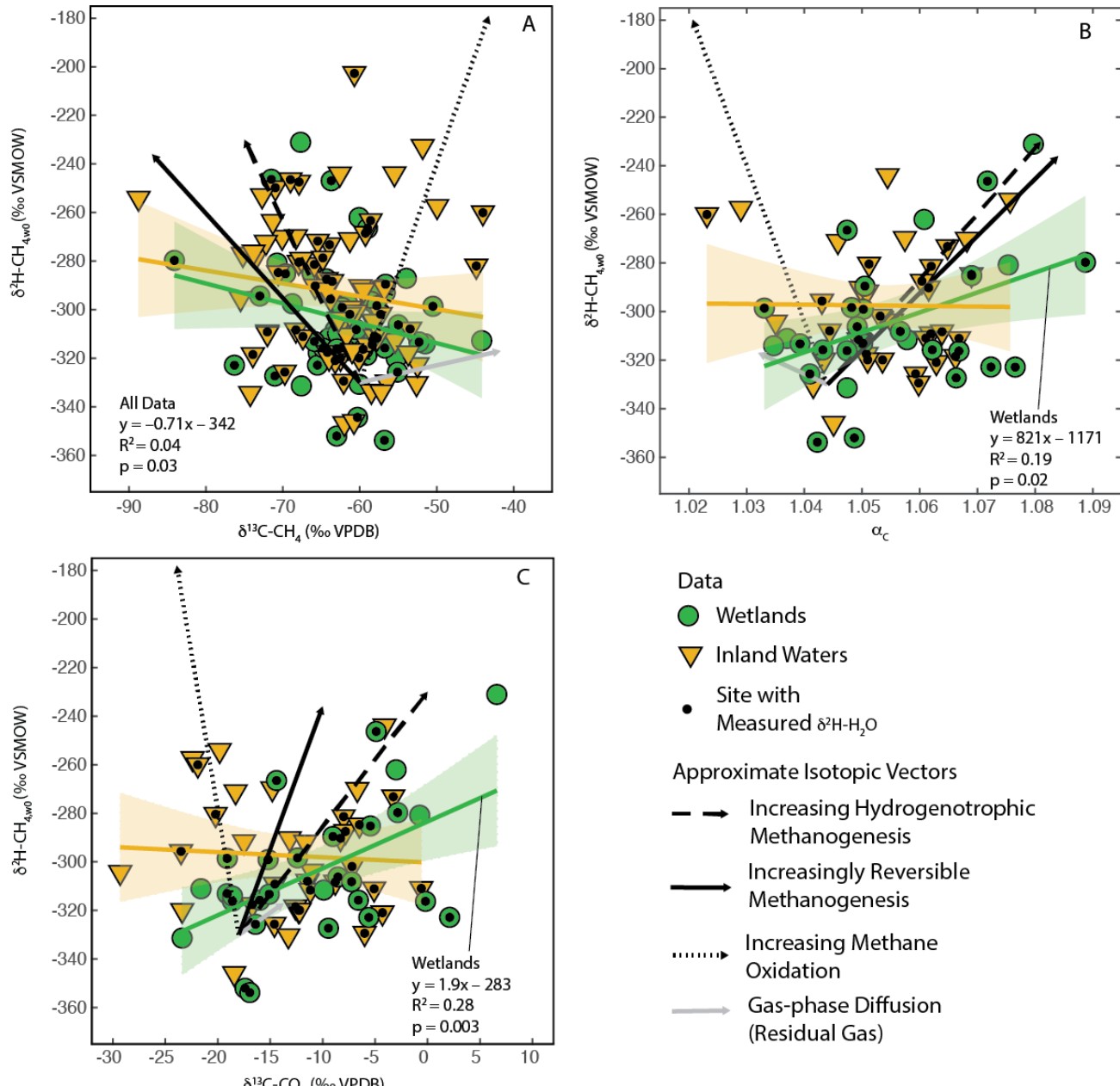

Figure 6: Scatter plots of $\delta^2$H-CH$_{4,w0}$ vs. (A) $\delta^{13}$C-CH$_4$, (B) $\alpha_c$, and (C) $\delta^{13}$C-CO$_2$. Approximate vectors for isotopic co-
variation related to four biogeochemical variables are shown. See details in Sect. 3.4 and the supplemental text. Regression
relationships are shown for wetland and inland water sites, with envelopes indicating 95% confidence intervals. Regression
statistics are shown here for relationships with significant correlations (p < 0.05). All regression statistics are detailed in
Supplemental Table S4.

In order to facilitate interpretation of isotopic co-variation, we estimated approximate vectors of predicted isotopic co-variation for the four variables being considered (Fig. 6). We emphasize that these vectors are uncertain, and while they can be considered indicators for the sign of the slope of co-variation and the relative magnitude of expected isotopic variability, they are not precise representations of the slope or intercept of isotopic co-variation. In reality, isotopic co-variance associated with these processes likely varies depending on specific environmental conditions, although the sign of co-variance should be consistent. The starting point for the vectors is arbitrarily set to typical isotopic values for inferred acetoclastic methanogenesis in freshwater systems (Whiticar, 1999). We based the vectors for differences in the dominant methanogenic pathway and methane oxidation on Figures 8, 5, and 10 in Whiticar (1999). These figures are widely applied to interpret environmental isotopic data related to $CH_4$ cycling. However, we note that both environmental and experimental research has questioned whether differences in the dominant methanogenic pathway has an influence on $\delta^2H$-$CH_4$ (Waldron et al., 1998;Waldron et al., 1999a). Differences in $\delta^2H$-$CH_4$ between hydrogenotrophic and acetoclastic methanogenesis are likely highly dependent on both the $\delta^2H$ of acetate and the carbon and hydrogen kinetic isotope effects for both methanogenic pathways, both of which are poorly constrained in natural environments and are likely to vary between sites (see Sect. 3.3.1). We did not differentiate between anaerobic and aerobic methane oxidation, and the vectors shown are similar to experimental results for aerobic methane oxidation (Wang et al., 2016).

The vector for isotopic fractionation related to gas-phase diffusion is based on the calculations of Chanton (2005), and indicates isotopic change for residual gas following a diffusive loss. Gas-liquid diffusion is predicted to have a much smaller isotopic effect (Chanton, 2005). The vector for differences in enzymatic reversibility are based on experiments where $CH_4$ and $CO_2$ isotopic compositions were measured together with changes in methane production rate or Gibbs free energy (Valentine et al., 2004b;Penning et al., 2005). We note that these studies did not measure $\delta^2H$-$CH_4$ in the same experiments as $\delta^{13}C$-$CH_4$ or $\delta^{13}C$-$CO_2$, implying large uncertainty in the co-variance vectors. More detail on the estimated vectors is provided in the Supplementary Text.

We observe significant positive correlations between $\delta^2H$-$CH_{4,W0}$, calculated using best estimate $\delta^2H$-$H_2O$, and both $\delta^{13}C$-$CO_2$, and $\alpha_C$ for wetland sites (Fig. 6B,C; Supplemental Table S4). We do not observe a significant correlation between these variables for inland water sites or for the dataset as a whole. We also observe a very weak, but significant, negative correlation, between $\delta^2H$-$CH_{4,W0}$ and $\delta^{13}C$-$CH_4$ for all sites, but not for data disaggregated into wetlands and inland water categories (Fig. 6A). The significant correlations shown in Figure 6 should be interpreted with caution, since repeating this analysis only using sites with measured $\delta^2H$-$H_2O$ does not result in any significant correlations (Supplemental Table S4). It is unclear whether this different result when using best-estimate or measured $\delta^2H$-$H_2O$ represents a bias related to estimating $\delta^2H$-$H_2O$ using $\delta^2H_p$, or is an effect of the much smaller sample size for sites with $\delta^2H$-$H_2O$ measurements. If accurate, the observed significant positive correlations in Figures 6B and C suggest that residual variability in $\delta^2H$-$CH_4$ in wetlands is more strongly controlled by biogeochemical variables related to

methanogenesis, namely differences in methanogenic pathway or thermodynamic favorability, than post-production processes such as diffusive transport and $CH_4$ oxidation. However, the residual variability in $\delta^2H$-$CH_4$ explained by $\delta^{13}C$-$CO_2$ and $\alpha_C$ in wetlands is relatively small, specifically between 19 to 28% based on the $R^2$ values in Figures 6B and C. For inland water sites our analysis suggests that no single biogeochemical variable has clear effect in controlling residual variability in $\delta^2H$-$CH_4$. It is intriguing that we observe the strongest correlation in wetlands between $\delta^2H$-$CH_{4,W0}$ and $\delta^{13}C$-$CO_2$, since it is probable that a wide range of biotic and abiotic processes unrelated to methane cycling influence $\delta^{13}C$-$CO_2$. This suggests that measurements of $\delta^{13}C$-$CO_2$ are important for future research on environmental variables controlling wetland $\delta^2H$-$CH_4$.

Overall, our results are not consistent with arguments that residual variability in freshwater $\delta^2H$-$CH_4$ is dominantly controlled by either differences in methanogenic pathway (Chanton et al., 2006), or post-production processes (Waldron et al., 1999a). Instead they highlight the combined influence of a complex set of variables and processes that are difficult to disentangle on an inter-site basis using $\delta^{13}C$ measurements alone. It is also important to note the likely importance of variables that could influence $\delta^{13}C$-$CH_4$ or $\delta^{13}C$-$CO_2$ but not necessarily affect $\delta^2H$-$CH_4$, including variance in the $\delta^{13}C$ of soil or sediment organic matter (Conrad et al., 2011;Ganesan et al., 2018), diverse metabolic and environmental sources and sinks of $CO_2$ in aquatic environments, and Rayleigh fractionation associated with $CH_4$ carbon substrate depletion (Whiticar, 1999). Finally, the possible role of other carbon substrates, such as methanol, in $CH_4$ production could be important in controlling isotope variability. Culture experiments suggest that $CH_4$ produced from methanol has low $\delta^{13}C$ and $\delta^2H$ values relative to other pathways (Krzycki et al., 1987;Penger et al., 2012;Gruen et al., 2018), although the importance of this difference in environmental $CH_4$ is unclear.

Further research examining intra-site isotopic co-variation, which largely avoids complications associated with estimating $\delta^2H$-$H_2O$, would help to more clearly resolve the relative importance of these processes, and how they vary between environments. Expanded research using methyl fluoride to inhibit acetoclastic methanogenesis (Penning et al., 2005;Penning and Conrad, 2007;Conrad et al., 2011), with a particular focus on $\delta^2H$-$CH_4$ measurements, would also help to clarify the importance of methanogenic pathway on isotopic co-variation. Finally, an expanded application of measurements of clumped isotopes, which have distinctive patterns of variation related to these processes (Douglas et al., 2016; Douglas et al., 2017; Young et al., 2017; Douglas et al., 2020), would also be of value in determining their relative importance in controlling $\delta^2H$-$CH_4$ values in freshwater environments.

**3.5 Differences in $\delta^2H$-$CH_4$ and $\delta^{13}C$-$CH_4$ by latitude**

When analysing all sites together we found a significant difference in the distribution of $\delta^2H$-$CH_4$ between high-latitude sites (median: -351‰) and both low (median: -298‰) and mid-latitude sites (median: -320 ‰) (Fig. 7A). However, we did not find a significant difference in the distribution of low- and mid-latitude sites. Similar differences

were found when the data were disaggregated into wetland and inland water sites. We also found that the distribution of $\delta^{13}C$-$CH_4$ for low latitude sites (median: -61.6‰) was significantly higher than for high latitude sites (median: −63.0‰), but that mid-latitude sites (median: -60.3‰) were not significantly different from the other two latitudinal zones (Fig. 7B). The observed difference by latitudinal zone in $\delta^{13}C$-$CH_4$ appears to be driven primarily by latitudinal differences between inland water sites, where a similar pattern is found. In wetland sites we found no significant differences in the distribution of $\delta^{13}C$-$CH_4$ by latitude.

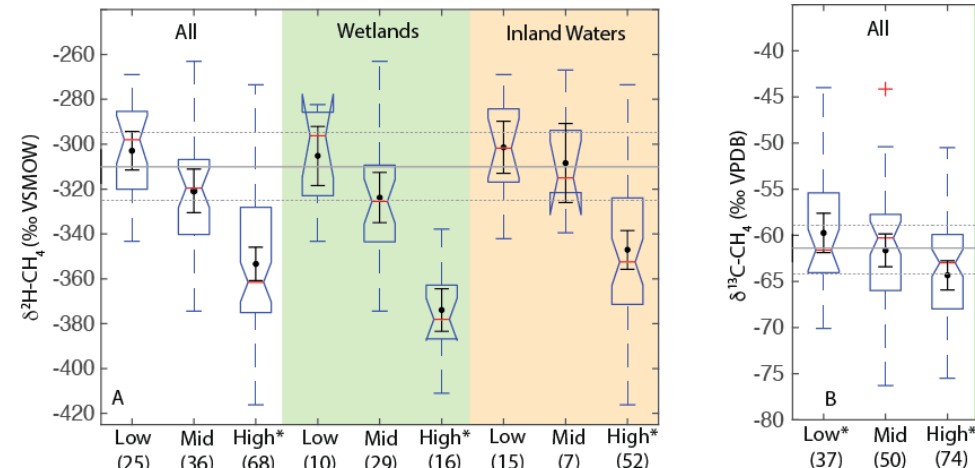
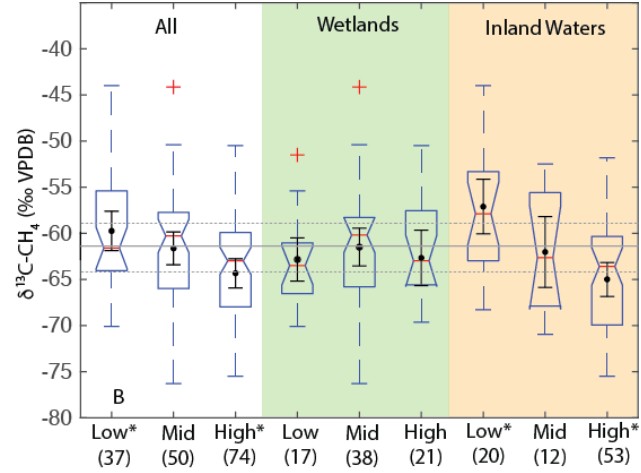

**Figure 7: Boxplots of (A) $\delta^{2}H$-$CH_4$ and (B) $\delta^{13}C$-$CH_4$ for sites differentiated by latitude, for all data, wetlands, and inland waters. Numbers in parentheses indicate the number of sites for each category. Red lines indicate medians, boxes indicate 25th and 75th percentiles, whiskers indicate 95th and 5th percentiles, and outliers are shown as red crosses. Notches indicate the 95% confidence intervals of the median value; where notches overlap the edges of the box this indicates the median confidence interval exceeds the 75th or 25th percentile. Black points and error bars indicate the category mean and 95% confidence interval of the mean. Gray lines indicate the estimated flux-weighted mean values for global freshwater $CH_4$, and dashed lines indicate the 95% confidence interval of this value.. Asterisks in (A) indicate that high-latitude sites have significantly different distributions from other latitudinal bands. Asterisks in (B) indicate groups that have significantly different distributions from one another, within a specific environmental category. Two extremely low outliers (<-80‰; high latitude wetland and inland water) are not shown in (B).**

Estimates of flux-weighted mean freshwater $\delta^{2}H$-$CH_4$ and $\delta^{13}C$-$CH_4$, calculated using the Monte Carlo approach described in Sect. 2.4, are -310±15‰ (Fig. 7A) and -61.5±3‰ (Fig. 7B) respectively. Flux weighted mean values for natural wetlands (not including inland waters or rice paddies) are -310±25‰ for $\delta^{2}H$-$CH_4$ and -63.9±3.3‰ for $\delta^{13}C$-$CH_4$. Flux weighted mean values for inland waters are -309±31‰ for $\delta^{2}H$-$CH_4$ and -60±5.7‰ for $\delta^{13}C$-$CH_4$. As discussed in Sect. 2.4 there are limited data in our dataset or that of Sherwood et al., (2017) from $C_4$ plant dominated wetlands, and therefore our low-latitude and flux-weighted mean $\delta^{13}C$-$CH_4$ values for wetlands are probably biased towards low values.

Differences in $\delta^{2}H$-$CH_4$ by latitude has the potential to aid in geographic discrimination of freshwater methane sources, both because it is based on a clear mechanistic linkage with $\delta^{2}H$-$H_2O$ (Figs. 3 and 4), and because geographic variation in $\delta^{2}H$-$H_2O$ is relatively well understood (Bowen and Revenaugh, 2003;Bowen et al., 2005). However, recent studies of atmospheric $\delta^{2}H$-$CH_4$ variation have typically not accounted for geographic variation in source signals. As an

example, Rice et al., (2016) apply a constant $\delta^2$H-CH$_4$ of -322‰ for both low-latitude (0-30º N) and high latitude (30-90º N) wetland emissions. Based on our dataset this estimate is an inaccurate representation of wetland $\delta^2$H-CH$_4$ for either 0-30º N (mean: -305±13‰) or 30-90º N (mean: -345±11‰). Studies of ice core measurements have more frequently differentiated freshwater $\delta^2$H-CH$_4$ values as a function of latitude. For example, Bock et al., (2010) differentiated $\delta^2$H-CH$_4$ between tropical (-320‰) and boreal (-370‰) wetlands. This tropical wetland signature is significantly higher than our estimate of low-latitude wetland $\delta^2$H-CH$_4$, although the boreal wetland signature is similar to our mean value for high-latitude wetlands (-374±10‰). Overall, our results imply that accounting for latitudinal variation in freshwater $\delta^2$H-CH$_4$, along with accurate latitudinal flux estimates, is important for developing accurate estimates of global freshwater $\delta^2$H-CH$_4$ source signatures.

Our analysis indicates significant differences in the distribution of freshwater $\delta^{13}$C-CH$_4$ between the low- and high-latitudes, but mid-latitude sites cannot be differentiated. Furthermore our results do not indicate significant latitudinal differences in $\delta^{13}$C-CH$_4$ for wetland sites in particular, whereas we do observe significant differences between the low- and high-latitudes for inland water sites. This is in contrast to previous studies that have inferred significant differences in wetland $\delta^{13}$C-CH$_4$ by latitude (Bock et al., 2010;Rice et al., 2016;Ganesan et al., 2018). An important caveat is that we have not analyzed a comprehensive dataset of freshwater $\delta^{13}$C-CH$_4$, for which there are much more published data than for $\delta^2$H-CH$_4$. However, our analysis does comprise the largest dataset of freshwater $\delta^{13}$C-CH$_4$ compiled to date (See Sect. 2.3). In addition, our analysis does not take into account the geographic distribution of different ecosystem categories, although we do not find significant differences in $\delta^{13}$C-CH$_4$ between ecosystem categories (Fig. 8; Sect. 3.6). Low-latitude ecosystems dominated by C$_4$ plants are especially underrepresented both in our dataset and that of Sherwood et al., (2017), and accounting for this would likely lead to a more enriched low-latitude wetland $\delta^{13}$C-CH$_4$. In contrast, high-latitude ecosystems, including bogs, are relatively well represented in these datasets (Fig. 8), and we suggest that inferences of especially low $\delta^{13}$C-CH$_4$ in high-latitude wetlands (Bock et al., 2010;Rice et al., 2016;Ganesan et al., 2018) are not consistent with the compiled dataset of in-situ measurements. However, we note that atmospheric estimates of high-latitude wetland $\delta^{13}$C-CH$_4$ (~-68±4‰; Fisher et al., 2011) are lower than the median or mean value shown in Figure 7B, and are in close agreement with the relatively low values predicted by (Ganesan et al., 2018). Ombrotrophic and minerotrophic peatlands have distinctive $\delta^{13}$C-CH$_4$ signatures (Bellisario et al., 1999;Bowes and Hornibrook, 2006;Hornibrook, 2009), with lower signatures in ombrotrophic peatlands. We did not differentiate peatlands by trophic status, and it is possible that the dataset of high-latitude wetland in-situ measurements is biased towards minerotrophic peatlands with relatively high $\delta^{13}$C-CH$_4$.

Latitudinal differences in $\delta^{13}$C-CH$_4$ inferred by Ganesan et al. (2018) were based on two key mechanisms: (1) differences in methanogenic pathway between different types of wetlands, especially between minerotrophic fens and ombrotrophic bogs; and (2) differential inputs of organic matter from C$_3$ and C$_4$ plants. Because inferred latitudinal

differences in $\delta^{13}C$-$CH_4$ and $\delta^2H$-$CH_4$ are caused by different mechanisms, they could be highly complementary in
validating estimates of freshwater emissions by latitude. It is also important to note that previous assessments of
latitudinal differences in $\delta^{13}C$-$CH_4$ did not include inland water environments. Our analysis suggests that latitudinal
variation in $\delta^{13}C$-$CH_4$ in inland waters may be more pronounced than in wetlands, although the mechanisms causing this
difference will need to be elucidated with further study.  A benefit of geographic discrimination based on $\delta^2H$-$CH_4$ is that
the same causal mechanism applies to all freshwater emissions, including both wetlands and inland waters.
**3.5.1 Potential for geographic discrimination of other microbial methane sources based on $\delta^2H$-$CH_4$**
We speculate that latitudinal differences in $\delta^2H$-$CH_4$ should also be observed in other fluxes of microbial methane from
terrestrial environments, including enteric fermentation in livestock and wild animals, manure ponds, landfills, and
termites. This is because microbial methanogenesis in all of these environments will incorporate hydrogen from
environmental water, and therefore will be influenced by variation in precipitation $\delta^2H$. There are limited data currently
available to test this prediction, but $\delta^2H$-$CH_4$ data from cow rumen and landfills are available with either specified
locations or $\delta^2H$-$H_2O$ (Burke Jr, 1993;Levin et al., 1993;Liptay et al., 1998;Bilek et al., 2001;Wang et al., 2015;Teasdale
et al., 2019). These data plot in a range that is consistent with the $\delta^2H$-$CH_4$ vs. $\delta^2H$-$H_2O$ relationships for freshwater $CH_4$
(Fig. 5). Landfill data are only available for a very small range of estimated $\delta^2H$-$H_2O$, making it impossible to assess for
geographic variation currently. $\delta^2H$-$CH_4$ data from cow rumen span a much wider range, and express substantial variation
that is independent of $\delta^2H$-$H_2O$. However, the cow rumen data span a range that is similar to that observed in freshwater
environments. Based on these limited data, variation observed in incubation studies that simulate landfill conditions
(Schoell, 1980;Waldron et al., 1998), and our understanding of the influence of $\delta^2H$-$H_2O$ on microbial $\delta^2H$-$CH_4$ (Fig. 6),
we suggest that both landfill and cow rumen $\delta^2H$-$CH_4$ likely vary geographically as a function of $\delta^2H$-$H_2O$. If validated,
this variation could also be used to distinguish these $CH_4$ sources geographically. More data are clearly needed to test this
conjecture, and it will also be important to evaluate how closely annual or seasonal $\delta^2H_p$ corresponds to environmental
$\delta^2H$-$H_2O$ in both landfills and cow rumen. Relatedly, the $\delta^2H$ of $CH_4$ emitted by biomass burning or directly by plants
has also been shown to vary as a function of $\delta^2H$-$H_2O$ (Vigano et al., 2010;Umezawa et al., 2011).
**3.6 Differences in $\delta^2H$-$CH_4$ and $\delta^{13}C$-$CH_4$ by ecosystem**
When comparing ecosystems, we analyze $\delta^2H$-$CH_{4,W0}$ values to account for variability related to differences in $\delta^2H$-$H_2O$.
Ecosystem types are not evenly distributed by latitude, and therefore have different distributions of $\delta^2H$-$H_2O$ values.
There are differences in the median values by ecosystem, with rivers (-283‰) exhibiting relatively enriched median $\delta^2H$-
$CH_{4,W0}$, and fens (-310‰) and rice paddies (-314‰) exhibiting relatively low median values. However, given the small
sample sizes and large variance in most of these categories, our analysis does not infer a significant difference in the
distribution of $\delta^2$H-CH$_{4,W0}$ between ecosystems. Comparing the broader categories of inland waters and wetlands with a
we do find a significant difference in $\delta^2$H-CH$_{4,W0}$ distributions, with inland waters shifted towards higher values (median:
-296‰) than wetlands (median: -311‰). We repeated this analysis only including sites with measured $\delta^2$H-H$_2$O and
found the same results in terms of category differences (Supplemental Figure S1). We did not observe any significant
differences in $\delta^{13}$C-CH$_4$ distributions between ecosystems, ,nor was there a significant difference in $\delta^{13}$C-CH$_4$
distributions between inland waters and wetlands. The median $\delta^{13}$C-CH$_4$ value for bogs was relatively low (-66‰), while
median values for fens (-60.3‰) and rice paddies (-60.3‰) were relatively high, but there was a large range in values for
all of these ecosystems.

The significant difference in the distribution of $\delta^2$H-CH$_{4,W0}$ between the overarching categories of inland waters

and wetlands is primarily a result of the difference in $\delta^2$H-CH$_4$ between these environments in the high latitudes (Figs. 3,
4, and 7). We are unsure of the mechanism causing this difference, though it is likely related to a greater overall
prevalence of CH$_4$ oxidation in inland waters. As shown in Figure 6, the lack of positive co-variation between $\delta^2$H-
CH$_{4,W0}$ and $\delta^{13}$C-CO$_2$, and $\alpha_C$ could be interpreted to support a greater role for CH$_4$ oxidation to control $\delta^2$H-CH$_{4,W0}$ in
inland waters relative to wetlands, although this result requires further validation. In lakes that undergo seasonal
overturning and water column oxygenation there may be a greater overall effect of CH$_4$ oxidation than there are in
wetlands typically. The absence of significant differences in $\delta^2$H-CH$_{4,W0}$ distributions between specific ecosystem
categories could be the result of small samples sizes for most ecosystems. Further study could be targeted towards
verifying and testing the apparent differences shown in Figure 8A. Generally lower $\delta^2$H-CH$_{4,W0}$ in rice paddies and fens
could reflect a greater proportion of acetoclastic methanogenesis inferred for these ecosystems (Conrad and Klose,
1999;Hornibrook, 2009;Ganesan et al., 2018), or possibly more thermodynamically favorable methanogenesis related to
high carbon substrate, H$_2$, or nutrient concentrations. Both of these explanations would be consistent with the relatively
high median $\delta^{13}$C-CH$_4$ values in these ecosystems (Fig. 7B, see also Fig. 6A). High median values in river ecosystems, in
contrast, may be a function of generally greater rates of oxidation, given that these environments are also characterized
by relatively high $\delta^{13}$C-CH$_4$ (Fig. 8B), and the potential for greater water-column oxygenation in fluvial environments
with turbulent flow (Devol et al., 1987). However, our river dataset is highly biased towards the Amazon river basin, and
drawing firm conclusions will require a larger and more widely distributed dataset.

The absence of significant differences between ecosystems in terms of $\delta^{13}$C-CH$_4$ (Fig. 8B) is in contrast to

previous studies that have suggested that fens and bogs in particular have distinctive $\delta^{13}$C-CH$_4$ (Ganesan et al., 2018).
Bogs in particular have a very wide distribution of $\delta^{13}$C-CH$_4$ that could represent differences between minerotrophic and
ombrotrophic bogs (Hornibrook, 2009), which we did not differentiate in our dataset. This result should be interpreted
with caution given that our dataset is not a comprehensive compilation of published $\delta^{13}$C-CH$_4$ data, although it is the
largest compiled dataset available (Sect. 2.3). We argue that inferred differences in $\delta^{13}$C-CH$_4$ between wetland ecosystem
categories should be further verified with more comprehensive data assimilation and additional measurements.

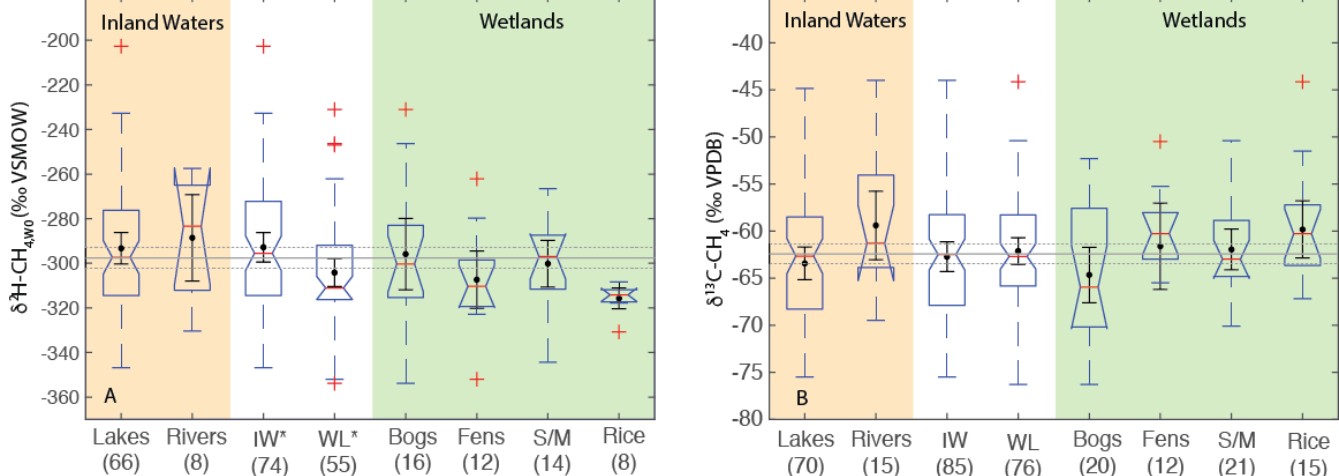

**Figure 5: Boxplots of (A) $\delta^2$H-CH$_{4,w0}$ and (B) $\delta^{13}$C-CH$_4$ for sites differentiated by ecosystem type. Numbers in parentheses**
**indicate the number of sites for each category. Boxplots parameters are as in Fig. 7. Black points and error bars indicate**
**the category mean and 95% confidence interval of the mean. Gray lines indicate the mean values across all categories and the**
**dashed lines indicate the 95% confidence interval of this value. Two extremely low outliers (<-80‰; lake and fen) are not**
**shown in (B). IW- Inland Waters; WL- Wetlands; S/M- Swamps and marshes. Asterisks in A indicate that inland waters and**
**wetlands have significantly different distributions.**

### 3.7 Differences in $\delta^2$H-CH$_4$ and $\delta^{13}$C-CH$_4$ by sample type

As with comparing ecosystems, when comparing sample types we analyze $\delta^2$H-CH$_{4,W0}$ values to normalize for variability
related to differences in $\delta^2$H-H$_2$O, since sample types are not distributed evenly by latitude. When comparing sample
types, dissolved CH$_4$ samples do not have a significantly different $\delta^2$H-CH$_{4,W0}$ distribution for the dataset as a whole, nor
is there a significant difference between these groups in wetland sites (Fig. 9A). There is, however, a significant
difference in inland water sites, with dissolved CH$_4$ samples having a more enriched distribution (median: -270‰) vs.
gas samples (median: -302‰). We repeated this analysis only including sites with measured $\delta^2$H-H$_2$O and found the
same results in terms of category differences (Supplemental Figure S2). We did not observe a significant difference in
the distribution of $\delta^{13}$C between dissolved and gas-phase CH$_4$ samples, either for the dataset as a whole or when the
dataset was disaggregated into wetlands and inland waters (Fig. 9B). We suggest that the higher $\delta^2$H-CH$_{4,W0}$ in dissolved
vs. gas samples for inland waters could be a result of generally greater oxidation of dissolved CH$_4$ in inland water
environments, potentially as a result of longer exposure to aerobic conditions in lake or river water columns. This is in
contrast to wetlands, where aerobic conditions are generally limited to the uppermost layers of wetlands proximate to the

water table. However, our dataset for inland water dissolved $CH_4$ is quite small (n=9), and more data are needed to test this hypothesis. Furthermore, it is unclear why oxidation in inland water dissolved $CH_4$ would be more strongly expressed in terms of $\delta^2H$-$CH_{4,W0}$ (Fig. 9A) than $\delta^{13}C$ (Fig. 9B).

Overall, our data imply that isotopic differences between dissolved and gas phase methane are relatively minor on a global basis, especially in wetlands. This result could imply that the relative balance of diffusive vs. ebullition gas fluxes doesn not have a large effect on the isotopic composition of freshwater $CH_4$ emissions. However, our study does not specifically account for isotopic fractionation occurring during diffusive or plant-mediated transport (Hornibrook, 2009), and most of our dissolved sample data are of *in-situ* dissolved $CH_4$ and not diffusive fluxes. More isotopic data specifically focused on diffusive methane emissions, for example using measurements of gas sampled from chambers, would help to resolve this question, as would more comparisons of the isotopic composition of diffusive and ebullition $CH_4$ emissions from the same ecosystem.

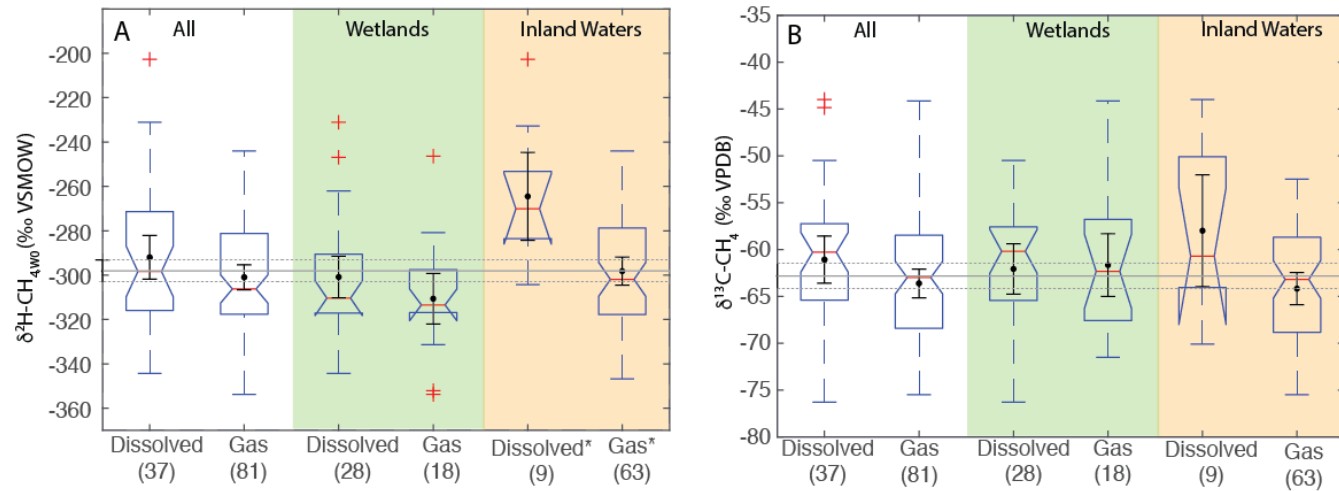

**Figure 9: Boxplots of (A) $\delta^2H$-$CH_{4,w0}$ and (B) $\delta^{13}C$-$CH_4$ for sites differentiated by sample type. Numbers in parentheses indicate the number of sites for each category. Boxplots parameters are as in Fig. 7. Black points and error bars indicate the category mean and 95% confidence interval of the mean. Gray lines indicate the mean values across all categories and the dashed lines indicate the 95% confidence interval of this value. Two extremely low outliers (<-80‰; dissolved wetland and gas inland water) are not shown in (B). Asterisks in A indicate that dissolved and gas-phase $CH_4$ samples from inland water sites have significantly different distributions.**

### 3.8 Estimates of global emissions source $\delta^2H$-$CH_4$ and $\delta^{13}C$-$CH_4$

Our mixing model and Monte Carlo analyses estimate a global source $\delta^2H$-$CH_4$ of -278±15‰, and a global source $\delta^{13}C$-$CH_4$ of -56.4±2.6‰ (Fig. 10). Monte Carlo sensitivity tests that only included uncertainty in either isotopic source signatures or flux estimates suggest that larger uncertainty is associated with isotopic source signatures (12‰ for $\delta^2H$; 2.2‰ for $\delta^{13}C$) than with flux estimates (8‰ for $\delta^2H$; 1.4‰ for $\delta^{13}C$). When correcting for wetland and biomass burning

emissions from C$_4$ plant ecosystems, as described in Section 2.4, our estimate of global source $\delta^{13}$C-CH$_4$ increases to -55.2±2.6‰.

Our estimate of global source $\delta^2$H-CH$_4$ is substantially higher than a previous bottom-up estimate using a similar approach (-295‰; Fig. 10) (Whiticar and Schaefer, 2007). This difference can be largely attributed by the application of more depleted $\delta^2$H-CH$_4$ source signatures for tropical wetlands (-360 ‰), and to a lesser extent boreal wetlands (-380 ‰), by Whiticar and Schaefer (2007). Another key difference is their inclusion of a relatively large flux and enriched $\delta^2$H-CH$_4$ signature from aerobic methane production from plants by Whiticar and Schaefer (2007), which is not included as a CH$_4$ source in our calculations.

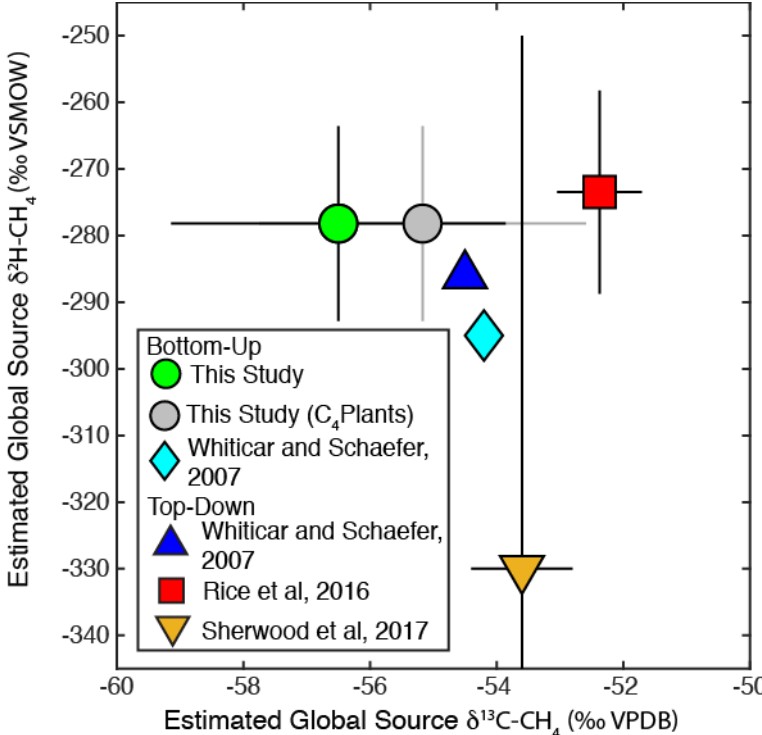

**Figure 10: Comparison of estimates of dual-isotope global source $\delta^2$H-CH$_4$ and $\delta^{13}$C-CH$_4$ from this and previous studies. Error bars from this study indicate the 2σ standard deviation from Monte Carlo analysis. Gray dot and error bars indicate an estimate corrected for the lack of data from wetlands and biomass burning in C$_4$ plant environments, as described in Sect. 2.4. Error bars for Rice et al., (2016), indicate the range of values estimated in that study between 1977-2005. Error bars for Sherwood et al., (2017) reflect the combined measurement uncertainty and uncertainty in sink fractionations reported in that study. Whiticar and Schaefer (2007) did not provide uncertainties for their estimates.**

Our bottom-up estimate of global source $\delta^2$H-CH$_4$ substantially overlaps the range of top-down estimates (-258 to -289‰) based on atmospheric $\delta^2$H-CH$_4$ measurements from 1977-2005 and a box model of sink fluxes and kinetic isotope effects (Rice et al., 2016) (Fig. 10). It is also within error of simpler top-down estimates calculated based on

mean atmospheric measurements and estimates of a constant sink fractionation factor (Whiticar and Schaefer,
2007;Sherwood et al., 2017). Sherwood et al., (2017) estimate a very wide range of possible global source $\delta^2$H-CH$_4$
values based on a relatively large atmospheric sink fractionation with large uncertainty (-235±80‰). This range overlaps
with our bottom up estimate, although its mid-point is substantially lower than our estimate. We argue that the box-model
method used to account for sink fractionations applied by Rice et al. (2016) probably provides a more accurate
representation of global-source isotopic composition than the other top-down estimates shown in Figure 10 (Whiticar and
Schaefer, 2007;Sherwood et al., 2017). The estimates of Rice et al. (2016) are also supported by the results of a global
inversion model. Overall, the overlap between our bottom-up estimate of global source $\delta^2$H-CH$_4$ with top-down estimates
is encouraging, and suggests that the estimates of emission source $\delta^2$H-CH$_4$ signatures applied in this study are
reasonably accurate. However, as discussed below, there is still substantial room to further constrain these estimates and
to reduce uncertainty.

Our bottom-up estimate of global source $\delta^{13}$C-CH$_4$ is lower than the other top-down and bottom-up estimates

shown in Figure 10. As discussed above, there is likely a bias in our freshwater CH$_4$ isotopic database in that it includes
very few wetland sites from C$_4$-plant dominated ecosystem. When correcting for this, as well as CH$_4$ emissions from
biomass burning (Fig. 10), our estimate shifts to a more enriched value that is within uncertainty of other estimates.
Clearly, accounting for the effect of C$_4$ plants in wetland and biomass burning CH$_4$ emissions, and potentially also in
enteric fermentation emissions, is important for accurate estimates of global source $\delta^{13}$C-CH$_4$. As discussed below, other
sources of error in both isotopic source signatures and inventory-based flux estimates could also partially account for our
relatively low global source $\delta^{13}$C-CH$_4$ estimate.

Previous studies have argued, on the basis of comparing atmospheric measurements and emissions source $\delta^{13}$C-

CH$_4$ signatures, that there are biases in global emissions inventories, specifically that fossil fuel emissions estimates are
too low, and that either microbial emissions estimates are too high (Schwietzke et al., 2016), or that biomass burning
estimates are too high (Worden et al., 2017). We argue that greater analysis of $\delta^2$H-CH$_4$ measurements could be valuable
for evaluating these and other emissions scenarios, as has been suggested previously (Rigby et al., 2012). This is
especially true for determining the relative proportion of fossil fuel and microbial emissions, since these sources have
widely differing $\delta^2$H-CH$_4$ signatures (Table 1). Currently, atmospheric $\delta^2$H-CH$_4$ measurements are not a routine
component of CH$_4$ monitoring programs, but we argue that based on both their value in constraining emissions sources
and sinks (Rigby et al., 2012), and the increasing practicality of high-frequency measurements (Chen et al.,
2016;Röckmann et al., 2016;Yacovitch et al., 2020), that there should be a renewed focus on these measurements.

The uncertainty in our bottom-up estimates, the overall greater uncertainty associated with isotopic source

signatures in our Monte Carlo calculations, and the apparent discrepancies for $\delta^{13}$C-CH$_4$ shown in Figure 10, also imply
that isotopic source signatures for specific sources could be greatly improved. As noted by Rigby et al. (2012), the impact
of improved isotopic source signatures increases as measurement precision improves. We have discussed above the
importance of increased data assimilation and measurements from tropical wetlands, with a particular focus on $C_4$ plant
dominated ecosystems. Using the isotopic source signal uncertainties and emissions fluxes shown in Table 1, we
identified the sources with the greatest flux-weighted uncertainty in isotopic signatures. Based on this analysis, the
greatest uncertainty for global source $\delta^2$H-$CH_4$ estimates comes from source signatures for enteric fermentation and
manure, low-latitude wetlands, onshore geological emissions, low-latitude and mid-latitude inland waters, termites, and
landfills. We identified the same source categories as having the greatest flux-weighted uncertainty for $\delta^{13}$C-$CH_4$, with
the exception of termites, but repeat the caveat that the underlying dataset is less comprehensive for $\delta^{13}$C-$CH_4$. We argue
that these source categories should be considered priorities for future emissions source isotopic characterization through
data assimilation and additional measurements. In particular, as discussed in Sect. 3.5.1, evaluation of possible latitudinal
variation in enteric fermentation and landfill $\delta^2$H-$CH_4$ is particularly promising.
**5 Conclusions**
Our analysis of an expanded isotopic dataset for freshwater $CH_4$ confirms the previous finding that $\delta^2$H-$H_2O$ is the
primary determinant of $\delta^2$H-$CH_4$ on a global scale (Waldron et al., 1999a), but also finds that the slope of this
relationship is probably flatter than was inferred previously (Fig. 3). This flatter slope is primarily the result of the
inclusion of a much larger number of high-latitude sites with low $\delta^2$H-$H_2O$ our dataset. We find that the inferred
relationship between $\delta^2$H-$CH_4$ and $\delta^2$H-$H_2O$ is not highly sensitive to whether measured $\delta^2$H-$H_2O$, modeled $\delta^2H_p$, or a
combination of the two (i.e. a best-estimate) is used to estimate $\delta^2$H-$H_2O$. This implies that gridded datasets of $\delta^2H_p$ or
isotope-enabled climate models could be used to predict the distribution of $\delta^2$H-$CH_4$ in the present, as well as under past
and future climates. Our analysis also suggests that annual $\delta^2H_p$ may be a better predictor for wetland $\delta^2$H-$CH_4$, while
seasonal $\delta^2H_p$ may be a better predictor of inland water $\delta^2$H-$CH_4$. The slope of $\delta^2$H-$CH_4$ vs. $\delta^2$H-$H_2O$ in both wetlands
and inland waters agrees well with that found in incubation experiments (Schoell, 1980;Sugimoto and Wada,
1995;Waldron et al., 1998;Waldron et al., 1999a), and we concur with previous inferences that this slope is partly
controlled by variation in the $\delta^2$H of acetate as a function of $\delta^2$H-$H_2O$ (Waldron et al., 1999a). Analysis of co-variation of
$\delta^2$H-$CH_{4,W0}$ with $\delta^{13}$C-$CH_4$, $\delta^{13}$C-$CO_2$, and $\alpha_C$ suggest that residual variation in $\delta^2$H-$CH_4$ is influenced by a complex set
of biogeochemical variables, including both variable isotopic fractionation related to methanogenesis, and post-
production isotopic fractionation related to $CH_4$ oxidation and diffusive gas transport. A significant positive correlation
between $\delta^2$H-$CH_{4,W0}$ and both $\delta^{13}$C-$CO_2$, and $\alpha_C$ in wetlands suggests that variable fractionation related to
methanogenesis pathway and thermodynamics may be more important in these environments, but this result is dependent
on the method used to estimate $\delta^2$H-$H_2O$ and requires further validation.

The dependence of $\delta^2$H-$CH_4$ on $\delta^2$H-$H_2O$ leads to clear latitudinal differences in $\delta^2$H-$CH_4$, with particularly low

values from high latitude sites (Fig. 4; Fig. 7A). The mechanism for latitudinal differences in $\delta^2$H-$CH_4$ is distinct from

proposed mechanisms for latitudinal differences in $\delta^{13}$C-CH$_4$ (Ganesan et al., 2018), implying that these two isotopic tracers are complementary in differentiating geographic emissions sources. We estimate a global flux-weighted $\delta^2$H-CH$_4$ signature from freshwater environments of -310±15‰, which is enriched relative to values used in previous source apportionment studies (Rice et al., 2016;Bock et al., 2017). We observe a significantly higher $\delta^2$H-CH$_{4,W0}$ distribution in inland waters relative to wetlands (Fig. 8A), which we suggest is a result of greater rates of CH$_4$ oxidation. We do not find significant differences between more specific ecosystem categories, but there are apparent differences between some wetland ecosystems that could be verified with larger datasets. We also do not find significant differences in $\delta^2$H-CH$_{4,W0}$ between sample types (Fig. 9A), with the exception of higher values in dissolved CH$_4$ relative to gas-phase CH$_4$ in inland water environments.

Our bottom-up estimate of the global $\delta^2$H-CH$_4$ source signature, -278±15‰, is higher than previous bottom-up estimates (Whiticar and Schaefer, 2007), but is within the range of top-down estimates based on atmospheric measurements and modeled sink fractionations (Rice et al., 2016), In contrast, our bottom-up estimate of global $\delta^{13}$C-CH$_4$, -56.4±2.6‰, is low relative to top-down estimates, which is partially explained by a lack of data from C$_4$ plant-dominated ecosystems in the freshwater CH$_4$ isotopic dataset. The agreement between bottom-up and top-down global $\delta^2$H-CH$_4$ estimates suggests that our current understanding of $\delta^2$H-CH$_4$ source signatures, when combined with inventory-based flux estimates (Saunois et al., 2020), is consistent with atmospheric measurements. This supports the argument that increased measurements and modeling of atmospheric $\delta^2$H-CH$_4$ could help to constrain global CH$_4$ budgets (Rigby et al., 2012). However, there is clearly a need to better constrain source signatures for both $\delta^2$H-CH$_4$ and $\delta^{13}$C-CH$_4$, especially from low-latitude microbial sources.

**Data Availability:** The datasets used in this paper (Supplementary Tables 1-4) are publicly available: Douglas, Peter; Stratigopoulos, Emerald; Park, Jenny; Phan, Dawson (2020): Data for geographic variability in freshwater methane hydrogen isotope ratios and its implications for emissions source apportionment and microbial biogeochemistry. figshare. Dataset. https://doi.org/10.6084/m9.figshare.13194833.v2

**Author Contribution:** PMJD designed the project, assisted with compiling the data, analyzed the data, and wrote the manuscript; ES and JP compiled the data, and assisted with analyzing the data and editing the manuscript; DP developed code for mixing model and Monte Carlo calculations, and assisted with analyzing the data and editing the manuscript.

**Competing Interests:** The authors declare they have no competing interests.

**Acknowledgments:** We thank all of the researchers whose published data made this analysis possible (See Supplemental Table 1). We also thank Susan Waldron, Edward Hornibrook, and an anonymous reviewer for constructive feedback.

This research was partially funded by McGill Science Undergraduate Research Awards to ES and JP and by NSERC
Discovery Grant 2017-03902 to PMJD.

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
