# Peer review of "Geographic variability in freshwater methane hydrogen isotope ratios and its implications for global isotopic source signatures"

_Biogeosciences, 2020_

## Referee Comment (RC1) · Susan Waldron (Referee) · 6 Dec 2020

Review of 'Global geographic variability in freshwater methane hydrogen isotope ratios and its implications for emissions source apportionment and microbial biogeochemistry' by Douglas et al.

Reviewer: Susan Waldron, University of Glasgow susan.waldron@glasgow.ac.uk

**Review questions:**

1.      Does the paper address relevant scientific questions within the scope of BG?
Yes

2.      Does the paper present novel concepts, ideas, tools, or data?

This manuscript tackles an overdue update of assessing how robustly $\delta D$-CH$_4$ can be described by $\delta D$-H$_2$O. Refining this relationships is valuable as isotope enabled Earth System Models could allow projection of $\delta D$-CH$_4$ source characterisation from locations where field measurements are not possible, and, when atmospheric $\delta D$-CH$_4$ dynamics are understood, then this may allow source apportionment to constrain better Earth Surface Fluxes. In addition, the authors explore if the variation in $\delta D$-CH$_4$ not described by $\delta D$-H$_2$O can be understood, exploring the hypothesis that differences in methanogenic pathway exhibit a control. The authors also apportion the database they have constructed into different habitats to explore if there is a habitat specific signature for both $\delta D$-CH$_4$ and $\delta^{13}$CH$_4$ and using these signatures and published flux estimates, upscale to calculate an atmospheric CH $\delta^{13}$CH$_4$ and $\delta D$-CH$_4$ paper. It is an ambitious paper of two halves (the controls on isotopic signatures vs. the habitat upscaling) and could be two papers, but their linkage is sensible

3.      Are substantial conclusions reached?

Substantial conclusions are reached, but the interrogative approach has weaknesses that propagate through substantial analytical reasoning and so the integrity of the conclusions is questionable. I detail this further below, but until the analytical approaches are reconsidered the conclusions are not securely reached

4.      Are the scientific methods and assumptions valid and clearly outlined?

My expertise is in understand the control on methane isotopic composition in the field and lab., and not on flux upscaling, so I can less securely comment on that.

With respect to understanding isotopic compositions: the methods are not all valid, particularly the reconstruction of missing $\delta D$-H$_2$O for a field measurement of $\delta D$-CH$_4$. The interrogation of this relationship (Fig. 2) lacks statistical rigour, and its propagation - a relationship that has bias and significant variability - is unconsidered in all analysis thereafter (as represented by Figs. 3-9 and possibly 10) and so this reasoning is flawed and the interpretations may be wrong.

The authors are not consistent in identifying when processes they are interpreting are based on hypothesised relationships and the impression is given such processes are certain (detailed below).

5.      Are the results sufficient to support the interpretations and conclusions?

No due to the flaw above, and its propagation in subsequent analysis. But this can be revisited.

6.      Is the description of experiments and calculations sufficiently complete and precise to allow their reproduction by fellow scientists (traceability of results)?

I found it difficult to follow the calculations behind $\alpha$C – an important part of the manuscript – when I was trying to compare other data sets with their approach.

7.      Do the authors give proper credit to related work and clearly indicate their own new/original contribution?

Largely bit not always, for example there is a large section in 4.31. that is repeating suggestions made in section 1.1. of Waldron et al 1999, but this work is unreferenced and so as written implies the review m/s is the first to have suggested this; the abstract does not make clear refining an existing phenomena observed and described similarly previously.

8.      Does the title clearly reflect the contents of the paper?

Broadly but not sure how "geographic variability in freshwater methane hydrogen isotope ratios has implications for microbial biogeochemistry" - the microbes are active with no knowledge $\delta D$…so this can be refined.

9.    Does the abstract provide a concise and complete summary?

Yes, but incorrect interpretation for the reasons above

10.    Is the overall presentation well structured and clear?

Yes, it flows quite well.

11.    Is the language fluent and precise?
Mostly – some language could be better constrained.

12.    Are mathematical formulae, symbols, abbreviations, and units correctly defined and used?
Yes

13.    Should any parts of the paper (text, formulae, figures, tables) be clarified, reduced, combined, or eliminated?

It is a paper with a lot of detail and so to follow it all the reader has to concentrate deeply for the results section. As such, and maybe in addition, the discussion from section 4 onwards seems in places repetitive.

14.    Are the number and quality of references appropriate?
Broadly yes – I suggest a group whose work may be missing in the intro.

15.    Is the amount and quality of supplementary material appropriate?
Yes, very helpful, but sheet 2 could make it clearer if the data offered is used in $\alpha_C$ or these are summarised data from other sources.

**Additional feedback**

Context: I have not anonymised this review as I believe we should be prepared to stand by our comments, and open reviews promote a fairer review. Thus, for the sake of clarity clear I am lead author in Waldron et al 1999, which is a key paper this BG paper builds on.

As outlined above the aims of this paper are admirable and the ambition to join field research with isotope-enabled ESM an important aspiration. Thus, I very much welcome this paper.

I also recognise that although my research has been predicated towards the interpretation that $\delta D$-$CH_4$ (in shallow freshwater environments) is primarily controlled by $\delta D$-$H_2O$ - either directly by incorporation of hydrogen when $CO_2$ and acetate are used as substrate, or indirectly as organic matter which is turn source hydrogen from environmental water – the community have continued to draw heavily on the hypothesis that methanogenic pathway also imparts a control on $\delta D$-$CH_4$ (heavily influenced by the work of Whiticar), and so it is not surprising the authors explore this approach, although it is not consistent with my research and so I remain sceptical.

I approached this BG manuscript from this position but open to new evidence of the control of methanogenic pathway in explaining the additional variability unexplained by $\delta D$-$H_2O$. For this reason, I welcomed the use of paired $\alpha_C$ as I concur the evidence is strong that changes in methanogenic pathway are reflected by changes in $\delta^{13}C$-$CH_4$.

I consider I can comment expertly on what controls the isotopic signature of methane and field relationships, but am not expert in estimating global mean compositions using the approaches in the second part of the paper, although I do understand the principles and can assess the scientific communication Thus I comment less on the upscaling.

**More detailed review**
Expansion on my response to review Q3-5.
*The database used:*
The substantive conclusions in this manuscript rely on a data set where $\delta D$-$H_2O$ does not exist for more than half the data: 53% of the sites do not have field measured $\delta D$-$H_2O$ (L88). In these cases,

$\delta$D-$H_2O$ is inferred from a reputable global precipitation database and a correlation observed for sites where measured values exist. The authors consider this relationship sufficiently robust to proceed to use the reconstructed $\delta$D-$H_2O$ where measured values do not exist. I disagree this is the case.

Figure 2 shows that for a given projected $\delta$D-$H_2O_p$, the field (measured) $\delta$D-$H_2O$ could be up to 50‰ more depleted or enriched (for example, compare the range in the y-axis for the field data for a predicted $\delta$D-$H_2O_p$ for ~ - 125‰ and -150‰,) and can be more enriched than $\delta$D-$H_2O$ by up to 20‰ (offset in the x-axis from the 1:1 line). Thus, there is both uncertainty and bias in the relationship that is now used to create $\delta$D-$H_2O$ for a field $\delta$D-$CH_4$.

The statistical integrity shown elsewhere in the manuscript is lacking in this section on reconstructing $\delta$D-$H_2O$, with the authors describing their predictive relationship as showing "generally good agreement" and proceeding to use it. The bias and variability in a predictive $\delta$D-$H_2O_p$ and thus how far it may be from the true $\delta$D-$H_2O$ appear unconsidered in any further analysis (no errors propagated through for estimated $\delta$D-$H_2O$?).

Further, I note that the data in table S3 supplementary information for which there are measured $\delta$D-$CH_4$ $\delta$D$H_2O$ fit closely to the in-vitro line from which Waldron et al 1999 project a global relationship - but the data with estimated $\delta$D$H_2O$ in table S3 do not. This is important for two reasons:

1. It confirms the predictive relationship in Waldron et al 1999 for $\delta$D-$CH_4$ from $\delta$D-$H_2O$ still has integrity, more so by adding in another methane-producing environment (innocula), a significant time gap, and another geographic locality.

2. If statement 1 is considered sound, then the poor fit of paired $\delta$D-$CH_4$ $\delta$D-$H_2O$ with predicted $\delta$D-$H_2O$ supports the assertion above that the relationship the authors are using here to reconstruct $\delta$D-$H_2O$ is questionable.

*The revised global database of paired $\delta$D-$CH_4$ $\delta$D-$H_2O$*

With the greatest of respect, using the predicted data produces an outcome that is like a 'house of cards' – all subsequent analysis using this data is built on a shaky foundation. I therefore think that incorporating paired $\delta$D-$CH_4$ $\delta$D-$H_2O_p$ in further analysis is flawed and offer two examples why:

1. It creates a new global line for $\delta$D-$CH_4$ $\delta$D-$H_2O$ that may be wrong.

2. It could lead to artefact in interpretation, which indeed may be 'visible' in the dependent analysis. For example, the data in Fig. 3b visually also appears to separate between paired $\delta$D-$CH_4$ $\delta$D-$H_2O$ data that are predicted (inland waters) and measured (wetlands), and if this is the case interpreting a biome difference here, and later in the paper, is also questionable.

With respect to the redefining of a new global $\delta$D-$CH_4$ $\delta$D-$H_2O$ and consideration of how this has changed from the relationship offered in Waldron et al 1999: unless the authors can produce a more robust estimation of $\delta$D-$H_2O_p$, the data that uses $\delta$D-$CH_4$ paired with predicted $\delta$D-$H_2O$ needs to be removed - for as noted earlier, there is insufficient confidence this is an accurate representation of the field situation and may create a false outcome. I suspect this will change the global relationship and increase the slope as paired data with $\delta$D-$H_2O_p$ visually appears to dominate the enriched samples.

Then for the comparison with Waldron et al 1999 the following approach would be more robust:

o   Please plot both the in-vitro and in-vivo relationship, and for the former its prediction intervals - which are missing from 3b and so give the sense of a poorer fit of Waldron et al 1999 to the bgd expanded field data set here.

o   Compare whether the in-vivo line is statistically different to the relationship generated from the data set presented in the bgd manuscript. This will allow confidence in any further discussion on how the relationship has been redefined (than just comparing slopes etc). If the two relationships are indistinguishable statistically, nuanced statements about differences in slope etc are meaningless – all that has happened is that the expanded data set has redefined better the field relationship for $\delta$D-$CH_4$ $\delta$D-$H_2O$ (as indicated likely in Waldron et al, 1999) - noting that this field relationship does not wholly reflect the relationship at production (see next point).

o   Assess whether the expanded field data set is predominantly [13]C-enriched compared to the in-vivo relationship described in Waldron et al 1999, and therefore consistent with an interpretation that differences in field $\delta$D-$CH_4$ may be an artefact of fractionating processes post-production than pathway per se This is advocated as I am still unaware of experimental

evidence methanogenic pathway in shallow freshwaters changes $\delta D$-$CH_4$, but there is evidence of processes, oxidation and mixing, causing enrichment, and so this approach is consistent with scientific principle of parsimony and interpreting data using the simplest approach.

*The inference of methanogenic pathway from $\alpha_C$*

To explore why the paired $\delta D$-$CH_4$-$\delta D$-$H_2O$ measurements are not fully described by the best fit line, the authors explore whether a difference in (dominant) methanogenic pathway is evident in the data. With no evidence from paired $\delta D$-$CH_4$-$\delta^{13}CH_4$ the authors draw on $\alpha_c$ as a proxy for methanogenic pathway to assess this. Step-wise regression is used to explore this. I think this is interesting and something to revisit when the paired data relying in predicted $\delta D$-$H_2O$ has been removed, but currently it is the next floor in the 'house-of-cards', reliant on data that we do not know to be accurate, and therefore the significant relationships that the authors infer changes in methanogenic pathway from, we do not know to be true.

The authors in their revision should be careful in the value of thinking about $\alpha_c$ for the following reasons: some of the literature generating $\alpha_c$ relies on assumption of differences in methanogenic pathway interpreted from differences in $\delta D$-$CH_4$, but there is competng evidence $\delta D$-$CH_4$ cannot be interpreted in this way (so $\alpha_c$ using $\alpha_c$ to infer methanogenic pathway in $\delta D$-$CH_4$ when $\delta D$-$CH_4$ has been used to infer methanogenic pathway becomes a circular, self-supporting and flawed approach).

To help here I would advise the authors to consider Waldron et al 1998 (Geomicrobiology, 15, 157-169), which contributes to the in-vitro line in Waldron et al 1999, but the authors do not cite so I am unsure if they are aware of the detail in this.

Here dominance of methanogenic pathway was changed in mixed culture (as would be found in the field) incubations, and $\delta D$-$CH_4$ monitored with time – so not just one measurement as may be misinterpreted from Waldron et al 1999. Except for one measurement broadly within analytical uncertainty, $\delta D$-$CH_4$ remained constant. However, $\delta^{13}CH_4$ did change and consistently with fractionation ranges for the methanogenic pathways thought to be dominant (as assessed from independent measurements of substrate turnover). I advise the authors to consult Waldron 1998 for two reasons:

1. The authors approach in the bgd paper to draw on $\delta D$-$CH_4$ to represent differences in methanogenic pathway would be stronger if they can provide an explanation for the constancy in $\delta D$-$CH_4$ while $\delta^{13}CH_4$ changes.

2. Waldron et al can also be used to calculate $\alpha_c$ (both from $CO_2$ and from estimated substrate composition). $\alpha_c$ $CO_2$-$CH_4$ generates values of 1.057 for the period when $CO_2$ reduction is considered dominant (i) and 1.055 when acetoclastic methanogenesis is considered dominant (ii). These are very similar and it would be valuable to understand how the authors interpret this when they infer much wider ranges in $\alpha_c$. For clarity $\delta^{13}CO_2$ and $\delta^{13}CH_4$ respectively for (i) were -8.3 ‰ and -62‰ , and for (ii) were 1.55‰ and -47.5‰

This concludes my main comment on the parts of the manuscript that I consider are not yet robust enough to concur with the interpretation.
In working through the manuscript, I made the following other comments that the authors should also consider.

**Abstract generally**: Is clear and summarises the paper but projects a future methane emissions scenario (L25-26) before the modelling and assessment of how well this approach can reconstruct current estimates ( L27-30) and this seems in the wrong order to me, given the former has a reliance on the latter. Further, the abstract does not acknowledge this research is augmenting the research that historically first documented the global relationship between $\delta D$-$CH_4$ and $\delta D$-$H_2O$ easily addressed for example by changing L12 to 'We have refined the existing global relationship between $\delta D$-$CH_4$ - $\delta D$-$H_2O$ by the compilation of a more extensive global dataset…."

L28: The authors postulate the mismatch is dependent only on the work of others (emission inventories, etc) and not possibly an error in their approach. Scientifically this is not correct – both 'sides' could have errors.

L19: results do not imply; one interprets data to generate a 'result'.

L22: high (more $^{13}$C-enriched) in rivers and bogs - this is the dataset that has more $\delta$D-H$_2$O projected, so is this an artefact of the modelling than a real biome-specific difference?

L27: integrated (by mass balance) not combined (which is used when sources are added) – which I know the authors have done (L204) but the descriptor is incorrect here.

**Intro:**

L36: I think the following references is missing: Variability in Atmospheric Methane From Fossil Fuel and Microbial Sources Over the Last Three Decades. / Thompson et al: Geophysical Research Letters, Vol. 45, No. 20, 28.10.2018, p. 11499-11508 (and I invite the authors to wonder if also some of the work from the Royal Holloway group should augment L47-51)

L59 & L83 Citations are given in chronological order of 1999b and 1999a which seems not typical convention to me (uncertain of the referencing convention for BG but for example the two references for Walter K are not in chronological order in the reference list so the in-paper citations would not be b then a due to this convention in the reference list?)

L68: Logic only follows that impact on $\delta^{13}$CH$_4$ can affect geographic provenancing if reader knows it can also affect $\delta$D-CH$_4$, so does this need to be made explicit?

L70: this implies that different ecosystems have different methanogenic pathways. More accurate text would be "differentiated geographically based on ecosystem differences in the relative strengths of different methanogenic pathways and $\delta^{13}$C of source organic matter" (as per the introduction of the Ganesam paper). Noting relative strengths is important, as a common mistake propagated in the literature and again here (L???) is to assume methanogenesis proceeds by one methanogenic pathway only – this would be rare, with field-based methane production contemporaneous from CO$_2$ and acetate, and varying temporally in strength as input of fresh OM changes seasonally (or not).

L84-85: sounds a bit defensive? How about "We have advanced existing compilations of freshwater $\delta$D-CH$_4$ by 1,2,3 …? I would remove significantly (statistical connotations) and just say larger as the number speak for themselves.

L91: The aims are clear (good) but 'then' and 'potential' not needed – the latter as embedded in implications that there is a potential for impact

**Methods:**

L106 & L117, 9L206 and possibly elsewhere): small w for where, as this follows from an unfinished sentence in both cases with the equation used in between

L136: the five ecosystem categories are not clear from this sentence: 'lakes' and 'rivers' and then there are five wetlands listed. Further, it is debatable that floodplains are aligned with rivers as CH$_4$ production would only occur when sediments are deoxygenated from standing water. So I would say more with ponds as the recession of water can be slow and could be like a pond drying in some situations. Noteworthy here is that gas loss from rivers is velocity dependent (see Long et al (2015) Hydraulics are a first order control on CO$_2$ efflux from fluvial systems Journal of Geophysical Research – Biogeosciences, 120, (doi:10.1002/2015JG002955), and similar references. This will also be the case with methane – possibly more so as insoluble, and may cause an isotope fractionation independent of degassing, and may also be a reason the Amazon rivers in Fig. 5 plot differently.

L139: Similarly, I question the scientific integrity in lumping lakes with rivers here – gas loss from river systems is controlled by hydrological processes primarily and there could be fractionations during emission from lotic systems that are different to lentic systems where diffusion and wind of lake thermal orographic processes control turnover. This starts to become important where these mean sources are used to simulate a resultant atmospheric composition e.g. L227. Thus, the authors should think about how to provide added confidence of the robustness of their catergorisation.

L145-148: Such categorisation is good, and the open access data set is very welcome. This categorisation relies on the integrity of the interpretation, but this integrity is important as the data analysis relies on this. With 131 sites it is impossible for the reviewer to know each site and so as a

check I can only look at my own data: L61 in the excel files. These methane samples were collected in-situ from porewater diffusing into samplers embedded in the peat (the GBC abstract notes in-situ and the methods clarifies at depth sampling) so I would classify as more aligned with dissolved porewater than diffusive flux (which is normally associated with the potential for oxidation and change in $\delta$ values ). Further I comment in the GBC paper there is a dynamic zone and interpret that is the section from which gas can be emitted. Mean $\delta$D-CH$_4$ here is -332 ± 17‰, more depleted the -294 ± 39‰ used in the table and subsequent data analysis. Thus, some feedback from the authors in the revised manuscript that their interpretations are not sensitive to the variation their interpretation of environment and which data to use would be valuable.

L152 – typically small – as this manuscript relies on several source of data estimation (here, $\delta^2$H$_2$O, it would be good to provide estimates as to what the maximum is this would manifest in $\delta$D (recognising that it changes with resolution and scale of figure and so this is challenging, but saying small is insufficient).

L177: the authors need to unpick for the reader the statement more as they have with L179 onwards. I am thus left to interpret the reasoning. I assume it is based on considerations that methanogenic pathway influences $\delta$D-CH$_4$? If so please see earlier substantive comments on this and decide whether to proceed in the revised manuscript.

L200: Clarify where the flux estimate comes from at this point – I presume from Saunois et al as in L209, but this should be clarified when first introduced. I am not expert enough to judge if the methodology for the bottom up flux section is sound, but it seems reasonable to me.

L 267: given the statistical approaches such as Monte Carlo bootstrapping used with the flux estimate section previously I would have expected more rigorous comparison should be undertaken here to show if there is a statistical offset between measured and predicted $\delta$D-H$_2$O than relying on descriptors of "generally good agreement" and using RMSE. The RMSE is a red herring if the lines generating 19 and 23 ‰ do not overlap - ?

Fig 2: Should the predicted (postulated and therefore dependent) not be regressed onto the measured (the true field value, so measured and independent and as a control of $\delta$D-CH$_4$ the one to get as close to the true value as possible)?

Fig 3B: this needs revisited once the δD-CH$_4$ -δD-H$_2$O predicted data has been removed as described above. There may still be an inland water specific difference here, but again that this may not be controlled by anything more complex than lentic and lotic freshwater systems having generalised differences in gas transport mechanism (ebullition or diffusion). These would be influenced by atmospheric and sediment interface boundary layer dynamics, transit time, depth of oxidative zone, lake stratification, and surface roughness, with the latter in turn influenced by wind speed, depth of water, and river flow velocity, slope. In other words, considerable methane isotope fractionation (enrichment) is possible, or not.

Fig 4. It is good to see this plotted but not surprising given δD-H$_2$O varies with latitude and δD-CH$_4$ varies with δD-H$_2$O. The same difficulties in estimating field δD-H$_2$O from modelled δD-H$_2$O are evident when considering δD-CH$_4$ as a function of predicted δD-H$_2$O. The authors need to note here that there may be an imbalance of where methane is sampled from globally and so if more measurements existed from the higher latitudes then there may be as much scatter as with the lower latitudes.

Section 3.4 jumps to something completely different with L313 "shifts to being controlled by changes in methanogenic pathway to being controlled by ….". There has not been clear discussion from the authors to date they are considering changes in methanogenic pathway of δD-CH$_4$ so this seems out of context. And yet L317 goes on to consider this in more detail. The key message in the Waldron et al 1999 paper is that considering methanogenic pathway a control on $\delta$D-CH$_4$ is misplaced and that "that 50% of the variation in natural δD-CH$_4$ samples can be explained by δD-H$_2$O, with isotopic fractionation post-production, or mixing with gas already fractionated likely responsible for most of the noise in the natural system". The analysis prior to section 3.4 may be more likely to support this interpretation than refute it, particularly when the data in Fig. 3.2. is appropriately compared (as described earlier), and so now considering data as a function of methanogenic pathway seems to be ignoring this. Indeed the authors observe they find no relationship between δ$^{13}$C-CH$_4$ and δ$^2$H-

$CH_{4,W0}$ which would be expected if $\delta^2H\text{-}CH_4$ was influenced by methanogenic pathway as $\delta^{13}C\text{-}CH_4$ is (Fig. 5a). Thus, the authors should not make clearer statements such as L312 of "shifts from being controlled by variation in methanogenesis pathway" are inferred controls.

Figs. 5b=c. The uncertainty around what $\alpha_c$ should be for different methanogenic pathways has been described earlier in this review. But additionally, although breakpoint analysis was used, there is a high dependence in this on data set that has enriched $\delta^2H\text{-}CH_4$ to generate opposing trends. The eye is drawn by the projected pathways, but if these was not included as we cannot be sure it is oxidation[1] and all the remaining data was considered in a weighted regression would there be trends?

[1]If the high $\delta^2H\text{-}CH_4$ is from the Amazonian rivers, there are shales in this basin that fuel C cycling (Vihermaa et al) and this could be thermogenic: $\delta^2H\text{-}CH_4$ is also consistent with this.

Vihermaa L.E., Waldron S. , Garnett M.H., and Newton J. (2014) Old carbon contributes to aquatic emissions of carbon dioxide in the Amazon. Biogeosciences, 11, 3635-3645. (doi: 10.5194/bgd-11-1773-2014).

It is remarkable Fig 7 is so consistent – this is very interesting. Is it what we would expect?

L370 discussion is over-interpretations given the differences between sites are not statistically significant. It would be ok to say the prevalence of more depleted $CH_4$ is greater in the ecosystems sampled but for example this could represent accessibility of field sites, or differential investment into research measurements in these areas, than group compositional differences per se. Ecosystem types are not evenly distributed by latitude (L370) – nor is resource for investment in field research with tropical regions of the Earth lacking measurement due to access or financial constraints – we need to start recognising what we have not measured is as important as what we measure.

Fig. 10 is tiny and needs to be bigger

L426 "roughly as strong a predictor". Too big a leap: explain how – from ice core gases "roughly is a colloquialism"

L487 – as noted earlier, the paired measured values plot on Waldron et al 1999 In-vitro line, consolidating further the significant of this line. Please acknowledge this.

L508 – in the revised manuscript please detail the % variation explained by $\delta D\text{-}H_2O$ and then additionally by $\alpha_c$ should this prove to still be important

L510 – this is the crux of what is new to explore in isotope biogeochemistry of methane and also the role of methanol substrates.

L519 – same comments as before about is there really a relationship, but why more points classified as oxidised with this pairing than with $\alpha_c$?

L551- Much of 4.31. is repeating statements first described in Waldron et al 1999 section 1.1., paragraph starting "In addition…" but this is not referenced and as written implies the authors are the primary source of this thinking. This is not the case and should be referenced appropriately to indicate this was first noted 20+ years ago.

L564 – please note pure cultures are not representative of the field processes of methane production and thus the batch cultures and other experimental data collated in Waldron et al 1998, 1999 are. This is not clear from the statement.

L569, please reverse the order of the references or remove Whiticar 1999. The Waldron 1999 paper is the one that is particularly focussed on the global relationship between $\delta D\text{-}CH_4$ $\delta D\text{-}H_2O$, and constructs the first global relationship, which this paper finds with new data is similar. This gives appropriate credit to the conceptual understanding. The Whiticar paper coplots $\delta D\text{-}CH_4$ $\delta D\text{-}H_2O$ but does not assert that " $\delta^2H\text{-}H_2O$ is a primary determinant of $\delta^2H\text{-}CH_4$ on a global scale", rather the focus is on the interpretation of how $\delta^2H\text{-}CH_4$ reflects methanogenic pathway or marine vs. freshwater.

To conclude: this has been an uncomfortable review for me to undertake as my position of not anonymising the review puts me up for public scrutiny, and  a misinterpreted that I am trying to defend my work and am unwilling to accept an addition to this. This does not represent my professional scientific principles, I would urge the authors to accept this is not the case - indeed in

the 1999 GCA paper I welcome refinement of my work. However, the authors have still not presented here compelling evidence that $\delta D\text{-}CH_4$ can represent well different methanogenic pathways and so the reliance of this in the manuscript I find troubling. I consider the $\alpha_c$ approach may be valuable in helping constrain the signal in $\delta D\text{-}CH_4$ that is not defined by $\delta D\text{-}H_2O$, but the current manuscript is not constraining uncertainty sufficiently and the approach is therefore flawed. I would urge the authors to find a way to better constrain projected $\delta D\text{-}H_2O$ and revisit this, or work with only measured data and revisit this. Their refined analysis should undertake rigorous statistical comparison with the existing field $\delta D\text{-}CH_4$ $\delta D\text{-}H_2O$ relationship from Waldron et al 1999 to say whether it is different (although the new larger dataset will likely be a more representative relationship that the community can go forward with), and adopt a parsimonious interpretation of variation within the data set, as that is least likely to induce an erroneous interpretation. The biome specific considerations and upscaling should also be revisited if the removal of biased and inaccurate data pairings changes the source bulk compositions, and further thought should be given to the basis for source differentiation based on scenarios of methane production and loss in this upscaling.

---

## Referee Comment (RC2) · Anonymous Referee #2 · 14 Dec 2020

Review of "Global geographic variability in freshwater methane hydrogen isotope ratios and its implications for emissions source apportionment and microbial biogeochemistry" by Douglas et al.

The paper investigates the relation between the hydrogen isotopic composition of methane emitted from freshwaters on the global scale and the isotopic composition of water and/or modeled precipitation, as well the carbon isotopic composition of methane and carbon dioxide. The authors analyze data from a large number of previous studies and apply statistical methods in order to evaluate correlations between the various sig-

natures. The statistics are applied in a straightforward manner. I am missing a more detailed/critical scientific analysis of differences between the results of this study and previous studies. This has two aspects: 1) The study uses more sites than previous studies for dD, and it uses modeled fields of dD in precipitation. Which of these differences is primarily responsible for the differences to the previous literature (or is it both)? 2) The study uses less sites than previous studies for d13C. Are the results from these sites still adequate to be used in a global extrapolation? The derived global average 13C source signature derived by the authors is almost certainly too light, given what we know about the fractionation in the sinks. Furthermore, I think that the errors assumed for the bottom-up determination of the global average the source signatures are too optimistic, and the discussion on the implications for the atmospheric isotope budget in section 4.6 and too simplistic. See detailed comments below.

Specific comments:

L37: I suggest citing Worden et al., 2017, where this point is shown particularly well.

L64: Maybe you want to include here, or later in the discussion section, that there are also other lines of evidence that the hydrogen isotopic composition of CH4 (and other trace gases) depends on the isotopic composition of the precipitation, e.g., CH4 from biomass burning across climatic zones (Umezawa et al.2011), CH4 produced by UV irradiation of leaves that were grown with isotopically distinct waters (Vigano et al., 2010) or molecular H2 produced in the combustion of wood from different climatic zones (Röckmann et al., 2010).

L109: Replace the factor 1000 by 1, the delta value is defined the correct way in line 105, and no factor 1000 is necessary.

L136: What are the 5 categories? This is not clear, to me it sounds like 4 categories.

L159: Is the annual average dD value of precipitation really the best estimator for a source that very likely has a strong seasonality?

[Figure]

L253, Figure 1: Many of the sites are hidden behind others so I cannot see the colors. Would this improve if the figure is enlarged? It may be useful to show by color or shape for which of the sites you have measured dD-H2O and for which not.

L244, Table 1: The d13C signatures for wetland have an opposite "latitudinal order" compared to what is usually assumed, i.e. they are higher at high latitudes and lower at low latitudes. The data in Table 1 for wetlands do not agree with the data presented in Figure 7. Please explain the difference. You mention that the dataset evaluated here is different from what other studies have used for d13C, so is your dataset now representative? Should this limited set of values be used in the upscaling later? The errors presented for the different source categories are too optimistic, especially for the fossil sources at the bottom of the table, but probably also for the wetland category.

L276, Fig 2 and related text: This is a key figure for the following analysis. In principle it is an interesting approach to use modeled dD values in case measurements are not available, but it is also a source of error. Although there is a generally good agreement, the slope is lower than 1 and this may contribute to the differences and thus may affect some of the further analysis.

L284: Maybe you could state briefly whether you can reproduce the slope of Waldron et al. when you use the same dataset. Just as a baseline.

L292: Figure 3a: It looks like the lower slope is caused by a lot of points where you have only modeled but no measured dD data near the low dD-H2O end. And these are mostly inland waters (Figure 3b). Can you evaluate this in more detail? Can this be caused by a bias in the modeled dDp? Probably not, but it is useful to investigate further to strengthen your argument.

L308: Would you find a correlation if you took the slope of Waldron et al. for calculating CH4,W0?

L323, Figure 5: Does it make sense that in b) only few points are classified as oxidation

influenced and in c) many more points? Does it make sense that in c) the very lowest dD value is in the group of the oxidation influenced points? I find the "pathway trend" concept a bit confusing, this indicates a smooth transition of dD-CH4,W0 with alpha_C or d13C_CO2. Is this a real trend, or rather a consequence of two different groups of data (acetoclastic and hydrogenotrophic sites)? Wouldn't it be useful in this case to show these two groups with two different colors, separated by the potential break points, rather than the trend areas?

L350 and Figure 7b, wetlands: These numbers do not agree with the data in Table 1.

L374-379: I get a bit confused by the diverging statements on significance with different tests, please try to reformulate, or add a sentence to synthesize.

L395-397: See points above: Are the uncertainties for the different categories adequate? Is there an issue with the difference between values in the text and table 1? Is the rather heavy d13C value for high latitude wetlands appropriate?

L406: Figure 10: This figure may require a bit more explanation. What does the x axis "emission flux change" mean for the points from Rice et al.? I think I can guess it, but it could be presented more clearly.

L431 ff: The differences to the previously published values from Waldron et al. should be discussed in some more detail. E.g., is there an influence from the modeled dDp values, or a certain sampling region? L439 ff: Same for the discussion of the environment type

L465, section 4.2.1: See comments above on the representativeness of the dataset analyzed here and possible consequences. You write that the dataset is not comprehensive or d13C, so should it be considered as representative? In this case, what have other studies potentially missed?

L483 ff: You may want to refer here to the studies I mentioned in the beginning that looked at other (non-microbial) sources.

L503 ff: See comments on Fig. 5 regarding the samples affected by oxidation.

L519 ff: The authors state that they do not observe a correlation between dD and d13C of CH4. Nevertheless, the vast majority of the points in Fig 5a seem to fall in the range of the "pathway trend" (I find the term misleading, see comments above). Does this not mean that the two groups (acetate fermentation and CO2 reduction) still form distinct distributions?

L549: the remark on the intercepts does not add much and is rather trivial when the slope is different.

L555 - 561: I am also not aware of dD measurements in natural acetate, but the method from Greule et al. (2008) has been used in Vigano et al. (2010) to measure dD in methoxyl groups which were compared to produced CH4 and modeled dD in water.

L574 – 578: Why do you explain the variability for bogs by the pathway difference, and the high values in rivers by oxidation. Can oxidation not also cause large differences for bogs?

L599: Why should the oxidation signal only be apparent for dD and not for d13C (L603-604)?

L606: I do not understand how you can conclude that ". . .that the relative balance of diffusive vs. ebullition gas fluxes should not have a large effect on the isotopic composition of freshwater CH4 emissions.". The chance for oxidative effects is much larger for a slow process like diffusion compared to the fast process of ebullition.

L611: The analysis in this section has much less scientific rigor than the previous sections and presents some sensitivity calculations involving highly improbable assumptions, see following points.

L619 ff: See comments above on the depleted d13C source signature. Here you argue that three factors may explain this difference. I am quite convinced that the first one (errors in the sink fractionation factors) cannot explain the large difference. The two

published studies for the fractionation in the CH4 + OH reaction (Cantrell et al, 1990, Saueressig et al, 2001) are 5.4 and 3.9 per mill, respectively. A contribution from Cl may increase this a bit, but not enough to support a global average source signature of -56.4 per mill. So I think that the reason should come from the other two processes mentioned. Given the discrepancy to previous studies I wonder whether it is not mainly the choice of signatures in this study. In line 625 you already show that changing one parameter leads to a change of the global average source signature of 1.3 per mill, which is almost the entire uncertainty range reported.

L628: Rather arbitrarily changing big sources by a factor of 2 is a huge adjustment of the atmospheric CH4 budget. This investigation on the effect on the atmospheric isotopic composition is too simplistic.

L634 ff: Same comment for the bb source, this should be discussed in a more detailed way. Worden et al. (2017) illustrate the strong influence of the bb source.

L660f: The statement "This flatter slope may be the result of the inclusion of a greater proportion of inland water sites in our dataset." requires more underlying analysis. I think that the "may be" can be replaced by "is likely", but this should be investigated. See also other points above.

L662: If possible make more concrete after reevaluation of the impact of modeled data.

L686: Here the second argument of the three presented before (see comment on L619) has disappeared, but as argued above it may be the most important one and particularly the sink argument does likely not explain (at least exclusively) the difference.

L687: Cite Worden et al. (2017), who precisely did that.

References:

Cantrell, CA, et al., Carbon kinetic isotope effect in the oxidation of methane by the hydroxyl radical, J. Geophys. Res., 95 D, 22,455/462, 1990.

Greule M, et al., A rapid and precise method for determination of D/H ratios of plant methoxyl groups; Rapid Commun. Mass Spectrom. 22(24),3983–3988, 2008.

Röckmann, T, et al., The isotopic composition of H2 from biomass burning - dependency on combustion efficiency, moisture content and ïĄďD of local precipitation, J. Geophys. Res., 115, D17308, doi:10.1029/2009JD013188, 2010. Saueressig, G, et al., Carbon 13 and D kinetic isotope effects in the reactions of CH4 with O(1D) and OH: New laboratory measurements and their implications for the isotopic composition of stratospheric methane, J. Geophys. Res., 106(D19), 23127-23138, 2001.

Umezawa, T, et al., Carbon and hydrogen stable isotopic ratios of methane emitted from wetlands and wildfires in Alaska: Aircraft observations and bonfire experiments, J. Geophys. Res., 116, D15305, doi:10.1029/2010JD015545, 2011.

Vigano, I, et al., Water drives the deuterium content of the methane emitted from plants, Geochim. Cosmochim. Act., 74, 3865-3873, 2010.

Worden, JR, et al., Reduced biomass burning emissions reconcile conflicting estimates of the post-2006 atmospheric methane budget, Nature Commun., doi: 10.1038/s41467-017-02246-0, 2017.

---

## Referee Comment (RC3) · Edward Hornibrook (Referee) · 25 Dec 2020

During the past two decades, there has been limited progress in advancing understanding of controls on $\delta^2H(CH_4)$ values in freshwater environments and improving estimates of $\delta^2H$ values of $CH_4$ emissions. This study: (i) updates and attempts to refine the relationship between $\delta^2H(H_2O)$ and $\delta^2H(CH_4)$ first reported by Waldron et al. (1999b), (ii) evaluates the extent to which factors other than $\delta^2H(H_2O)$ may influence $\delta^2H(CH_4)$ values in freshwater environments, (iii) uses the refined relationships to estimate new $\delta^2H$ values for $CH_4$ emissions from freshwater sources, and (iv) weights $CH_4$ fluxes reported by Saunois et al. (2020) with a mixture of old and new $\delta^2H$ and $\delta^{13}C$ values to estimate global $\delta^2H$ and $\delta^{13}C$ values for atmospheric $CH_4$. In my opinion, the study offers new insights that are worthy of publication pending revision.

**General comments**

***Site level mean values*** - The study has produced a thorough compilation of stable isotope data related to $CH_4$ from freshwater environments. The availability of $\delta^2H(CH_4)$ values presumably was the key criterion for inclusion in the data base. The supplemental file contains a summary of the data, showing the number of samples from each site and site-level mean isotopic values as described in section 2.3.1.

While I appreciate the motivation to avoid introducing bias towards sites that have larger datasets, this approach does limit the extent to which the study can comment meaningfully on differences between environments. $\delta^2H(CH_4)$, $\delta^{13}C(CH_4)$ and $\delta^{13}C(CO_2)$ values all exhibit significant ranges and trends with depth in the subsurface of wetlands. That information is lost when profiles of $\delta$-values are averaged. In peatlands where $CH_4$ production pathways change with depth or $CH_4$ oxidation occurs, $\delta$-values determined from an average of shallow and deep layers has little meaning in the context of production pathways or evidence for $CH_4$ alteration. The pooled $\delta$-values also do not take into account differences in the amount of $CH_4$ or $CO_2$ at different depths. Moreover, $\delta$-values from deep peat typically will have little bearing on the stable isotope composition of $CH_4$ emitted from a wetland. Venting of accumulated gas bubbles from deep peat can occur (e.g., Glaser et al, 2004) but there is little evidence that such events are common. The bulk of $CH_4$ production occurs at shallow depths (from water table level to ~50 cm depth) where the supply of labile substrates from plant roots is greatest and temperature is highest during summer. The residence time of $CH_4$ at those depths is shortest (e.g., Lombardi et al., 1997; Bowes and Hornibrook, 2006) and most of the $CH_4$ produced seasonally is either consumed or evaded to the atmosphere.

If subsurface data must be averaged to avoid bias, then I suggest using a consistent depth range (e.g., 0 to 50 cm) to (i) generate mean $\delta$-values that are more likely to represent $\delta$-values of $CH_4$ emissions, and (ii) enable analysis of $\alpha_C$ and $\alpha_H$ values that are more likely to be related to one methanogenic pathway or exhibit the influence of methane oxidation rather than a blend of pathways and processes across a range of depths. An important advance in this study was the attempt to discern the relative impact of factors other than $\delta^2H(H_2O)$ on $\delta^2H(CH_4)$ values. Use of site level means for $\delta$-values raises concern about the validity of the $\alpha_C$ and $\alpha_H$ values calculated to assess breakpoints in $CH_4$ production pathways and oxidation.

***'Bottom-up' mixing model*** - I appreciate that considerable effort was invested in attempting to upscale $\delta^2H(CH_4)$ and $\delta^{13}C(CH_4)$ values; however, it is questionable whether that portion of the manuscript has potential to advance discourse on global isotope-weighted $CH_4$ budgets. A more valuable outcome of this work would have been the one identified by that authors in lines 441-

443: "A logical next step in predicting global freshwater $\delta^2$H-$CH_4$ source signatures would be to combine high-resolution mapping of wetlands and inland waters, maps of the global distribution of $\delta^2H_p$, and regression relationships between $\delta^2$H-$CH_4$ vs. $\delta^2H_p$." In my view, production of a global gridded map of $\delta^2$H($CH_4$) values for freshwater environments would have a more suitable application of the outcomes from the data analysis. It would provide a useful counterpart to the $\delta^{13}$C($CH_4$) global map for wetlands published by Ganesan et al. (2018). I realize at this stage in the process that would take the second half of the manuscript in a very different direction. As things stand, the weighted atmospheric $\delta^2$H($CH_4$) and $\delta^{13}$C($CH_4$) values that were calculated are difficult to reconcile with atmospheric data and KIEs associated with sinks for atmospheric $CH_4$. It's possible that the values may be offering new insights but it seems more likely that there are issues with attribution of $\delta^2$H and $\delta^{13}$C values to $CH_4$ sources.

These are my two main concerns with the manuscript in its present form. I am supportive of publication in a revised form. The work has potential to be a useful contribution and stimulate further efforts to characterise $\delta^2$H values of $CH_4$ produced and emitted from freshwater environments.

**Specific comments**
Citations within the text do not appear to be listed consistently either alphabetically or chronologically.

Line 38: 'clearly' = 'unequivocally' ?

Lines 51-52: 'recent technological developments'. An additional sentence or two about laser-based methods would be helpful for a broader readership.

Lines 53-57: Rigby *et al.* (2012) also demonstrated the utility of a multi-isotope approach for global methane cycle characterization.

Lines 87-88 (and elsewhere): 'data is' should be 'data are'

Line 105: A citation for Coplen (2011) could be added for the definition of delta that (correctly) does not include a 'x 1000' factor.

L129: The citation for John Lansdown's thesis should be:

Lansdown J. M. (1992) The carbon and hydrogen stable isotope composition of methane released from natural wetlands and ruminants. Ph.D. dissertation, Univ. of Washington.

(The citation can be confirmed at: https://dggs.alaska.gov/pubs/id/28259)

L156 – Is the annual estimate of $\delta^2H_p$ weighted by the relative amounts of precipitation during different seasons?

L200: $\delta^2$H (superscript missing)

L258-L259 "55 sites are classified as wetlands, including 16 bogs, 14 swamps and marshes, 12 fens, and 8 rice paddies."
>> Are the classifications for bogs and fens based upon pore water chemistry and vegetation surveys? The word 'bog' sometimes is used in site names that are other wetland types, in particular, fens.

Table 1: Origins of some data are unclear. When indicated as 'no specific measurement in database', what does it mean to say 'we used the isotopic values and uncertainties for X'? Which literature source? Also, only C3 $\delta^{13}C$ values appear to be used for biomass burning. Grassland and savanna wildfires presumably generate $CH_4$ that has more positive $\delta^{13}C$ values from burning of C4 grasses.

L266-L271 The comparison of modelled $\delta^2H_p$ values and measured $\delta^2H(H_2O)$ values for 62 sites is important for validating the approach on which estimating $\delta^2H(CH_4)$ relies. The text is not clear though with respect to causes in deviation from a 1:1 relationship. Presumably "$\delta^2H$-$H_2O$ is generally higher" means $^2H$-enrichment is evident in the measured data. Is the statement about 'overall smaller water volumes' meant to infer evaporative enrichment of $^2H$?

L282-L283 "Both relationships result in a large amount of unexplained residual variability, implying the importance of other variables in controlling $\delta^2H$-$CH_4$."

I'll expand here on the point raised in my general comments. The extent to which residual variability exists is likely underestimated because of the use of site-level means. There are relatively few data sets globally that contain subsurface profiles of both $\delta^2H(H_2O)$ and $\delta^2H(CH_4)$ values. Four of those data sets are shown in the enclosed figure which was published in Hornibrook and Aravena (2010): Turnagain Bog (open triangles; Chanton et al. 2006), Sifton Bog (open diamonds; Hornibrook et al. 1997), Point Pelee Marsh (open circles; Hornibrook et al. 1997) and Ellergower Moss (open squares; Waldron et al. 1999a). The arrows indicate the direction of increasing depth in peat for Turnagain Bog, Sifton Bog, Point Pelee Marsh and Ellergower Marsh. The figure also includes $\delta^2H$ values of coexisting $CH_4$ and $H_2O$ values from Alaskan peatlands along a N-S transect (filled triangles; Chanton et al. 2006) and regression equations (Table 6.2 from Hornibrook and Aravena, 2010 also enclosed) from a number of studies including Waldron et al. (1999b; line 5) and Whiticar et al. (1986; lines 1 and 2).
   The approach of using site-level means reduces each of those depth trends to a single point in $\delta^2H(H_2O)$ vs. $\delta^2H(CH_4)$ space. The $\delta^2H$ values of $CH_4$ emitted to the atmosphere are likely to be similar to the most $^2H$-depleted values in each trend which corresponds to $CH_4$ in shallow peat near the water-air interface and within the root zone where $CH_4$ may be transported to the atmosphere via plant aerenchyma. Averaging $\delta^2H(CH_4)$ values from all depths (2 m for Sifton Bog and Pelee Marsh; 6 m for Ellergower moss) yields a mean that is substantially more $^2H$-rich.
   Again, I appreciate the goal of not biasing the analysis to these larger data sets but a single mean for each site does not reflect the considerable residual variability that exists with depth as $\delta^2H(CH_4)$ values shift away from the global $\delta^2H(H_2O)$ vs. $\delta^2H(CH_4)$ regression line. Moreover, the $\delta^{13}C(CH_4)$ and $\delta^{13}C(CO_2)$ depth trends from these sites yield systematic shifts in $\alpha_C$ values that are lost when the $\delta^{13}C$ values similarly are reduced to unitary site-level means.

[Figure]

**TABLE 6.2**

**Equations Relating $\delta^2H$ Values of Coexisting $CH_4$ and $H_2O$ Shown in Figure 6.1**

| No. | Equation | Origin | Source |
|---|---|---|---|
| 1 | $\delta D\text{-}CH_4 = 1.000\ \delta D\text{-}H_2O - 180(\pm10)\text{‰}$ | $CO_2$ reduction in situ | Whiticar, Faber, & Schoell (1986) |
| 2 | $\delta D\text{-}CH_4 = 0.250\ \delta D\text{-}H_2O - 321\text{‰}$ | acetate fermentation in situ | Whiticar et al. (1986) |
| 3 | $\delta D\text{-}CH_4 = 0.19\ \delta D\text{-}H_2O - 259\text{‰}$ | *Methanosaeta thermophila* | Valentine et al. (2004) |
| 4 | $\delta D\text{-}CH_4 = 1.55(\pm0.46)\ \delta D\text{-}H_2O - 145(\pm30)\text{‰}$ | Alaskan peatland transect | Chanton, Fields, & Hines (2006) |
| 5 | $\delta D\text{-}CH_4 = 0.675(\pm0.10)\ \delta D\text{-}H_2O - 284(\pm6)\text{‰}$ | global freshwater in situ | Waldron et al. (1999) |
| 6 | $\delta D\text{-}CH_4 = 0.437(\pm0.05)\ \delta D\text{-}H_2O - 302(\pm15)\text{‰}$ | acetate fermentation in vivo | Sugimoto & Wada (1995) |
| 7 | $\delta D\text{-}CH_4 = 0.444(\pm0.03)\ \delta D\text{-}H_2O - 321(\pm4)\text{‰}$ | mesophilic incubations in vivo | Waldron et al. (1998) |
| 8 | $\delta D\text{-}CH_4 = 0.683(\pm0.02)\ \delta D\text{-}H_2O - 317(\pm20)\text{‰}$ | $CO_2$ reduction in vivo | Sugimoto & Wada (1995) |

L308-L309 "We do not find evidence for a piece-wise linear relationship between $\delta^{13}C\text{-}CH_4$ and $\delta^2H\text{-}CH_{4,W0}$ (Fig. 5a), nor did we find a significant simple linear correlation between these variables."

>> It may be worth exploring whether any relationships exist in the full data sets rather than site-level means.

L377 – L378. "Similarly, we did not observe any significant differences in $\delta^{13}$C-CH$_4$ values between wetland ecosystems in this dataset based on a Kruskal-Wallis test, nor between inland waters and wetlands based on a U-test."
>> I recommend examining whether this is the case if CH$_4$ data are used from a common depth interval rather than site-level means.

L441-L443: "A logical next step in predicting global freshwater $\delta^2$H-CH$_4$ source signatures would be to combine high-resolution mapping of wetlands and inland waters, maps of the global distribution of $\delta^2$H$_p$, and regression relationships between $\delta^2$H-CH$_4$ vs. $\delta^2$H$_p$."
>> I agree with the authors and suggest this would be a worthwhile output to include in this manuscript instead of the global upscaling estimate.

L445-L464 Section 4.2. This section would benefit from acknowledging and discussing the study by Rigby et al. (2012).

L500-L504 In addition to the caveat noted that CH$_4$ data exhibiting $^2$H-enrichment due to methane oxidation are uncommon, the amount of CH$_4$ emitted to the atmosphere bearing the effects of methanotrophy is likely to be small. Bacteria oxidation is highly efficient in the subsurface of wetlands and little CH$_4$ tends to escape to the atmosphere via diffusion through porewater. This comment applies to peatlands. The situation is different in inland water environments.

L510–L518 I was pleased to see incorporation of these alternate explanations for relationships between $\delta^2$H and $\delta^{13}$C values of CH$_4$. Methanogenic pathways are not the only potential explanation.

L592-L593 – Bellisario et al. (1999) provides a good example of how $\delta^{13}$C(CH$_4$) values vary along a trophic gradient in a wetland complex. Differences in $\delta^{13}$C values of CH$_4$ emissions and porewater CH$_4$ values in minerotrophic vs. ombrotrophic wetland are demonstrated in Hornibrook and Bowes (2007) and Hornibrook (2009). Landscape scale measurements (atmospheric inversions and aircraft measurements; Fisher et al., 2017) also show that northern wetlands contain sources of $^{13}$C-poor CH$_4$ that differ from values of ~-62 to -58 permil typically attributed to northern peatlands in isotope-weight CH$_4$ budgets. Characterization of sites as ombrotrophic or minerotrophic on the basis of water chemistry and vegetation surveys is essential for making these distinctions.

L617 to L622 It is unclear how a more negative than expected value for estimated $\delta^{13}$C(CH$_4$) can be explained by (2) source signatures being biased toward more positive $\delta^{13}$C values.

*Ed Hornibrook*
*24 December 2020*

**References**

Bellisario, L. M., J. L. Bubier, T. R. Moore, and J. P. Chanton (1999), Controls on $CH_4$ emissions from a northern peatland, Global Biogeochem. Cycles, 13, 81–91.

Bowes, H.L., and Hornibrook, E.R.C. 2006. Emission of highly $^{13}$C-depleted methane from an upland blanket mire. *Geophys. Res. Lett.* 33:L04401, doi:10.1029/2005GL025209.

Coplen, T.B. (2011). Guidelines and recommended terms for expression of stable-isotope-ratio and gas-ratio measurement results. *Rapid Communications in Mass Spectrometry* 25, 2538-2560.

Fisher, R. E., France, J. L., Lowry, D., Lanoisellé, M., Brownlow, R., Pyle, J. A., et al. (2017). Measurement of the $^{13}$C isotopic signature of methane emissions from northern European wetlands. *Global Biogeochemical Cycles*, 31, 605–623.

Glaser, P.H., Chanton, J.P., Morin, P., Rosenberry, D.O., Siegel, D.I., Ruud, O., Chasar, L.I. and Reeve, A.S. (2004) Surface deformations as indicators of deep ebullition fluxes in a large northern peatland. Global Biogeochemical Cycles 18, art. no.-GB1003.

Hornibrook E.R.C. and Aravena R. (2010). Isotopes and methane cycling. In *Environmental Isotopes in Bioremediation and Biodegradation* (eds. C. M. Aelion, R. Aravena, P. Höhener, and D. Hunkeler). Taylor and Francis, CRC Press, 167-201.

Hornibrook E.R.C. and Bowes H.L. (2007). Trophic status impacts both the magnitude and stable carbon isotope composition of methane flux from peatlands. *Geophysical Research Letters* 34, L21401, 1-5, doi:10.1029/2007 GL031231.

Hornibrook E.R.C., F.J. Longstaffe and W.S. Fyfe (2000) Evolution of stable carbon-isotope compositions for methane and carbon dioxide in freshwater wetlands and other anaerobic environments. *Geochimica et Cosmochimica Acta*, 64(6), 1013-1027.

Lombardi, J. E., et al. (1997), Investigation of the methyl fluoride technique for determining rhizospheric methane oxidation, Biogeochemistry, 36, 153–172.

Rigby, M., A. J. Manning, and R. G. Prinn (2012). The value of high-frequency, high-precision methane isotopologue measurements for source and sink estimation, J. *Geophys. Res.*, 117, D12312, doi:10.1029/2011JD017384.

Waldron, S., Hall, A.J., and Fallick, A.E. (1999a). Enigmatic stable isotope dynamics of deep peat methane. *Global Biogeochem. Cy.* 13:93-100.

Waldron, S., Lansdown, J. M., Scott, E. M., Fallick, A. E., and Hall, A. J. (1999b). The global influence of the hydrogen isotope composition of water on that of bacteriogenic methane from shallow freshwater environments. *Geochim. Cosmochim. Ac.* 63:2237-2245.

---

## Author Comment (AC1) · 15 Feb 2021

**Response to reviewer 1**

All of the reviewers provided excellent suggestions and feedback on the paper, and we think that by addressing their concerns the paper will be greatly improved. Many of their comments were complementary. Therefore we will first summarize the major revisions we plan to make to the paper before responding to each reviewer in detail:

**Planned Major Revisions**
**1)** We have revised the $\delta^2$H-$CH_4$ dataset in response to comments from Reviewer 1 and Reviewer 3. (i) For peatland sites with depth stratified sampling we have decided to only include samples from the upper 50 cm, as suggested by reviewer 3, since this is the depth range that is most likely to emit $CH_4$ to the atmosphere. This affects a total of 8 sites. (ii) Reviewer 1 noted that an outlier sample from the Amazon River with very high $\delta^2$H-$CH_4$ and $\delta^{13}$C-$CH_4$ could be derived from thermogenic methane. We agree that this outlier is suspect, and therefore have decided not to include it. (iii) We also noted that one site (Mirror Lake, Florida, USA) was analyzed in two separate studies, and therefore was included twice in the dataset. We have combined the data from the two studies into one site entry.

**2)** As suggested by all three reviewers, we have performed much more rigorous analysis of the relationship between measured and modeled $\delta^2$H-$H_2O$ values. Specifically we have done the following: (i) In addition to annual precipitation $\delta^2$H values, we now also analyze growing season precipitation $\delta^2$H, which is defined as the amount-weighted mean $\delta^2$H of months with mean temperature greater than $0°$ C. This provides an opportunity to assess whether seasonal variation in precipitation in the mid to high-latitudes is important in controlling the environmental $\delta^2$H-$H_2O$ value; (ii) separately analyzing inland water and wetland environments, since these are different hydrological environments and the controls on $\delta^2$H-$H_2O$ are potentially different.

This analysis led to the following key results (see Table R1 below for summary of results): A) growing season modeled precipitation $\delta^2$H is a better predictor of inland water $\delta^2$H-$H_2O$ than annual precipitation $\delta^2$H, in that the regression curve is indistinguishable from the 1:1 line. B) Annual modeled precipitation $\delta^2$H is a better predictor of wetland $\delta^2$H-$H_2O$, in that the slope of the regression is indistinguishable from 1, and the $R^2$ value is higher. However, the regression line is offset from the 1:1 line by 18.6±9‰. We interpret this as an indicator of likely widespread evaporative effects on $\delta^2$H-$H_2O$ in wetland environments.

We use these results to then develop a 'best estimate' for comparing $\delta^2$H-$H_2O$ with d2H-CH4. (i) For sites with measured $\delta^2$H-$H_2O$ values we use the measured value. (ii) For inland water sites without measured $\delta^2$H-$H_2O$ we use modeled growing season precipitation, since as discussed above the regression of this against measured $\delta^2$H-$H_2O$ is indistinguishable from the 1:1 line. (iii) For wetland sites without measured $\delta^2$H-$H_2O$ we estimate the $\delta^2$H-$H_2O$ using the regression relationship with annual precipitation $\delta^2$H-$H_2O$ shown in Table R2. We feel this approach combining measured and modeled data is

most consistent with that of Waldron et al., 1999, who we note also analyzed a combination of sites with measured $\delta^2$H-$H_2O$ (29 out of 51 sites) and estimated $\delta^2$H-$H_2O$ based on precipitation isotopic measurements (22 out of 51 sites).

Table R1: Comparison of regression relationships between modeled $\delta^2H_p$ and measured $\delta^2$H-$H_2O$

|  | Slope | Intercept | $R^2$ | RMSE | $p$ | $n$ |
|---|---|---|---|---|---|---|
| **Inland waters** | | | | | | |
| *Growing season $\delta^2H_p$* | 1.05±0.09 | -0.3±8 | 0.82 | 22.3 | 4.81E-13 | 33 |
| *Annual $\delta^2H_p$* | 0.85±0.06 | -2.1±7 | 0.84 | 20.5 | 3.17E-14 | 33 |
| **Wetlands** | | | | | | |
| *Growing season $\delta^2H_p$* | 1.24±0.09 | 14.8±10 | 0.87 | 16.5 | 4.46E-13 | 28 |
| *Annual $\delta^2H_p$* | 1.057±0.08 | 18.6±9 | 0.88 | 15.7 | 1.20E-13 | 28 |

**3).** As suggested by all three reviewers, it is important to consider the effects of modeled $\delta^2$H-$H_2O$ on the regression between $\delta^2$H-$H_2O$ and $\delta^2$H-$CH_4$. To do this carefully we performed the regression analysis using four different estimates of $\delta^2$H-$H_2O$:
 (i) the 'best-estimate' of $\delta^2$H-$H_2O$ as described above in Planned Major Revision 2; (ii) measured $\delta^2$H-$H_2O$, only analyzing sites with this measurement; (iii) modeled annual precipitation $\delta^2$H; and (iv) modeled growing season precipitation $\delta^2$H. We think it is valuable to continue to include the regression relationships for modeled precipitation because these relationships could be used in future studies using Earth Systems Models to predict the distribution of $\delta^2$H-$CH_4$. For each of these cases we analyzed all sites, inland waters, and wetlands. We also compare these relationships with those of Waldron et al., (1999), both for the total dataset in that study, and for the dataset that only includes sites with measurements of $\delta^2$H-$H_2O$ (29 out of 51 sites). A summary of the results of this analysis is shown in Table R2 below.

A key point is that we have decided to use unweighted, as opposed to weighted, regression. Comments by Reviewer 1 made us realize that weighting by standard error was causing a few sites to strongly bias the regression results. Statistical research has found that for environmental data with poorly constrained error variance unweighted regression is frequently less biased than weighted regression (Fletcher and Dixon, 2012). Using a statistical test proposed by that study we find that unweighted regression is a good choice for our dataset. Note that in Table R2 we apply unweighted regression to the dataset of Waldron et al., (1999), in part because the specific weighting methodology was not specified in that study. This produces a small difference in the regression relationship shown in Table R2 with that reported by Waldron et al., (1999), but the two regression relationships are within error.

We then used analysis of covariance (ANCOVA) to examine differences between the regression relationships shown in the table. Based on a multiple comparison test, none of

the regression relationships shown in Table R2 are significantly different from one another. Therefore we conclude that (i) using modeled $\delta^2$H-$H_2O$ does not have a significant effect on the estimate of the relationship between $\delta^2$H-$H_2O$ vs. $\delta^2$H-$CH_4$; (ii) Differences in the slope of this relationship between inland waters and wetland sites are not conclusive; and (iii) that since all of the regression relationships using the larger dataset produce a flatter slope than that of Waldron et al., (1999), the true global slope is likely to be flatter than inferred in that study, but confirmation of this flatter global slope will require more data and further analysis.

Table R2: Comparison of regression relationships between $\delta^2$H-$H_2O$ and $\delta^2$H-$CH_4$ using different estimates of $\delta^2$H-$H_2O$

| | Slope | Intercept | $R^2$ | RMSE | $p$ | $n$ |
|---|---|---|---|---|---|---|
| **Best Estimate $\delta^2$H-$H_2O$** | | | | | | |
| *All* | 0.44±0.05 | -298±5 | 0.42 | 27.4 | 1.33E-16 | 129 |
| *Wetlands* | 0.51±0.06 | -300±5 | 0.59 | 23.7 | 7.85E-12 | 55 |
| *Inland Waters* | 0.42±0.07 | -295±7 | 0.34 | 29.1 | 6.72E-08 | 74 |
| **Measured $\delta^2$H-$H_2O$** | | | | | | |
| *All* | 0.5±0.08 | -292±8 | 0.43 | 28.6 | 1.22E-08 | 61 |
| *Wetlands* | 0.53±0.11 | -298±13 | 0.44 | 26.2 | 6.58E-05 | 28 |
| *Inland Waters* | 0.42±0.1 | -291±10 | 0.37 | 28.9 | 0.000156 | 33 |
| **Modeled Annual $\delta^2H_p$** | | | | | | |
| *All* | 0.42±0.04 | -293±5 | 0.44 | 26.9 | 1.17E-17 | 129 |
| *Wetlands* | 0.57±0.06 | -287±6 | 0.65 | 21.9 | 1.01E-13 | 55 |
| *Inland Waters* | 0.37±0.06 | -293±7 | 0.36 | 28.4 | 1.25E-08 | 74 |
| **Modeled Growing Season $\delta^2H_p$** | | | | | | |
| *All* | 0.51±0.05 | -292±5 | 0.41 | 27.6 | 2.55E-16 | 129 |
| *Wetlands* | 0.71±0.07 | -285±6 | 0.63 | 22.4 | 4.05E-13 | 55 |
| *Inland Waters* | 0.44±0.07 | -294±8 | 0.33 | 29.3 | 1.03E-07 | 74 |
| **Waldron et al. (1999)** | | | | | | |
| *All data* | 0.74±0.1 | -284±6 | 0.5 | 26.3 | 6.56E-09 | 51 |
| *Measured $\delta^2$H-$H_2O$* | | | | | | |
| *only* | 0.79±0.2 | -279±10 | 0.44 | 29.6 | 9.21E-05 | 29 |

**4)** We then used the 'best-estimate' $\delta^2$H-$H_2O$ values and the regression based on those values, shown in Table R2, to calculate a revised $\delta^2$H-$CH_{4,w0}$ value for each site. These

analyses were then applied in the subsequent analyses in the paper shown in Figures 5,8 and 9. We also calculated an alternate value for sites with measured $\delta^2$H-H$_2$O, using the values and regression curve for those sites.

Notably, for the comparison between $\delta^2$H-CH$_{4,w0}$ and $\alpha_C$ we have found that there continues to be evidence for a segmented linear relationship. However, the breakpoint of this relationship is not consistent when analyzing all sites or only sites with measured $\delta^2$H-H$_2$O. Furthermore, the regression relationships for the two components of the segmented linear relationship were weaker than in our original analysis, and were not consistently statistically significant. Therefore in our revised analysis we will place less emphasis on this result, and less emphasis on the relationship between methanogenic pathway and $\delta^2$H-CH$_4$ generally, as suggested by reviewer 1. Instead we will discuss four processes or variables that have the potential to influence $\delta^2$H-CH$_4$ in freshwater environments: (i) differences in methanogenic pathway, including possible use of methanol as a substrate; (ii) methane oxidation; (iii) isotopic fractionation due to diffusion; and (iv) differential thermodynamic favorability of methanogenesis, or differential enzymatic reversibility. Ultimately, our conclusion is that $\delta^{13}$C-CH$_4$ or $\alpha_C$ cannot fully resolve the effects of these processes on $\delta^2$H-CH$_4$ on a global basis, and other approaches will be necessary to determine their relative importance, or the possible importance of other processes.

We will continue to present these results in Figure 5, given that we feel it is important to show co-variation, or lack thereof, between these isotopic measurements. Given the findings mentioned above, we will substantially revise Figure 6. Instead of distinguishing samples by inferred methanogenic pathway in this figure, we will distinguish samples by environment (wetland vs. inland water), and also show available data for cow rumen and landfills. We may reverse the order of Figures 5 and 6.

**5)** Reviewer 2 made numerous comments about the representativeness of our $\delta^{13}$C-CH$_4$ dataset. We want to make clear that to our knowledge this is the largest database of freshwater methane $\delta^{13}$C-CH$_4$ currently compiled. For comparison, the second largest dataset, that of Sherwood et al., (2017), includes 48 freshwater sites (including rice paddies), of which 16 are also included in our database. However our $\delta^{13}$C-CH$_4$ database is not comprehensive (unlike the $\delta^2$H-CH$_4$ database), in that it does not include many measurements that are not paired with $\delta^2$H-CH$_4$ measurements and that have not yet been compiled into a database. It is also probably not representative, because some important environments, namely C$_4$ plant dominated ecosystems, are not well represented.

Since the primary focus of this paper is $\delta^2$H-CH$_4$, it is not within its scope to provide a comprehensive database of freshwater $\delta^{13}$C-CH$_4$, although that would be a worthwhile goal for future research. In order to make our analysis as complete as possible, in our revised manuscript we will include the 32 freshwater sites from Sherwood et al., (2017) that were not included in our original analysis in our calculations for the upscaling exercise, as well as Figures 7, 8, and 9. We will also carefully discuss the likely biases in

this dataset, especially in terms of $C_4$ plant environments, and their implications for our interpretations.

**6)** Both reviewers 2 and 3 expressed some concerns with the upscaling analysis. We acknowledge that the upscaling analysis is relatively simplistic, and that some of the interpretations were speculative. However, we still think it is valuable to use the estimates of freshwater $CH_4$ isotopic composition, differentiated by latitude, produced in this study to estimate global source $\delta^2 H\text{-}CH_4$ and $\delta^{13}C\text{-}CH_4$, and to compare that with other estimates. We wish to make clear that given uncertainties and complexity in estimating sink fractionations, particularly for $\delta^2 H\text{-}CH_4$, we are not attempting to estimate atmospheric values, but instead the integrated source $\delta^2 H\text{-}CH_4$ and $\delta^{13}C\text{-}CH_4$ prior to sink fractionations. We think there is value in comparing this with (i) previous bottom-up estimates of these values; and (ii) with the top-down estimates reported by Rice et al., (2016). We concur with Reviewer 2 that the discussion of alternate emissions scenarios is too speculative and simplistic, and therefore we will remove this discussion. Instead, we will focus on likely sources of error in the isotopic source signatures, and the best ways to address these errors in future studies.

We disagree with Reviewer 2 that the error estimates for isotopic source signatures are generally too optimistic, which we will discuss in more detail in our response to that reviewer.

Given comments from all three reviewers we will revise Figure 10 to only include panel C, and make the comparison with other estimates of global source $\delta^2 H\text{-}CH_4$ and $\delta^{13}C\text{-}CH_4$ clearer in this figure.

*Specific Responses to Reviewer 1:* Reviewer comments are in plain text. **Responses are in bold text.**

**We thank Dr. Waldron for her careful and thorough review. We appreciate that this review presented a challenging situation, and we value her honesty and openness. We are confident that we can address her concerns in the revised manuscript.**

Substantial conclusions are reached, but the interrogative approach has weaknesses that propagate through substantial analytical reasoning and so the integrity of the conclusions is questionable. I detail this further below, but until the analytical approaches are reconsidered the conclusions are not securely reached

**We understand this critique, and in response we will strengthen the statistical analyses and interrogation of the data, in particular with regards to the 1) comparison of measured and modeled $\delta^2 H\text{-}H_2O$ values, 2) the inference of a regression slope between $\delta^2 H\text{-}H_2O$ and $\delta^2 H\text{-}CH_4$, and 3) the application of carbon isotope fractionation factors to evaluate the potential effects of methane oxidation, methanogenesis pathway, or other biogeochemical effects on $\delta^2 H\text{-}CH_4$. See the planned major revisions 2, 3, and 4 above.**

With respect to understanding isotopic compositions: the methods are not all valid, particularly

the reconstruction of missing δD-H2O for a field measurement of δD-CH4. The interrogation of this relationship (Fig. 2) lacks statistical rigour, and its propagation - a relationship that has bias and significant variability - is unconsidered in all analysis thereafter (as represented by Figs. 3-9 and possibly 10) and so this reasoning is flawed and the interpretations may be wrong.

**We will thoroughly re-analyze the relationship between measured and modeled $\delta^2$H-H$_2$O in the revised manuscript. See Planned Major revision 2 above. We will then carry this revised analysis forward to the remainder of the manuscript. See Planned Major Revisions 3 and 4.**

The authors are not consistent in identifying when processes they are interpreting are based on hypothesised relationships and the impression is given such processes are certain (detailed below).

**We will make it clearer in the revised manuscript where we are discussing hypothesized relationships, as discussed in more detail below. In particular regarding hypotheses regarding the effects of methanogenic pathway on $\delta^2$H-CH$_4$ we will be more circumspect in the revised manuscript, and discuss alternate hypotheses in greater detail. See Planned Major Revision 4.**

I found it difficult to follow the calculations behind αC – an important part of the manuscript – when I was trying to compare other data sets with their approach.

**We are uncertain what aspect of this calculation was unclear, but we will endeavor to make the description of this calculation clearer.**

Largely bit not always, for example there is a large section in 4.31. that is repeating suggestions made in section 1.1. of Waldron et al 1999, but this work is unreferenced and so as written implies the review m/s is the first to have suggested this; the abstract does not make clear refining an existing phenomena observed and described similarly previously.

**We regret the omission of references and acknowledgment to Waldron et al., 1999, in the ideas presented in section 4.3.1. We will thoroughly revise this section to provide proper credit for these ideas, and integrate the discussion with that previously published by Waldron et al., (1999).**

Broadly but not sure how "geographic variability in freshwater methane hydrogen isotope ratios has implications for microbial biogeochemistry" - the microbes are active with no knowledge δD…so this can be refined.

**We will follow this suggestion and modify the title to: *Geographic variability in freshwater methane hydrogen isotope ratios and its implications global isotopic source signatures*.**

It is a paper with a lot of detail and so to follow it all the reader has to concentrate deeply for the results section. As such, and maybe in addition, the discussion from section 4 onwards seems in places repetitive.

**We agree the results are very detailed. Given the critiques of the reviewers we will need to add additional statistical analyses and data interrogation to the results,**

**leading to an overall increase in detail. However, we will present this in as streamlined and clear manner as possible. We will also carefully review and revise the discussion sections to avoid repetition.**

Broadly yes – I suggest a group whose work may be missing in the intro.

**We will add the suggested citations in the introduction**

Yes, very helpful, but sheet 2 could make it clearer if the data offered is used in $\alpha_C$ or these are summarised data from other sources.

**Sheet 2 will be omitted from the revised manuscript, as multiple reviewers have questioned the value of the predicted fields for pathway and oxidation dependent isotopic variation. See Planned Major Revision 4.**

The substantive conclusions in this manuscript rely on a data set where $\delta D$-$H_2O$ does not exist for more than half the data: 53% of the sites do not have field measured $\delta D$-$H_2O$ (L88). In these cases, $\delta D$-$H_2O$ is inferred from a reputable global precipitation database and a correlation observed for sites where measured values exist. The authors consider this relationship sufficiently robust to proceed to use the reconstructed $\delta D$-$H_2O$ where measured values do not exist. I disagree this is the case.

**We acknowledge this is an important critique. In the revised manuscript we will take steps to strengthen the analysis of the relationship between modeled and measured $\delta^2H$-$H_2O$, and to carefully evaluate if using modeled values of d2H-H2O leads to a bias in the inferred relationship between $\delta^2H$-$H_2O$ and $\delta^2H$-$CH_4$. See Planned Major Revisions 2 and 3. Our analysis indicates that using modeled $\delta^2H$-$H_2O$ does not lead to a significant difference in the regression relationship with $\delta^2H$-$CH_4$. See Table R2 above.**

The statistical integrity shown elsewhere in the manuscript is lacking in this section on reconstructing $\delta D$-$H_2O$, with the authors describing their predictive relationship as showing "generally good agreement" and proceeding to use it. The bias and variability in a predictive $\delta D$-$H_2O_p$ and thus how far it may be from the true $\delta D$-$H_2O$ appear unconsidered in any further analysis (no errors propagated through for estimated $\delta D$-$H_2O$?).

**As described above, in the revised manuscript we will more carefully evaluate this relationship. See Planned Major Revision 2.**

Further, I note that the data in table S3 supplementary information for which there are measured $\delta D$-$CH_4$ - $\delta DH_2O$ fit closely to the in-vitro line from which Waldron et al 1999 project a global relationship - but the data with estimated $\delta DH_2O$ in table S3 do not. This is important for two reasons:

**The reviewer has pointed to an interesting observation. The difference observed by the reviewer is at odds with Figure 3A in the original manuscript, which clearly showed that the regression lines for $\delta^2H$-$H_2O$ vs $\delta^2H$-$CH_4$ fully overlap whether modeled (black regression**

line) or measured (blue regression line) $\delta^2$H-H$_2$O is applied, and was not in agreement with the in vivo line of Waldron et al., (1999).

On further analysis of the data, we identified that the discrepancy observed by the reviewer was probably caused by two factors:

1) the weighted regression method that we used in the original manuscript was strongly influenced by a few sites at high latitudes that both (a) have a large number of measurements (and therefore a low standard error and a higher weight); and b) relatively high $\delta^2$H-CH$_4$ values. We infer that this strong weighting at these sites leads to a strong bias in the regression. Based on statistical research involving environmental samples (Fletcher and Dixon, 2012) we infer that unweighted regression is preferable for this dataset (See Planned Major Revision 3).

2) The unweighted regression performed by the reviewer was likely strongly influenced by an outlier site from the Amazon with very high $\delta^2$H-CH$_4$. When this point is removed, as suggested by reviewer 1, the regression slope becomes flatter.

The two factors above effectively cancel each other out. In re-analyzing the data after accounting for these two changes (see Table R2 above) we find that (a) there is not a large or significant difference in the regression slope if measured $\delta^2$H-H$_2$O, 'best-estimate' $\delta^2$H-H$_2$O, or modeled precipitation $\delta^2$H-H$_2$O is used and (b) all of these regression slopes are flatter than that of Waldron et al., (1999).

1. It confirms the predictive relationship in Waldron et al 1999 for $\delta$D-CH$_4$ from $\delta$D-H$_2$O still has integrity, more so by adding in another methane-producing environment (innocula), a significant time gap, and another geographic locality.

As mentioned above, our revised regrewssion analysis continues to result in flatter slopes than the predictive relationship proposed by Waldron et al., (1999), regardless of the method of estimating $\delta^2$H-H$_2$O. However, as we also discuss above, given the wide confidence intervals of these relationships we do not find a significant difference with the prediction of Waldron et al., 1999 using multiple group comparison ANCOVA. However, given that every analysis of the larger dataset presented here results in a flatter slope, we think it is highly likely that the global relationship has a somewhat flatter slope than that inferred by Waldron et al., (1999).

2. If statement 1 is considered sound, then the poor fit of paired $\delta$D-CH$_4$- $\delta$D-H$_2$O with predicted $\delta$D-H$_2$O supports the assertion above that the relationship the authors are using here to reconstruct $\delta$D-H$_2$O is questionable.

As noted above, we do not see evidence that there is a significant difference in $\delta^2$H -CH$_4$ vs. $\delta^2$H -H$_2$O based on the method of estimating/inferring $\delta^2$H-H$_2$O. Therefore we disagree with the assertion that the methods used to reconstruct $\delta^2$H-H$_2$O are questionable. However, we have revised our approach to predicting $\delta^2$H-H$_2$O, as explained in more detail in Planned Major Revisions 2 and 3.

**We think it is important to note here that the in-vivo relationship of Waldron et al., (1999) is not based purely on sites with measured $\delta^2$H-H$_2$O. Instead, that study used a combination of sites with $\delta^2$H-H$_2$O measurements (57%) measurements and estimates based on precipitation isotope measurements (43%).**

**Quoting from that study:** "*Where paired δD(CH4)–δD(H2O) measurements were not published δD(H2O) was sourced from measured precipitation values for the area, for example, the weighted mean of the precipitation samples collected in south Florida over a 3-yr period (Swart et al., 1989) was used as an appropriate value for δD(H2O) for St. Marks Swamp, Florida (Happell et al., 1994).*

  *Other unknown δD(H2O) signatures (e.g., the Alaskan Lakes; Martens et al., 1992) were estimated from the weighted mean value of sites close to the area sampled, that participated in the global network, Isotopes in Precipitation (IAEA, 1992) or from the meteoric water line (Craig, 1961). We are aware that δD(groundwater) can differ by up to 30‰ frommeasured δD(precipitation) (e.g., Hornibrook et al., 1997; E. R. C. Hornibrook, pers. comm.), but such fractionation is difficult to quantify and the logical approach we have adopted provides the best estimate for dD(H2O) where measured values are unavailable.*"

**We agree with Waldron et al., (1999) that differences between precipitation and groundwater (or lake water) can be large, and these differences can be difficult to quantify. However, we do not think these potential differences negate the value of estimating $\delta^2$H-H$_2$O using available estimates of precipitation $\delta^2$H, along with accounting for the effects of precipitation seasonality and evaporation, as described in Planned Major Revision 2. Indeed, we think our revised approach of combining measured and estimated water $\delta^2$H into a 'best-estimate' is a logical extension of the approach used by Waldron et al., (1999). However, we agree that a more careful evaluation of this approach is warranted, and we are planning to add this to the revised manuscript as described above (Planned Major Revision 2).**

With the greatest of respect, using the predicted data produces an outcome that is like a 'house of cards' – all subsequent analysis using this data is built on a shaky foundation. I therefore think that incorporating paired δD-CH4 -δD-H2Op in further analysis is flawed and offer two examples why:
1. It creates a new global line for δD-CH4 -δD-H2O that may be wrong.
2. It could lead to artefact in interpretation, which indeed may be 'visible' in the dependent analysis. For example, the data in Fig. 3b visually also appears to separate between paired δD-CH4 -δD-H2O data that are predicted (inland waters) and measured (wetlands), and if this is the case interpreting a biome difference here, and later in the paper, is also questionable.

**We do not agree with the house of cards analogy, but we do agree that it is imporotant to provide more confidence in our underlying analyses. As mentioned above (Planned Major Revision 3; Table R2), we do not observe a significant difference in $\delta^2$H -CH$_4$ vs. $\delta^2$H-H$_2$O whether modeled or measured $\delta^2$H-H$_2$Ois used. Therefore we disagree that there is a 'shaky foundation' to our subsequent analysis.**

**However, as discussed above we have revised our data analysis to use the 'best-estimate' $\delta^2H$-$H_2O$ value, including measured values where available.**

**We note that in Figure 3B the reviewer likely misinterpreted the data presented. All of the data shown in this figure are based on modeled $\delta^2H$-$H_2O$. Therefore the observed difference between inland waters and wetlands cannot be because of differences in the source of $\delta^2H$-$H_2O$ data. See Table R2 above, which shows that there are consistent differences between wetlands and inland waters in the slope of the regression line regardless of the method used to estimate $\delta^2H$-$H_2O$, but also that these differences are small and statistically insignificant. Therefore in our revised manuscript we will state that we cannot confidently infer a difference in the relationship between these environments.**

With respect to the redefining of a new global $\delta D$-$CH_4$-$\delta D$-$H_2O$ and consideration of how this has changed from the relationship offered in Waldron et al 1999: unless the authors can produce a more robust estimation of $\delta D$-$H_2O_P$, the data that uses $\delta D$-$CH_4$ paired with predicted $\delta D$-$H_2O$ needs to be removed - for as noted earlier, there is insufficient confidence this is an accurate representation of the field situation and may create a false outcome. I suspect this will change the global relationship and increase the slope as paired data with $\delta D$-$H_2O_P$ visually appears to dominate the enriched samples.

**We believe the approach taken by our Planned Major Revision 3 effectively addresses this critique. We now take an approach similar to that of Waldron et al., (1999), namely combining measured and modeled $\delta^2H$-$H_2O$ values to produce a best-estimate value for each site.**

**As noted above, we do not observe a significant difference in the $\delta^2H$ -$CH_4$ vs. $\delta^2H$-$H_2O$ relationship whether modeled or measured $\delta^2H$-$H_2O$ is used (See Table R2)**

Please plot both the in-vitro and in-vivo relationship, and for the former its prediction intervals - which are missing from 3b and so give the sense of a poorer fit of Waldron et al 1999 to the bgd expanded field data set here.

**We are planning to include a more robust comparison with the data from Waldron et al (1999) in Figure 3 in the revised manuscript, including confidence intervals. We argue that confidence intervals are the more appropriate metric, since this gives the uncertainty of the regression relationship, as opposed to the predicted range of observations. We are more interested in comparing the underlying regression relationships, as opposed to the predicted range of observations.**

**Also, the new figures will be highly complex with multiple regression relationships and therefore we think that including the in-vitro relationship, which does not significantly overlap with the data, would only further complicate the figures. However, we will include a comparison with the in-vitro relationship in the revised version of Figure 6, as was done in the original manuscript.**

Compare whether the in-vivo line is statistically different to the relationship generated from the data set presented in the bgd manuscript. This will allow confidence in any further discussion on how the relationship has been redefined (than just comparing slopes etc). If the two relationships

are indistinguishable statistically, nuanced statements about differences in slope etc are meaningless – all that has happened is that the expanded data set has redefined better the field relationship for $\delta D\text{-}CH_4$ $\text{-}\delta D\text{-}H_2O$ (as indicated likely in Waldron et al, 1999) - noting that this field relationship does not wholly reflect the relationship at production (see next point).

**We will use analysis of covariance (ANCOVA) to statistically compare differences in regression relationships. As discussed in Planned Major Revision 3, multiple group comparison with ANCOVA does not indicate a significant difference in slope between our dataset and that of Waldron et al. (1999). However, regardless of the method of estimating $\delta^2H\text{-}H_2O$ used, the analysis of the larger dataset produces a flatter slope. Therefore we infer that the 'true' global slope is likely to be flatter than that inferred by Waldron et al., 1999, but also that more data and further analysis is needed to confirm this and reduce the uncertainty of the slope.**

Assess whether the expanded field data set is predominantly 13C-enriched compared to the in-vivo relationship described in Waldron et al 1999, and therefore consistent with an interpretation that differences in field $\delta D\text{-}CH_4$ may be an artefact of fractionating processes post-production than pathway per se This is advocated as I am still unaware of experimental evidence methanogenic pathway in shallow freshwaters changes $\delta D\text{-}CH_4,$ but there is evidence of processes, oxidation and mixing, causing enrichment, and so this approach is consistent with scientific principle of parsimony and interpreting data using the simplest approach.

**We have assessed this and in fact the opposite is the case. The data from sites included in Waldron et al., (1999) is somewhat higher in $\delta^{13}C\text{-}CH_4$ (-60.8±0.9‰ SEM) relative to the total dataset (-62.6±0.6‰) or the sites that were not included in Waldron et al., (1999) (-63.4±0.8‰). We do not think it is likely that there is systematic difference in these sets of sites in terms of post-production processes, which we take to mean oxidation, diffusive fractionation, and mixing of different $CH_4$ reservoirs. We agree that such processes can lead to variation in $\delta^2H\text{-}CH_4$ (see planned major revision 4 above), but see no evidence that this explains the difference between our dataset and that of Waldron et al., (1999).**

To explore why the paired $\delta D\text{-}CH_4\text{-}\delta D\text{-}H_2O$ measurements are not fully described by the best fit line, the authors explore whether a difference in (dominant) methanogenic pathway is evident in the data. With no evidence from paired $\delta D\text{-}CH_4\text{-}\delta 13CH_4$ the authors draw on $a_c$ as a proxy for methanogenic pathway to assess this. Step-wise regression is used to explore this. I think this is interesting and something to revisit when the paired data relying in predicted $\delta D\text{-}H_2O$ has been removed, but currently it is the next floor in the 'house-of-cards', reliant on data that we do not know to be accurate, and therefore the significant relationships that the authors infer changes in methanogenic pathway from, we do not know to be true.

**We have re-assessed the step-wise regression relationship between $\alpha_C$ and d2H-CH4,wO using revised values for the latter as described above (Planned Major Revision 4). Indeed to ensure this relationship is robust we tested it using two different approaches: 1) $\delta^2H\text{-}CH_{4,w0}$ using the 'best estimate' for $\delta^2H\text{-}H_2O$, as described above; 2) $\delta^2H\text{-}CH_{4,w0}$ using only sites with measured $\delta^2H\text{-}H_2O$. Both of these approaches indicated a step-wise linear relationship. However, the two relationships generated were not consistent in the breakpoint, and the linear relationships were not all statistically significant. Given this**

**result we agree that it is prudent to focus less on methanogenic pathway as an explanation of residual variability in $\delta^2$H-CH$_4$, and instead discuss the complex interrelationship of multiple variables and processes that can influence $\delta^2$H-CH$_4$, namely i) methanogenic pathway; (ii) methane oxidation; (iii) isotopic fractionation due to diffusion; and (iv) differential thermodynamic favorability of methanogenesis, or differential enzymatic reversibility.**

The authors in their revision should be careful in the value of thinking about $a_c$ for the following reasons: some of the literature generating $a_c$ relies on assumption of differences in methanogenic pathway interpreted from differences in $\delta$D-CH$_4$, but there is competng evidence $\delta$D-CH$_4$ cannot be interpreted in this way (so $a_c$ using $a_c$ to infer methanogenic pathway in $\delta$D-CH$_4$ when $\delta$D-CH$_4$ has been used to infer methanogenic pathway becomes a circular, self-supporting and flawed approach).

**We do not agree with the reviewer's contention of circular reasoning here. While both $\alpha_c$ and $\delta^2$H-CH$_4$ have in the past been used to infer methanogenic pathway, the use of $\alpha_c$ is primarily based on theoretical predictions of fractionation factors for these pathways, and to our knowledge has not been 'validated' via analysis of $\delta^2$H-CH$_4$. There is evidence from culturing studies (e.g. Valentine et al., 2004;Penning et al., 2006a), and from studies that isolate specific pathways in the environment (e.g. Penning et al., 2006a,b; Galand et al., 2010), that $\alpha_c$ varies in relation to differences in methanogenic pathway.**

**However, we do note that other variables related to methanogenesis have the potential to influence $\alpha_C$, including enzymatic reversibility and the thermodynamic favorability of methanogenesis, as well as diffusive fractionation and methane oxidation. In addition, sources and sinks of CO$_2$ in natural environments that are independent of methanogenesis will also influence measured $\alpha_C$. We discussed this possibility in the original manuscript (Lines 510 to 518), but we will give more emphasis to this in the revised manuscript. See Planned Major Revision 4 above.**

To help here I would advise the authors to consider Waldron et al 1998 (Geomicrobiology, 15, 157-169), which contributes to the in-vitro line in Waldron et al 1999, but the authors do not cite so I am unsure if they are aware of the detail in this.
Here dominance of methanogenic pathway was changed in mixed culture (as would be found in the field) incubations, and $\delta$D-CH$_4$ monitored with time – so not just one measurement as may be misinterpreted from Waldron et al 1999. Except for one measurement broadly within analytical uncertainty, $\delta$D-CH$_4$ remained constant. However, $\delta$13CH$_4$ did change and consistently with fractionation ranges for the methanogenic pathways thought to be dominant (as assessed from independent measurements of substrate turnover). I advise the authors to consult Waldron 1998 for two reasons:
1. The authors approach in the bgd paper to draw on $\delta$D-CH$_4$ to represent differences in methanogenic pathway would be stronger if they can provide an explanation for the constancy in $\delta$D-CH$_4$ while $\delta$13CH$_4$ changes.

**This is an important study and we appreciate the reviewer highlighting it. We will certainly cite and discuss it in the revised manuscript. The finding of constant $\delta^2$H-CH$_4$ is intriguing. Recent pure culture studies have clearly shown that acetoclastic methanogenesis differs in hydrogen isotope fractionation from hydrogenotrophic methanogenesis under the same conditions (i.e. Gruen et al., 2018), implying that acetate-methyl hydrogen does not fully**

exchange with water during methanogenesis. Therefore we infer that the effect observed in Waldron et al., (1998) likely results from hydrogen isotope exchange with water during fermentation of acetate from butyrate or other substrates. The constancy of the $\delta^2$H-CH$_4$ would therefore imply that the isotopic fractionation of H-exchange between water and the acetate methyl group effectively compensates for the difference in hydrogen isotope fractionation between acetoclastic methanogenesis and hydrogenotrophic methanogenesis. Clearly, this is an interesting result that merits further study to resolve with the results of pure culture experiments. However, we do not feel that this study on its own negates the potential for differential net hydrogen isotope fractionation between acetoclastic methanogenesis and hydrogenotrophic methanogenesis.

As discussed above in Planned Major Revision 4, in the revised manuscript we will focus less on the role methanogenic pathway in controlling $\delta^2$H-CH$_4$, and emphasize the complexities induced by multiple mechanisms influencing hydrogen isotope fractionation during and after methanogenesis.

2. Waldron et al can also be used to calculate a$_c$ (both from CO$_2$ and from estimated substrate composition). a$_c$ CO$_2$-CH$_4$ generates values of 1.057 for the period when CO$_2$ reduction is considered dominant (i) and 1.055 when acetoclastic methanogenesis is considered dominant (ii). These are very similar and it would be valuable to understand how the authors interpret this when they infer much wider ranges in a$_c$. For clarity $\delta_{13}$CO$_2$ and $\delta_{13}$CH$_4$ respectively for (i) were -8.3 ‰ and -62‰ , and for (ii) were 1.55‰ and -47.5‰

This is also an interesting result. We think there could be some complicating factors that influence $\delta^{13}$C-CO$_2$ in this study in particular. We note that the headspace concentration of CO$_2$ decreased through the experiment, which would not be the expected stoichiometric result of a net shift from hydrogenotrophic to acetoclastic methanogenesis. This suggests that there were additional sinks of CO$_2$ in the experiment that became more prevalent as the experiment proceeded, and this may have led to the observed enrichment in $\delta^{13}$C-CO$_2$. In particular we are curious about the possible role of increased homoacetogenesis, although this is difficult to evaluate based on the results of the study.

Overall, we do not think this finding necessarily negates the use of $\alpha_C$ as an indicator of differences in methanogenic pathway, which is supported by other studies (e.g. Penning et al., 2006a,b; Galand et al., 2010). But it does point to the potential for other variables to complicate the relationship between $\alpha_C$ and the relative proportion of different pathways. In our revised manuscript we will highlight these complications, including citing this paper. See planned major revision 4.

Abstract: Is clear and summarises the paper but projects a future methane emissions scenario (L25-26) before the modelling and assessment of how well this approach can reconstruct current estimates ( L27-30) and this seems in the wrong order to me, given the former has a reliance on the latter. Further, the abstract does not acknowledge this research is augmenting the research that historically first documented the global relationship between δD-CH$_4$ and δD-H$_2$O easily addressed for example by changing L12 to 'We have refined the existing global relationship between δD-CH$_4$ - δD-H$_2$O by the compilation of a more extensive global dataset…."

We agree with these suggestions, and we will revise the abstract accordingly. We note that based on the suggestions of the other reviewers there will be other changes to the abstract, including a modification of the description of the upscaling component of the manuscript.

L28: The authors postulate the mismatch is dependent only on the work of others (emission inventories, etc) and not possibly an error in their approach. Scientifically this is not correct – both 'sides' could have errors.

**We will change this as part of revising the upscaling results and analysis.**

L19: results do not imply; one interprets data to generate a 'result'.

**We will change the language here to make clear this is an interpretation of the data.**

L22: high (more $^{13}$C-enriched) in rivers and bogs - this is the dataset that has more $\delta$D-H$_2$O projected, so is this an artefact of the modelling than a real biome-specific difference?

**We are not sure what the reviewer means by 'this is the dataset that has more d2H-H2O projected.' 81% of bog sites have $\delta^2$H-H$_2$O measurements, while 37% of river sites have this measurement. For the dataset as a whole the percentage is 48%.**

**As discussed above (Planned Major Revision 3) we carefully assess the use of modeled precipitation $\delta^2$H-H$_2$O, and find it does not have a major impact on the regression relationship between $\delta^2$H-H$_2$O and $\delta^2$H-CH$_4$. Therefore we do not believe this result is an artifact. Regardless, we will revise our analysis of differences by ecosystem using the 'best-estimate' $\delta^2$H-H$_2$O and resulting $\delta^2$H-CH$_{4,w0}$.**

L27: integrated (by mass balance) not combined (which is used when sources are added) – which I know the authors have done (L204) but the descriptor is incorrect here.

**We will change the wording here.**

L36: I think the following references is missing: Variability in Atmospheric Methane From Fossil Fuel and Microbial Sources Over the Last Three Decades. / Thompson et al: Geophysical Research Letters, Vol. 45, No. 20, 28.10.2018, p. 11499-11508 (and I invite the authors to wonder if also some of the work from the Royal Holloway group should augment L47-51)

**We thank the reviewer for this suggestion, and we will add the suggested reference and also see if other work from the Royal Holloway group would be good to add in the suggested part of the introduction.**

L59 & L83 Citations are given in chronological order of 1999b and 1999a which seems not typical convention to me (uncertain of the referencing convention for BG but for example the two references for Walter K are not in chronological order in the reference list so the in-paper citations would not be b then a due to this convention in the reference list?)

**We were relying on EndNote for citation management, and there may have been some errors with the citation format in the software. We will check this carefully.**

L68: Logic only follows that impact on $\delta^{13}$CH$_4$ can affect geographic provenancing if reader knows it can also affect $\delta$D-CH$_4$, so does this need to be made explicit?

**We do not understand the reviewer's comment here. The hypothesized geographic variation in $\delta^{13}$C-CH$_4$ is independent of variation in $\delta^2$H-CH$_4$, as they are controlled by different mechanisms. We will make this clearer in the revised text.**

L70: this implies that different ecosystems have different methanogenic pathways. More accurate text would be "differentiated geographically based on ecosystem differences in the relative strengths of different methanogenic pathways and $\delta_{13}$C of source organic matter" (as per the introduction of the Ganesam paper). Noting relative strengths is important, as a common mistake propagated in the literature and again here (L???) is to assume methanogenesis proceeds by one methanogenic pathway only – this would be rare, with field-based methane production contemporaneous from CO$_2$ and acetate, and varying temporally in strength as input of fresh OM changes seasonally (or not).

**The reviewer raises an important point here, and we will make it clear that the difference is in the relative strength of the pathways operating in difference ecosystems, and not different pathways per se.**

L84-85: sounds a bit defensive? How about "We have advanced existing compilations of freshwater $\delta$D-CH$_4$ by 1,2,3 …? I would remove significantly (statistical connotations) and just say larger as the number speak for themselves.

**We agree with this suggestion and will make the proposed change.**

L91: The aims are clear (good) but 'then' and 'potential' not needed – the latter as embedded in implications that there is a potential for impact

**We will make this change**

L106 & L117, 9L206 and possibly elsewhere): small w for where, as this follows from an unfinished sentence in both cases with the equation used in between

**We will make this change**

L136: the five ecosystem categories are not clear from this sentence: 'lakes' and 'rivers' and then there are five wetlands listed. Further, it is debatable that floodplains are aligned with rivers as CH$_4$ production would only occur when sediments are deoxygenated from standing water. So I would say more with ponds as the recession of water can be slow and could be like a pond drying in some situations. Noteworthy here is that gas loss from rivers is velocity dependent (see Long et al (2015) Hydraulics are a first order control on CO$_2$ efflux from fluvial systems Journal of Geophysical Research – Biogeosciences, 120, (doi:10.1002/2015JG002955), and similar references. This will also be the case with methane – possibly more so as insoluble, and may cause an isotope fractionation independent of degassing, and may also be a reason the Amazon rivers in Fig. 5 plot differently.

**We note that there was an error in this text and it is actually six categories. We will add a numbered list to make this clearer. We note that essentially all of the river data come from floodplain lakes or deltas, with one exception, and most of the data are from the Amazon. We think it is valid to continue to differentiate these environments from other lakes and ponds, since they are different from typical lakes and ponds in a number of ways (overturning and redox regimes, nutrient inputs, dynamics of gas loss and hydraulics). We will make the nature of the river/floodplain/delta sites clearer in the revised manuscript.**

**The reviewer mentions an important point about degassing dynamics in rivers, and we will include this in our discussion of differences between environments in Section 4.4.**

L139: Similarly, I question the scientific integrity in lumping lakes with rivers here – gas loss from river systems is controlled by hydrological processes primarily and there could be fractionations during emission from lotic systems that are different to lentic systems where diffusion and wind of lake thermal orographic processes control turnover. This starts to become important where these mean sources are used to simulate a resultant atmospheric composition e.g. L227. Thus, the authors should think about how to provide added confidence of the robustness of their catergorisation.

**See above. We are primarily analyzing floodplain lakes and deltas in the river category. We will make this clearer in the revised manuscript.**

**However, the basis of this categorization is essentially to align with flux inventories (Saunois et al., 2020), which specifically define an 'inland water' category that includes rivers and lakes, as well as reservoirs. We keep to this categorization in order to be able to compare with the flux estimates. We do clearly discuss possible differences between the river and lake sites in the Discussion.**

L145-148: Such categorisation is good, and the open access data set is very welcome. This categorisation relies on the integrity of the interpretation, but this integrity is important as the data analysis relies on this. With 131 sites it is impossible for the reviewer to know each site and so as a 6 check I can only look at my own data: L61 in the excel files. These methane samples were collected in-situ from porewater diffusing into samplers embedded in the peat (the GBC abstract notes in-situ and the methods clarifies at depth sampling) so I would classify as more aligned with dissolved porewater than diffusive flux (which is normally associated with the potential for oxidation and change in $\delta$ _v_a_l_u_e_s_ _). Further I comment in the GBC paper there is a dynamic zone and interpret that is the section from which gas can be emitted. Mean $\delta D\text{-}CH_4$ here is -332 ± 17‰, more depleted the -294 ± 39‰ used in the table and subsequent data analysis. Thus, some feedback from the authors in the revised manuscript that their interpretations are not sensitive to the variation their interpretation of environment and which data to use would be valuable.

**The reviewer raises a valid point about the complexities of each site. This complements the comments of reviewer 3 about the validity of including data from deep peat samples. As discussed in our response to reviewer 3 we think this is a valid concern and we have decided to limit our data from peatlands to the uppermost 50 cm. See Planned Major Revision 1. This coincides approximately with the dynamic zone mentioned here.**

**We note that peatlands (bogs and fens) were the only environments to be sampled on depth gradients, and that similar issues are unlikely to affect the interpretation of data from other ecosystem categories. We will change the entry in the Table for this Waldron et al. 1999b to Dissolved-Pore Water. Since we group diffusive flux and dissolved pore water, this distinction does not make a difference to our analysis.**

L152 – typically small – as this manuscript relies on several source of data estimation (here, $\delta_2H_2O$, it would be good to provide estimates as to what the maximum is this would manifest in $\delta D$ (recognising that it changes with resolution and scale of figure and so this is challenging, but saying small is insufficient).

**We will provide a quantitative estimate of the likely error produced by this analysis, as a percent error, and then translate that to $\delta^2H$ values.**

L177: the authors need to unpick for the reader the statement more as they have with L179 onwards. I am thus left to interpret the reasoning. I assume it is based on considerations that methanogenic pathway influences $\delta D\text{-}CH_4$? If so please see earlier substantive comments on this and decide whether to proceed in the revised manuscript.

**Based on the reviewers comments here and below we plan to thoroughly revise section 2.3.2 to discuss in more detail how $\delta^2H\text{-}CH_4$ is predicted to vary with carbon isotope composition, and the different processes that would influence that co-variation, considering an expanded set of processes. See Planned Major Revision 4 above.**

L200: Clarify where the flux estimate comes from at this point – I presume from Saunois et al as in L209, but this should be clarified when first introduced. I am not expert enough to judge if the methodology for the bottom up flux section is sound, but it seems reasonable to me.

**We will clarify the source of the flux estimates earlier in this section. It is indeed Saunois et al., 2020.**

L 267: given the statistical approaches such as Monte Carlo bootstrapping used with the flux estimate section previously I would have expected more rigorous comparison should be undertaken here to show if there is a statistical offset between measured and predicted $\delta D\text{-}H_2O$ than relying on descriptors of "generally good agreement" and using RMSE. The RMSE is a red herring if the lines generating 19 and 23 ‰ do not overlap - ?

**As discussed above, we will provide a more detailed analysis of the comparison of measured and modeled $\delta^2H\text{-}H_2O$. See Planned Major Revision 1.**

Fig 2: Should the predicted (postulated and therefore dependent) not be regressed onto the measured (the true field value, so measured and independent and as a control of $\delta D\text{-}CH_4$ the one to get as close to the true value as possible)?

**This depends on the goal of the regression. In this case we are attempting to develop a regression relationship that predicts measured $\delta^2H\text{-}H_2O$ as a function of modeled $\delta^2H\text{-}H_2O$, in order to use this as a predictive tool for sites without $\delta^2H\text{-}H_2O$ measurements, and to assess the goodness of fit. Therefore it makes more sense to have measured $\delta^2H\text{-}H_2O$ on the y-axis in this case, and we will keep this orientation for the revised Figure 2.**

Fig 3B: this needs revisited once the $\delta D\text{-}CH_4$ - $\delta D\text{-}H_2O$ predicted data has been removed as described above. There may still be an inland water specific difference here, but again that this may not be controlled by anything more complex than lentic and lotic freshwater systems having generalised differences in gas transport mechanism (ebullition or diffusion). These would be influenced by atmospheric and sediment interface boundary layer dynamics, transit time, depth of oxidative zone, lake stratification, and surface roughness, with the latter in turn influenced by wind speed, depth of water, and river flow velocity, slope. In other words, considerable methane isotope fractionation (enrichment) is possible, or not.

**Based on our revised analysis, we find that we cannot detect a significant difference in the regression relationship between inland waters and wetlands (See Planned Major Revision 3**

**above). The difference inferred in the original manuscript is likely partly a result of the hydrological differences in these environments, and resulting differences in the regression of modeled vs measured $\delta^2$H-H$_2$O (see Planned Major Revision 2). Therefore we will revise the manuscript here to reflect this revised understanding.**

Fig 4. It is good to see this plotted but not surprising given $\delta$D-H$_2$O varies with latitude and $\delta$D-CH$_4$ varies with $\delta$D-H$_2$O. The same difficulties in estimating field $\delta$D-H$_2$O from modelled $\delta$D-H$_2$O are evident when considering $\delta$D-CH$_4$ as a function of predicted $\delta$D-H$_2$O. The authors need to note here that there may be an imbalance of where methane is sampled from globally and so if more measurements existed from the higher latitudes then there may be as much scatter as with the lower latitudes.

**We are glad the reviewer agrees with us on the utility of plotting the data in this way. We did note the uneven geographic distribution of data at several points in the manuscript, but can further emphasize the likelihood of similar scatter at all latitudes with more sampling. We are assuming the reviewer meant to say there is greater scatter at high latitudes, which is what we observe.**

Section 3.4 jumps to something completely different with L313 "shifts to being controlled by changes in methanogenic pathway to being controlled by ….". There has not been clear discussion from the authors to date they are considering changes in methanogenic pathway of $\delta$D-CH$_4$ so this seems out of context. And yet L317 goes on to consider this in more detail. The key message in the Waldron et al 1999 paper is that considering methanogenic pathway a control on $\delta$D-CH$_4$ is misplaced and that "that 50% of the variation in natural $\delta$D-CH$_4$ samples can be explained by $\delta$D-H$_2$O, with isotopic fractionation post-production, or mixing with gas already fractionated likely responsible for most of the noise in the natural system". The analysis prior to section 3.4 may be more likely to support this interpretation than refute it, particularly when the data in Fig. 3.2. is appropriately compared (as described earlier), and so now considering data as a function of methanogenic pathway seems to be ignoring this. Indeed the authors observe they find no relationship between $\delta_{13}$C-CH$_4$ and $\delta_2$H- CH$_{4,W0}$ which would be expected if $\delta_2$H-CH$_4$ was influenced by methanogenic pathway as $\delta_{13}$C-CH$_4$ is (Fig. 5a). Thus, the authors should not make clearer statements such as L312 of "shifts from being controlled by variation in methanogenesis pathway" are inferred controls.

**This is clearly a key point of concern for the reviewer, and we understand the reservations about inferring that variability is a function of methanogenic pathway. However, we think it is unlikely that all of the remaining variability not explained by $\delta^2$H-H$_2$O is controlled by "isotopic fractionation post-production, or mixing with gas already fractionated". First, it is important to be clear about what these post-production processes are. To our knowledge there are two key post-production processes that can affect methane isotopic composition: methane oxidation (either aerobic or anaerobic) or isotopic fractionation caused by diffusion. We are unaware of other important processes. Both of these processes would be likely to lead to higher $\delta^2$H-CH$_4$, and lead to positive co-variation with $\delta^{13}$C-CH$_4$. Oxidation will also lead to negative co-variation with $\alpha_c$, because CH$_4$ is invariably oxidized to CO$_2$, leading to a smaller isotopic difference between these gases. However, diffusion would lead to positive co-variation with $\alpha_c$, because diffusion is expected to have a smaller isotopic effect on CO$_2$, as a result of smaller mass difference between CO$_2$ isotopologues (Chanton, 2005). Mixing effects will depend on the mixing end-members. Unless there is a large proportion of non-microbial methane present, which we argue is unlikely in most circumstances, mixing will not alter the overall isotopic signature of microbial methane in**

the ecosystem. It is possible to have mixing with 'gas already fractionated', but in this case the underlying fractionation is the key process controlling the isotopic composition of the resulting gas, and again to our knowledge this would have to be the result of oxidation or diffusion.

It is unclear on what basis Waldron et al (1999) ascribed the remaining ~50% of variability in $\delta^2$H-CH$_4$ to these post-production processes, and we would argue that this assertion is untested.

We do agree that our focus on methanogenic pathway did not include other plausible mechanisms for co-variation between $\delta^2$H-CH$_4$ and $\alpha_C$. We plan to expand our discussion to take other processes, namely (i) diffusion and (ii) differences in enzymatic reversibility, into account.

Figs. 5b=c. The uncertainty around what $a_c$ should be for different methanogenic pathways has been described earlier in this review. But additionally, although breakpoint analysis was used, there is a high dependence in this on data set that has enriched $\delta_2$H-CH$_4$ to generate opposing trends. The eye is drawn by the projected pathways, but if these was not included as we cannot be sure it is oxidation[1] and all the remaining data was considered in a weighted regression would there be trends?
[1]If the high $\delta_2$H-CH$_4$ is from the Amazonian rivers, there are shales in this basin that fuel C cycling (Vihermaa et al) and this could be thermogenic: $\delta_2$H-CH$_4$ is also consistent with this. Vihermaa L.E., Waldron S. , Garnett M.H., and Newton J. (2014) Old carbon contributes to aquatic emissions of carbon dioxide in the Amazon. Biogeosciences, 11, 3635-3645. (doi: 10.5194/bgd-11-1773-2014).

We acknowledge concerns about the 'predicted trends', both by reviewer 1 and 2, and therefore we will remove them from the revised manuscript. We will instead focus on the patterns of co-variation, and potential explanations for them. As discussed above (Planned Major Revision 4) we will focus less on methanogenic pathway, and increase our focus on other mechanisms.

We agree that the one outlying point with very high $\delta^2$H-CH$_4$ (and $\delta^{13}$C-CH$_4$) is questionable, and may be thermogenic methane. It is indeed from the Amazon. We will therefore remove this from our dataset and repeat the analyses (Planned Major Revision 1)

As noted above, weighted regression is leading to biases in this analysis, and is not generally preferable to unweighted regression (Fletcher and Dixon, 2012), and therefore we are proceeding with unweighted regression in the revised manuscript.

It is remarkable Fig 7 is so consistent – this is very interesting. Is it what we would expect?

We assume the reviewer is referring to Figure 7B. This is not necessarily what we would expect based on other studies. We have already provided some discussion of this in section 4.4, but plan to revise this in response to questions from reviewer 2, especially focusing on possible biases in the $\delta^{13}$C-CH$_4$ dataset.

L370 discussion is over-interpretations given the differences between sites are not statistically significant. It would be ok to say the prevalence of more depleted CH$_4$ is greater in the ecosystems sampled but for example this could represent accessibility of field sites, or differential

investment into research measurements in these areas, than group compositional differences per se. Ecosystem types are not evenly distributed by latitude (L370) – nor is resource for investment in field research with tropical regions of the Earth lacking measurement due to access or financial constraints – we need to start recognising what we have not measured is as important as what we measure.

**We agree that this analysis is preliminary given the small sample sizes for each ecosystem. We tried to emphasize this in the original manuscript, and noted that the possible differences represented hypotheses that merited further testing. But we will further emphasize this. We will also note clearly here that more investigation of tropical ecosystems is especially important.**

Fig. 10 is tiny and needs to be bigger

**We are going to revise Figure 10 to simplify it based on comments from reviewers 2 and 3 on the upscaling exercise. We will reduce it to a single panel (equivalent to Figure 10C), which will make it more legible.**

L426 "roughly as strong a predictor". Too big a leap: explain how – from ice core gases "roughly is a colloquialism"

**We do not fully understand this comment, but we agree that this language is imprecise, and we will make a more quantitative statement. We are not sure what the reviewer is saying about ice-core gases.**

L487 – as noted earlier, the paired measured values plot on Waldron et al 1999 In-vitro line, consolidating further the significant of this line. Please acknowledge this.

**We are assuming the reviewer meant the in-vivo line here. As discussed above, we will provide a more thorough comparison of the Waldron in-vivo line with the results of this study, and will represent this in the revised discussion. Our analysis (Table R2) shows that the paired measured values do not plot on the Waldron et al (1999) in vivo line, and have a flatter slope. This difference is not significant based on ANCOVA, but we infer that the larger dataset implies a flatter global slope.**

L508 – in the revised manuscript please detail the % variation explained by $\delta D$-$H_2O$ and then additionally by $a_c$ should this prove to still be important

**We will perform a re-assessment of this analysis, and include it if $\alpha_c$ is still a significant predictor of $\delta^2H$-$CH_4$. If so, we will clearly detail the % variation that is explained by these two variables.**

L510 – this is the crux of what is new to explore in isotope biogeochemistry of methane and also the role of methanol substrates.

**We agree with the reviewer that $CH_4$ isotopic variability related to enzymatic reversibility is an important topic, and based on other comments we will expand discussion of this here, as well as in the methods and results. At this point there is little we can say about methanol substrates, but we will mention it briefly as another variable that merits consideration.**

L519 – same comments as before about is there really a relationship, but why more points classified as oxidised with this pairing than with $a_c$?

**This is an interesting question, and we don't know the answer. We would speculate that it is because sources of $CO_2$ can be very variable, and this may be adding noise to Figure 5C that is not present in figure 5B. As discussed above we will be substantially revising this section, but will point out the differences in these two plots as an indicator of complex interactions of different biogeochemical processes.**

L551- Much of 4.31. is repeating statements first described in Waldron et al 1999 section 1.1., paragraph starting "In addition…" but this is not referenced and as written implies the authors are the primary source of this thinking. This is not the case and should be referenced appropriately to indicate this was first noted 20+ years ago.

**We regret that we did not acknowledge the earlier statement of these ideas. We will thoroughly revise this section to provide credit to Waldron et al., (1999) for the ideas that are presented there.**

L564 – please note pure cultures are not representative of the field processes of methane production and thus the batch cultures and other experimental data collated in Waldron et al 1998, 1999 are. This is not clear from the statement.

**We agree that pure cultures are not representative of methanogenesis in the environment. We are not sure that batch cultures or incubations are truly representative either, in that they do not necessarily fully represent the processes occurring in natural environments, but agree they are clearly a closer approximation than pure cultures. We did try to make this distinction clear in the original manuscript, but will further clarify in the revised manuscript. However, we do feel that inferences from pure cultures are important for understanding the more complex processes that occur in batch cultures or natural environments. For example, differential hydrogen isotope fractionation between water and methane by methanogenic pathway has been clearly observed in pure culture experiments (i.e. Gruen et al., 2018). This implies that acetate methyl hydrogen is not fully equilibrated with water during the methanogenesis reaction itself.**

L569, please reverse the order of the references or remove Whiticar 1999. The Waldron 1999 paper is the one that is particularly focussed on the global relationship between $\delta D\text{-}CH_4$ - $\delta D\text{-}H_2O$, and constructs the first global relationship, which this paper finds with new data is similar. This gives appropriate credit to the conceptual understanding. The Whiticar paper coplots $\delta D\text{-}CH_4$ - $\delta D$-$H_2O$ but does not assert that " $\delta_2 H\text{-}H_2O$ is a primary determinant of $\delta_2 H\text{-}CH_4$ on a global scale", rather the focus is on the interpretation of how $\delta_2 H\text{-}CH_4$ reflects methanogenic pathway or marine vs. freshwater.

**There seems to be an error in the page numbering, and we are not sure which citation the reviewer is referring to. However, we will make clear in the revised manuscript that Waldron et al., (1999) first proposed and found evidence for the global relationship between $\delta D\text{-}CH_4$ - $\delta D\text{-}H_2O$.**

To conclude: this has been an uncomfortable review for me to undertake as my position of not anonymising the review puts me up for public scrutiny, and a misinterpreted that I am trying to defend my work and am unwilling to accept an addition to this. This does not represent my professional scientific principles, I would urge the authors to accept this is not the case - indeed in

the 1999 GCA paper I welcome refinement of my work. However, the authors have still not presented here compelling evidence that $\delta D\text{-}CH_4$ can represent well different methanogenic pathways and so the reliance of this in the manuscript I find troubling. I consider the $a_c$ approach may be valuable in helping constrain the signal in $\delta D\text{-}CH_4$ that is not defined by $\delta D\text{-}H_2O$, but the current manuscript is not constraining uncertainty sufficiently and the approach is therefore flawed. I would urge the authors to find a way to better constrain projected $\delta D\text{-}H_2O$ and revisit this, or work with only measured data and revisit this. Their refined analysis should undertake rigorous statistical comparison with the existing field $\delta D\text{-}CH_4$ - $\delta D\text{-}H_2O$ relationship from Waldron et al 1999 to say whether it is different (although the new larger dataset will likely be a more representative relationship that the community can go forward with), and adopt a parsimonious interpretation of variation within the data set, as that is least likely to induce an erroneous interpretation. The biome specific considerations and upscaling should also be revisited if the removal of biased and inaccurate data pairings changes the source bulk compositions, and further thought should be given to the basis for source differentiation based on scenarios of methane production and loss in this upscaling.

**Once again, we regret that this has been an uncomfortable review process. We appreciate the frank and detailed signed review, and the collaborative nature of the comments. We do not agree with all arguments made by the reviewer, but do agree with many of the suggested improvements to the manuscript, and will make these changes in a thoroughly revised manuscript. See the Planned Major Revisions above for a summary of these changes.**

**We believe that these changes will address the reviewer's concerns and will greatly strengthen the conclusions of the manuscript.**

**References Cited:**

Chanton, Jeffrey P. "The effect of gas transport on the isotope signature of methane in wetlands." *Organic Geochemistry* 36.5 (2005): 753-768.

Fletcher, D., & Dixon, P. M. (2012). Modelling data from different sites, times or studies: weighted vs. unweighted regression. *Methods in Ecology and Evolution*, *3*(1), 168-176.

Galand, P. E., Kim Yrjälä, and Ralfi Conrad. "Stable carbon isotope fractionation during methanogenesis in three boreal peatland ecosystems." *Biogeosciences* 7.11 (2010): 3893-3900.

Gruen, Danielle S., David T. Wang, Martin Könneke, Begüm D. Topçuoğlu, Lucy C. Stewart, Tobias Goldhammer, James F. Holden, Kai-Uwe Hinrichs, and Shuhei Ono. "Experimental investigation on the controls of clumped isotopologue and hydrogen isotope ratios in microbial methane." *Geochimica et Cosmochimica Acta* 237 (2018): 339-356.

Penning, H., Claus, P., Casper, P., & Conrad, R. (2006). Carbon isotope fractionation during acetoclastic methanogenesis by Methanosaeta concilii in culture and a lake sediment. *Applied and Environmental Microbiology*, *72*(8), 5648-5652.

Penning, H., S. C. Tyler, and R. Conrad. "Determination of isotope fractionation factors and

quantification of carbon flow by stable carbon isotope signatures in a methanogenic rice root model system." *Geobiology* 4.2 (2006): 109-121.

Rice, A. L., Butenhoff, C. L., Teama, D. G., Röger, F. H., Khalil, M. A. K., and Rasmussen, R. A.: Atmospheric methane isotopic record favors fossil sources flat in 1980s and 1990s with recent increase, Proceedings of the National Academy of Sciences, 113, 10791-10796, 2016.

Saunois, M., Stavert, A. R., Poulter, B., Bousquet, P., Canadell, J. G., Jackson, R. B., Raymond, P. A., Dlugokencky, E. J., Houweling, S., and Patra, P. K.: The global methane budget 2000–2017, Earth System Science Data, 12, 1561-1623, 2020.

Sherwood, O. A., Schwietzke, S., Arling, V. A., and Etiope, G.: Global inventory of gas geochemistry data from fossil fuel, microbial and burning sources, version 2017, Earth System Science Data, 9, 2017.

Valentine, D. L., Chidthaisong, A., Rice, A., Reeburgh, W. S., & Tyler, S. C. (2004). Carbon and hydrogen isotope fractionation by moderately thermophilic methanogens. *Geochimica et Cosmochimica Acta*, *68*(7), 1571-1590.

Waldron, S., Lansdown, J., Scott, E., Fallick, A., and Hall, A.: The global influence of the hydrogen iostope composition of water on that of bacteriogenic methane from shallow freshwater environments, Geochim Cosmochim Ac, 63, 2237-2245, 1999a.

---

## Author Comment (AC2) · 15 Feb 2021

**Response to reviewer 2**

All of the reviewers provided excellent suggestions and feedback on the paper, and we think that by addressing their concerns the paper will be greatly improved. Many of their comments were complementary. Therefore we will first summarize the major revisions we plan to make to the paper before responding to each reviewer in detail:

**Planned Major Revisions**
**1)** We have revised the $\delta^2$H-CH$_4$ dataset in response to comments from Reviewer 1 and Reviewer 3. (i) For peatland sites with depth stratified sampling we have decided to only include samples from the upper 50 cm, as suggested by reviewer 3, since this is the depth range that is most likely to emit CH$_4$ to the atmosphere. This affects a total of 8 sites. (ii) Reviewer 1 noted that an outlier sample from the Amazon River with very high $\delta^2$H-CH$_4$ and $\delta^{13}$C-CH$_4$ could be derived from thermogenic methane. We agree that this outlier is suspect, and therefore have decided not to include it. (iii) We also noted that one site (Mirror Lake, Florida, USA) was analyzed in two separate studies, and therefore was included twice in the dataset. We have combined the data from the two studies into one site entry.

**2)** As suggested by all three reviewers, we have performed much more rigorous analysis of the relationship between measured and modeled $\delta^2$H-H$_2$O values. Specifically we have done the following: (i) In addition to annual precipitation $\delta^2$H values, we now also analyze growing season precipitation $\delta^2$H, which is defined as the amount-weighted mean $\delta^2$H of months with mean temperature greater than 0º C. This provides an opportunity to assess whether seasonal variation in precipitation in the mid to high-latitudes is important in controlling the environmental $\delta^2$H-H$_2$O value; (ii) separately analyzing inland water and wetland environments, since these are different hydrological environments and the controls on $\delta^2$H-H$_2$O are potentially different.

This analysis led to the following key results (see Table R1 below for summary of results): A) growing season modeled precipitation $\delta^2$H is a better predictor of inland water $\delta^2$H-H$_2$O than annual precipitation $\delta^2$H, in that the regression curve is indistinguishable from the 1:1 line. B) annual modeled precipitation $\delta^2$H is a better predictor of wetland $\delta^2$H-H$_2$O, in that the slope of the regression is indistinguishable from 1, and the R$^2$ value is higher. However, the regression line is offset from the 1:1 line by 18.6±9‰. We interpret this as an indicator of likely widespread evaporative effects on $\delta^2$H-H$_2$O in wetland environments.

We use these results to then develop a 'best estimate' for comparing $\delta^2$H-H$_2$O with d2H-CH4. (i) For sites with measured $\delta^2$H-H$_2$O values we use the measured value. (ii) For inland water sites without measured $\delta^2$H-H$_2$O we use modeled growing season precipitation, since as discussed above the regression of this against measured $\delta^2$H-H$_2$O is indistinguishable from the 1:1 line. (iii) For wetland sites without measured $\delta^2$H-H$_2$O we estimate the $\delta^2$H-H$_2$O using the regression relationship with annual precipitation $\delta^2$H-H$_2$O shown in Table R2. We feel this approach combining measured and modeled data is

most consistent with that of Waldron et al., 1999, who we note also analyzed a combination of sites with measured $\delta^2$H-$H_2O$ (29 out of 51 sites) and estimated $\delta^2$H-$H_2O$ based on precipitation isotopoic measurements (22 out of 51 sites).

Table R1: Comparison of regression relationships between modeled $\delta^2H_p$ and measured $\delta^2$H-$H_2O$

|  | Slope | Intercept | $R^2$ | RMSE | $p$ | $n$ |
|---|---|---|---|---|---|---|
| **Inland waters** | | | | | | |
| *Growing season $\delta^2H_p$* | 1.05±0.09 | -0.3±8 | 0.82 | 22.3 | 4.81E-13 | 33 |
| *Annual $\delta^2H_p$* | 0.85±0.06 | -2.1±7 | 0.84 | 20.5 | 3.17E-14 | 33 |
| **Wetlands** | | | | | | |
| *Growing season $\delta^2H_p$* | 1.24±0.09 | 14.8±10 | 0.87 | 16.5 | 4.46E-13 | 28 |
| *Annual $\delta^2H_p$* | 1.057±0.08 | 18.6±9 | 0.88 | 15.7 | 1.20E-13 | 28 |

**3).** As suggested by all three reviewers, it is important to consider the effects of modeled $\delta^2$H-$H_2O$ on the regression between $\delta^2$H-$H_2O$ and $\delta^2$H-$CH_4$. To do this carefully we performed the regression analysis using four different estimates of $\delta^2$H-$H_2O$:
 (i) the 'best-estimate' of $\delta^2$H-$H_2O$ as described above in Planned Major Revision 2; (ii) measured $\delta^2$H-$H_2O$, only analyzing sites with this measurement; (iii) modeled annual precipitation $\delta^2$H; and (iv) modeled growing season precipitation $\delta^2$H. We think it is valuable to continue to include the regression relationships for modeled precipitation because these relationships could be used in future studies using Earth Systems Models to predict the distribution of $\delta^2$H-$CH_4$. For each of these cases we analyzed all sites, inland waters, and wetlands. We also compare these relationships with those of Waldron et al., (1999), both for the total dataset in that study, and for the dataset that only includes sites with measurements of $\delta^2$H-$H_2O$ (29 out of 51 sites). A summary of the results of this analysis are shown in Table R2 below.

A key point is that we have decided to use unweighted, as opposed to weighted, regression. Comments by Reviewer 1 made us realize that weighting by standard error was causing a few sites to strongly bias the regression results. Statistical research has found that for environmental data with poorly constrained error variance unweighted regression is frequently less biased than weighted regression (Fletcher and Dixon, 2012). Using a statistical test proposed by that study we find that unweighted regression is a good choice for our dataset. Note that in Table R2 we apply unweighted regression to the dataset of Waldron et al., (1999), in part because the specific weighting methodology was not specified in that study. This produces a small difference in the regression relationship shown in Table R2 with that reported by Waldron et al., (1999), but the two regression relationships are within error.

We then used analysis of covariance (ANCOVA) to examine differences between the regression relationships shown in the table. Based on a multiple comparison test, none of

the regression relationships shown in Table R2 are significantly different from one another. Therefore we conclude that (i) using modeled $\delta^2$H-$H_2O$ does not have a significant effect on the estimate of the relationship between $\delta^2$H-$H_2O$ vs $\delta^2$H-$CH_4$; (ii) Differences in the slope of this relationship between inland waters and wetland sites are not conclusive; and (iii) that since all of the regression relationships using the larger dataset produce a flatter slope than that of Waldron et al., (1999), the true global slope is likely to be flatter than inferred in that study, but confirmation of this flatter global slope will require more data and further analysis.

Table R2: Comparison of regression relationships between $\delta^2$H-$H_2O$ and $\delta^2$H-$CH_4$ using different estimates of $\delta^2$H-$H_2O$

| | Slope | Intercept | $R^2$ | RMSE | $p$ | $n$ |
|---|---|---|---|---|---|---|
| **Best Estimate $\delta^2$H-$H_2O$** | | | | | | |
| *All* | 0.44±0.05 | -298±5 | 0.42 | 27.4 | 1.33E-16 | 129 |
| *Wetlands* | 0.51±0.06 | -300±5 | 0.59 | 23.7 | 7.85E-12 | 55 |
| *Inland Waters* | 0.42±0.07 | -295±7 | 0.34 | 29.1 | 6.72E-08 | 74 |
| **Measured $\delta^2$H-$H_2O$** | | | | | | |
| *All* | 0.5±0.08 | -292±8 | 0.43 | 28.6 | 1.22E-08 | 61 |
| *Wetlands* | 0.53±0.11 | -298±13 | 0.44 | 26.2 | 6.58E-05 | 28 |
| *Inland Waters* | 0.42±0.1 | -291±10 | 0.37 | 28.9 | 0.000156 | 33 |
| **Modeled Annual $\delta^2$H_p** | | | | | | |
| *All* | 0.42±0.04 | -293±5 | 0.44 | 26.9 | 1.17E-17 | 129 |
| *Wetlands* | 0.57±0.06 | -287±6 | 0.65 | 21.9 | 1.01E-13 | 55 |
| *Inland Waters* | 0.37±0.06 | -293±7 | 0.36 | 28.4 | 1.25E-08 | 74 |
| **Modeled Growing Season $\delta^2$H_p** | | | | | | |
| *All* | 0.51±0.05 | -292±5 | 0.41 | 27.6 | 2.55E-16 | 129 |
| *Wetlands* | 0.71±0.07 | -285±6 | 0.63 | 22.4 | 4.05E-13 | 55 |
| *Inland Waters* | 0.44±0.07 | -294±8 | 0.33 | 29.3 | 1.03E-07 | 74 |
| **Waldron et al. (1999)** | | | | | | |
| *All data* | 0.74±0.1 | -284±6 | 0.5 | 26.3 | 6.56E-09 | 51 |
| *Measured $\delta^2$H-$H_2O$* | | | | | | |
| *only* | 0.79±0.2 | -279±10 | 0.44 | 29.6 | 9.21E-05 | 29 |

**4)** We then used the 'best-estimate' $\delta^2$H-$H_2O$ values and the regression based on those values, shown in Table R2, to calculate a revised $\delta^2$H-$CH_{4,w0}$ value for each site. These

analyses were then applied in the subsequent analyses in the paper shown in Figures 5,8 and 9. We also calculated an alternate value for sites with measured $\delta^2$H-H$_2$O, using the values and regression curve for those sites.

Notably, for the comparison between $\delta^2$H-CH$_{4,w0}$ and $\alpha_C$ we have found that there continues to be evidence for a segmented linear relationship. However, the breakpoint of this relationship is not consistent when analyzing all sites or only sites with measured $\delta^2$H-H$_2$O. Furthermore, the regression relationships for the two components of the segmented linear relationship were weaker than in our original analysis, and were not consistently statistically significant. Therefore in our revised analysis we will place less emphasis on this result, and less emphasis on the relationship between methanogenic pathway and $\delta^2$H-CH$_4$ generally, as suggested by reviewer 1. Instead we will discuss four processes or variables that have the potential to influence $\delta^2$H-CH$_4$ in freshwater environments: (i) differences in methanogenic pathway, including possible use of methanol as a substrate; (ii) methane oxidation; (iii) isotopic fractionation due to diffusion; and (iv) differential thermodynamic favorability of methanogenesis, or differential enzymatic reversibility. Ultimately, our conclusion is that $\delta^{13}$C-CH$_4$ or $\alpha_C$ cannot fully resolve the effects of these processes on $\delta^2$H-CH$_4$ on a global basis, and other approaches will be necessary to determine their relative importance, or the possible importance of other processes.

We will continue to present these results in Figure 5, given that we feel it is important to show co-variation, or lack thereof, between these isotopic measurements. Given the findings mentioned above, we will substantially revise Figure 6. Instead of distinguishing samples by inferred methanogenic pathway in this figure, we will distinguish samples by environment (wetland vs inland water), and also show available data for cow rumen and landfills. We may reverse the order of Figures 5 and 6.

**5)** Reviewer 2 made numerous comments about the representativeness of our $\delta^{13}$C-CH$_4$ dataset. We want to make clear that to our knowledge this is the largest database of freshwater methane $\delta^{13}$C-CH$_4$ currently compiled. For comparison, the second largest dataset, that of Sherwood et al., (2017), includes 48 freshwater sites (including rice paddies), of which 16 are also included in our database. However our $\delta^{13}$C-CH$_4$ database is not comprehensive (unlike the $\delta^2$H-CH$_4$ database), in that it does not include many measurements that are not paired with $\delta^2$H-CH$_4$ measurements and that have not yet been compiled into a database. It is also probably not representative, because some important environments, namely C$_4$ plant dominated ecosystems, are not well represented.

Since the primary focus of this paper is $\delta^2$H-CH$_4$, it is not within its scope to provide a comprehensive database of freshwater $\delta^{13}$C-CH$_4$, although that would be a worthwhile goal for future research. In order to make our analysis as complete as possible, in our revised manuscript we will include the 32 freshwater sites from Sherwood et al., (2017) that were not included in our original analysis in our calculations for the upscaling exercise, as well as Figures 7, 8, and 9. We will also carefully discuss the likely biases in

this dataset, especially in terms of $C_4$ plant environments, and their implications for our interpretations.

**6)** Both reviewers 2 and 3 expressed some concerns with the upscaling analysis. We acknowledge that the upscaling analysis is relatively simplistic, and that some of the interpretations were speculative. However, we still think it is valuable to use the estimates of freshwater $CH_4$ isotopic composition, differentiated by latitude, produced in this study to estimate global source $\delta^2 H$-$CH_4$ and $\delta^{13}C$-$CH_4$, and to compare that with other estimates. We wish to make clear that given uncertainties and complexity in estimating sink fractionations, particularly for $\delta^2 H$-$CH_4$, we are not attempting to estimate atmospheric values, but instead the integrated source $\delta^2 H$-$CH_4$ and $\delta^{13}C$-$CH_4$ prior to sink fractionations. We think there is value in comparing this with (i) previous bottom-up estimates of these values; and (ii) with the top-down estimates reported by Rice et al., (2016). We concur with Reviewer 2 that the discussion of alternate emissions scenarios is too speculative and simplistic, and therefore we will remove this discussion. Instead, we will focus on likely sources of error in the isotopic source signatures, and the best ways to address these errors in future studies.

We disagree with Reviewer 2 that the error estimates for isotopic source signatures are generally too optimistic, which we will discuss in more detail in our response to that reviewer.

Given comments from all three reviewers we will revise Figure 10 to only include panel C, and make the comparison with other estimates of global source $\delta^2 H$-$CH_4$ and $\delta^{13}C$-$CH_4$ clearer in this figure.

*Specific Responses to Reviewer 2:* Reviewer comments are in plain text. **Responses are in bold text.**

The paper investigates the relation between the hydrogen isotopic composition of methane emitted from freshwaters on the global scale and the isotopic composition of water and/or modeled precipitation, as well the carbon isotopic composition of methane and carbon dioxide. The authors analyze data from a large number of previous studies and apply statistical methods in order to evaluate correlations between the various signatures. The statistics are applied in a straightforward manner.

**We thank the reviewer for their assessment.**

I am missing a more detailed/critical scientific analysis of differences between the results of this study and previous studies. This has two aspects: 1) The study uses more sites than previous studies for dD, and it uses modeled fields of dD in precipitation. Which of these differences is primarily responsible for the differences to the previous literature (or is it both)?

**This is a good question and similar questions were raised by reviewers 1 and 3.**

**In response to these questions we will present a much more detailed comparison of the previous literature (Waldron et al., 1999) in comparison with our study. See Planned Major Revisions 2 and 3 for more details on this. The short answer is that regardless of which water isotope values are used, our dataset produces a flatter slope between $\delta^2$H-H$_2$Oand $\delta^2$H-CH$_4$ than that of Waldron et al., (1999). However, analysis of covariance (ANCOVA) indicates this difference in slope is not significant. We ascribe this difference to the inclusion of many more sites from high-latitude environments in this study. Our analysis is that the relatively small number of high-latitude sites analyzed by Waldron et al., (1999) were skewed toward relatively low $\delta$2H-CH4 values. We will expand on this explanation in the revised manuscript.**

2) The study uses less sites than previous studies for d13C. Are the results from these sites still adequate to be used in a global extrapolation?

**These are important points for clarification. However, we disagree that this study uses less sites than previous studies for $\delta^{13}$C-CH$_4$. See our comments on Planned Major Revision 5. We noted that the dataset was not comprehensive for $\delta^{13}$C-CH$_4$ (i.e. it does not include all published data), whereas it is comprehensive (to the best of our knowledge) for $\delta^2$H-CH$_4$. However, our $\delta^{13}$C-CH$_4$ dataset for freshwater environments is substantially larger than the largest previously published dataset that we are aware of (Sherwood et al., 2017). We include $\delta^{13}$C-CH$_4$ data for 129 freshwater sites, whereas the database of Sherwood et al. (2017) included 48. Of these, 16 are included in both databases. In order to make our $\delta^{13}$C-CH$_4$ analysis more accurate we will include all sites from Sherwood et al., ( 2017)in our analysis of $\delta^{13}$C-CH$_4$ variability. This expands the number of sites included to 161. There is a clear need for a larger effort to compile freshwater CH4 $\delta^{13}$C-CH$_4$ data into a comprehensive database, but such an effort is beyond the scope of this paper. We will highlight the importance of this for future research in our revised discussion.**

The derived global average 13C source signature derived by the authors is almost certainly too light, given what we know about the fractionation in the sinks. Furthermore, I think that the errors assumed for the bottom-up determination of the global average the source signatures are too optimistic, and the discussion on the implications for the atmospheric isotope budget in section 4.6 and too simplistic. See detailed comments below.

**We agree that it is too light, which was a key point of our analysis in the original Discussion (Line numbers 617-638). Based on the comments of reviewer 2, as well as reviewer 3, it is clear that the upscaling exercise in the current version of the paper is too limited to provide new insights into atmospheric methane budgets. However, we also feel that a more detailed upscaling exercise is beyond the scope of this paper, which as mentioned by Reviewer 1 is long and ambitious in scope. We think it is still worthwhile to perform the mixing model calculations for global methane source isotope signatures, and to compare these with previous estimates. See Planned Major Revision 6 for more details on this. Instead of the comparison to atmospheric budgets that we originally discussed, our revised discussion will focus on the likely**

**sources of error or bias in isotopic source signatures, and make recommendations to improve isotopic source signal estimates.**
**We disagree in general that our uncertainties for the isotopic source signatures are too optimistic. We will provide more details on this below.**

L37: I suggest citing Worden et al., 2017, where this point is shown particularly well.

**We thank the reviewer for bringing this to our attention, and we will cite this paper and modify the text accordingly.**

L64: Maybe you want to include here, or later in the discussion section, that there are also other lines of evidence that the hydrogen isotopic composition of CH4 (and other trace gases) depends on the isotopic composition of the precipitation, e.g., CH4 from biomass burning across climatic zones (Umezawa et al.2011), CH4 produced by UV irradiation of leaves that were grown with isotopically distinct waters (Vigano et al., 2010) or molecular H2 produced in the combustion of wood from different climatic zones (Röckmann et al., 2010).

**We appreciate this suggestion. We will reference these studies in both the introduction and the discussion**

L109: Replace the factor 1000 by 1, the delta value is defined the correct way in line 105, and no factor 1000 is necessary.

**Thanks for this reminder, we will make the suggested change.**

L136: What are the 5 categories? This is not clear, to me it sounds like 4 categories.

**The list of categories will be clarified in the manuscript with a numbered list. In fact it is six categories: 1) lakes and ponds; 2) rivers and floodplains; 3) bogs; 4) fens; 5) swamps and marshes; and 6) rice paddies.**

L159: Is the annual average dD value of precipitation really the best estimator for a source that very likely has a strong seasonality?

**This is an important question, and given this comment as well as those of reviewer 1 clearly needs more attention. See our detailed comments on Planned Major Revision 2 that discuss this at length. In short, we plan to take seasonality into account in our revised manuscript, and we find that it is important for inland water environments in particular.**

L253, Figure 1: Many of the sites are hidden behind others so I cannot see the colors. Would this improve if the figure is enlarged? It may be useful to show by color or shape for which of the sites you have measured dD-H2O and for which not.

This is challenging because many of the sites are very close to one another, and it is difficult to resolve the individual sites, while also showing the global distribution. We included the colors to give a sense of how the values vary globally, but for a more in-depth picture of geographic variability Figure 4 may be more useful.

To respond to the reviewer's comment we will provide inset maps of specific areas with many measurements, specifically Eastern North America and Alaska. We will also show the sites with water $\delta^2H$ measurements with a different shape, i.e. a triangle.

L244, Table 1: The d13C signatures for wetland have an opposite "latitudinal order" compared to what is usually assumed, i.e. they are higher at high latitudes and lower at low latitudes. The data in Table 1 for wetlands do not agree with the data presented in Figure 7. Please explain the difference. You mention that the dataset evaluated here is different from what other studies have used for d13C, so is your dataset now representative? Should this limited set of values be used in the upscaling later? The errors presented for the different source categories are too optimistic, especially for the fossil sources at the bottom of the table, but probably also for the wetland category.

The reviewer raises some key aspects of the table that are not clear.

The opposite order of the $\delta^{13}C$-$CH_4$ data in the wetlands is simply what the data indicate. The uncertainties overlap, and our analysis therefore implies that we cannot confidently infer a latitudinal difference in $\delta^{13}C$-$CH_4$ in wetlands based on currently compiled data. This is also shown in Figure 7. We note here and elsewhere in our response that there is an important absence of data from $C_4$ plantecosystems in this dataset and other databases. Including more data from such ecosystems would probably lead tropical sites to have a higher $\delta^{13}C$-$CH_4$ value. We plan to discuss this in more detail in the methods, the results, and discussion. As discussed in Planned Major Revision 5, we will include addition $\delta^{13}C$-$CH_4$ data from Sherwood et al., (2017). However, our analysis indicates this will not change the observation of no significant latitudinal differences in wetland $\delta^{13}C$-$CH_4$ values.

The differences between Table 1 and Figure 7 are a result of the Table presenting mean values, whereas Figure 7 presents median values. We presented mean values in Table 1 because it is simpler to express uncertainty for the mean, and because when thinking about atmospheric contributions we think the mean is the best estimate of the isotopic source signal. In boxplots like Figure 7 it is more common to depict the median value. However, to avoid confusion and for the sake of comparison we will also plot the mean and its standard error in Figure 7 (and also do so in Figures 8 and 9).

It is not clear to us what the reviewer means when they say the errors are too optimistic for the fossil fuel categories. The error estimates are 95% confidence intervals for the mean values for these categories based on the fossil fuel database of Sherwood et al., (2017). We consider the 95% confidence interval of the mean to be

**a well-established metric for characterizing the uncertainty in the mean value of these sources. We have categorized the fossil fuel sources slightly differently than Sherwood et al., (2017), to align with the inventory categories of Saunois et al., (2020), but our uncertainty estimates are essentially the same as, and actually somewhat larger than, those of the original study (see Table 5 in Sherwood et al., 2017). Note Sherwood et al., (2017) presents standard errors of the mean. 95% CI is derived by multiplying this value by 1.96. In addition, our uncertainties for the $\delta^{13}$C-CH$_4$ source signal for fossil fuels is very similar to those used by Worden et al., (2017). Without further details, it is unclear why the reviewer considers these error estimates to be too small or optimistic.**

**We used the same approach in our estimates of uncertainty in the wetland source signatures, and other source categories, and therefore also disagree that these estimates are too optimistic.**

L276, Fig 2 and related text: This is a key figure for the following analysis. In principle it is an interesting approach to use modeled dD values in case measurements are not available, but it is also a source of error. Although there is a generally good agreement, the slope is lower than 1 and this may contribute to the differences and thus may affect some of the further analysis.

**We agree this is a key figure and requires more in-depth analysis, which we will provide in the revised manuscript. See our Planned Major Revision 2. We agree with the reviewer that the slope being lower than 1 is concerning. In our revised analysis we find that applying annual precipitation $\delta^2$H to wetland environments, and growing season precipitation $\delta^2$H to inland water environments, results in slopes that are within error of 1.**

L284: Maybe you could state briefly whether you can reproduce the slope of Waldron et al. when you use the same dataset. Just as a baseline.

**This is a valuable suggestion. Please see our response in Planned Major Revision 3. We have included a much more careful comparison of our dataset with that of Waldron et al (1999). It is important to note that the analysis of Waldron et al. (1999) also included key assumptions that influence the regression relationship produced with that dataset. Specifically, that study included sites with measured water $\delta^2$H (57%) and sites with estimated water $\delta^2$H based on regional precipitation measurements (43%). To perform a robust comparison we re-analyze the Waldron et al dataset, which is discussed in our Planned Major Revision 3. Because the exact details of the weighted regression method used by Waldron et al., 1999 are not provided, we did not precisely reproduce their regression relationship [see Table R2]. But using unweighted regression we produced a relationship that is statistically indistinguishable.**

L292: Figure 3a: It looks like the lower slope is caused by a lot of points where you have only modeled but no measured dD data near the low dD-H2O end. And these

are mostly inland waters (Figure 3b). Can you evaluate this in more detail? Can this be caused by a bias in the modeled dDp? Probably not, but it is useful to investigate further to strengthen your argument.

**The reviewer correctly noted that the reported regression line for inland waters was not a good visual fit to the data, and this influenced the overall regression line. This was also noted by reviewer 1. We note that in the original Figure 4a the two regression lines were very similar, so this effect was not a result of bias in modeled $\delta^2$Hp, since a very similar regression was produced when only analyzing sites with measured water $\delta^2$H-H$_2$O.**

**After analyzing this more closely we realized that this is a result of the weighted regression methods we were using. Specifically, a few high-latitude sites with 1) many measurements (and therefore a low standard error) and 2) high $\delta^2$H-CH$_4$ values, were heavily weighted and had a large effect on the regression relationship. We therefore decided that a more accurate regression relationship would be produced using unweighted regression. This is supported by studies on the efficacy of unweighted regression in analyzing environmental data, which in many cases is less biased than weighted regression (Fletcher and Dixon, 2012). See more details in Planned Major Revision 3.**

**The unweighted regression provides a somewhat steeper slope for the overall dataset, as well as for inland waters. It also indicates there is not a significant difference in the regression whether measured $\delta^2$H-H$_2$O or modeled $\delta^2$H-H$_2$O, or a combination of the two (i.e. a 'best-estimate') is used. See Planned Major Revision 3 and Table R2 above.**

L308: Would you find a correlation if you took the slope of Waldron et al. for calculating CH4,W0?

**We have significantly revised this analysis, as discussed in Planned Major Revision 4 above. The result of this is that the relationship between $\alpha_C$ and $\delta^2$H-CH$_{4,w0}$ is observed regardless of how $\delta^2$H-H$_2$O is estimated. However, the specifics of this relationship are not robust to the method of estimating $\delta^2$H-H$_2$O, and therefore we will emphasize this relationship to a lesser dgree in the revised manuscript.**

**The slope of Waldron et al., (1999) is not a good fit to the overall dataset, and therefore we do not think it makes sense to apply this to calculate $\delta^2$H-CH$_{4,w0}$. However, we have performed the suggested analysis as a test. It still results in a segmented relationship with $\alpha_C$, but this relationship is less strong than the one presented in the original manuscript. This finding has contributed to our decision to focus less on the relationship between $\alpha_C$ and $\delta^2$H-CH$_{4,w0}$ as a signal of differences in methanogenic pathway and methane oxidation.**

L323, Figure 5: Does it make sense that in b) only few points are classified as oxidation

influenced and in c) many more points? Does it make sense that in c) the very lowest dD value is in the group of the oxidation influenced points? I find the "pathway trend" concept a bit confusing, this indicates a smooth transition of dD-CH4,W0 with alpha_C or d13C_CO2. Is this a real trend, or rather a consequence of two different groups of data (acetoclastic and hydrogenotrophic sites)? Wouldn't it be useful in this case to show these two groups with two different colors, separated by the potential break points, rather than the trend areas?

**The reviewer raises important questions about the predicted trends that we presented in Figure 5. Reviewer 1 also raised important questions about this, and given the overall lack of agreement on the predicted patterns we have decided that we should not present predicted trends, as there is not a strong consensus on these predictions. Instead, we will focus on the co-variance (or lack thereof) between $\delta^2$H-CH$_{4,w0}$, $\delta^{13}$C-CH$_4$, $\alpha_C$, and $\delta^{13}$C-CO$_2$, and multiple mechanisms that could influence this co-variation in freshwater ecosystems. Our ultimate conclusion is that patterns of co-variation cannot definitively resolve which mechanisms for $\delta^2$H-CH$_4$ are most important when comparing between sites.**

L350 and Figure 7b, wetlands: These numbers do not agree with the data in Table 1.

**As noted above, these are median values, whereas Table 1 presents mean values. To clarify this we will also plot mean values in Figure 7.**

L374-379: I get a bit confused by the diverging statements on significance with different tests, please try to reformulate, or add a sentence to synthesize.

**We will re-write to clarify the significance tests, focusing on the pair-wise comparison between wetlands and inland waters first (Mann-Whitney test), and then the multiple group comparison (Kruskal-Wallis test).**

L395-397: See points above: Are the uncertainties for the different categories adequate? Is there an issue with the difference between values in the text and table 1? Is the rather heavy d13C value for high latitude wetlands appropriate?

**See our response to comments on Table 1. It is unclear what difference between the table and text is being referred to- we assume this is the difference between median values (Figure 7) and mean values (Table 1). The heavy value for high latitude wetlands is the mean value of this dataset, and therefore we argue it is appropriate. In our revised manuscript we will include additional data from Sherwood et al., (2017), as discussed above, which includes 5 additional high latitude wetland sites. This makes the mean $\delta^{13}$C-CH$_4$ value 0.5‰ lower, but does not change the median value. We will include this value in a revised Monte Carlo analysis, but in essence this additional data does not change our conclusion. Based on our analysis, an assumption of low $\delta^{13}$C-CH$_4$ in high latitude wetlands is not supported by the available data, and we think this assumption requires further empirical validation.**

L431 ff: The differences to the previously published values from Waldron et al. should be discussed in some more detail. E.g., is there an influence from the modeled dDp values, or a certain sampling region? L439 ff: Same for the discussion of the environment type

**See our responses above and Planned Major Revisions 2 and 3. Our conclusion is that the difference is largely controlled by the small number of high-latitude sites in the Waldron et al (1999) dataset, and that those sites were skewed towards relatively low $\delta^2$H-CH$_4$ values. We do not observe a significant difference in the regression relationship when modeled or measured $\delta^2$H-H$_2$O values are used (see Table R2 above, as well as Figure 4a in the original manuscript).**

L465, section 4.2.1: See comments above on the representativeness of the dataset analyzed here and possible consequences. You write that the dataset is not comprehensive
or d13C, so should it be considered as representative? In this case, what have other studies potentially missed?

**See Planned Major Revision 5 above. As mentioned above, it is the largest compiled dataset available, but it is not comprehensive because there is a large amount of $\delta^{13}$C-CH$_4$ data that has not yet been compiled into a database. It is also probably not representative, with a notable lack of data from C$_4$ plant ecosystems. Given that it is the largest dataset available, we proceed with analyzing it. However, in the revised manuscript we will give more attention to the likely sources of error, and key data gaps that should be addressed.**

L483 ff: You may want to refer here to the studies I mentioned in the beginning that looked at other (non-microbial) sources.

**Thanks for this suggestion, we will mention these studies here.**

L519 ff: The authors state that they do not observe a correlation between dD and d13C of CH4. Nevertheless, the vast majority of the points in Fig 5a seem to fall in the range of the "pathway trend" (I find the term misleading, see comments above). Does this not mean that the two groups (acetate fermentation and CO2 reduction) still form distinct distributions?

**As mentioned above, there are concerns with the 'pathway trend' noted by Reviewer 2, as well as Reviewer 1 and we have decided to omit this from the revised manuscript. Our primary concern is whether $\delta^{13}$C-CH$_4$ is a strong predictor of $\delta^2$H-CH$_4$, and our analysis indicates that it is not. It is still possible that different pathways form different distributions in terms of $\delta^{13}$C-CH$_4$, but these distributions do not correspond to clear differences in $\delta^2$H-CH$_4$.**

L549: the remark on the intercepts does not add much and is rather trivial when the slope is different.

**This discussion will now be heavily modified, as discussed in Planned Major Revision 5. We will not focus on the role of methanogenic pathway as much in the revised manuscript. We will use analysis of covariance (ANCOVA) for any comparison of regression relationships in the revised manuscript.**

L555 - 561: I am also not aware of dD measurements in natural acetate, but the method from Greule et al. (2008) has been used in Vigano et al. (2010) to measure dD in methoxyl groups which were compared to produced CH4 and modeled dD in water.

**We appreciate these suggested references. We will include them in our revised discussion.**

L574 – 578: Why do you explain the variability for bogs by the pathway difference, and the high values in rivers by oxidation. Can oxidation not also cause large differences for bogs?

**This inference was based on the differences in both $\delta^{13}$C-CH$_4$ and $\delta^2$H-CH$_4$. Since bogs have higher $\delta^2$H-CH$_4$ on average, but lower $\delta^{13}$C-CH$_4$, we inferred this was related to a pathway difference. We were also influenced by previous studies (i.e. Ganesan et al., 2018) that had suggested bogs have a higher proportion of hydrogenotrophic methanogenesis. In contrast, rivers are higher in both $\delta^2$H-CH$_4$ and $\delta^{13}$C-CH$_4$, which we inferred to be a signal of oxidation. We will make this analysis clearer in the revised manuscript. In addition, as shown in Figure 5a, co-variation in $\delta^{13}$C-CH$_4$ and $\delta^2$H-CH$_4$ is not necessarily indicative of mechanisms for isotopic variability, so we will moderate our interpretations here in the revised manuscript.**

L599: Why should the oxidation signal only be apparent for dD and not for d13C (L603-604)?

**Overall dissolved CH$_4$ from inland waters is also shifted to higher $\delta^{13}$C-CH$_4$ values, although this is not a significant difference. We will note in the revised manuscript that greater oxidation would be expected to lead to higher $\delta^{13}$C-CH$_4$ values, and the absence of a strong signal in $\delta^{13}$C-CH$_4$ may be inconsistent with our hypothesis. We will also discuss other possible mechanisms for the observation of high $\delta^2$H-CH$_4$ in dissolved inland water samples, including different water sources and effects of diffusion on isotopic fractionation.**

L606: I do not understand how you can conclude that ": : :that the relative balance of diffusive vs. ebullition gas fluxes should not have a large effect on the isotopic composition of freshwater CH4 emissions.". The chance for oxidative effects is much larger for a slow process like diffusion compared to the fast process of ebullition.

**This statement is simply a reflection of the available data, as shown in Figure 9a and b, which do not show a clear difference between these two gas sample types in their**

**isotopic composition. We note below this (lines 607-610) several caveats that moderate this conclusion, and that the question deserves more study. We will add the likely greater effect of oxidation on diffusive fluxes as an additional area that requires further empirical validation.**

L611: The analysis in this section has much less scientific rigor than the previous sections and presents some sensitivity calculations involving highly improbable assumptions, see following points.

**We acknowledge that the sensitivity calculations and scenarios are somewhat simplistic and loosely defined. As discussed above, we think the solution to this is to scale back this section to focus on the results of a global source mixing model calculation, to compare that with previous estimates of global source signals, and to discuss key data gaps that are likely leading to biases in this estimate (See planned major revision 6). Therefore the revised manuscripts would not include the sensitivity calculations, which would be left for future work.**

L619 ff: See comments above on the depleted d13C source signature. Here you argue that three factors may explain this difference. I am quite convinced that the first one (errors in the sink fractionation factors) cannot explain the large difference. The two published studies for the fractionation in the CH4 + OH reaction (Cantrell et al, 1990, Saueressig et al, 2001) are 5.4 and 3.9 per mill, respectively. A contribution from Cl may increase this a bit, but not enough to support a global average source signature of -56.4 per mill. So I think that the reason should come from the other two processes mentioned. Given the discrepancy to previous studies I wonder whether it is not mainly the choice of signatures in this study. In line 625 you already show that changing one parameter leads to a change of the global average source signature of 1.3 per mill, which is almost the entire uncertainty range reported.

**We acknowledge the point the reviewer is making. As discussed above, we will revise this section to limit our interpretation to comparison with previous estimates and possible biases in isotopic source signals, and not focus on sink fractionations, which are not a focus of this study. We will mention errors in flux inventories, which we think is probably partly responsible for the discrepancy.**

L628: Rather arbitrarily changing big sources by a factor of 2 is a huge adjustment of the atmospheric CH4 budget. This investigation on the effect on the atmospheric isotopic composition is too simplistic.

**We understand this critique, and as discussed above we will avoid performing this analysis in the revised paper. This analysis was based on the work of Schwietzke et al., (2016), who make a similar, but more precise adjustment. We will mention the possibility of higher fossil fuel emissions than in inventories, as discussed by Schwietzke et al., (2016), but leave a detailed analysis resolving this with $\delta^2$H-CH$_4$ measurements to future studies.**

L634 ff: Same comment for the bb source, this should be discussed in a more detailed way. Worden et al. (2017) illustrate the strong influence of the bb source.

**As discussed above, we feel it is best to omit the discussion of specific different emissions scenarios from the discussion. We will briefly discuss the results of the Worden et al., (2017) study, and mention biomass burning emissions as an influential variable for isotopic source signatures that merits further study, particularly in terms of $\delta^2$H.**

L660f: The statement "This flatter slope may be the result of the inclusion of a greater proportion of inland water sites in our dataset." requires more underlying analysis. I think that the "may be" can be replaced by "is likely", but this should be investigated. See also other points above.

**Based on the comments of all three reviewers we will thoroughly revise our comparison of our results with that of Waldron et al (1999). Therefore this part of the conclusions will be changed to reflect this revised comparison, and likely causes of the different slope. Our revised analysis implies that differences between inland waters and wetlands is probably not primarily responsible for this difference (see Table R2), and that a greater amount of data from high-latitude environments is more important.**

L662: If possible make more concrete after reevaluation of the impact of modeled data.

**We will also revise this statement after a more thorough analysis of the differences in the regression relationship for modeled and measured $\delta^2$H-H$_2$O. Our revised analysis shows that using modeled $\delta^2$H-H$_2$O provides a good estimate of the relationship between $\delta^2$H-H$_2$O and $\delta^2$H-CH$_4$, and supports the use of isotope-enabled Earth Systems Models to predict $\delta^2$H-CH$_4$.**

L686: Here the second argument of the three presented before (see comment on L619) has disappeared, but as argued above it may be the most important one and particularly the sink argument does likely not explain (at least exclusively) the difference.

**As discussed above, we will substantially revise and scale back the upscaling estimates. Therefore these conclusions will be thoroughly changed. We will focus primarily on the uncertainties in the source signature estimates.**

**References Cited:**

Fletcher, D., & Dixon, P. M. (2012). Modelling data from different sites, times or studies: weighted vs. unweighted regression. *Methods in Ecology and Evolution*, *3*(1), 168-176.

Rice, A. L., Butenhoff, C. L., Teama, D. G., Röger, F. H., Khalil, M. A. K., and Rasmussen, R. A.: Atmospheric methane isotopic record favors fossil sources flat in

1980s and 1990s with recent increase, Proceedings of the National Academy of Sciences, 113, 10791-10796, 2016.

Schwietzke, S., Sherwood, O. A., Bruhwiler, L. M., Miller, J. B., Etiope, G., Dlugokencky, E. J., Michel, S. E., Arling, V. A., Vaughn, B. H., and White, J. W.: Upward revision of global fossil fuel methane emissions based on isotope database, Nature, 538, 88-91, 2016.

Sherwood, O. A., Schwietzke, S., Arling, V. A., and Etiope, G.: Global inventory of gas geochemistry data from fossil fuel, microbial and burning sources, version 2017, Earth System Science Data, 9, 2017.

Waldron, S., Lansdown, J., Scott, E., Fallick, A., and Hall, A.: The global influence of the hydrogen iostope composition of water on that of bacteriogenic methane from shallow freshwater environments, Geochim Cosmochim Ac, 63, 2237-2245, 1999a.

---

## Author Comment (AC3) · 15 Feb 2021

Response to reviewer 3:

All of the reviewers provided excellent suggestions and feedback on the paper, and we think that by addressing their concerns the paper will be greatly improved. Many of their comments were complementary. Therefore we will first summarize the major revisions we plan to make to the paper before responding to each reviewer in detail:

**Planned Major Revisions**
**1)** We have revised the $\delta^2$H-CH$_4$ dataset in response to comments from Reviewer 1 and Reviewer 3. (i) For peatland sites with depth stratified sampling we have decided to only include samples from the upper 50 cm, as suggested by reviewer 3, since this is the depth range that is most likely to emit CH$_4$ to the atmosphere. This affects a total of 8 sites. (ii) Reviewer 1 noted that an outlier sample from the Amazon River with very high $\delta^2$H-CH$_4$ and $\delta^{13}$C-CH$_4$ could be derived from thermogenic methane. We agree that this outlier is suspect, and therefore have decided not to include it. (iii) We also noted that one site (Mirror Lake, Florida, USA) was analyzed in two separate studies, and therefore was included twice in the dataset. We have combined the data from the two studies into one site entry.

**2)** As suggested by all three reviewers, we have performed much more rigorous analysis of the relationship between measured and modeled $\delta^2$H-H$_2$O values. Specifically we have done the following: (i) In addition to annual precipitation $\delta^2$H values, we now also analyze growing season precipitation $\delta^2$H, which is defined as the amount-weighted mean $\delta^2$H of months with mean temperature greater than 0º C. This provides an opportunity to assess whether seasonal variation in precipitation in the mid to high-latitudes is important in controlling the environmental $\delta^2$H-H$_2$O value; (ii) separately analyzing inland water and wetland environments, since these are different hydrological environments and the controls on $\delta^2$H-H$_2$O are potentially different.

This analysis led to the following key results (see Table R1 below for summary of results): A) growing season modeled precipitation $\delta^2$H is a better predictor of inland water $\delta^2$H-H$_2$O than annual precipitation $\delta^2$H, in that the regression curve is indistinguishable from the 1:1 line. B) annual modeled precipitation $\delta^2$H is a better predictor of wetland $\delta^2$H-H$_2$O, in that the slope of the regression is indistinguishable from 1, and the R$^2$ value is higher. However, the regression line is offset from the 1:1 line by 18.6±9‰. We interpret this as an indicator of likely widespread evaporative effects on $\delta^2$H-H$_2$O in wetland environments.

We use these results to then develop a 'best estimate' for comparing $\delta^2$H-H$_2$O with d2H-CH4. (i) For sites with measured $\delta^2$H-H$_2$O values we use the measured value. (ii) For inland water sites without measured $\delta^2$H-H$_2$O we use modeled growing season precipitation, since as discussed above the regression of this against measured $\delta^2$H-H$_2$O is indistinguishable from the 1:1 line. (iii) For wetland sites without measured $\delta^2$H-H$_2$O we estimate the $\delta^2$H-H$_2$O using the regression relationship with annual precipitation $\delta^2$H-H$_2$O shown in Table R2. We feel this approach combining measured and modeled data is

most consistent with that of Waldron et al., 1999, who we note also analyzed a combination of sites with measured $\delta^2$H-$H_2O$ (29 out of 51 sites) and estimated $\delta^2$H-$H_2O$ based on precipitation isotopoic measurements (22 out of 51 sites).

Table R1: Comparison of regression relationships between modeled $\delta^2H_p$ and measured $\delta^2$H-$H_2O$

|  | Slope | Intercept | $R^2$ | RMSE | $p$ | $n$ |
|---|---|---|---|---|---|---|
| **Inland waters** | | | | | | |
| *Growing season $\delta^2H_p$* | 1.05±0.09 | -0.3±8 | 0.82 | 22.3 | 4.81E-13 | 33 |
| *Annual $\delta^2H_p$* | 0.85±0.06 | -2.1±7 | 0.84 | 20.5 | 3.17E-14 | 33 |
| **Wetlands** | | | | | | |
| *Growing season $\delta^2H_p$* | 1.24±0.09 | 14.8±10 | 0.87 | 16.5 | 4.46E-13 | 28 |
| *Annual $\delta^2H_p$* | 1.057±0.08 | 18.6±9 | 0.88 | 15.7 | 1.20E-13 | 28 |

**3).** As suggested by all three reviewers, it is important to consider the effects of modeled $\delta^2$H-$H_2O$ on the regression between $\delta^2$H-$H_2O$ and $\delta^2$H-$CH_4$. To do this carefully we performed the regression analysis using four different estimates of $\delta^2$H-$H_2O$:
 (i) the 'best-estimate' of $\delta^2$H-$H_2O$ as described above in Planned Major Revision 2; (ii) measured $\delta^2$H-$H_2O$, only analyzing sites with this measurement; (iii) modeled annual precipitation $\delta^2$H; and (iv) modeled growing season precipitation $\delta^2$H. We think it is valuable to continue to include the regression relationships for modeled precipitation because these relationships could be used in future studies using Earth Systems Models to predict the distribution of $\delta^2$H-$CH_4$. For each of these cases we analyzed all sites, inland waters, and wetlands. We also compare these relationships with those of Waldron et al., (1999), both for the total dataset in that study, and for the dataset that only includes sites with measurements of $\delta^2$H-$H_2O$ (29 out of 51 sites). A summary of the results of this analysis are shown in Table R2 below.

A key point is that we have decided to use unweighted, as opposed to weighted, regression. Comments by Reviewer 1 made us realize that weighting by standard error was causing a few sites to strongly bias the regression results. Statistical research has found that for environmental data with poorly constrained error variance unweighted regression is frequently less biased than weighted regression (Fletcher and Dixon, 2012). Using a statistical test proposed by that study we find that unweighted regression is a good choice for our dataset. Note that in Table R2 we apply unweighted regression to the dataset of Waldron et al., (1999), in part because the specific weighting methodology was not specified in that study. This produces a small difference in the regression relationship shown in Table R2 with that reported by Waldron et al., (1999), but the two regression relationships are within error.

We then used analysis of covariance (ANCOVA) to examine differences between the regression relationships shown in the table. Based on a multiple comparison test, none of

the regression relationships shown in Table R2 are significantly different from one another. Therefore we conclude that (i) using modeled $\delta^2$H-$H_2O$ does not have a significant effect on the estimate of the relationship between $\delta^2$H-$H_2O$ vs $\delta^2$H-$CH_4$; (ii) Differences in the slope of this relationship between inland waters and wetland sites are not conclusive; and (iii) that since all of the regression relationships using the larger dataset produce a flatter slope than that of Waldron et al., (1999), the true global slope is likely to be flatter than inferred in that study, but confirmation of this flatter global slope will require more data and further analysis.

Table R2: Comparison of regression relationships between $\delta^2$H-$H_2O$ and $\delta^2$H-$CH_4$ using different estimates of $\delta^2$H-$H_2O$

| | Slope | Intercept | $R^2$ | RMSE | $p$ | $n$ |
|---|---|---|---|---|---|---|
| **Best Estimate $\delta^2$H-$H_2O$** | | | | | | |
| *All* | 0.44±0.05 | -298±5 | 0.42 | 27.4 | 1.33E-16 | 129 |
| *Wetlands* | 0.51±0.06 | -300±5 | 0.59 | 23.7 | 7.85E-12 | 55 |
| *Inland Waters* | 0.42±0.07 | -295±7 | 0.34 | 29.1 | 6.72E-08 | 74 |
| **Measured $\delta^2$H-$H_2O$** | | | | | | |
| *All* | 0.5±0.08 | -292±8 | 0.43 | 28.6 | 1.22E-08 | 61 |
| *Wetlands* | 0.53±0.11 | -298±13 | 0.44 | 26.2 | 6.58E-05 | 28 |
| *Inland Waters* | 0.42±0.1 | -291±10 | 0.37 | 28.9 | 0.000156 | 33 |
| **Modeled Annual $\delta^2H_p$** | | | | | | |
| *All* | 0.42±0.04 | -293±5 | 0.44 | 26.9 | 1.17E-17 | 129 |
| *Wetlands* | 0.57±0.06 | -287±6 | 0.65 | 21.9 | 1.01E-13 | 55 |
| *Inland Waters* | 0.37±0.06 | -293±7 | 0.36 | 28.4 | 1.25E-08 | 74 |
| **Modeled Growing Season $\delta^2H_p$** | | | | | | |
| *All* | 0.51±0.05 | -292±5 | 0.41 | 27.6 | 2.55E-16 | 129 |
| *Wetlands* | 0.71±0.07 | -285±6 | 0.63 | 22.4 | 4.05E-13 | 55 |
| *Inland Waters* | 0.44±0.07 | -294±8 | 0.33 | 29.3 | 1.03E-07 | 74 |
| **Waldron et al. (1999)** | | | | | | |
| *All data* | 0.74±0.1 | -284±6 | 0.5 | 26.3 | 6.56E-09 | 51 |
| *Measured $\delta^2$H-$H_2O$* | | | | | | |
| *only* | 0.79±0.2 | -279±10 | 0.44 | 29.6 | 9.21E-05 | 29 |

**4)** We then used the 'best-estimate' $\delta^2$H-$H_2O$ values and the regression based on those values, shown in Table R2, to calculate a revised $\delta^2$H-$CH_{4,w0}$ value for each site. These

analyses were then applied in the subsequent analyses in the paper shown in Figures 5,8 and 9. We also calculated an alternate value for sites with measured $\delta^2$H-H$_2$O, using the values and regression curve for those sites.

Notably, for the comparison between $\delta^2$H-CH$_{4,w0}$ and $\alpha_C$ we have found that there continues to be evidence for a segmented linear relationship. However, the breakpoint of this relationship is not consistent when analyzing all sites or only sites with measured $\delta^2$H-H$_2$O. Furthermore, the regression relationships for the two components of the segmented linear relationship were weaker than in our original analysis, and were not consistently statistically significant. Therefore in our revised analysis we will place less emphasis on this result, and less emphasis on the relationship between methanogenic pathway and $\delta^2$H-CH$_4$ generally, as suggested by reviewer 1. Instead we will discuss four processes or variables that have the potential to influence $\delta^2$H-CH$_4$ in freshwater environments: (i) differences in methanogenic pathway, including possible use of methanol as a substrate; (ii) methane oxidation; (iii) isotopic fractionation due to diffusion; and (iv) differential thermodynamic favorability of methanogenesis, or differential enzymatic reversibility. Ultimately, our conclusion is that $\delta^{13}$C-CH$_4$ or $\alpha_C$ cannot fully resolve the effects of these processes on $\delta^2$H-CH$_4$ on a global basis, and other approaches will be necessary to determine their relative importance, or the possible importance of other processes.

We will continue to present these results in Figure 5, given that we feel it is important to show co-variation, or lack thereof, between these isotopic measurements. Given the findings mentioned above, we will substantially revise Figure 6. Instead of distinguishing samples by inferred methanogenic pathway in this figure, we will distinguish samples by environment (wetland vs inland water), and also show available data for cow rumen and landfills. We may reverse the order of Figures 5 and 6.

**5)** Reviewer 2 made numerous comments about the representativeness of our $\delta^{13}$C-CH$_4$ dataset. We want to make clear that to our knowledge this is the largest database of freshwater methane $\delta^{13}$C-CH$_4$ currently compiled. For comparison, the second largest dataset, that of Sherwood et al., (2017), includes 48 freshwater sites (including rice paddies), of which 16 are also included in our database. However our $\delta^{13}$C-CH$_4$ database is not comprehensive (unlike the $\delta^2$H-CH$_4$ database), in that it does not include many measurements that are not paired with $\delta^2$H-CH$_4$ measurements and that have not yet been compiled into a database. It is also probably not representative, because some important environments, namely C$_4$ plant dominated ecosystems, are not well represented.

Since the primary focus of this paper is $\delta^2$H-CH$_4$, it is not within its scope to provide a comprehensive database of freshwater $\delta^{13}$C-CH$_4$, although that would be a worthwhile goal for future research. In order to make our analysis as complete as possible, in our revised manuscript we will include the 32 freshwater sites from Sherwood et al., (2017) that were not included in our original analysis in our calculations for the upscaling exercise, as well as Figures 7, 8, and 9. We will also carefully discuss the likely biases in

this dataset, especially in terms of $C_4$ plant environments, and their implications for our interpretations.

**6)** Both reviewers 2 and 3 expressed some concerns with the upscaling analysis. We acknowledge that the upscaling analysis is relatively simplistic, and that some of the interpretations were speculative. However, we still think it is valuable to use the estimates of freshwater $CH_4$ isotopic composition, differentiated by latitude, produced in this study to estimate global source $\delta^2H\text{-}CH_4$ and $\delta^{13}C\text{-}CH_4$, and to compare that with other estimates. We wish to make clear that given uncertainties and complexity in estimating sink fractionations, particularly for $\delta^2H\text{-}CH_4$, we are not attempting to estimate atmospheric values, but instead the integrated source $\delta^2H\text{-}CH_4$ and $\delta^{13}C\text{-}CH_4$ prior to sink fractionations. We think there is value in comparing this with (i) previous bottom-up estimates of these values; and (ii) with the top-down estimates reported by Rice et al., (2016). We concur with Reviewer 2 that the discussion of alternate emissions scenarios is too speculative and simplistic, and therefore we will remove this discussion. Instead, we will focus on likely sources of error in the isotopic source signatures, and the best ways to address these errors in future studies.

We disagree with Reviewer 2 that the error estimates for isotopic source signatures are generally too optimistic, which we will discuss in more detail in our response to that reviewer.

Given comments from all three reviewers we will revise Figure 10 to only include panel C, and make the comparison with other estimates of global source $\delta^2H\text{-}CH_4$ and $\delta^{13}C\text{-}CH_4$ clearer in this figure.

*Specific Responses to Reviewer 3:* Reviewer comments are in plain text. **Responses are in bold text.**

During the past two decades, there has been limited progress in advancing understanding of controls on **d2**H(**CH4**) values in freshwater environments and improving estimates of **d2**H values of $CH_4$ emissions. This study: (i) updates and attempts to refine the relationship between **d2**H(**H2**O) and **d2**H(**CH4**) first reported by Waldron et al. (1999b), (ii) evaluates the extent to which factors other than **d2**H(**H2**O) may influence **d2**H(**CH4**) values in freshwater environments, (iii) uses the refined relationships to estimate new **d2**H values for $CH_4$ emissions from freshwatersources, and (iv) weights $CH_4$ fluxes reported by Saunois et al. (2020) with a mixture of old and new **d2**H and **d13**C values to estimate global **d2**H and **d13**C values for atmospheric $CH_4$. In my opinion, the study offers new insights that are worthy of publication pending revision.

**We thank Dr. Hornibrook for his detailed review, and we are heartened to hear his opinion that the study is worthy of publication pending revision.**

Site level mean values - The study has produced a thorough compilation of stable isotope data related to $CH_4$ from freshwater environments. The availability of **d2**H(**CH4**) values presumably was the key criterion for inclusion in the data base. The supplemental file

contains a summary of the data, showing the number of samples from each site and site-level mean isotopic values as described in section 2.3.1. While I appreciate the motivation to avoid introducing bias towards sites that have larger datasets, this approach does limit the extent to which the study can comment meaningfully on differences between environments. **d₂H(CH₄)**, **d₁₃C(CH₄)** and **d₁₃C(CO₂)** values all exhibit significant ranges and trends with depth in the subsurface of wetlands. That information is lost when profiles of **d**-values are averaged. In peatlands where CH₄ production pathways change with depth or CH₄ oxidation occurs, **d**-values determined from an average of shallow and deep layers has little meaning in the context of production pathways or evidence for CH₄ alteration. The pooled **d**-values also do not take into account differences in the amount of CH₄ or CO₂ at different depths. Moreover, **d**-values from deep peat typically will have little bearing on the stable isotope composition of CH₄ emitted from a wetland. Venting of accumulated gas bubbles from deep peat can occur (e.g., Glaser et al, 2004) but there is little evidence that such events are common. The bulk of CH₄ production occurs at shallow depths (from water table level to ~50 cm depth) where the supply of labile substrates from plant roots is greatest and temperature is highest during summer. The residence time of CH₄ at those depths is shortest (e.g., Lombardi et al., 1997; Bowes and Hornibrook, 2006) and most of the CH₄ produced seasonally is either consumed or evaded to the atmosphere. If subsurface data must be averaged to avoid bias, then I suggest using a consistent depth range (e.g., 0 to 50 cm) to (i) generate mean **d**-values that are more likely to represent **d**-values of CH₄ emissions, and (ii) enable analysis of **a**C and **a**H values that are more likely to be related to one methanogenic pathway or exhibit the influence of methane oxidation rather than a blend of pathways and processes across a range of depths. An important advance in this study was the attempt to discern the relative impact of factors other than **d₂H(H₂O)** on **d₂H(CH₄)** values. Use of site level means for **d**-values raises concern about the validity of the **a**C and **a**H values calculated to assess breakpoints in CH₄ production pathways and oxidation.

**The reviewer raises an important point about $\delta^2$H-CH₄ variability with depth in peatlands, and potential biases that are introduced by averaging values across depth profiles. The primary goal of our study is to investigate spatial variability between sites, and therefore we think it is important to provide a single value for each site. In addition, one of the key goals is to characterize the $\delta^2$H values of CH₄ emitted to the atmosphere. Therefore, we agree with the reviewer's suggestion to use a consistent depth range (0-50 cm) when averaging data from peatlands with depth-resolved sampling. See Planned Major Revision 1 above. This change affects 8 sites, from 5 publications (Hornibrook et al., 1997; Waldron et al., 1999; Chasar et al., 2000; Chanton et al., 2006; Alstad and Whiticar, 2011). Other studies included in our dataset sampled peatlands at shallow depths. To our knowledge all studies in other wetland environments also sampled shallow (< 50 cm) soils.**

'Bottom-up' mixing model - I appreciate that considerable effort was invested in attempting to upscale d₂H(CH₄) and d₁₃C(CH₄) values; however, it is questionable whether that portion of the manuscript has potential to advance discourse on global isotope-weighted CH₄ budgets. A more valuable outcome of this work would have been

the one identified by that authors in lines 441- 443: "A logical next step in predicting global freshwater $\delta_2$H-CH$_4$ source signatures would be to combine high-resolution mapping of wetlands and inland waters, maps of the global distribution of $\delta_2$H$_p$, and regression relationships between $\delta_2$H-CH$_4$ vs. $\delta_2$H$_p$." In my view, production of a global gridded map of **d**$_2$H(CH$_4$) values for freshwater environments would have a more suitable application of the outcomes from the data analysis. It would provide a useful counterpart to the **d**$_{13}$C(CH$_4$) global map for wetlands published by Ganesan et al. (2018). I realize at this stage in the process that would take the second half of the manuscript in a very different direction. As things stand, the weighted atmospheric **d**$_2$H(CH$_4$) and **d**$_{13}$C(CH$_4$) values that were calculated are difficult to reconcile with atmospheric data and KIEs associated with sinks for atmospheric CH$_4$. It's possible that the values may be offering new insights but it seems more likely that there are issues with attribution of **d**$_2$H and **d**$_{13}$C values to CH$_4$ sources.

**We recognize the reviewer's concerns that the upscaling results presented in this paper may not advance discourse on isotope weighted CH$_4$ budgets. Reviewer 2 made similar comments. We note that we specifically did not try to resolve these results with atmospheric data, given the uncertainties related to sink KIEs, but instead compared them with past estimates of global source isotopic values that are based on atmospheric data and previously published models of sink fractionations (Rice et al., 2016, Figure 10C in the original manuscript). We will make this clearer in the revised manuscript, and highlight the associated uncertainties to a greater degree.**

**We have decided to substantially revise this part of the manuscript. See Planned Major Revision 6. We think it is still worthwhile to present estimates of global methane source $\delta^2$H and $\delta^{13}$C that include the results of our data analysis, and to compare this with previous bottom-up estimates of global isotopic source signatures, as well as the top-down estimates from Rice et al., (2016) mentioned above. We will then focus on an assessment of the largest areas of uncertainty in the isotopic source signatures, and not dwell on uncertainties in sink fractionations, since these are not the focus of this paper. We will mention possible errors in flux inventories, but will devote less focus to this than possible biases in isotopic signatures. In particular we will direct more focus on the problem of a lack of data from C$_4$ plant dominated ecosystems in synthetic datasets, which may compromise data-based estimates of freshwater $\delta^{13}$C-CH$_4$ signatures.**

**Creating a gridded map of freshwater $\delta^2$H-CH$_4$ values entails a substantial amount of work and additional expertise in GIS methods, and this is beyond the scope of the revisions for this paper, which as reviewer 1 noted is already quite extensive and ambitious. However, this is the goal of collaborative research that is currently in development. This research in development will also look more closely at comparisons with atmospheric data.**

Citations within the text do not appear to be listed consistently either alphabetically or chronologically.

**We thank the reviewer for noting this. It is probably a problem with the EndNote citation style, and we will check this carefully in the revised version.**

Line 38: 'clearly' = 'unequivocally' ?

**We agree this would make this sentence clearer and will make the change.**

Lines 51-52: 'recent technological developments'. An additional sentence or two about laser based methods would be helpful for a broader readership.

**That is a good idea and we will add a sentence or two about new laser based methodologies.**

Lines 53-57: Rigby et al. (2012) also demonstrated the utility of a multi-isotope approach for global methane cycle characterization.

**We thank the reviewer for bringing this paper to our attention. We will revise this paragraph to include this conclusions of that study.**

Lines 87-88 (and elsewhere): 'data is' should be 'data are'

**We will adjust this here and throughout the manuscript.**

Line 105: A citation for Coplen (2011) could be added for the definition of delta that (correctly)
does not include a 'x 1000' factor.

**We will add the suggested citation**

L129: The citation for John Lansdown's thesis should be:
Lansdown J. M. (1992) The carbon and hydrogen stable isotope composition of methane released from natural wetlands and ruminants. Ph.D. dissertation, Univ. of Washington.
(The citation can be confirmed at: https://dggs.alaska.gov/pubs/id/28259)

**We thank the reviewer for this correction, and will edit the references and citations**

L156 – Is the annual estimate of $\delta^2H_p$ weighted by the relative amounts of precipitation during different seasons?

**Yes, the annual estimates from the model are amount-weighted values (See Bowen and Wilkinson 2002 for specifics on the methodology).**

L200: $d^2H$ (superscript missing)

**We will fix this error**

L258-L259 "55 sites are classified as wetlands, including 16 bogs, 14 swamps and marshes, 12fens, and 8 rice paddies."
>> Are the classifications for bogs and fens based upon pore water chemistry and vegetation surveys? The word 'bog' sometimes is used in site names that are other wetland types, in particular, fens.

**This is a good point. We have done our best to be careful about the wetland classifications, but we have primarily relied on the classification of the original study. Of the 16 bog sites, 14 came from studies that specifically differentiate between bogs and fens (Chanton et al., 2006; Lansdown, 1992 (thesis), Alstad and Whiticar, 2011, Waldron et al., 1999; Chasar et al., 2000), or provide detailed information on vegetation and/or soil pH (Lansdown et al., 1992; Hornibrook et al., 1996). One other paper (Whiticar et al., 1986) provides data from Volo Bog, Illinois, which based on other references is an ombrotrophic, sphagnum-dominated bog. The only remaining bog site is a West Virginia Bog, from Wahlen, (1994), which did not provide enough information to verify this classification. Given that this original classification is all we have to go on we continue to use it for this sample.**

Table 1: Origins of some data are unclear. When indicated as 'no specific measurement indatabase', what does it mean to say 'we used the isotopic values and uncertainties for X'? Which literature source? Also, only C3 $d_{13}$C values appear to be used for biomass burning. Grassland and savanna wildfires presumably generate $CH_4$ that has more positive $d_{13}$C values from burning of C4 grasses.

**Thank you for raising these ambiguities in Table 1. Reviewer 2 has brought up similar concerns and we will make this table and the underlying data clearer in the revised manuscript. The database being referred to is the Gas Geochemistry Isotope Database (Sherwood et al., 2017), as referenced in section 2.4. This was the source for all isotopic estimates, with the exception of biogenic marine methane, which we derived from Whiticar et al., (1999).**

**The Global Gas Geochemistry Database was our basis for the biomass burning $\delta^{13}C$-$CH_4$ values. Out of 24 biomass burning $\delta^{13}C$-$CH_4$ values, only 2 are ostensibly from $C_4$ plants and have a higher $\delta^{13}C$-$CH_4$ value. These were included in our analysis. In keeping with our data centered approach, and the lack of definitive estimates of the relative proportion of biomass burning $CH_4$ emissions from $C_4$ plants, we did not attempt to weight these values in our analysis. However, in the revised manuscript we will mention this as a possible source of error in our discussion, and highlight the importance of more data on methane from $C_4$ plant ecosystems, both for biomass burning and microbial emissions.**

L266-L271 The comparison of modelled $\delta_2H_p$ values and measured $d_2H(H_2O)$ values for 62 sites is important for validating the approach on which estimating $d_2H(CH_4)$ relies. The text is not clear though with respect to causes in deviation from a 1:1 relationship. Presumably "$d_2H$-$H_2O$ is generally higher" means $_2H$-enrichment is evident in the measured data. Is the statement about 'overall smaller water volumes' meant to infer

evaporative enrichment of $_2$H?

**This comment, as well as those of reviewers 1 and 2, make it clear that we need to more thoroughly evaluate the relationship between emipirical $\delta^2$H-H$_2$O and modeled $\delta^2$H$_p$ values in this paper. We have done so, including considering wetlands and inland waters separately, and examining whether modeled annual precipitation or growing season precipitation is a better predictor of the empirical $\delta^2$H-H$_2$O values. See our Planned Major Revision 2.**

**The comment about higher $\delta^2$H-H$_2$O in mid-latitude sites was based on our expectation that in wetlands the residence time of water is lower, and therefore there is more seasonal variability in $\delta^2$H-H$_2$O. Since almost all samples were collected in summer, when $\delta^2$H$_p$ is higher than average in higher-latitude settings, this would lead these values to be higher than annual precipitation. However, our more detailed analysis does not support this contention, and instead implies that evaporation is likely leading to water $\delta^2$H-H$_2$O values that are higher than precipitation in wetlands specifically. See Planned Major Revision 2 and Table R1.**

L282-L283 "Both relationships result in a large amount of unexplained residual variability, implying the importance of other variables in controlling $\delta_2$H-CH$_4$."

I'll expand here on the point raised in my general comments. The extent to which residual variability exists is likely underestimated because of the use of site-level means. There are relatively few data sets globally that contain subsurface profiles of both **d$_2$H(H$_2$O)** and **d$_2$H(CH$_4$)** values. Four of those data sets are shown in the enclosed figure which was published in Hornibrook and Aravena (2010): Turnagain Bog (open triangles; Chanton et al. 2006), Sifton Bog (open diamonds; Hornibrook et al. 1997), Point Pelee Marsh (open circles; Hornibrook et al. 1997) and Ellergower Moss (open squares; Waldron et al. 1999a). The arrows indicate the direction of increasing depth in peat for Turnagain Bog, Sifton Bog, Point Pelee Marsh and Ellergower Marsh. The figure also includes **d$_2$H** values of coexisting CH$_4$ and H$_2$O values from Alaskan peatlands along a N-S transect (filled triangles; Chanton et al. 2006) and regression equations (Table 6.2 from Hornibrook and Aravena, 2010 also enclosed) from a number of studies including Waldron et al. (1999b; line 5) and Whiticar et al. (1986; lines 1 and 2).
The approach of using site-level means reduces each of those depth trends to a single point in **d$_2$H(H$_2$O)** vs. **d$_2$H(CH$_4$)** space. The **d$_2$H** values of CH$_4$ emitted to the atmosphere are likely to be similar to the most $_2$H-depleted values in each trend which corresponds to CH$_4$ in shallow peat near the water-air interface and within the root zone where CH$_4$ may be transported to the atmosphere via plant aerenchyma. Averaging **d$_2$H(CH$_4$)** values from all depths (2 m for Sifton Bog and Pelee Marsh; 6 m for Ellergower moss) yields a mean that is substantially more $_2$H-rich. Again, I appreciate the goal of not biasing the analysis to these larger data sets but a single mean for each site does not reflect the considerable residual variability that exists with depth as **d$_2$H(CH$_4$)** values shift away from the global **d$_2$H(H$_2$O)** vs. **d$_2$H(CH$_4$)** regression line. Moreover, the **d$_{13}$C(CH$_4$)** and **d$_{13}$C(CO$_2$)** depth trends from these sites yield systematic shifts in **a$_C$** values that are lost when the **d$_{13}$C** values similarly are reduced to unitary site-level means.

**We thank the reviewer for the detailed explanation of their argument on this issue. As we discussed above, the primary goals of this paper are to explore inter-site geographic variability in the $\delta^2$H-CH$_4$ emitted to the atmosphere. Therefore, while intra-site variability is of great interest, we do not want to add an additional layer of complexity to this paper by considering this. We feel the reviewer's earlier suggestion of limiting samples from the upper 50 cm of peat is a good solution to this issue, and we have followed this suggestion in our revised analysis. See Planned Major Revision 1.**

L308-L309 "We do not find evidence for a piece-wise linear relationship between $\delta_{13}$C-CH$_4$ and $\delta_2$H-CH$_4$,$_{w0}$ (Fig. 5a), nor did we find a significant simple linear correlation between these variables."
>> It may be worth exploring whether any relationships exist in the full data sets rather than site level means.

**This is an interesting suggestion, though we have concerns that such an analysis might be biased by over-representing sites that have a large number of measurements. It will also require a large amount of additional data analysis, since the $\delta^2$H-CH$_{4,w0}$ data are not currently disaggregated on a per sample basis. Given that the focus of this work is on variability between sites, we will leave this analysis for future work focused on intra-site isotopic variation.**

L441-L443: "A logical next step in predicting global freshwater $\delta_2$H-CH$_4$ source signatures would be to combine high-resolution mapping of wetlands and inland waters, maps of the global distribution of $\delta_2$H$_p$, and regression relationships between $\delta_2$H-CH$_4$ vs. $\delta_2$H$_p$." >> I agree with the authors and suggest this would be a worthwhile output to include in this manuscript instead of the global upscaling estimate.

**We appreciate this suggestion. As mentioned above, adding this output to this manuscript would entail substantial additional work, as well as additional expertise beyond that of the authors. We have however begun a collaboration with another research group to perform this analysis, and this will be the focus of a future publication.**

L445-L464 Section 4.2. This section would benefit from acknowledging and discussing the study by Rigby et al. (2012).

**We thank the reviewer again for this suggestion. We will acknowledge and discuss this work in the revised manuscript.**

L500-L504 In addition to the caveat noted that CH$_4$ data exhibiting $_2$H-enrichment due to methane oxidation are uncommon, the amount of CH$_4$ emitted to the atmosphere bearing the effects of methanotrophy is likely to be small. Bacteria oxidation is highly efficient in the subsurface of wetlands and little CH$_4$ tends to escape to the atmosphere via diffusion through porewater. This comment applies to peatlands. The situation is different in inland

water environments.

**This is an important point, and we will revise this section to make this clear. As noted in Planned Major Revision 4, we are now less confident that the observed variation in $\delta^2$H-CH$_{4,w0}$ can be primarily ascribed to differences in methanogenic pathway. Therefore our discussion of relative importance of these mechanisms, as well as other possibly influential processes, will be quite different in the revised manuscript.**

L510–L518 I was pleased to see incorporation of these alternate explanations for relationships between **d2**H and **d13**C values of CH4. Methanogenic pathways are not the only potential explanation.

**We are glad to see that there is a positive reception to this. Based on this comment and those of reviewer 1 we are planning to focus on alternate explanations to a greater degree in the revised manuscript. See Planned Major Revision 4.**

L592-L593 – Bellisario et al. (1999) provides a good example of how **d13**C(CH4) values vary along a trophic gradient in a wetland complex. Differences in **d13**C values of CH4 emissions and porewater CH4 values in minerotrophic vs. ombrotrophic wetland are demonstrated in Hornibrook and Bowes (2007) and Hornibrook (2009). Landscape scale measurements (atmospheric inversions and aircraft measurements; Fisher et al., 2017) also show that northern wetlands contain sources of 13C-poor CH4 that differ from values of ~-62 to -58 permil typically attributed to northern peatlands in isotope-weight CH4 budgets. Characterization of sites as ombrotrophic or minerotrophic on the basis of water chemistry and vegetation surveys is essential for making these distinctions.

**We thank the reviewer for these insights. We will expand this paragraph to include the ideas and references mentioned by the reviewer. While it is difficult for us to make these distinctions in this dataset, we will note these points. In addition to the absence of C$_4$ plant ecosystems, this is an additional potential bias in the d13C database assembled in this study, and we will acknowledge this and discuss how it could be addressed with future research.**

L617 to L622 It is unclear how a more negative than expected value for estimated **d13**C(CH4) can be explained by (2) source signatures being biased toward more positive **d13**C values.

**This was a mistake. We meant to say '$^{13}$C *depleted* values' and '$^{13}$C *depleted* sources'. Regardless, this section of the discussion will be heavily revised based on the suggestions of reviewers 2 and 3, with less emphasis on discrepancies with atmospheric measurements. See planned major revision 6.**

**References Cited:**

Alstad, K. P., and Whiticar, M. J.: Carbon and hydrogen isotope ratio characterization of methane dynamics for Fluxnet Peatland Ecosystems, Org Geochem, 42, 548-558, 2011.

Bowen, G. J., and Wilkinson, B.: Spatial distribution of δ18O in meteoric precipitation, Geology, 30, 315-318, 2002.

Chasar, L., Chanton, J., Glaser, P. H., and Siegel, D.: Methane concentration and stable isotope distribution as evidence of rhizospheric processes: Comparison of a fen and bog in the Glacial Lake Agassiz Peatland complex, Annals of Botany, 86, 655-663, 2000.

Chanton, J. P., Fields, D., and Hines, M. E.: Controls on the hydrogen isotopic composition of biogenic methane from high-latitude terrestrial wetlands, Journal of Geophysical Research: Biogeosciences (2005–2012), 111, 2006.

Fletcher, D., & Dixon, P. M. (2012). Modelling data from different sites, times or studies: weighted vs. unweighted regression. *Methods in Ecology and Evolution*, *3*(1), 168-176.

Lansdown, J., Quay, P., and King, S.: CH4 production via CO2 reduction in a temperate bog: A source of 13C-depIeted CH4, Geochim Cosmochim Ac, 56, 3493-3503, 1992.

Lansdown J. M. (1992) The carbon and hydrogen stable isotope composition of methane released from natural wetlands and ruminants. Ph.D. dissertation, Univ. of Washington.

Hornibrook, E. R., Longstaffe, F. J., and Fyfe, W. S.: Spatial distribution of microbial methane production pathways in temperate zone wetland soils: stable carbon and hydrogen isotope evidence, Geochim Cosmochim Ac, 61, 745-753, 1997.

Rice, A. L., Butenhoff, C. L., Teama, D. G., Röger, F. H., Khalil, M. A. K., and Rasmussen, R. A.: Atmospheric methane isotopic record favors fossil sources flat in 1980s and 1990s with recent increase, Proceedings of the National Academy of Sciences, 113, 10791-10796, 2016.

Sherwood, O. A., Schwietzke, S., Arling, V. A., and Etiope, G.: Global inventory of gas geochemistry data from fossil fuel, microbial and burning sources, version 2017, Earth System Science Data, 9, 2017.

Wahlen, M.: Carbon dioxide, carbon monoxide and methane in the atmosphere: abundance and isotopic composition, Stable isotopes in ecology and environmental science, 93-113, 1994.

Waldron, S., Lansdown, J., Scott, E., Fallick, A., and Hall, A.: The global influence of the hydrogen iostope composition of water on that of bacteriogenic methane from shallow freshwater environments, Geochim Cosmochim Ac, 63, 2237-2245, 1999a.

Waldron, S., Hall, A. J., and Fallick, A. E.: Enigmatic stable isotope dynamics of deep peat methane, Global Biogeochem Cy, 13, 93-100, 1999b.

Whiticar, M. J., Faber, E., and Schoell, M.: Biogenic methane formation in marine and freshwater environments: $CO_2$ reduction *vs.* acetate fermentation—Isotope evidence, Geochim Cosmochim Ac, 50, 693-709, 1986.

Whiticar, M. J.: Carbon and hydrogen isotope systematics of bacterial formation and oxidation of methane, Chem Geol, 161, 291-314, 199

---

## Author Response (AR1)

Response to reviewers for **Geographic variability in freshwater methane hydrogen isotope ratios and its implications for global isotopic source signatures**

Peter M.J. Douglas[1], Emerald Stratigopoulos[1], Jenny Park[1], Dawson Phan[1]

[1]Earth and Planetary Sciences, McGill University, Montreal, H3A 0E8, Canada

*Correspondence to*: Peter M. J. Douglas (peter.douglas@mcgill.ca)

All of the reviewers provided excellent suggestions and feedback on the paper, and we think that by addressing their concerns the paper will be greatly improved. Many of their comments were complementary. Therefore we will first summarize the major revisions we made to the paper before responding to each reviewer in detail:

**Major Revisions**

**1)** We have revised the freshwater isotopic dataset in response to comments from Reviewer 1 and Reviewer 3. (i) For peatland sites with depth stratified sampling we have decided to only include samples from the upper 50 cm, as suggested by reviewer 3, since this is the depth range that is most likely to emit $CH_4$ to the atmosphere. This affects a total of 8 sites. (ii) Reviewer 1 noted that an outlier sample from the Amazon River with very high $\delta^2$H-$CH_4$ and $\delta^{13}$C-$CH_4$ could be derived from thermogenic methane. We agree that this outlier is suspect, and therefore have decided not to include it. (iii) We also noted that one sites (Mirror Lake, Florida, USA) were analyzed in two separate studies, and therefore was included twice in the dataset. We have combined the data from the two studies into one entry.

**2)** As suggested by all three reviewers, we have performed more rigorous analysis of the relationship between measured and modeled $\delta^2$H-$H_2O$ values. Specifically we have done the following: (i) In addition to annual precipitation $\delta^2$H values, we now also analyze growing season precipitation $\delta^2$H, which is defined as the amount-weighted mean $\delta^2$H of months with mean temperature greater than 0º C. This provides an opportunity to assess whether seasonal variation in precipitation in the mid to high-latitudes is important in controlling the environmental $\delta^2$H-$H_2O$ value; (ii) separately analyzing inland water and wetland environments, since these are very different hydrological environments and the controls on $\delta^2$H-$H_2O$ are potentially different.

This analysis led to the following key results (see the revised Figure 2): i) growing season modeled precipitation $\delta^2$H is a better predictor of inland water $\delta^2$H-$H_2O$ than annual precipitation $\delta^2$H, in that the regression curve is indistinguishable from the 1:1 line. ii) Annual modeled precipitation $\delta^2$H is a better predictor of wetland $\delta^2$H-$H_2O$, in that the slope of the regression is indistinguishable from 1, and the $R^2$ value is higher. However, the regression line is offset from the 1:1 line by 18.6±9‰. We interpret this as an indicator of likely widespread evaporative effects on $\delta^2$H-$H_2O$ in wetland environments. These results are consistent with isotope hydrology studies, as discusses in section 3.2.

We use these results to then develop a 'best estimate' for comparing $\delta^2$H-$H_2O$ with $\delta^2$H -
CH4. (i) For sites with measured $\delta^2$H-$H_2O$ values we use the measured value. (ii) For
inland water sites without measured $\delta^2$H-$H_2O$ we use modeled growing season
precipitation, since as discussed above the regression of this against measured $\delta^2$H-$H_2O$
is indistinguishable from the 1:1 line. (iii) For wetland sites without measured $\delta^2$H-$H_2O$
we estimate the $\delta^2$H-$H_2O$ using the regression relationship with annual precipitation $\delta^2$H-
$H_2O$ shown in Figure 2A. We feel this approach combining measured and modeled data
is consistent with that of Waldron et al., (1999a), who we note also analyzed a
combination of sites with measured $\delta^2$H-$H_2O$ (29 out of 51 sites) and estimated $\delta^2$H-$H_2O$
based on precipitation isotopic measurements or estimates (22 out of 51 sites).
**3).** As suggested by all three reviewers, it is important to consider the effects of modeled
$\delta^2$H-$H_2O$ on the regression between $\delta^2$H-$H_2O$ and $\delta^2$H-CH$_4$. To do this carefully we
performed the regression analysis using four different estimates of $\delta^2$H-$H_2O$:
(i) the 'best-estimate' of $\delta^2$H-$H_2O$ as described above in Major Revision 2; (ii) measured
$\delta^2$H-$H_2O$, only analyzing sites with this measurement; (iii) modeled annual precipitation
$\delta^2$H; and (iv) modeled growing season precipitation $\delta^2$H. We think it is valuable to
continue to include the regression relationships for modeled precipitation because these
relationships could be used in future studies using Earth Systems Models to predict the
distribution of $\delta^2$H-CH$_4$. For each of these cases we analyzed all sites, inland waters, and
wetlands (See Supplemental Table 2). We compare each of these relationships with the
'in-vivo' line of Waldron et al., (1999a)
A key point is that we have decided to use unweighted, as opposed to weighted,
regression. Comments by Reviewer 1 made us realize that weighting by standard error
was causing a few sites to strongly bias the regression results. Statistical research has
found that for environmental data with poorly constrained error variance unweighted
regression is frequently less biased than weighted regression (Fletcher and Dixon, 2012).
Using a statistical test proposed by that study we find that unweighted regression is a
good choice for our dataset. Note that in Supplemental Table 2 we apply unweighted
regression to the dataset of Waldron et al., (1999), in part because the specific weighting
methodology was not specified in that study. This produces a small difference in the
regression relationship shown in Supplemental Table 2 with that reported by Waldron et
al., (1999), but the two regression relationships are within error.
We then used analysis of covariance (ANCOVA) to examine differences between the
regression relationships shown in the table. Based on a multiple comparison test, none of
the regression relationships shown with our dataset are significantly different, nor are
they significantly different from the regression of Waldron et al., (1999a). Therefore we
conclude that (i) using modeled precipitation $\delta^2$H-$H_2O$ does not have a significant effect
on the estimate of the relationship between $\delta^2$H-$H_2O$ vs. $\delta^2$H-CH$_4$; (ii) Differences in the
slope of this relationship between inland waters and wetland sites are not conclusive; and
(iii) that since all of the regression relationships using the larger dataset produce a flatter
slope than that of Waldron et al., (1999a), the true global slope is likely to be flatter than inferred in that study, but confirmation of this flatter global slope will require more data
and further analysis.
**4)** We then used the 'best-estimate' $\delta^2$H-$H_2O$ values and the regression based on those
values, shown in Figure 3A and Supplemental Table S2, to calculate a revised $\delta^2$H-
$CH_{4,w0}$ value for each site. These analyses were then applied in the subsequent analyses in
the paper shown in Figures 6, 8, and 9. We also calculated an alternate value for sites
with measured $\delta^2$H-$H_2O$, using the values and regression curve for those sites (Figure 3B
and Supplemental Table S2).
We have decided to substantially change and revise our analysis of co-variation between
$\delta^2$H-$CH_{4,w0}$ and $\delta^{13}$C-$CH_4$, $\delta^{13}$C-$CO_2$, and $\alpha_C$. Using the revised $\delta^2$H-$CH_{4,w0}$ values we
found inconsistent results of the breakpoint regression analysis applied in the original
manuscript. Specifically, the identified breakpoint is not consistent when analyzing all
sites or only sites with measured $\delta^2$H-$H_2O$. Given this inconsistency, and the complexity
of this analysis, we decided to omit this analysis from the revised manuscript. Instead we
focus on simple linear regression between these variables, both for the dataset as a whole
and for sites disaggregated into wetlands and inland waters. This analysis implies a
significant correlation between $\delta^2$H-$CH_{4,w0}$ and both $\delta^{13}$C-$CO_2$, and $\alpha_C$ for wetlands in
particular, but only when all sites are analyzed. These relationships are not apparent when
only sites with measured $\delta^2$H-$H_2O$ are included. Therefore these correlations are clearly
preliminary and require further verification.
Therefore in our revised analysis we will place less emphasis on these apparent
correlations, and less emphasis on the relationship between methanogenic pathway and
$\delta^2$H-$CH_4$ generally, as suggested by reviewer 1. Instead we discuss four processes or
variables that have the potential to influence $\delta^2$H-$CH_4$ in freshwater environments: (i)
differences in methanogenic pathway; (ii) methane oxidation; (iii) isotopic fractionation
due to diffusion; and (iv) differential thermodynamic favorability or differential
enzymatic reversibility of methanogenesis. Ultimately, our conclusion is co-variation
with $\delta^{13}$C-$CH_4$, $\delta^{13}$C-$CO_2$, and $\alpha_C$ cannot fully resolve the complex interactions between
these processes on $\delta^2$H-$CH_4$ on a global or inter-site basis, and other approaches will be
necessary to determine their relative importance, or the possible importance of other
processes.
Given the findings mentioned above, we will also substantially revise the original Figure
6, which is now Figure 5. Instead of distinguishing samples by inferred methanogenic
pathway in this figure, we distinguish samples by environment (wetland vs. inland
water), show available data for cow rumen and landfills, and show data and regression
lines for incubation and pure culture experiments. We feel this revised analysis is very
informative about the likely processes controlling the slope of the regression between
$\delta^2$H-$H_2O$ and $\delta^2$H-$CH_4$, and supports the application of the 'in-vitro' line of Waldron et
al. (1999a) as an analogue for environmental samples. We have revised our discussion of
this to provide proper credit to the ideas presented in that paper.
**5)** Reviewer 2 made numerous comments about the representativeness of our $\delta^{13}$C-$CH_4$

dataset. We want to make clear that to our knowledge this is the largest database of
freshwater methane $\delta^{13}$C-CH$_4$ currently compiled. For comparison, the second largest
dataset, that of Sherwood et al., (2017), includes 48 freshwater sites (including rice
paddies), of which 16 are also included in our database. However our $\delta^{13}$C-CH$_4$
database is not comprehensive (unlike the $\delta^{2}$H-CH$_4$ database), in that it does not include
many measurements that are not paired with $\delta^{2}$H-CH$_4$ measurements and that have not
yet been compiled into a database. It is also probably not representative, because some
important environments, notably C$_4$ plant dominated ecosystems, are not well
represented.
Since the primary focus of this paper is $\delta^{2}$H-CH$_4$, it is not within its scope to provide a
comprehensive database of freshwater $\delta^{13}$C-CH$_4$, although that would be a worthwhile
goal for future research. In order to make our analysis as complete as possible, in our
revised manuscript we will include the 32 freshwater sites from Sherwood et al., (2017)
that were not included in our original analysis in our calculations for the upscaling
exercise, as well as Table 1 and Figures 7 and 8. Sherwood et al., (2017) do not provide
information on sample type, and we therefore did not include these additional data in the
analysis for Figure 9.  We will also carefully discuss the likely biases in this dataset,
especially in terms of C$_4$ plant environments, and their implications for our
interpretations.
**6)** Both reviewers 2 and 3 expressed some concerns with the upscaling analysis. We
acknowledge that the upscaling analysis is relatively simplistic, and that some of the
interpretations were speculative. However, we still think it is valuable to use the
estimates of freshwater CH$_4$ isotopic composition, differentiated by latitude, produced in
this study to estimate global source $\delta^{2}$H-CH$_4$ and $\delta^{13}$C-CH$_4$, and to compare that with
other estimates. We wish to make clear that given uncertainties and complexity in
estimating sink fractionations, particularly for $\delta^{2}$H-CH$_4$, we are not attempting to
estimate atmospheric $\delta^{2}$H-CH$_4$ and $\delta^{13}$C-CH$_4$, but instead the integrated source $\delta^{2}$H-CH$_4$
and $\delta^{13}$C-CH$_4$ prior to sink fractionations. We think there is value in comparing this with
(i) a previous bottom-up estimate of these values (Whiticar and Schaefer, 2007); and (ii)
with top-down estimates reported by Rice et al., (2016), as well as simpler estimates
provided by Whiticar and Schaefer (2007) and Sherwood et al., (2017). We concur with
Reviewer 2 that the discussion of alternate emissions scenarios is too speculative and
simplistic, and therefore we will omit this discussion. We have also omitted our analysis
of the sensitivity of global source $\delta^{2}$H-CH$_4$ and $\delta^{13}$C-CH$_4$ to varying emissions fluxes by
latitude. Some of this analysis will instead appear in another paper currently in
preparation.  We do mention previous papers that suggest errors in emissions inventories
based on $\delta^{13}$C measurements, but do not attempt to resolve the findings of those studies
using our simple upscaling estimate. Instead, we focus on likely sources of error in the
isotopic source signatures, and the best ways to address these errors in future studies. We
note that we now express uncertainty for the Monte Carlo analysis as  2 $\sigma$ standard
deviation, which is a more conservative estimate of uncertainty.

We disagree with Reviewer 2 that the error estimates for isotopic source signatures are
generally too optimistic, which we will discuss in more detail in our response to that
reviewer.
Given comments from all three reviewers we will revise Figure 10 to only include panel
C, and make the comparison with other estimates of global source $\delta^2$H-CH$_4$ and $\delta^{13}$C-
CH$_4$ clearer in this figure.
**Changes to manuscript structure:** Given the comment of Reviewer 1 on the
redundancy of the Results and Discussion sections we have decided to combine these
sections. We feel this simplifies the manuscript and improves the flow.
**Revisions to Table 1:** As described above in Major Revisions 1 and 4, we have changed
the data inputs to Table 1, which has changed some of the isotopic values and
uncertainties shown in this table.
**Additional and Revised Supplemental Material:** We have added supplemental text that
describes in detail isotopic vectors for different biogeochemical processes that are
depicted in the new version of Figure 6. We have added two supplemental figures, which
are versions of Figure 8A and 9A that only include sites with measured $\delta^2$H-H$_2$O. We
have added two additional supplemental tables that detail regression statistics for i)
regression analyses of $\delta^2$H-H$_2$O vs. $\delta^2$H-CH$_4$ (Supplemental Table 2); and ii) regression
analyses of $\delta^2$H-CH$_{4,w0}$ vs. $\delta^{13}$C-CH$_4$, $\delta^{13}$C-CO$_2$, and $\alpha_C$ (Supplemental Table 4). We
have omitted the original Supplemental Table 2, which is replaced by the supplemental
text described above.
*Specific Responses to Reviewer 1:* Reviewer comments are in plain text. **Responses are**
**in bold text.**
**We thank Dr. Waldron for her careful and thorough review. We appreciate that**
**this review presented a challenging situation, and we value her honesty and**
**openness. We are confident that we can address her concerns in the revised**
**manuscript.**
Substantial conclusions are reached, but the interrogative approach has weaknesses that propagate
through substantial analytical reasoning and so the integrity of the conclusions is questionable. I
detail this further below, but until the analytical approaches are reconsidered the conclusions are
not securely reached
**We understand this critique, and in response we have strengthened the statistical analyses**
**and interrogation of the data, in particular with regards to the 1) comparison of measured**
**and modeled $\delta^2$H-H$_2$O values, 2) the inference of a regression slope between $\delta^2$H-H$_2$O and**
**$\delta^2$H-CH$_4$, and 3) the application of carbon isotope fractionation factors to evaluate the**
**potential effects of methane oxidation, methanogenesis pathway, or other biogeochemical**
**effects on $\delta^2$H-CH$_4$.  See major revisions 2, 3, and 4 above.**
With respect to understanding isotopic compositions: the methods are not all valid, particularly the reconstruction of missing δD-H2O for a field measurement of δD-CH4. The interrogation of
this relationship (Fig. 2) lacks statistical rigour, and its propagation - a relationship that has bias
and significant variability - is unconsidered in all analysis thereafter (as represented by Figs. 3-9
and possibly 10) and so this reasoning is flawed and the interpretations may be wrong.
**We have thoroughly re-analyzed the relationship between measured and modeled**
**$\delta^2$H-H$_2$O in the revised manuscript. See Major Revision 2 above. We then applied**
**this revised analysis forward to the remainder of the manuscript. See Major**
**Revisions 3 and 4.**
The authors are not consistent in identifying when processes they are interpreting are based on
hypothesised relationships and the impression is given such processes are certain (detailed
below).
**We have made it clearer in the revised manuscript where we are discussing hypothesized**
**relationships, as discussed in more detail below. In particular, regarding hypotheses**
**regarding the effects of methanogenic pathway on $\delta^2$H-CH$_4$ we are more circumspect in the**
**revised manuscript, and discuss alternate hypotheses in greater detail. See Major Revision**
**4.**
I found it difficult to follow the calculations behind αC – an important part of the manuscript –
when I was trying to compare other data sets with their approach.
**We are uncertain what aspect of this calculation was unclear, but we have tried to make the**
**description of this calculation clearer. See lines 193-194.**
Largely bit not always, for example there is a large section in 4.31. that is repeating suggestions
made in section 1.1. of Waldron et al 1999, but this work is unreferenced and so as written
implies the review m/s is the first to have suggested this; the abstract does not make clear refining
an existing phenomena observed and described similarly previously.
**We regret the omission of references and acknowledgment to Waldron et al., 1999, in the**
**ideas presented in section 4.3.1. We thoroughly revised this section to provide proper credit**
**for these ideas, and integrated the discussion with that previously published by Waldron et**
**al., (1999a). This material is now incorporated in Section 3.3.1.**
Broadly but not sure how "geographic variability in freshwater methane hydrogen isotope ratios
has implications for microbial biogeochemistry" - the microbes are active with no knowledge
δD…so this can be refined.
**We will follow this suggestion and modify the title to:** *Geographic variability in freshwater*
*methane hydrogen isotope ratios and its implications for global isotopic source signatures.*
It is a paper with a lot of detail and so to follow it all the reader has to concentrate deeply for the
results section. As such, and maybe in addition, the discussion from section 4 onwards seems in
places repetitive.
**We agree the results are very detailed. Given the critiques of the reviewers we will**
**need to add additional statistical analyses and data interrogation to the results,**

275 **leading to an overall increase in detail. However, we will present this in as**
276 **streamlined and clear manner as possible. To do so we have combined the results**
277 **and discussion sections, which we feel has streamlined and simplified the article,**
278 **and reduces redundancy.**
280 Broadly yes – I suggest a group whose work may be missing in the intro.
282 **We have added the suggested citations in the introduction**
284 Yes, very helpful, but sheet 2 could make it clearer if the data offered is used in $\alpha_C$ or these are
285 summarised data from other sources.
287 **Sheet 2 is omitted from the revised supplemental tables, as multiple reviewers have**
288 **questioned the value of the predicted fields for pathway and oxidation dependent**
289 **isotopic variation. Instead we provide approximate vectors of isotopic variability for**
290 **different biogeochemical processes. We emphasize that these are guidelines and are**
291 **not precise. These are summarized in Section 3.4, and in more detail in the**
292 **supplemental text.**
294 The substantive conclusions in this manuscript rely on a data set where $\delta D$-$H_2O$ does not exist
295 for more than half the data: 53% of the sites do not have field measured $\delta D$-$H_2O$ (L88). In these
296 cases, $\delta D$-$H_2O$ is inferred from a reputable global precipitation database and a correlation
297 observed for sites where measured values exist. The authors consider this relationship sufficiently
298 robust to proceed to use the reconstructed $\delta D$-$H_2O$ where measured values do not exist. I disagree
299 this is the case.
301 **We acknowledge this is an important critique. In the revised manuscript we take**
302 **steps to strengthen the analysis of the relationship between modeled and measured**
303 **$\delta^2H$-$H_2O$, and carefully evaluate if using modeled values of $\delta^2H$-$H_2O$ leads to a bias**
304 **in the inferred relationship between $\delta^2H$-$H_2O$ and $\delta^2H$-$CH_4$. See Major Revisions 2**
305 **and 3. Our analysis indicates that using modeled $\delta^2H$-$H_2O$ does not lead to a**
306 **significant difference in the regression relationship with $\delta^2H$-$CH_4$. See Figure 3 and**
307 **Supplemental Table 2.**
309 The statistical integrity shown elsewhere in the manuscript is lacking in this section on
310 reconstructing $\delta D$-$H_2O$, with the authors describing their predictive relationship as showing
311 "generally good agreement" and proceeding to use it. The bias and variability in a predictive $\delta D$-
312 $H_2O_p$ and thus how far it may be from the true $\delta D$-$H_2O$ appear unconsidered in any further
313 analysis (no errors propagated through for estimated $\delta D$-$H_2O$?).
315 **As described above, in the revised manuscript we now carefully evaluate this**
316 **relationship. See Major Revision 2.**
318 Further, I note that the data in table S3 supplementary information for which there are measured
319 $\delta D$-$CH_4$ -$\delta DH_2O$ fit closely to the in-vitro line from which Waldron et al 1999 project a global
320 relationship - but the data with estimated $\delta DH_2O$ in table S3 do not. This is important for two
321 reasons:

**The reviewer has pointed to an interesting observation. The difference observed by the**
**reviewer is at odds with Figure 3A in the original manuscript, which clearly showed that the**
**regression lines for $\delta^2$H-H$_2$O vs $\delta^2$H-CH$_4$ fully overlap whether modeled (black regression**
**line) or measured (blue regression line) $\delta^2$H-H$_2$O is applied, and was not in agreement with**
**the in vivo line of Waldron et al., (1999).**
**On further analysis of the data, we identified that the discrepancy observed by the reviewer**
**was probably caused by two factors:**
**1) The weighted regression method that we used in the original manuscript was strongly**
**influenced by a few sites at high latitudes that both (a) have a large number of**
**measurements (and therefore a low standard error and a higher weight); and b) relatively**
**high $\delta^2$H-CH$_4$ values. We infer that this strong weighting at these sites led to a strong bias**
**in the regression. Based on statistical research involving environmental samples (Fletcher**
**and Dixon, 2012) we have decided that unweighted regression is preferable for this dataset**
**(See Major Revision 3). We assume the reviewer applied unweighted regression when**
**analyzing these data**
**2) The unweighted regression performed by the reviewer was likely strongly influenced by**
**the outlier site from the Amazon with very high $\delta^2$H-CH$_4$. When this point is removed, as**
**suggested by reviewer 1, the regression slope becomes flatter.**
**The two factors above effectively cancel each other out. In re-analyzing the data after**
**accounting for these two changes (See Figure 3 and Supplemental Table 2) we find that (a)**
**there is not a large or significant difference in the regression slope if measured $\delta^2$H-H$_2$O,**
**'best-estimate' $\delta^2$H-H$_2$O (see Major Revision 2), or modeled precipitation $\delta^2$H-H$_2$O is**
**used and (b) all of these regression slopes are flatter than that of Waldron et al., (1999).**
1. It confirms the predictive relationship in Waldron et al 1999 for $\delta$D-CH$_4$ from $\delta$D-H$_2$O still has
integrity, more so by adding in another methane-producing environment (innocula), a significant
time gap, and another geographic locality.
**As mentioned above (Major Revision 3), our revised regression analysis continues to**
**result in flatter slopes than the predictive relationship proposed by Waldron et al.,**
**(1999), regardless of the method of estimating $\delta^2$H-H$_2$O (See Figure 3 and**
**Supplemental Table 2). However, as we also discuss above, given the wide confidence**
**intervals of these relationships we do not find a significant difference with the**
**prediction of Waldron et al., 1999 using multiple group comparison ANCOVA.**
**However, given that every analysis of the larger dataset presented here results in a**
**flatter slope, we think it is probable that the global relationship has a somewhat**
**flatter slope than was inferred by Waldron et al., (1999).**
2. If statement 1 is considered sound, then the poor fit of paired $\delta$D-CH$_4$- $\delta$D-H$_2$O with predicted
$\delta$D-H$_2$O supports the assertion above that the relationship the authors are using here to
reconstruct $\delta$D-H$_2$O is questionable.

**As noted above, we do not see evidence that there is a significant difference in $\delta^2$H -**
**CH4 vs. $\delta^2$H -H2O based on the method of estimating/inferring $\delta^2$H-H2O. Therefore we**
**disagree with the assertion that the methods used to reconstruct $\delta^2$H-H$_2$O**
**are questionable. However, we have revised our approach to predicting $\delta^2$H-H$_2$O, as**
**explained in more detail in Major Revisions 2 and 3.**
**We think it is important to note that the in-vivo relationship of Waldron et al., (1999a) was**
**not based purely on sites with measured $\delta^2$H-H$_2$O. Instead, that study used a combination**
**of sites with $\delta^2$H-H$_2$O measurements (57%) measurements and estimates based on**
**precipitation isotope measurements (43%).**
**Quoting from that Waldron et al., (1999a): "***Where paired δD(CH4)–δD(H2O)*
*measurements were not published δD(H2O) was sourced from measured precipitation*
*values for the area, for example, the weighted mean of the precipitation samples*
*collected in south Florida over a 3-yr period (Swart et al., 1989) was used as an*
*appropriate value for δD(H2O) for St. Marks Swamp, Florida (Happell et al., 1994).*
*Other unknown δD(H2O) signatures (e.g., the Alaskan Lakes; Martens et al., 1992)*
*were estimated from the weighted mean value of sites close to the area sampled, that*
*participated in the global network, Isotopes in Precipitation (IAEA, 1992) or from the*
*meteoric water line (Craig, 1961). We are aware that δD(groundwater) can differ by up*
*to 30‰ from measured δD(precipitation) (e.g., Hornibrook et al., 1997; E. R. C.*
*Hornibrook, pers. comm.), but such fractionation is difficult to quantify and the logical*
*approach we have adopted provides the best estimate for dD(H2O)*
*where measured values are unavailable."*
**We agree with Waldron et al., (1999a) that differences between precipitation and**
**groundwater (or lake water) can be large, and these differences can be difficult to**
**quantify. However, we do not think these potential differences negate the value of**
**estimating $\delta^2$H-H$_2$O using available estimates of precipitation $\delta^2$H, along with**
**accounting for the effects of precipitation seasonality and evaporation, as described**
**in Major Revision 2. Indeed, we think our revised approach of combining measured**
**and estimated water $\delta^2$H into a 'best-estimate' is a logical extension of the approach**
**used by Waldron et al., (1999a). However, we agree that a more careful evaluation**
**of this approach is warranted, and we have added this to the revised manuscript as**
**described above (Major Revisions 2, 3, and 4).**
With the greatest of respect, using the predicted data produces an outcome that is like a 'house of
cards' – all subsequent analysis using this data is built on a shaky foundation. I therefore think
that incorporating paired δD-CH4 -δD-H2Op in further analysis is flawed and offer two examples
why:
1. It creates a new global line for δD-CH4 -δD-H2O that may be wrong.
2. It could lead to artefact in interpretation, which indeed may be 'visible' in the dependent
analysis. For example, the data in Fig. 3b visually also appears to separate between paired δD-
CH4 -δD-H2O data that are predicted (inland waters) and measured (wetlands), and if this is the
case interpreting a biome difference here, and later in the paper, is also questionable.

**We do not agree with the house of cards analogy, but we do agree that it is**
**important to provide more confidence in our underlying analyses. As mentioned**
**above (Major Revision 3; Supplemental Table 2), we do not observe a significant**
**difference in $\delta^2$H -CH$_4$ vs. $\delta^2$H-H$_2$O whether modeled or measured $\delta^2$H-H$_2$O is used.**
**Therefore we disagree that there is a 'shaky foundation' to our subsequent analysis.**
**However, as discussed above we have revised our data analysis to use the 'best-estimate'**
**$\delta^2$H-H$_2$O value, including measured values where available.**
**We note that in the original Figure 3B the reviewer likely misinterpreted the data**
**presented. All of the data shown in this figure are based on modeled $\delta^2$H-H$_2$O. Therefore**
**the observed difference between inland waters and wetlands cannot be related to**
**differences in the source of $\delta^2$H-H$_2$O data. Supplemental Table 2 shows that there are**
**consistent differences between wetlands and inland waters in the slope of the regression line**
**regardless of the method used to estimate $\delta^2$H-H$_2$O, but also that these differences are**
**small and statistically insignificant. Therefore in our revised manuscript we state that we**
**cannot confidently infer a difference in the relationship between these environments.**
With respect to the redefining of a new global $\delta$D-CH$_4$-$\delta$D-H$_2$O and consideration of how this has
changed from the relationship offered in Waldron et al 1999: unless the authors can produce a
more robust estimation of $\delta$D-H$_2$O$_p$, the data that uses $\delta$D-CH$_4$ paired with predicted $\delta$D-H$_2$O
needs to be removed - for as noted earlier, there is insufficient confidence this is an accurate
representation of the field situation and may create a false outcome. I suspect this will change the
global relationship and increase the slope as paired data with $\delta$D-H$_2$O$_p$ visually appears to
dominate the enriched samples.
**We believe the approach taken by our Major Revision 2 and 3 effectively addresses**
**this critique. We now take an approach similar to that of Waldron et al., (1999a),**
**namely combining measured and modeled $\delta^2$H-H$_2$O values to produce a best-**
**estimate value for each site.**
**As noted above, we do not observe a significant difference in the $\delta^2$H -CH$_4$ vs. $\delta^2$H-**
**H$_2$O relationship whether modeled or measured $\delta^2$H-H$_2$O, or a combination of the two is**
**used (See Figure 3 and Supplemental Table 2). Furthermore, when only analyzing sites with**
**measured $\delta^2$H-H$_2$O we still observe a slope flatter than that of Waldron et al.,**
**(1999a) (Figure 3B).**
Please plot both the in-vitro and in-vivo relationship, and for the former its prediction intervals -
which are missing from 3b and so give the sense of a poorer fit of Waldron et al 1999 to the bgd
expanded field data set here.
**We have included a more robust comparison with the data from Waldron et al**
**(1999) in the revised Figure 3 including confidence intervals. We argue that**
**confidence intervals are the more appropriate metric, since this gives the**
**uncertainty of the regression relationship, as opposed to the predicted range of**
**observations. We are more interested in comparing the underlying regression**
**relationships, as opposed to the predicted range of observations.**

**Figure 3 is relatively complex did not include the in-vitro relationship in this figure. However, we have included a comparison with the in-vitro relationship in the revised Figure 5, and make a strong point of its similarity with the inferred environmental regression relationships for both wetlands and inland waters, especially in terms of the slope.**

Compare whether the in-vivo line is statistically different to the relationship generated from the data set presented in the bgd manuscript. This will allow confidence in any further discussion on how the relationship has been redefined (than just comparing slopes etc). If the two relationships are indistinguishable statistically, nuanced statements about differences in slope etc are meaningless – all that has happened is that the expanded data set has redefined better the field relationship for $\delta D\text{-}CH_4$ - $\delta D\text{-}H_2O$ (as indicated likely in Waldron et al, 1999) - noting that this field relationship does not wholly reflect the relationship at production (see next point).

**We used analysis of covariance (ANCOVA) to statistically compare differences in regression relationships. As discussed in Major Revision 3, multiple group comparison with ANCOVA does not indicate a significant difference in slope between our dataset and that of Waldron et al. (1999a). However, regardless of the method of estimating $\delta^2H\text{-}H_2O$ used, the analysis of the larger dataset produces a flatter slope. Therefore we infer that the 'true' global slope is likely to be flatter than that inferred by Waldron et al., (1999a), but also that more data and further analysis is needed to confirm this and reduce the uncertainty of the slope.**

Assess whether the expanded field data set is predominantly 13C-enriched compared to the in-vivo relationship described in Waldron et al 1999, and therefore consistent with an interpretation that differences in field $\delta D\text{-}CH_4$ may be an artefact of fractionating processes post-production than pathway per se This is advocated as I am still unaware of experimental evidence methanogenic pathway in shallow freshwaters changes $\delta D\text{-}CH_4$, but there is evidence of processes, oxidation and mixing, causing enrichment, and so this approach is consistent with scientific principle of parsimony and interpreting data using the simplest approach.

**We have assessed this and in fact the opposite is the case. The data from sites included in Waldron et al., (1999a) is somewhat higher in $\delta^{13}C\text{-}CH_4$ (-60.8±0.9‰ SEM) relative to the total dataset (-62.6±0.6‰) or to the sites that were not included in Waldron et al., (1999) (-63.4±0.8‰). We do not think it is likely that there is systematic difference in these sets of sites in terms of post-production processes, which we take to mean oxidation, diffusive fractionation, and mixing of different $CH_4$ reservoirs. We agree that such processes can lead to variation in $\delta^2H\text{-}CH_4$ (see Major revision 4 above), but see no evidence that this explains the difference between our dataset and that of Waldron et al., (1999a). Instead this difference is most likely a function of the much larger dataset from the high latitudes in this study, which we discuss in the revised manuscript.**

To explore why the paired $\delta D\text{-}CH_4$-$\delta D\text{-}H_2O$ measurements are not fully described by the best fit line, the authors explore whether a difference in (dominant) methanogenic pathway is evident in the data. With no evidence from paired $\delta D\text{-}CH_4$-$\delta 13CH_4$ the authors draw on $a_c$ as a proxy for methanogenic pathway to assess this. Step-wise regression is used to explore this. I think this is interesting and something to revisit when the paired data relying in predicted $\delta D\text{-}H_2O$ has been removed, but currently it is the next floor in the 'house-of-cards', reliant on data that we do not
know to be accurate, and therefore the significant relationships that the authors infer changes in
methanogenic pathway from, we do not know to be true.

**We have re-assessed the step-wise regression relationship between $\alpha_C$ and d2H-CH4,wO**
**using revised values for the latter as described above (Major Revision 4). Indeed to ensure**
**this relationship is robust we tested it using two different approaches: 1) $\delta^2$H-CH$_{4,w0}$ using**
**the 'best estimate' for $\delta^2$H-H$_2$O, as described above; 2) $\delta^2$H-CH$_{4,w0}$ using only sites with**
**measured $\delta^2$H-H$_2$O. Both of these approaches indicated a step-wise linear relationship.**
**However, the two relationships generated were not consistent in the breakpoint, and the**
**linear relationships were not all statistically significant. Given this result we agree that it is**
**prudent to focus less on methanogenic pathway as an explanation of residual variability in**
**$\delta^2$H-CH$_4$, and instead discuss the complex interrelationship of multiple variables and**
**processes that can influence $\delta^2$H-CH$_4$, namely i) methanogenic pathway; (ii) methane**
**oxidation; (iii) isotopic fractionation due to diffusion; and (iv) differential**
**thermodynamic favorability of methanogenesis, or differential enzymatic**
**reversibility.**

**We have also decided to omit the step-wise regression results, as they were**
**inconsistent and difficult to interpret with this revised analysis. We have instead**
**focused on simple linear regression, and have focused on an interpretation that the**
**residual variation in $\delta^2$H-CH$_4$ is complex and cannot be explained by a single**
**biogeochemical variable or process.**

The authors in their revision should be careful in the value of thinking about a$_c$ for the following
reasons: some of the literature generating a$_c$ relies on assumption of differences in methanogenic
pathway interpreted from differences in $\delta$D-CH$_4$, but there is competng evidence $\delta$D-CH$_4$ cannot
be interpreted in this way (so a$_c$ using a$_c$ to infer methanogenic pathway in $\delta$D-CH$_4$ when $\delta$D-CH$_4$
has been used to infer methanogenic pathway becomes a circular, self-supporting and flawed
approach).

**We do not agree with the reviewer's contention of circular reasoning here. While both $\alpha_c$**
**and $\delta^2$H-CH$_4$ have in the past been used to infer methanogenic pathway, the use of $\alpha_c$ is**
**primarily based on theoretical predictions of fractionation factors for these pathways, and**
**to our knowledge has not been 'validated' via analysis of $\delta^2$H-CH$_4$. There is evidence from**
**culturing studies (e.g. Valentine et al., 2004;Penning et al., 2006a), and from studies that**
**isolate specific pathways in the environment (e.g. Penning et al., 2006a,b; Galand et al.,**
**2010), that $\alpha_c$ varies in relation to differences in methanogenic pathway.**

**However, we do note that other variables related to methanogenesis have the potential to**
**influence $\alpha_C$, including enzymatic reversibility and the thermodynamic favorability of**
**methanogenesis, as well as diffusive fractionation and methane oxidation. In addition,**
**sources and sinks of CO$_2$ in natural environments that are independent of methanogenesis**
**will also influence measured $\alpha_C$. We discussed this possibility in the original manuscript**
**(Lines 510 to 518), and have given more emphasis to this in the revised manuscript (Section**
**3.4). See Planned Major Revision 4 above.**

To help here I would advise the authors to consider Waldron et al 1998 (Geomicrobiology, 15,

157-169), which contributes to the in-vitro line in Waldron et al 1999, but the authors do not cite
so I am unsure if they are aware of the detail in this.
Here dominance of methanogenic pathway was changed in mixed culture (as would be found in
the field) incubations, and δD-CH4 monitored with time – so not just one measurement as may be
misinterpreted from Waldron et al 1999. Except for one measurement broadly within analytical
uncertainty, δD-CH4 remained constant. However, δ13CH4 did change and consistently with
fractionation ranges for the methanogenic pathways thought to be dominant (as assessed from
independent measurements of substrate turnover). I advise the authors to consult Waldron 1998
for two reasons:
1. The authors approach in the bgd paper to draw on δD-CH4 to represent differences in
methanogenic pathway would be stronger if they can provide an explanation for the constancy in
δD-CH4 while δ13CH4 changes.
**This is an important study and we appreciate the reviewer highlighting it. We cite and**
**discuss it in the revised manuscript. The finding of constant $\delta^2$H-CH$_4$ is intriguing. Recent**
**pure culture studies have clearly shown that acetoclastic methanogenesis differs in**
**hydrogen isotope fractionation from hydrogenotrophic methanogenesis under the same**
**conditions (i.e. Gruen et al., 2018), implying that acetate-methyl hydrogen does not fully**
**exchange with water during methanogenesis. See the revised Figure 5 which demonstrates**
**this. Therefore we infer that the effect observed in Waldron et al., (1998) likely results from**
**hydrogen isotope exchange with water during production of acetate from butyrate or other**
**substrates. The constancy of the $\delta^2$H-CH$_4$ would therefore imply that the isotopic**
**fractionation of H-exchange between water and the acetate methyl group effectively**
**compensates for the difference in hydrogen isotope fractionation between acetoclastic**
**methanogenesis and hydrogenotrophic methanogenesis. Clearly, this is an interesting result**
**that merits further study to resolve with the results of pure culture experiments. However,**
**we do not feel that this study on its own negates the potential for differential net hydrogen**
**isotope fractionation between acetoclastic methanogenesis and hydrogenotrophic**
**methanogenesis. But as we note in Section 3.3.1, this difference likely varies in different**
**environments as a function of differences in the $\delta^2$H of acetate, as well as differences in net**
**kinetic isotope effects associated with both pathways of methanogenesis.**
**As discussed above in Major Revision 4, the revised manuscript focuses less on the role**
**methanogenic pathway in controlling $\delta^2$H-CH$_4$, and emphasize the complexities induced by**
**multiple mechanisms influencing hydrogen isotope fractionation during and after**
**methanogenesis.**
2. Waldron et al can also be used to calculate ac (both from CO2 and from estimated substrate
composition). ac CO2-CH4 generates values of 1.057 for the period when CO2 reduction is
considered dominant (i) and 1.055 when acetoclastic methanogenesis is considered dominant (ii).
These are very similar and it would be valuable to understand how the authors interpret this when
they infer much wider ranges in ac. For clarity δ13CO2 and δ13CH4 respectively for (i) were -8.3 ‰
and -62‰ , and for (ii) were 1.55‰ and -47.5‰
**This is also an interesting result. We think there could be some complicating factors that**
**influence $\delta^{13}$C-CO$_2$ in this study in particular. We note that the headspace concentration of**
**CO$_2$ decreased through the experiment, which would not be the expected stoichiometric**
**result of a net shift from hydrogenotrophic to acetoclastic methanogenesis. This suggests**
**that there were additional sinks of CO$_2$ in the experiment that became more prevalent as**
**the experiment proceeded, and this may have led to the observed enrichment in $\delta^{13}$C-CO$_2$.**

**In particular we are curious about the possible role of increased homoacetogenesis,**
**although this is difficult to evaluate based on the results of the study.**
**Overall, we do not think this finding necessarily negates the use of $\alpha_C$ as an indicator of**
**differences in methanogenic pathway, which is supported by other studies (e.g. Penning et**
**al., 2006a,b; Galand et al., 2010). But it does point to the potential for other variables to**
**complicate the relationship between $\alpha_C$ and the relative proportion of different pathways.**
**In the revised manuscript we will highlight these complications to some extent, including**
**citing this paper. See major revision 4.**
Abstract: Is clear and summarises the paper but projects a future methane emissions scenario
(L25-26) before the modelling and assessment of how well this approach can reconstruct current
estimates ( L27-30) and this seems in the wrong order to me, given the former has a reliance on
the latter. Further, the abstract does not acknowledge this research is augmenting the research that
historically first documented the global relationship between $\delta D\text{-}CH_4$ and $\delta D\text{-}H_2O$ easily
addressed for example by changing L12 to 'We have refined the existing global relationship
between $\delta D\text{-}CH_4$ - $\delta D\text{-}H_2O$ by the compilation of a more extensive global dataset…."
**We agree with these suggestions, and we have revised the abstract accordingly. We note**
**that based on the suggestions of the other reviewers there will be other changes to the**
**abstract, including a modification of the description of the upscaling component of the**
**manuscript. In particular we have omitted a discussion of future emissions scenarios, as this**
**was highly speculative.**
L28: The authors postulate the mismatch is dependent only on the work of others (emission
inventories, etc) and not possibly an error in their approach. Scientifically this is not correct –
both 'sides' could have errors.
**We have changed this language, and have focused on possible errors in our analysis, and**
**errors in isotopic signals generally.**
L19: results do not imply; one interprets data to generate a 'result'.
**This line was deleted.**
L22: high (more [13]C-enriched) in rivers and bogs - this is the dataset that has more $\delta D\text{-}H_2O$
projected, so is this an artefact of the modelling than a real biome-specific difference?
**We are not sure what the reviewer means by 'this is the dataset that has more $\delta D\text{-}H_2O$**
**projected.' 81% of bog sites have $\delta^2 H\text{-}H_2O$ measurements, while 37% of river sites have**
**this measurement. For the dataset as a whole the percentage is 48%.**
**As discussed above (Major Revision 3) we carefully assess the use of modeled precipitation**
**$\delta^2 H\text{-}H_2O$, and find it does not have a major impact on the regression relationship between**
**$\delta^2 H\text{-}H_2O$ and $\delta^2 H\text{-}CH_4.$ Therefore we do not believe this result is an artifact. But the result**
**is not significant, and therefore remains preliminary. Regardless, we have revised our**
**analysis of differences by ecosystem using the 'best-estimate' $\delta^2 H\text{-}H_2O$ and resulting**
**$\delta^2 H\text{-}CH_{4,w0}.$ We also provide an additional analysis using only sites with measured**
**$\delta^2 H\text{-}H_2O$, shown in Supplemental Figure 1.**

L27: integrated (by mass balance) not combined (which is used when sources are added) – which
I know the authors have done (L204) but the descriptor is incorrect here.
**This line was deleted.**
L36: I think the following references is missing: Variability in Atmospheric Methane From Fossil
Fuel and Microbial Sources Over the Last Three Decades. / Thompson et al: Geophysical
Research Letters, Vol. 45, No. 20, 28.10.2018, p. 11499-11508 (and I invite the authors to
wonder if also some of the work from the Royal Holloway group should augment L47-51)
**We thank the reviewer for this suggestion, and we have added the suggested reference and**
**citations to other work from the Royal Holloway group in the introduction.**
L59 & L83 Citations are given in chronological order of 1999b and 1999a which seems not
typical convention to me (uncertain of the referencing convention for BG but for example the two
references for Walter K are not in chronological order in the reference list so the in-paper
citations would not be b then a due to this convention in the reference list?)
**We were relying on EndNote for citation management, and there were errors with the**
**citation format in the software. We have corrected this.**
L68: Logic only follows that impact on $\delta^{13}CH_4$ can affect geographic provenancing if reader
knows it can also affect $\delta D$-$CH_4$, so does this need to be made explicit?
**We do not understand the reviewer's comment here. The hypothesized geographic variation**
**in $\delta^{13}C$-$CH_4$ is independent of variation in $\delta^2H$-$CH_4$, as they are controlled by different**
**mechanisms. We will make this clearer in the revised text.**
L70: this implies that different ecosystems have different methanogenic pathways. More accurate
text would be "differentiated geographically based on ecosystem differences in the relative
strengths of different methanogenic pathways and $\delta^{13}C$ of source organic matter" (as per the
introduction of the Ganesam paper). Noting relative strengths is important, as a common mistake
propagated in the literature and again here (L???) is to assume methanogenesis proceeds by one
methanogenic pathway only – this would be rare, with field-based methane production
contemporaneous from $CO_2$ and acetate, and varying temporally in strength as input of fresh OM
changes seasonally (or not).
**The reviewer raises an important point here, and we have made it clear that the difference**
**is in the relative strength of the pathways operating in difference ecosystems, and not**
**different pathways per se.**
L84-85: sounds a bit defensive? How about "We have advanced existing compilations of
freshwater $\delta D$-$CH_4$ by 1,2,3 …? I would remove significantly (statistical connotations) and just
say larger as the number speak for themselves.
**We agree with this suggestion and have made the suggested change.**
L91: The aims are clear (good) but 'then' and 'potential' not needed – the latter as embedded in
implications that there is a potential for impact
**We have made this change**

L106 & L117, 9L206 and possibly elsewhere): small w for where, as this follows from an
unfinished sentence in both cases with the equation used in between
**We have made this change**
L136: the five ecosystem categories are not clear from this sentence: 'lakes' and 'rivers' and then
there are five wetlands listed. Further, it is debatable that floodplains are aligned with rivers as
$CH_4$ production would only occur when sediments are deoxygenated from standing water. So I
would say more with ponds as the recession of water can be slow and could be like a pond drying
in some situations. Noteworthy here is that gas loss from rivers is velocity dependent (see Long et
al (2015) Hydraulics are a first order control on $CO_2$ efflux from fluvial systems Journal of
Geophysical Research – Biogeosciences, 120, (doi:10.1002/2015JG002955), and similar
references. This will also be the case with methane – possibly more so as insoluble, and may
cause an isotope fractionation independent of degassing, and may also be a reason the Amazon
rivers in Fig. 5 plot differently.
**We note that there was an error in this text and it is actually six categories. We have added**
**a numbered list to make this clearer. We note that essentially all of the river data come**
**from floodplain lakes or deltas, with one exception, and most of the data are from the**
**Amazon. We think it is valid to continue to differentiate these environments from other**
**lakes and ponds, since they are different from typical lakes and ponds in a number of ways**
**(overturning and redox regimes, nutrient inputs, dynamics of gas loss and hydraulics). The**
**reference on gas loss being velocity dependent is interesting, but we doubt this process has a**
**large effect on our dataset since, as mentioned above, very few data are from fluvial systems,**
**and of those data almost all are from low-velocity environments like floodplain lakes or**
**deltas.**
L139: Similarly, I question the scientific integrity in lumping lakes with rivers here – gas loss
from river systems is controlled by hydrological processes primarily and there could be
fractionations during emission from lotic systems that are different to lentic systems where
diffusion and wind of lake thermal orographic processes control turnover. This starts to become
important where these mean sources are used to simulate a resultant atmospheric composition e.g.
L227. Thus, the authors should think about how to provide added confidence of the robustness of
their catergorisation.
**See above. We are primarily analyzing floodplain lakes and deltas in the river category. We**
**make this clearer in the revised manuscript (line 143)**
**However, the basis of this categorization is essentially to align with flux inventories (Saunois**
**et al., 2020), which specifically define an 'inland water' category that includes rivers and**
**lakes, as well as reservoirs. We keep to this categorization in order to be able to compare**
**with the flux estimates. We do discuss possible differences between the river and lake sites**
**in Section 3.6 of the revised manuscript.**
L145-148: Such categorisation is good, and the open access data set is very welcome. This
categorisation relies on the integrity of the interpretation, but this integrity is important as the data
analysis relies on this. With 131 sites it is impossible for the reviewer to know each site and so as
a 6 check I can only look at my own data: L61 in the excel files. These methane samples were
collected in-situ from porewater diffusing into samplers embedded in the peat (the GBC abstract
notes in-situ and the methods clarifies at depth sampling) so I would classify as more aligned with dissolved porewater than diffusive flux (which is normally associated with the potential for
oxidation and change in δ _v_a_l_u_e_s_ _). Further I comment in the GBC paper there is a
dynamic zone and interpret that is the section from which gas can be emitted. Mean δD-CH$_4$ here
is -332 ± 17‰, more depleted the -294 ± 39‰ used in the table and subsequent data analysis.
Thus, some feedback from the authors in the revised manuscript that their interpretations are not
sensitive to the variation their interpretation of environment and which data to use would be
valuable.
**The reviewer raises a valid point about the complexities of each site. This complements the**
**comments of reviewer 3 about the validity of including data from deep peat samples. As**
**discussed in our response to reviewer 3 we think this is a valid concern and we have decided**
**to limit our data from peatlands to the uppermost 50 cm. See Major Revision 1. This**
**coincides approximately with the dynamic zone mentioned by reviewer 1.**
**We note that peatlands (bogs and fens) were the only environments to be sampled on depth**
**gradients, and that similar issues are unlikely to affect the interpretation of data from other**
**ecosystem categories. We will change the entry in the Table for Waldron et al. 1999b to**
**Dissolved-Pore Water. Since we group diffusive flux and dissolved pore water, this**
**distinction does not make a difference to our analysis.**
L152 – typically small – as this manuscript relies on several source of data estimation (here,
δ$_2$H$_2$O, it would be good to provide estimates as to what the maximum is this would manifest in
δD (recognising that it changes with resolution and scale of figure and so this is challenging, but
saying small is insufficient).
**We have provided a quantitative estimate of the likely error produced by this analysis, as a**
**percent error, and then translate that to $\delta^2H$ values (lines 166-168)**
L177: the authors need to unpick for the reader the statement more as they have with L179
onwards. I am thus left to interpret the reasoning. I assume it is based on considerations that
methanogenic pathway influences δD-CH$_4$? If so please see earlier substantive comments on this
and decide whether to proceed in the revised manuscript.
**Based on the reviewers comments here and below we have omitted this section from the**
**methods, and instead discuss likely effects of different biogeochemical variables on $\delta^2$H-**
**CH$_4$ in Section 3.4, with more detail provided in the Supplemental Text. In genral**
**we place less emphasis on methanogenic pathway in the revised manuscript. See**
**Major Revision 4 above.**
L200: Clarify where the flux estimate comes from at this point – I presume from Saunois et al as
in L209, but this should be clarified when first introduced. I am not expert enough to judge if the
methodology for the bottom up flux section is sound, but it seems reasonable to me.
**We have clarified the source of the flux estimates earlier in this section. It is indeed Saunois**
**et al., 2020.**
L 267: given the statistical approaches such as Monte Carlo bootstrapping used with the flux
estimate section previously I would have expected more rigorous comparison should be
undertaken here to show if there is a statistical offset between measured and predicted δD-H$_2$O
than relying on descriptors of "generally good agreement" and using RMSE. The RMSE is a red herring if the lines generating 19 and 23 ‰ do not overlap - ?
**As discussed above, we now provide a more detailed analysis of the comparison of measured**
**and modeled $\delta^2$H-H$_2$O. See Major Revision 1.**
Fig 2: Should the predicted (postulated and therefore dependent) not be regressed onto the
measured (the true field value, so measured and independent and as a control of $\delta$D-CH$_4$ the one
to get as close to the true value as possible)?
**This depends on the goal of the regression. In this case we are attempting to develop a**
**regression relationship that predicts measured $\delta^2$H-H$_2$O as a function of modeled**
**precipitation $\delta^2$H-H$_2$O, in order to use this as a predictive tool for sites without $\delta^2$H-H$_2$O**
**measurements, and to assess the goodness of fit. Therefore it makes sense to have**
**measured $\delta^2$H-H$_2$O on the y-axis in this case, and we have kept this orientation for the**
**revised Figure 2.**
Fig 3B: this needs revisited once the $\delta$D-CH$_4$ -$\delta$D-H$_2$O predicted data has been removed as
described above. There may still be an inland water specific difference here, but again that this
may not be controlled by anything more complex than lentic and lotic freshwater systems having
generalised differences in gas transport mechanism (ebullition or diffusion). These would be
influenced by atmospheric and sediment interface boundary layer dynamics, transit time, depth of
oxidative zone, lake stratification, and surface roughness, with the latter in turn influenced by
wind speed, depth of water, and river flow velocity, slope. In other words, considerable methane
isotope fractionation (enrichment) is possible, or not.
**Based on our revised analysis, we find that we cannot detect a significant difference in the**
**regression relationship between inland waters and wetlands (See Major Revision 3 above).**
**The difference inferred in the original manuscript is likely partly a result of the**
**hydrological differences in these environments, and resulting differences in the regression**
**of modeled vs measured $\delta^2$H-H$_2$O (see Major Revision 2). Therefore we revised this section**
**of the manuscript to reflect this revised understanding (now section 3.3).**
Fig 4. It is good to see this plotted but not surprising given $\delta$D-H$_2$O varies with latitude and $\delta$D-
CH$_4$ varies with $\delta$D-H$_2$O. The same difficulties in estimating field $\delta$D-H$_2$O from modelled $\delta$D-
H$_2$O are evident when considering $\delta$D-CH$_4$ as a function of predicted $\delta$D-H$_2$O. The authors need
to note here that there may be an imbalance of where methane is sampled from globally and so if
more measurements existed from the higher latitudes then there may be as much scatter as with
the lower latitudes.
**We are glad the reviewer agrees with us on the utility of plotting the data in this way. We**
**did note the uneven geographic distribution of data at several points in the manuscript, but**
**further emphasize the likelihood of similar scatter at all latitudes with more sampling in the**
**revised manuscript. We are assuming the reviewer meant to say there is greater scatter at**
**high latitudes, which is what we observe.**
Section 3.4 jumps to something completely different with L313 "shifts to being controlled by
changes in methanogenic pathway to being controlled by ….". There has not been clear
discussion from the authors to date they are considering changes in methanogenic pathway of $\delta$D-
CH$_4$ so this seems out of context. And yet L317 goes on to consider this in more detail. The key
message in the Waldron et al 1999 paper is that considering methanogenic pathway a control on

δD-CH4 is misplaced and that "that 50% of the variation in natural δD-CH4 samples can be
explained by δD-H2O, with isotopic fractionation post-production, or mixing with gas already
fractionated likely responsible for most of the noise in the natural system". The analysis prior to
section 3.4 may be more likely to support this interpretation than refute it, particularly when the
data in Fig. 3.2. is appropriately compared (as described earlier), and so now considering data as a
function of methanogenic pathway seems to be ignoring this. Indeed the authors observe they find
no relationship between $\delta^{13}C\text{-}CH_4$ and $\delta^2H\text{-}CH_{4,W0}$ which would be expected if $\delta^2H\text{-}CH_4$ was
influenced by methanogenic pathway as $\delta^{13}C\text{-}CH_4$ is (Fig. 5a). Thus, the authors should not make
clearer statements such as L312 of "shifts from being controlled by variation in methanogenesis
pathway" are inferred controls.
**This is clearly a key point of concern for the reviewer, and we understand the reservations**
**about inferring that variability is a function of methanogenic pathway. However, we think it**
**is unlikely that all of the remaining variability not explained by $\delta^2H\text{-}H_2O$ is controlled by**
**"isotopic fractionation post-production, or mixing with gas already fractionated". First, it is**
**important to be clear about what these post-production processes are. To our knowledge**
**there are two key post-production processes that can affect methane isotopic composition:**
**methane oxidation (either aerobic or anaerobic) or isotopic fractionation caused by**
**diffusive gas transport. We are unaware of other important processes. Both of these**
**processes would be likely to lead to higher $\delta^2H\text{-}CH_4$, and lead to positive co-variation with**
**$\delta^{13}C\text{-}CH_4$. Oxidation will also lead to negative co-variation with $\alpha_c$, because $CH_4$ is**
**invariably oxidized to $CO_2$, leading to a smaller isotopic difference between these gases.**
**Diffusion will also lead to negative co-variation with $\alpha_c$ (See Revised Figure 6). Mixing**
**effects will depend on the mixing end-members. Unless there is a large proportion of non-**
**microbial methane present, which we argue is unlikely in most circumstances, mixing will**
**not alter the overall isotopic signature of microbial methane in the ecosystem. It is possible**
**to have mixing with 'gas already fractionated', but in this case the underlying fractionation**
**is the key process controlling the isotopic composition of the resulting gas, and again to our**
**knowledge this would have to be the result of oxidation or diffusion.**
**It is unclear on what basis Waldron et al (1999) ascribed the remaining ~50% of variability**
**in $\delta^2H\text{-}CH_4$ to these post-production processes, and we would argue that this assertion is**
**untested.**
**We do agree that our focus on methanogenic pathway did not include other plausible**
**mechanisms for co-variation between $\delta^2H\text{-}CH_4$ and $\alpha_C$. We have expanded our discussion**
**to take other processes, namely (i) diffusion and (ii) differences in enzymatic reversibility,**
**into account in Section 3.4. Ultimately our revised conclusion is that these processes or**
**variables cannot be clearly differentiated on a global scale on the basis of isotopic data.**
Figs. 5b=c. The uncertainty around what $a_c$ should be for different methanogenic pathways has
been described earlier in this review. But additionally, although breakpoint analysis was used,
there is a high dependence in this on data set that has enriched $\delta^2H\text{-}CH_4$ to generate opposing
trends. The eye is drawn by the projected pathways, but if these was not included as we cannot be
sure it is oxidation[1] and all the remaining data was considered in a weighted regression would
there be trends?
[1]If the high $\delta^2H\text{-}CH_4$ is from the Amazonian rivers, there are shales in this basin that fuel C
cycling (Vihermaa et al) and this could be thermogenic: $\delta^2H\text{-}CH_4$ is also consistent with this.

Vihermaa L.E., Waldron S. , Garnett M.H., and Newton J. (2014) Old carbon contributes to aquatic emissions of carbon dioxide in the Amazon. Biogeosciences, 11, 3635-3645. (doi: 10.5194/bgd-11-1773-2014).

**We acknowledge concerns about the 'predicted trends', both by reviewer 1 and 2, and therefore we will remove them from the revised manuscript. We will instead focus on the patterns of co-variation, and potential explanations for them. We do present approximate vectors of isotopic co-variation for four different biogeochemical variables (mentioned in Major Revision 4), but emphasize these are approximate and imprecise. We think they are valuable to indicate the direction and likely magnitude of co-variation. As discussed above (Major Revision 4) we will focus less on methanogenic pathway, and increase our focus on other mechanisms.**

**We agree that the one outlying point with very high $\delta^2$H-CH$_4$ (and $\delta^{13}$C-CH$_4$) is questionable, and may be thermogenic methane. It is indeed from the Amazon. We therefore removes this from our dataset (Major Revision 1)**

**As noted above, weighted regression is leading to biases in this analysis, and is not generally preferable to unweighted regression (Fletcher and Dixon, 2012), and therefore we used unweighted regression in the revised manuscript.**

It is remarkable Fig 7 is so consistent – this is very interesting. Is it what we would expect?

**We assume the reviewer is referring to Figure 7B. This is not necessarily what we would expect based on other studies. We already provided some discussion of this in section 4.4 of the original manuscript, but have revised this in response to questions from reviewer 2, especially focusing on possible biases in the $\delta^{13}$C-CH$_4$ dataset. See Major Revision 5, and the revised Section 3.5.**

L370 discussion is over-interpretations given the differences between sites are not statistically significant. It would be ok to say the prevalence of more depleted CH$_4$ is greater in the ecosystems sampled but for example this could represent accessibility of field sites, or differential investment into research measurements in these areas, than group compositional differences per se. Ecosystem types are not evenly distributed by latitude (L370) – nor is resource for investment in field research with tropical regions of the Earth lacking measurement due to access or financial constraints – we need to start recognising what we have not measured is as important as what we measure.

**We agree that this analysis is preliminary given the small sample sizes for each ecosystem. We emphasized this in the original manuscript, and noted that the possible differences represented hypotheses that merited further testing. We further emphasize this uncertainty in the revised manuscript. We emphasize that more investigation of tropical ecosystems is especially important, namely in Section 3.8.**

Fig. 10 is tiny and needs to be bigger

**We have revised Figure 10 to simplify it based on comments from reviewers 2 and 3 on the upscaling exercise. We have reduced it to a single panel (equivalent to Figure 10C), which makes it more legible.**

L426 "roughly as strong a predictor". Too big a leap: explain how – from ice core gases "roughly is a colloquialism"

**We do not fully understand this comment, but we agree that this language is imprecise, and we have changed this (Section 3.3).**

L487 – as noted earlier, the paired measured values plot on Waldron et al 1999 In-vitro line, consolidating further the significant of this line. Please acknowledge this.

**We are assuming the reviewer meant the in-vivo line here. As discussed above, we have provided a more thorough comparison of the Waldron in-vivo line with the results of this study, and have represented this in the revised Figure 3 and Section 3.3. Our analysis (Figure 3B) shows that the paired measured values do not plot on the Waldron et al (1999) in vivo line, and have a flatter slope. This difference is not significant based on ANCOVA, but we infer that the larger dataset consistently implies a flatter global slope.**

L508 – in the revised manuscript please detail the % variation explained by $\delta D$-$H_2O$ and then additionally by $a_c$ should this prove to still be important

**We have decided to omit the multivariate regression given the complications described in Major Revision 4 above.**

L510 – this is the crux of what is new to explore in isotope biogeochemistry of methane and also the role of methanol substrates.

**We agree with the reviewer that $CH_4$ isotopic variability related to enzymatic reversibility is an important topic, and based on other comments we have expanded discussion of this in the revised manuscript, especially in Section 3.4. At this point there is little we can say about methanol substrates, but we mention it briefly as another variable that merits consideration in Section 3.4.**

L519 – same comments as before about is there really a relationship, but why more points classified as oxidised with this pairing than with $a_c$?

**This is an interesting question, and we don't know the answer. We would speculate that it is because sources of $CO_2$ can be very variable, and this may be adding noise to the original Figure 5C that is not present in figure 5B. As discussed above we have substantially revised this aspect of the paper (Major Revision 4) and no longer attempt to differentiate samples influenced by oxidation.**

L551- Much of 4.31. is repeating statements first described in Waldron et al 1999 section 1.1., paragraph starting "In addition…" but this is not referenced and as written implies the authors are the primary source of this thinking. This is not the case and should be referenced appropriately to indicate this was first noted 20+ years ago.

**We regret that we did not acknowledge the earlier statement of these ideas. We have thoroughly revised this discussion to provide credit to Waldron et al., (1999) for the ideas that are presented there, which now appears in Section 3.3.1**

L564 – please note pure cultures are not representative of the field processes of methane
production and thus the batch cultures and other experimental data collated in Waldron et al 1998,
1999 are. This is not clear from the statement.
**We agree that pure cultures are not representative of methanogenesis in the environment.**
**We are not sure that batch cultures or incubations are truly representative either, in that**
**they do not necessarily fully represent the processes occurring in natural environments, but**
**agree they are clearly a closer approximation than pure cultures. We did try to make this**
**distinction clear in the original manuscript, and have further clarified in the revised**
**manuscript (Section 3.3.1). However, we do feel that inferences from pure cultures are**
**important for understanding the more complex processes that occur in batch cultures or**
**natural environments. For example, a very flat slope for $\delta^2$H-H$_2$O vs. $\delta^2$H-CH$_4$ is**
**observed in pure culture experiments with acetoclastic methanogenesis (i.e. Gruen et al.,**
**2018; Valentine et al., 2004). This implies that acetate methyl hydrogen is not fully**
**equilibrated with water during the methanogenesis reaction itself. This is reflected in the**
**revised Figure 5, and Section 3.3.1.**
L569, please reverse the order of the references or remove Whiticar 1999. The Waldron 1999
paper is the one that is particularly focussed on the global relationship between $\delta$D-CH$_4$ -$\delta$D-H$_2$O,
and constructs the first global relationship, which this paper finds with new data is similar. This
gives appropriate credit to the conceptual understanding. The Whiticar paper coplots $\delta$D-CH$_4$ -$\delta$D-
H$_2$O but does not assert that " $\delta_2$H-H$_2$O is a primary determinant of $\delta_2$H-CH$_4$ on a global scale",
rather the focus is on the interpretation of how $\delta_2$H-CH$_4$ reflects methanogenic pathway or marine
vs. freshwater.
**There seems to be an error in the page numbering, and we are not sure which citation the**
**reviewer is referring to. However, we have made clear in the revised manuscript that**
**Waldron et al., (1999) first proposed and found evidence for the global relationship between**
**$\delta$D-CH$_4$ -$\delta$D-H$_2$O.**
To conclude: this has been an uncomfortable review for me to undertake as my position of not
anonymising the review puts me up for public scrutiny, and a misinterpreted that I am trying to
defend my work and am unwilling to accept an addition to this. This does not represent my
professional scientific principles, I would urge the authors to accept this is not the case - indeed in
the 1999 GCA paper I welcome refinement of my work. However, the authors have still not
presented here compelling evidence that $\delta$D-CH$_4$ can represent well different methanogenic
pathways and so the reliance of this in the manuscript I find troubling. I consider the a$_c$ approach
may be valuable in helping constrain the signal in $\delta$D-CH$_4$ that is not defined by $\delta$D-H$_2$O, but the
current manuscript is not constraining uncertainty sufficiently and the approach is therefore
flawed. I would urge the authors to find a way to better constrain projected $\delta$D-H$_2$O and revisit
this, or work with only measured data and revisit this. Their refined analysis should undertake
rigorous statistical comparison with the existing field $\delta$D-CH$_4$ -$\delta$D-H$_2$O relationship from
Waldron et al 1999 to say whether it is different (although the new larger dataset will likely be a
more representative relationship that the community can go forward with), and adopt a
parsimonious interpretation of variation within the data set, as that is least likely to induce an
erroneous interpretation. The biome specific considerations and upscaling should also be revisited
if the removal of biased and inaccurate data pairings changes the source bulk compositions, and
further thought should be given to the basis for source differentiation based on scenarios of
methane production and loss in this upscaling.

Once again, we regret that this has been an uncomfortable review process. We appreciate the frank and detailed signed review, and the collaborative nature of the comments. We agree with many of the suggested improvements to the manuscript, and have made these changes in a thoroughly revised manuscript. See the Major Revisions above for a summary of these changes.

We believe that these changes will address the reviewer's concerns and will greatly strengthen the conclusions of the manuscript.

*Specific Responses to Reviewer 2:* Reviewer comments are in plain text. **Responses are in bold text.**

The paper investigates the relation between the hydrogen isotopic composition of methane emitted from freshwaters on the global scale and the isotopic composition of water and/or modeled precipitation, as well the carbon isotopic composition of methane and carbon dioxide. The authors analyze data from a large number of previous studies and apply statistical methods in order to evaluate correlations between the various signatures. The statistics are applied in a straightforward manner.

**We thank the reviewer for their assessment.**

I am missing a more detailed/critical scientific analysis of differences between the results of this study and previous studies. This has two aspects: 1) The study uses more sites than previous studies for dD, and it uses modeled fields of dD in precipitation. Which of these differences is primarily responsible for the differences to the previous literature (or is it both)?

**This is a good question and similar questions were raised by reviewers 1 and 3. In response to these questions we present a much more detailed comparison of the previous literature (Waldron et al., 1999a) in comparison with our study. See Major Revisions 2 and 3 for more details on this. The short answer is that regardless of which water isotope values are used, our dataset produces a flatter slope between $\delta^2$H-H$_2$Oand $\delta^2$H-CH$_4$ than that of Waldron et al., (1999). However, analysis of covariance (ANCOVA) indicates this difference in slope is not significant. We ascribe this apparent difference to the inclusion of many more sites from high-latitude environments in this study. Our analysis is that the relatively small number of high-latitude sites analyzed by Waldron et al., (1999a) were biased toward relatively low $\delta$2H-CH4 values.**

2) The study uses less sites than previous studies for d13C. Are the results from these sites still adequate to be used in a global extrapolation?

**These are important points for clarification. However, we disagree that this study uses less sites than previous studies for $\delta^{13}$C-CH$_4$. See our comments on Major Revision 5. We noted that the dataset was not comprehensive for $\delta^{13}$C-CH$_4$ (i.e. it does not include all published data), whereas it is comprehensive for $\delta^2$H-CH$_4$. However, our $\delta^{13}$C-CH$_4$ dataset for freshwater environments is substantially larger than the largest previously published dataset that we are aware of (Sherwood et al., 2017). We include $\delta^{13}$C-CH$_4$ data for 129 freshwater sites, whereas the database of Sherwood et al. (2017) included 48. Of these, 16 are included in both databases. In order to make our $\delta^{13}$C-CH$_4$ analysis more accurate we now include all sites from Sherwood et al., ( 2017) in our analysis of $\delta^{13}$C-CH$_4$ variability. This expands the number of sites included to 161. There is a clear need for a larger effort to compile freshwater CH4 $\delta^{13}$C-CH$_4$ data into a comprehensive database, but such an effort is beyond the scope of this paper. We highlight the importance of this for future research in our revised manuscript in Sections 3.5, 3.6, 3.7, and 3.8.**

The derived global average 13C source signature derived by the authors is almost certainly too light, given what we know about the fractionation in the sinks. Furthermore, I think that the errors assumed for the bottom-up determination of the global average the source signatures are too optimistic, and the discussion on the implications for the atmospheric isotope budget in section 4.6 and too simplistic. See detailed comments below.

**We agree that it is too light, which was a key point of our analysis in the original manuscript (Line numbers 617-638). Based on the comments of reviewer 2, as well as reviewer 3, it is clear that the upscaling exercise in the original version of the paper is too speculative. However, we also feel that a more detailed upscaling exercise is beyond the scope of this paper, which as mentioned by Reviewer 1 is long and ambitious in scope. We think it is still worthwhile to perform the mixing model calculations for global methane source isotope signatures, and to compare these with previous estimates. See Major Revision 6 for more details on this. Our revised analysis (Section 3.8) focuses on the likely sources of error or bias in isotopic source signatures, and make recommendations to improve isotopic source signal estimates. We disagree in general that our uncertainties for the isotopic source signatures are too optimistic. We will provide more details on this below.**

L37: I suggest citing Worden et al., 2017, where this point is shown particularly well.

**We thank the reviewer for bringing this to our attention, and we have cited this paper and modified the text accordingly (line 41).**

L64: Maybe you want to include here, or later in the discussion section, that there are also other lines of evidence that the hydrogen isotopic composition of CH4 (and other trace gases) depends on the isotopic composition of the precipitation, e.g., CH4 from biomass burning across climatic zones (Umezawa et al.2011), CH4 produced by UV irradiation of leaves that were grown with isotopically distinct waters (Vigano et al., 2010) or molecular H2 produced in the combustion of wood from different climatic zones (Röckmann et al., 2010).
**We appreciate this suggestion. We reference these studies in the introduction and**
**two of them in the results (Section 3.3.1)**
L109: Replace the factor 1000 by 1, the delta value is defined the correct way in line
105, and no factor 1000 is necessary.
**Thanks for this reminder, we have made the suggested change.**
L136: What are the 5 categories? This is not clear, to me it sounds like 4 categories.
**The list of categories is clarified in the manuscript with a numbered list. In fact it is**
**six categories: 1) lakes and ponds; 2) rivers and floodplains; 3) bogs; 4) fens; 5)**
**swamps and marshes; and 6) rice paddies.**
L159: Is the annual average dD value of precipitation really the best estimator for a
source that very likely has a strong seasonality?
**This is an important question, and given this comment as well as those of reviewer 1**
**clearly needs more attention. See our detailed comments on Major Revision 2 that**
**discuss this at length. In short, we take seasonality into account in our revised**
**manuscript, and we find that it is important for inland water environments in**
**particular.**
L253, Figure 1: Many of the sites are hidden behind others so I cannot see the colors.
Would this improve if the figure is enlarged? It may be useful to show by color or shape
for which of the sites you have measured dD-H2O and for which not.
**This is challenging because many of the sites are very close to one another, and it is**
**difficult to resolve the individual sites, while also showing the global distribution.**
**We included the colors to give a sense of how the values vary globally, but for a**
**more in-depth picture of geographic variability Figure 4 is probably more useful.**
**To respond to the reviewer's comment we provided a higher resolution map of**
**North America, which encompasses the majority of sites. We also show the sites**
**with water $\delta^2H$ measurements with a different shape, (i.e. a triangle).**
L244, Table 1: The d13C signatures for wetland have an opposite "latitudinal order"
compared to what is usually assumed, i.e. they are higher at high latitudes and lower
at low latitudes. The data in Table 1 for wetlands do not agree with the data presented
in Figure 7. Please explain the difference. You mention that the dataset evaluated
here is different from what other studies have used for d13C, so is your dataset now
representative? Should this limited set of values be used in the upscaling later? The
errors presented for the different source categories are too optimistic, especially for the
fossil sources at the bottom of the table, but probably also for the wetland category.

**The reviewer raises some key aspects of the table that are not clear.**

**The opposite order of the $\delta^{13}$C-CH$_4$ data in the wetlands is simply what the data**
**indicate. The uncertainties overlap, and our analysis therefore implies that we**
**cannot confidently infer a latitudinal difference in $\delta^{13}$C-CH$_4$ in wetlands based on**
**currently compiled data. This is also shown in Figure 7. We note here and elsewhere**
**in our response that there is an important absence of data from C$_4$ plant cosystems**
**in this dataset and other databases. Including more data from such ecosystems**
**would probably lead tropical sites to have a higher $\delta^{13}$C-CH$_4$ value. We discuss this**
**in more detail in the methods, and the results/discussion. As discussed in Major**
**Revision 5, we include additional $\delta^{13}$C-CH$_4$ data from Sherwood et al., (2017).**
**However, this does not change the observation of no significant latitudinal**
**differences in wetland $\delta^{13}$C-CH$_4$ values.**

**The differences between Table 1 and Figure 7 are a result of the Table presenting**
**mean values, whereas the original Figure 7 presented median values. We presented**
**mean values in Table 1 because it is simpler to express uncertainty for the mean,**
**and because when thinking about atmospheric contributions we think the mean is**
**the best estimate of the isotopic source signal. In boxplots like Figure 7 it is more**
**common to depict the median value. However, to avoid confusion and for the sake of**
**comparison we now also plot the mean and its standard error in Figure 7 (and also**
**do so in Figures 8 and 9).**

**It is not clear to us what the reviewer means when they say the errors are too**
**optimistic for the fossil fuel categories. The error estimates are 95% confidence**
**intervals for the mean values for these categories based on the fossil fuel database of**
**Sherwood et al., (2017). We consider the 95% confidence interval of the mean to be**
**a well-established metric for characterizing the uncertainty in the mean value of**
**these sources. We have categorized the fossil fuel sources slightly differently than**
**Sherwood et al., (2017), to align with the emissions categories of Saunois et al.,**
**(2020), but our uncertainty estimates are essentially the same as, and actually**
**somewhat larger than, those of the original study (see Table 5 in Sherwood et al.,**
**2017). Note Sherwood et al., (2017) presents standard errors of the mean. 95% CI is**
**derived by multiplying this value by 1.96. In addition, our uncertainties for the**
**$\delta^{13}$C-CH$_4$ source signal for fossil fuels is very similar to those used by Worden et al.,**
**(2017). Without further details, it is unclear why the reviewer considers these error**
**estimates to be too small or optimistic.**

**We used the same approach in our estimates of uncertainty in the wetland source**
**signatures, and other source categories, and therefore also disagree that these**
**estimates are too optimistic.**

L276, Fig 2 and related text: This is a key figure for the following analysis. In principle
it is an interesting approach to use modeled dD values in case measurements are not
available, but it is also a source of error. Although there is a generally good agreement, the slope is lower than 1 and this may contribute to the differences and thus may affect
some of the further analysis.

**We agree this is a key figure and requires more in-depth analysis, which we provide**
**in the revised manuscript. See our Major Revision 2. We agree with the reviewer**
**that the slope being lower than 1 is concerning. In our revised analysis we find that**
**applying annual precipitation $\delta^2$H to wetland environments, and growing season**
**precipitation $\delta^2$H to inland water environments, results in slopes that are within**
**error of 1.**

L284: Maybe you could state briefly whether you can reproduce the slope of Waldron
et al. when you use the same dataset. Just as a baseline.

**This is a valuable suggestion. Please see our response in Major Revision 3. We have**
**included a much more careful comparison of our dataset with that of Waldron et al**
**(1999). It is important to note that the analysis of Waldron et al. (1999a) also**
**included key assumptions that influence the regression relationship produced with**
**that dataset. Specifically, that study included sites with measured water $\delta^2$H (57%)**
**and sites with estimated water $\delta^2$H based on regional precipitation measurements**
**(43%). To perform a robust comparison we re-analyze the Waldron et al dataset,**
**which is discussed in our Planned Major Revision 3. Because the exact details of the**
**weighted regression method used by Waldron et al., 1999 are not provided, we did**
**not precisely reproduce their regression relationship [see Supplemental Table 2].**
**But when using unweighted regression we produced a relationship that is**
**statistically indistinguishable. We note that a previous paper that re-analyzed the**
**data of Waldron et al using unweighted regression (Chanton et al., 2006) found the**
**same regression relatioship that we did.**

L292: Figure 3a: It looks like the lower slope is caused by a lot of points where you
have only modeled but no measured dD data near the low dD-H2O end. And these
are mostly inland waters (Figure 3b). Can you evaluate this in more detail? Can this
be caused by a bias in the modeled dDp? Probably not, but it is useful to investigate
further to strengthen your argument.

**The reviewer correctly noted that the reported regression line for inland waters was**
**not a good visual fit to the data, and this influenced the overall regression line. This**
**was also noted by reviewer 1. We note that in the original Figure 4a the two**
**regression lines were very similar, so this effect was not a result of bias in modeled**
**$\delta^2$Hp, since a very similar regression was produced when only analyzing sites with**
**measured water $\delta^2$H-H$_2$O.**

**After analyzing this more closely we realized that this is a result of the weighted**
**regression methods we were using. Specifically, a few high-latitude sites with 1)**
**many measurements (and therefore a low standard error) and 2) high $\delta^2$H-CH$_4$**
**values, were heavily weighted and had a large effect on the regression relationship.**
**We therefore decided that a more accurate regression relationship would be**

**produced using unweighted regression. This is supported by studies on the efficacy**
**of unweighted regression in analyzing environmental data, which in many cases is**
**less biased than weighted regression (Fletcher and Dixon, 2012). See more details in**
**Major Revision 3.**
**The unweighted regression provides a somewhat steeper slope for the overall**
**dataset, as well as for inland waters. It also indicates there is not a significant**
**difference in the regression whether measured $\delta^2$H-H$_2$O or modeled $\delta^2$H-H$_2$O, or a**
**combination of the two (i.e. a 'best-estimate') is used. See Major Revision 3, Section**
**3.3 and Supplemental Table 2.**
L308: Would you find a correlation if you took the slope of Waldron et al. for calculating
CH4,W0?
**We have significantly revised this analysis, as discussed in Major Revision 4 above.**
**As described above, we have decided to omit the piece-wise regression analysis.**
**The slope of Waldron et al., (1999) is not a good fit to the overall dataset (Figure**
**3.3), and therefore we do not think it makes sense to apply this to calculate $\delta^2$H-**
**CH$_{4,w0}$.**
L323, Figure 5: Does it make sense that in b) only few points are classified as oxidation
influenced and in c) many more points? Does it make sense that in c) the very lowest
dD value is in the group of the oxidation influenced points? I find the "pathway trend"
concept a bit confusing, this indicates a smooth transition of dD-CH4,W0 with alpha_C
or d13C_CO2. Is this a real trend, or rather a consequence of two different groups
of data (acetoclastic and hydrogenotrophic sites)? Wouldn't it be useful in this case
to show these two groups with two different colors, separated by the potential break
points, rather than the trend areas?
**The reviewer raises important questions about the predicted trends that we**
**presented in Figure 5. Reviewer 1 also raised important questions about this, and**
**given these comments we have decided that we should not present predicted trends.**
**We do present approximate vectors of isotopic co-variation for different**
**biogeochemical variables (Revised Figure 6), but emphasize that these are guidelines**
**of the sign and magnitude of isotope effects and should not be interpreted as precise**
**predictions.**
**Our revised analysis focuses on the overall co-variance (or lack thereof) between**
**$\delta^2$H-CH$_{4,w0}$, $\delta^{13}$C-CH$_4$, $\alpha_C$, and $\delta^{13}$C-CO$_2$, and multiple mechanisms that could**
**influence this co-variation in freshwater ecosystems. Our ultimate conclusion is that**
**patterns of co-variation cannot definitively resolve which mechanisms for $\delta^2$H-CH$_4$**
** are most important when comparing between sites. See Section 3.4.**
L350 and Figure 7b, wetlands: These numbers do not agree with the data in Table 1.

**As noted above, these are median values, whereas Table 1 presents mean values. To**
**clarify this we will also plot mean values in Figure 7.**

 L374-379: I get a bit confused by the diverging statements on significance with different
tests, please try to reformulate, or add a sentence to synthesize.

**We have simplified this section to clarify that we are focusing on the pair-wise**
**comparison between wetlands and inland waters first and then the multiple group**
**comparison between all ecosystem categories. See Section 3.6.**

L395-397: See points above: Are the uncertainties for the different categories adequate?
Is there an issue with the difference between values in the text and table 1? Is
the rather heavy d13C value for high latitude wetlands appropriate?

**See our response to comments on Table 1. It is unclear what difference between the**
**table and text is being referred to- we assume this is the difference between median**
**values (Figure 7) and mean values (Table 1). The heavy value for high latitude**
**wetlands is the mean value of this dataset, and therefore we argue it is appropriate.**
**In our revised manuscript we include additional data from Sherwood et al., (2017),**
**as discussed above, which includes 5 additional high latitude wetland sites. This**
**makes the mean $\delta^{13}$C-CH$_4$ value 0.5‰ lower, but does not change the median value.**
**We include this value in the revised Monte Carlo analysis, but in essence this**
**additional data does not change our conclusion. Based on our analysis, an**
**assumption of low $\delta^{13}$C-CH$_4$ in high latitude wetlands is not supported by the**
**available data, and we think this assumption requires further empirical validation.**
**But we also note important caveats for this interpretation based on atmospheric**
**measurements (see Section 3.5)**

L431 ff: The differences to the previously published values from Waldron et al. should
be discussed in some more detail. E.g., is there an influence from the modeled dDp
values, or a certain sampling region? L439 ff: Same for the discussion of the environment
type

**See our responses above and Major Revisions 2 and 3. Our conclusion is that the**
**difference is largely controlled by the small number of high-latitude sites in the**
**Waldron et al (1999) dataset, and that those sites were skewed towards relatively**
**low $\delta^2$H-CH$_4$ values. We do not observe a significant difference in the regression**
**relationship when modeled or measured $\delta^2$H-H$_2$O values are used (see Figure 3 and**
**Supplemental Table 2).**

L465, section 4.2.1: See comments above on the representativeness of the dataset
analyzed here and possible consequences. You write that the dataset is not
comprehensive
or d13C, so should it be considered as representative? In this case, what have
other studies potentially missed?

**See Major Revision 5 above. As mentioned above, it is the largest compiled dataset**
**available, but it is not comprehensive because there is a large amount of $\delta^{13}$C-CH$_4$**
**data that has not yet been compiled into a database. It is also probably not**
**representative, with a notable lack of data from C$_4$ plant ecosystems. Given that it is**
**the largest dataset available, we proceed with analyzing it. However, in the revised**
**manuscript we give more attention to the likely sources of error, and key data gaps**
**that should be addressed.**
L483 ff: You may want to refer here to the studies I mentioned in the beginning that
looked at other (non-microbial) sources.
**Thanks for this suggestion, we mention the two studies focused on methane in this**
**discussion (now in Section 3.5.1).**
L519 ff: The authors state that they do not observe a correlation between dD and d13C
of CH4. Nevertheless, the vast majority of the points in Fig 5a seem to fall in the range
of the "pathway trend" (I find the term misleading, see comments above). Does this not
mean that the two groups (acetate fermentation and CO2 reduction) still form distinct
distributions?
**As mentioned above, there are concerns with the 'pathway trend' noted by**
**Reviewer 2, as well as Reviewer 1 and we have decided to omit this from the revised**
**manuscript. Our primary concern is whether $\delta^{13}$C-CH$_4$ is a strong predictor of $\delta^2$H-**
**CH$_4$, and our analysis indicates that it is not. We now note that there is a (very)**
**weak negative correlation when looking at all data. This is consistent with an effect**
**of methanogenesis pathway, but given the weakness of the correlation we do not**
**emphasize this.**
L549: the remark on the intercepts does not add much and is rather trivial when the
slope is different.
**This discussion is now be heavily modified, as discussed in Major Revision 5. We**
**will not focus on the role of methanogenic pathway as much in the revised**
**manuscript. We use analysis of covariance (ANCOVA) for any comparison of**
**regression relationships in the revised manuscript. We do discuss differences in**
**intercept when the slope is similar and the intercept is significantly different.**
L555 - 561: I am also not aware of dD measurements in natural acetate, but the method
from Greule et al. (2008) has been used in Vigano et al. (2010) to measure dD in
methoxyl groups which were compared to produced CH4 and modeled dD in water.
**We appreciate these suggested references. We include the Vigano reference in our**
**revised discussion (Section 3.3.1).**
L574 – 578: Why do you explain the variability for bogs by the pathway difference, and the high values in rivers by oxidation. Can oxidation not also cause large differences
for bogs?

**This inference was based on the differences in both $\delta^{13}C$-CH$_4$ and $\delta^2H$-CH$_4$. Since**
**bogs have higher $\delta^2H$-CH$_4$ on average, but lower $\delta^{13}C$-CH$_4$, we inferred this was**
**related to a pathway difference. We were also influenced by previous studies (i.e.**
**Ganesan et al., 2018) that had suggested bogs have a higher proportion of**
**hydrogenotrophic methanogenesis. In contrast, rivers are higher in both $\delta^2H$-CH$_4$**
**and $\delta^{13}C$-CH$_4$, which we inferred to be a signal of oxidation. We make this analysis**
**clearer in the revised manuscript, but also add additional caveats.**

L599: Why should the oxidation signal only be apparent for dD and not for d13C (L603-
604)?

**Overall dissolved CH$_4$ from inland waters is also shifted to higher $\delta^{13}C$-CH$_4$ values,**
**although this is not a significant difference. We note in the revised manuscript that**
**greater oxidation would be expected to lead to higher $\delta^{13}C$-CH$_4$ values, and the**
**absence of a strong signal in $\delta^{13}C$-CH$_4$ may be inconsistent with our hypothesis.**

L606: I do not understand how you can conclude that ": : :that the relative balance
of diffusive vs. ebullition gas fluxes should not have a large effect on the isotopic
composition of freshwater CH4 emissions.". The chance for oxidative effects is much
larger for a slow process like diffusion compared to the fast process of ebullition.

**This statement is simply a reflection of the available data, as shown in Figure 9a and**
**b, which do not show a clear difference between these two sample types in their**
**isotopic composition. We note that several caveats moderate this conclusion, and**
**that the question deserves more study (Section 3.7). We added that the likely greater**
**effect of oxidation on diffusive fluxes as an additional area that requires further**
**empirical validation.**

L611: The analysis in this section has much less scientific rigor than the previous sections
and presents some sensitivity calculations involving highly improbable assumptions,
see following points.

**We acknowledge that the sensitivity calculations and scenarios were somewhat**
**simplistic and loosely defined. As discussed above, our solution to this is to scale**
**back this section to focus on the results of a global source mixing model calculation,**
**to compare that with previous estimates of global source signals, and to discuss key**
**data gaps that are likely leading to biases in this estimate (See major revision 6).**
**Therefore the revised manuscripts does not include the sensitivity calculations,**
**which will be left for future work.**

L619 ff: See comments above on the depleted d13C source signature. Here you argue
that three factors may explain this difference. I am quite convinced that the first one
(errors in the sink fractionation factors) cannot explain the large difference. The two published studies for the fractionation in the CH4 + OH reaction (Cantrell et al, 1990,
Saueressig et al, 2001) are 5.4 and 3.9 per mill, respectively. A contribution from Cl
may increase this a bit, but not enough to support a global average source signature
of -56.4 per mill. So I think that the reason should come from the other two processes
mentioned. Given the discrepancy to previous studies I wonder whether it is not mainly
the choice of signatures in this study. In line 625 you already show that changing one
parameter leads to a change of the global average source signature of 1.3 per mill,
which is almost the entire uncertainty range reported.
**We acknowledge the point the reviewer is making. As discussed above, we have**
**revised this section (now Section 3.8) to limit our interpretation to comparison with**
**previous estimates and possible biases in isotopic source signals, and not focus on**
**sink fractionations, which are not a focus of this study. We will also mention**
**possible errors in flux inventories, but not highlight them as much as in the original**
**manuscript.**
L628: Rather arbitrarily changing big sources by a factor of 2 is a huge adjustment
of the atmospheric CH4 budget. This investigation on the effect on the atmospheric
isotopic composition is too simplistic.
**We understand this critique, and as discussed above we avoid performing this**
**analysis in the revised paper. This analysis was based on the work of Schwietzke et**
**al., (2016), who make a similar, but more precise adjustment. We now mention the**
**possibility of higher fossil fuel emissions than in inventories, as discussed by**
**Schwietzke et al., (2016), but leave a detailed analysis resolving this with $\delta^2$H-CH$_4$**
**measurements to future studies.**
L634 ff: Same comment for the bb source, this should be discussed in a more detailed
way. Worden et al. (2017) illustrate the strong influence of the bb source.
**As discussed above, we feel it is best to omit the discussion of specific different**
**emissions scenarios from the discussion. We now briefly discuss the results of the**
**Worden et al., (2017) study, and mention biomass burning emissions as an**
**influential variable for isotopic source signatures that merits further study.**
L660f: The statement "This flatter slope may be the result of the inclusion of a greater
proportion of inland water sites in our dataset." requires more underlying analysis. I
think that the "may be" can be replaced by "is likely", but this should be investigated.
See also other points above.
**Based on the comments of all three reviewers we have thoroughly revised our**
**comparison of our results with that of Waldron et al (1999a). Therefore this part of**
**the conclusions was changed to reflect this revised comparison, and likely causes of**
**the different slope. Our revised analysis implies that differences between inland**
**waters and wetlands is probably not primarily responsible for this difference (See**

**Figure 3 and Supplemental Table 2), and that a greater amount of data from high-**
**latitude environments is more important.**
L662: If possible make more concrete after reevaluation of the impact of modeled data.
**We also revised this statement after a more thorough analysis of the differences in**
**the regression relationship for modeled and measured $\delta^2$H-H$_2$O. Our revised**
**analysis shows that using modeled $\delta^2$H-H$_2$O provides a good estimate of the**
**relationship between $\delta^2$H-H$_2$O and $\delta^2$H-CH$_4$, and supports the use of isotope-**
**enabled Earth Systems Models to predict $\delta^2$H-CH$_4$.**
L686: Here the second argument of the three presented before (see comment on L619)
has disappeared, but as argued above it may be the most important one and particularly
the sink argument does likely not explain (at least exclusively) the difference.
**As discussed above, we have substantially revised and scale back the upscaling**
**estimates. Therefore this conclusion has been omitted.**
*Specific Responses to Reviewer 3:* Reviewer comments are in plain text. **Responses are**
**in bold text.**
During the past two decades, there has been limited progress in advancing understanding
of controls on **d$_2$H(CH$_4$)** values in freshwater environments and improving estimates of
**d$_2$H** values of CH$_4$ emissions. This study: (i) updates and attempts to refine the
relationship between **d$_2$H(H$_2$O)** and **d$_2$H(CH$_4$)** first reported by Waldron et al. (1999b),
(ii) evaluates the extent to which factors other than **d$_2$H(H$_2$O)** may influence **d$_2$H(CH$_4$)**
values in freshwater environments, (iii) uses the refined relationships to estimate new
**d$_2$H** values for CH$_4$ emissions from freshwatersources, and (iv) weights CH$_4$ fluxes
reported by Saunois et al. (2020) with a mixture of old and new **d$_2$H** and **d$_{13}$C** values to
estimate global **d$_2$H** and **d$_{13}$C** values for atmospheric CH$_4$. In my opinion, the study offers
new insights that are worthy of publication pending revision.
**We thank Dr. Hornibrook for his detailed review, and we are heartened to hear his**
**opinion that the study is worthy of publication pending revision.**
Site level mean values - The study has produced a thorough compilation of stable isotope
data related to CH$_4$ from freshwater environments. The availability of **d$_2$H(CH$_4$)** values
presumably was the key criterion for inclusion in the data base. The supplemental file
contains a summary of the data, showing the number of samples from each site and site-
level mean isotopic values as described in section 2.3.1. While I appreciate the
motivation to avoid introducing bias towards sites that have larger datasets, this approach
does limit the extent to which the study can comment meaningfully on differences
between environments. **d$_2$H(CH$_4$)**, **d$_{13}$C(CH$_4$)** and **d$_{13}$C(CO$_2$)** values all exhibit
significant ranges and trends with depth in the subsurface of wetlands. That information
is lost when profiles of **d**-values are averaged. In peatlands where CH$_4$ production
pathways change with depth or CH$_4$ oxidation occurs, **d**-values determined from an average of shallow and deep layers has little meaning in the context of production
pathways or evidence for $CH_4$ alteration. The pooled **d**-values also do not take into
account differences in the amount of $CH_4$ or $CO_2$ at different depths. Moreover, **d**-values
from deep peat typically will have little bearing on the stable isotope composition of $CH_4$
emitted from a wetland. Venting of accumulated gas bubbles from deep peat can occur
(e.g., Glaser et al, 2004) but there is little evidence that such events are common. The
bulk of $CH_4$ production occurs at shallow depths (from water table level to ~50 cm depth)
where the supply of labile substrates from plant roots is greatest and temperature is
highest during summer. The residence time of $CH_4$ at those depths is shortest (e.g.,
Lombardi et al., 1997; Bowes and Hornibrook, 2006) and most of the $CH_4$ produced
seasonally is either consumed or evaded to the atmosphere. If subsurface data must be
averaged to avoid bias, then I suggest using a consistent depth range (e.g., 0 to 50 cm) to
(i) generate mean **d**-values that are more likely to represent **d**-values of $CH_4$ emissions,
and (ii) enable analysis of **a**$_C$ and **a**$_H$ values that are more likely to be related to one
methanogenic pathway or exhibit the influence of methane oxidation rather than a blend
of pathways and processes across a range of depths. An important advance in this study
was the attempt to discern the relative impact of factors other than **d**$_2$H($H_2O$) on
**d**$_2$H($CH_4$) values. Use of site level means for **d**-values raises concern about the validity of
the **a**$_C$ and **a**$_H$ values calculated to assess breakpoints in $CH_4$ production pathways and
oxidation.
**The reviewer raises an important point about $\delta^2$H-$CH_4$ variability with depth in**
**peatlands, and potential biases that are introduced by averaging values across depth**
**profiles. The primary goal of our study is to investigate spatial variability between**
**sites, and therefore we think it is important to provide a single value for each site. In**
**addition, one of the key goals is to characterize the $\delta^2$H values of $CH_4$ emitted to the**
**atmosphere. Therefore, we agree with the reviewer's suggestion to use a consistent**
**depth range (0-50 cm) when averaging data from peatlands with depth-resolved**
**sampling. See Major Revision 1 above. This change affects 8 sites, from 5**
**publications (Hornibrook et al., 1997; Waldron et al., 1999; Chasar et al., 2000;**
**Chanton et al., 2006; Alstad and Whiticar, 2011). Other studies included in our**
**dataset sampled peatlands at shallow depths. To our knowledge all studies in other**
**wetland environments also sampled shallow (< 50 cm) soils.**
'Bottom-up' mixing model - I appreciate that considerable effort was invested in
attempting to upscale d$_2$H($CH_4$) and d$_{13}$C($CH_4$) values; however, it is questionable
whether that portion of the manuscript has potential to advance discourse on global
isotope-weighted $CH_4$ budgets. A more valuable outcome of this work would have been
the one identified by that authors in lines 441- 443: "A logical next step in predicting
global freshwater $\delta_2$H-$CH_4$ source signatures would be to combine high-resolution
mapping of wetlands and inland waters, maps of the global distribution of $\delta_2$H$_p$, and
regression relationships between $\delta_2$H-$CH_4$ vs. $\delta_2$H$_p$." In my view, production of a
global gridded map of **d**$_2$H($CH_4$) values for freshwater environments would have a more
suitable application of the outcomes from the data analysis. It would provide a useful
counterpart to the **d**$_{13}$C($CH_4$) global map for wetlands published by Ganesan et al. (2018).
I realize at this stage in the process that would take the second half of the manuscript in a very different direction. As things stand, the weighted atmospheric **d₂H(CH₄)** and
**d₁₃C(CH₄)** values that were calculated are difficult to reconcile with atmospheric data and
KIEs associated with sinks for atmospheric CH₄. It's possible that the values may be
offering new insights but it seems more likely that there are issues with attribution of **d₂H**
and **d₁₃C** values to CH₄ sources.
**We understand the reviewer's concerns about the upscaling results. Reviewer 2**
**made similar comments. We note that we specifically did not directly compare these**
**results with atmospheric data, given the uncertainties related to sink KIEs, but**
**instead compared them with previously published estimates of global source isotopic**
**values that are based on atmospheric data and models of sink fractionations (Rice et**
**al., 2016, Figure 10C in the original manuscript). We have made this clearer in the**
**revised manuscript, and have expanded the comparison to other top-down and**
**bottom-up estimates. We highlight the associated uncertainties to a greater degree.**
**We have decided to substantially revise this part of the manuscript. See Major**
**Revision 6. We think it is still worthwhile to present estimates of global methane**
**source $\delta^2$H and $\delta^{13}$C that include the results of our data analysis, and to compare**
**this with previous bottom-up and top-down estimates of global isotopic source**
**signatures. We then focus on an assessment of the largest areas of uncertainty in the**
**isotopic source signatures, and not dwell on uncertainties in sink fractionations,**
**since these are not the focus of this paper. We mention possible errors in flux**
**inventories, but devote less focus to this than possible biases in isotopic signatures.**
**In particular we direct more focus on the problem of a lack of data from C₄ plant**
**dominated ecosystems in synthetic datasets, which may compromise data-based**
**estimates of freshwater $\delta^{13}$C-CH₄ signatures.**
**Creating a gridded map of freshwater $\delta^2$H-CH₄ values entails a substantial amount**
**of additional work and additional expertise, and this is beyond the scope of the**
**revisions for this paper, which as reviewer 1 noted is already quite extensive and**
**ambitious. However, this is the goal of collaborative research that is currently**
**ongoing. This research in development will also look more closely at comparisons**
**with atmospheric data.**
Citations within the text do not appear to be listed consistently either alphabetically or
chronologically.
**We thank the reviewer for noting this. It was a problem with the EndNote citation**
**style, and we fix this in the revised version.**
Line 38: 'clearly' = 'unequivocally' ?
**We agree this makes this sentence clearer and made the change.**
Lines 51-52: 'recent technological developments'. An additional sentence or two about
laser based methods would be helpful for a broader readership.

**That is a good idea and we have added a sentence new laser based methodologies**
**(line 59).**
Lines 53-57: Rigby et al. (2012) also demonstrated the utility of a multi-isotope approach
for global methane cycle characterization.
**We thank the reviewer for bringing this paper to our attention. We have revised this**
**paragraph to include the conclusions of that study (line 66).**
Lines 87-88 (and elsewhere): 'data is' should be 'data are'
**We have adjusted this here and throughout the manuscript.**
Line 105: A citation for Coplen (2011) could be added for the definition of delta that
(correctly)
does not include a 'x 1000' factor.
**We have added the suggested citation**
L129: The citation for John Lansdown's thesis should be:
Lansdown J. M. (1992) The carbon and hydrogen stable isotope composition of methane released
from natural wetlands and ruminants. Ph.D. dissertation, Univ. of Washington.
(The citation can be confirmed at: https://dggs.alaska.gov/pubs/id/28259)
**We thank the reviewer for this correction, and have edited the references and**
**citations.**
L156 – Is the annual estimate of $\delta_2H_p$ weighted by the relative amounts of precipitation
during different seasons?
**Yes, the annual estimates from the model are amount-weighted values (See Bowen**
**and Wilkinson 2002 for specifics on the methodology). We have clarified this in the**
**methods (Line 175).**
L200: **d₂H** (superscript missing)
**We have fixed this error**
L258-L259 "55 sites are classified as wetlands, including 16 bogs, 14 swamps and
marshes, 12fens, and 8 rice paddies."
>> Are the classifications for bogs and fens based upon pore water chemistry and
vegetation surveys? The word 'bog' sometimes is used in site names that are other
wetland types, in particular, fens.
**This is a good point. We have done our best to be careful about the wetland**
**classifications, but we have primarily relied on the classification of the original**

**study. Of the 16 bog sites, 14 came from studies that specifically differentiate**
**between bogs and fens (Chanton et al., 2006; Lansdown, 1992 (thesis), Alstad and**
**Whiticar, 2011, Waldron et al., 1999; Chasar et al., 2000), or provide detailed**
**information on vegetation and/or soil pH (Lansdown et al., 1992; Hornibrook et al.,**
**1996). One other paper (Whiticar et al., 1986) provides data from Volo Bog, Illinois,**
**which based on other studies is an ombrotrophic, sphagnum-dominated bog. The**
**only remaining bog site is a West Virginia Bog, from Wahlen, (1994), which did not**
**provide enough information to verify this classification. Given that this original**
**classification is all we have to go on we continue to use it for this sample.**
Table 1: Origins of some data are unclear. When indicated as 'no specific measurement
indatabase', what does it mean to say 'we used the isotopic values and uncertainties for
X'? Which literature source? Also, only C3 $d_{13}$C values appear to be used for biomass
burning. Grassland and savanna wildfires presumably generate CH$_4$ that has more
positive $d_{13}$C values from burning of C4 grasses.
**Thank you for raising these ambiguities in Table 1. Reviewer 2 has brought up**
**similar concerns and we will make this table and the underlying data clearer in the**
**revised manuscript. The database being referred to is the Global Gas Geochemistry**
**Isotope Database (Sherwood et al., 2017), as referenced in section 2.4. We have**
**made this clearer in the notes for the table. This was the source for all isotopic**
**estimates, with the exception of biogenic marine methane, which we derived from**
**Whiticar et al., (1999).**
**The Global Gas Geochemistry Database was our basis for the biomass burning**
**$\delta^{13}$C-CH$_4$ values. Out of 24 biomass burning $\delta^{13}$C-CH$_4$ values, only 2 are ostensibly**
**from C$_4$ plants and have a higher $\delta^{13}$C-CH$_4$ value. These were included in our**
**analysis. In keeping with our data centered approach, we did not attempt to weight**
**these values in our analysis. However, in the revised manuscript we will mention**
**this as a possible source of error in our discussion, and highlight the importance of**
**more data on methane from C$_4$ plant ecosystems, both for biomass burning and**
**microbial emissions. We include an additional estimate of global source $\delta^{13}$C-CH$_4$**
**that accounts for emissions from C$_4$ plant dominated wetlands and biomass**
**burning, using estimates from Ganesan et al., (2018) and Schwietzke et al., (2016).**
L266-L271 The comparison of modelled $\delta_2 H_p$ values and measured $d_2 H(H_2 O)$ values for
62 sites is important for validating the approach on which estimating $d_2 H(CH_4)$ relies.
The text is not clear though with respect to causes in deviation from a 1:1 relationship.
Presumably "$d_2 H-H_2 O$ is generally higher" means $_2 H$-enrichment is evident in the
measured data. Is the statement about 'overall smaller water volumes' meant to infer
evaporative enrichment of $_2 H$?
**This comment, as well as those of reviewers 1 and 2, made it clear that we needed to**
**more thoroughly evaluate the relationship between empirical $\delta^2$H-H$_2$O and modeled**
**$\delta^2$H$_p$ values in this paper. We have done so, including considering wetlands and**

**inland waters separately, and examining whether modeled annual precipitation or growing season precipitation is a better predictor of the empirical $\delta^2$H-H$_2$O values. See our Major Revision 2 above.**

**The comment about higher $\delta^2$H-H$_2$O in mid-latitude sites was based on thinking that in wetlands the residence time of water would be lower, and therefore there is more seasonal variability in $\delta^2$H-H$_2$O. Since almost all samples were collected in summer, when $\delta^2$H$_p$ is higher than average in higher-latitude settings, this would lead these values to be higher than annual precipitation. However, our more detailed analysis, and further reading on this topic, does not support this contention, and instead implies that evaporation is likely leading to water $\delta^2$H-H$_2$O values that are higher than precipitation in wetlands specifically. In fact, seasonality is likely less important in wetlands than in inland waters. See Major Revision 2 above, Section 3.2, and Figure 2.**

L282-L283 "Both relationships result in a large amount of unexplained residual variability, implying the importance of other variables in controlling $\delta_2$H-CH$_4$."

I'll expand here on the point raised in my general comments. The extent to which residual variability exists is likely underestimated because of the use of site-level means. There are relatively few data sets globally that contain subsurface profiles of both **d$_2$H(H$_2$O)** and **d$_2$H(CH$_4$)** values. Four of those data sets are shown in the enclosed figure which was published in Hornibrook and Aravena (2010): Turnagain Bog (open triangles; Chanton et al. 2006), Sifton Bog (open diamonds; Hornibrook et al. 1997), Point Pelee Marsh (open circles; Hornibrook et al. 1997) and Ellergower Moss (open squares; Waldron et al. 1999a). The arrows indicate the direction of increasing depth in peat for Turnagain Bog, Sifton Bog, Point Pelee Marsh and Ellergower Marsh. The figure also includes **d$_2$H** values of coexisting CH$_4$ and H$_2$O values from Alaskan peatlands along a N-S transect (filled triangles; Chanton et al. 2006) and regression equations (Table 6.2 from Hornibrook and Aravena, 2010 also enclosed) from a number of studies including Waldron et al. (1999b; line 5) and Whiticar et al. (1986; lines 1 and 2).
The approach of using site-level means reduces each of those depth trends to a single point in **d$_2$H(H$_2$O)** vs. **d$_2$H(CH$_4$)** space. The **d$_2$H** values of CH$_4$ emitted to the atmosphere are likely to be similar to the most $_2$H-depleted values in each trend which corresponds to CH$_4$ in shallow peat near the water-air interface and within the root zone where CH$_4$ may be transported to the atmosphere via plant aerenchyma. Averaging **d$_2$H(CH$_4$)** values from all depths (2 m for Sifton Bog and Pelee Marsh; 6 m for Ellergower moss) yields a mean that is substantially more $_2$H-rich. Again, I appreciate the goal of not biasing the analysis to these larger data sets but a single mean for each site does not reflect the considerable residual variability that exists with depth as **d$_2$H(CH$_4$)** values shift away from the global **d$_2$H(H$_2$O)** vs. **d$_2$H(CH$_4$)** regression line. Moreover, the **d$_{13}$C(CH$_4$)** and **d$_{13}$C(CO$_2$)** depth trends from these sites yield systematic shifts in **a$_C$** values that are lost when the **d$_{13}$C** values similarly are reduced to unitary site-level means.

**We thank the reviewer for the detailed explanation of their argument on this issue. As we discussed above, the primary goals of this paper are to explore inter-site**

**geographic variability in the $\delta^2$H-CH$_4$ emitted to the atmosphere. Therefore, while**
**intra-site variability is of great interest, we do not want to add an additional layer of**
**complexity to this paper by considering this. We feel the reviewer's earlier**
**suggestion of limiting samples from the upper 50 cm of peat is a good solution to this**
**issue, and we have followed this suggestion in our revised analysis. See Major**
**Revision 1. There is certainly scope for considering intra-site variability in a**
**subsequent study, and we would like to do so.**
L308-L309 "We do not find evidence for a piece-wise linear relationship between $\delta_{13}$C-
CH$_4$ and $\delta_2$H-CH$_{4,w0}$ (Fig. 5a), nor did we find a significant simple linear correlation
between these variables."
>> It may be worth exploring whether any relationships exist in the full data sets rather
than site level means.
**This is an interesting suggestion, though we have concerns that such an analysis**
**might be biased by over-representing sites that have a large number of**
**measurements. It will also require a large amount of additional data analysis, since**
**the $\delta^2$H-CH$_{4,w0}$ estimates are not currently disaggregated on a per sample basis.**
**Given that the focus of this work is on variability between sites, we will leave this**
**analysis for future work focused on intra-site isotopic variation. Note that in the**
**revised manuscript we are no longer analyzing piece-wise regression results, as the**
**results of our revised analysis were inconclusive.**
L441-L443: "A logical next step in predicting global freshwater $\delta_2$H-CH$_4$ source
signatures would be to combine high-resolution mapping of wetlands and inland waters,
maps of the global distribution of $\delta_2$H$_\mathbf{p}$, and regression relationships between $\delta_2$H-CH$_4$ vs.
$\delta_2$H$_\mathbf{p}$."">> I agree with the authors and suggest this would be a worthwhile output to
include in this manuscript instead of the global upscaling estimate.
**We appreciate this suggestion. As mentioned above, adding this output to this**
**manuscript would entail substantial additional work, as well as additional expertise.**
**We have however begun a collaboration with another research group to perform**
**this analysis, and this will be the focus of a future publication.**
L445-L464 Section 4.2. This section would benefit from acknowledging and discussing
the study by Rigby et al. (2012).
**We thank the reviewer again for this suggestion. From our reading of Rigby et al.**
**(2012) there was not a focus on latitudinal variation in microbial or freshwater $\delta^2$H-**
**CH$_4$, so we did not reference this study here. But we do reference it at several points**
**when discussing upscaling and uncertainties in isotopic source signatures in Section**
**3.8.**
L500-L504 In addition to the caveat noted that CH$_4$ data exhibiting $_2$H-enrichment due to
methane oxidation are uncommon, the amount of CH$_4$ emitted to the atmosphere bearing
the effects of methanotrophy is likely to be small. Bacteria oxidation is highly efficient in the subsurface of wetlands and little $CH_4$ tends to escape to the atmosphere via diffusion
through porewater. This comment applies to peatlands. The situation is different in inland
water environments.
**This is an important point. As discussed in Major Revision 4, we are now less**
**confident that the observed variation in $\delta^2H$-$CH_{4,w0}$ can be primarily ascribed to**
**differences in methanogenic pathway. Therefore our discussion of relative**
**importance of these mechanisms, as well as other possibly influential processes, is**
**quite different in the revised manuscript. We do discuss that oxidation does not**
**seem to be a dominant factor in controlling $\delta^2H$-$CH_4$ in wetlands in the revised**
**Section 3.4.**
L510–L518 I was pleased to see incorporation of these alternate explanations for
relationships between $d_2H$ and $d_{13}C$ values of $CH_4$. Methanogenic pathways are not the
only potential explanation.
**We are glad to see that there is a positive reception to this. Based on this comment**
**and those of reviewer 1 we are planning to focus on alternate explanations to a**
**greater degree in the revised manuscript. See Major Revision 4.**
L592-L593 – Bellisario et al. (1999) provides a good example of how $d_{13}C(CH_4)$ values
vary along a trophic gradient in a wetland complex. Differences in $d_{13}C$ values of $CH_4$
emissions and porewater $CH_4$ values in minerotrophic vs. ombrotrophic wetland are
demonstrated in Hornibrook and Bowes (2007) and Hornibrook (2009). Landscape scale
measurements (atmospheric inversions and aircraft measurements; Fisher et al., 2017)
also show that northern wetlands contain sources of $_{13}C$-poor $CH_4$ that differ from values
of ~-62 to -58 permil typically attributed to northern peatlands in isotope-weight $CH_4$
budgets. Characterization of sites as ombrotrophic or minerotrophic on the basis of water
chemistry and vegetation surveys is essential for making these distinctions.
**We thank the reviewer for these insights. While it is difficult for us to make**
**distinctions between minerotrophic and ombrotrophic peatlands in this dataset, we**
**note the importance of this distinction in Sections 3.5 and 3.6. In addition to the**
**absence of $C_4$ plant ecosystems, this is an additional potential bias in the $\delta^{13}C$**
**database assembled in this study. We acknowledge this, including the observations**
**from atmospheric measurements that point to a depleted source in the high latitudes,**
**and discuss how it could be addressed with future research.**
L617 to L622 It is unclear how a more negative than expected value for estimated
$d_{13}C(CH_4)$ can be explained by (2) source signatures being biased toward more positive
$d_{13}C$ values.
**This was a mistake. We meant to say '$^{13}C$ _depleted_ values' and '$^{13}C$ _depleted_ sources'.**
**Regardless, this section of the discussion has been thoroughly revised based on the**
**suggestions of reviewers 2 and 3, with less emphasis on discrepancies with**
**atmospheric measurements. See major revision 6.**

**References Cited:**

Alstad, K. P., and Whiticar, M. J.: Carbon and hydrogen isotope ratio characterization of methane dynamics for Fluxnet Peatland Ecosystems, Org Geochem, 42, 548-558, 2011.

Bowen, G. J., and Wilkinson, B.: Spatial distribution of δ18O in meteoric precipitation, Geology, 30, 315-318, 2002.

Chasar, L., Chanton, J., Glaser, P. H., and Siegel, D.: Methane concentration and stable isotope distribution as evidence of rhizospheric processes: Comparison of a fen and bog in the Glacial Lake Agassiz Peatland complex, Annals of Botany, 86, 655-663, 2000.

Chanton, J. P., Fields, D., and Hines, M. E.: Controls on the hydrogen isotopic composition of biogenic methane from high-latitude terrestrial wetlands, Journal of Geophysical Research: Biogeosciences (2005–2012), 111, 2006.

Chanton, Jeffrey P. "The effect of gas transport on the isotope signature of methane in wetlands." *Organic Geochemistry* 36.5 (2005): 753-768.

Fletcher, D., & Dixon, P. M. (2012). Modelling data from different sites, times or studies: weighted vs. unweighted regression. *Methods in Ecology and Evolution*, *3*(1), 168-176.

Galand, P. E., Kim Yrjälä, and Ralf Conrad. "Stable carbon isotope fractionation during methanogenesis in three boreal peatland ecosystems." *Biogeosciences* 7.11 (2010): 3893-3900.

Gruen, Danielle S., David T. Wang, Martin Könneke, Begüm D. Topçuoğlu, Lucy C. Stewart, Tobias Goldhammer, James F. Holden, Kai-Uwe Hinrichs, and Shuhei Ono. "Experimental investigation on the controls of clumped isotopologue and hydrogen isotope ratios in microbial methane." *Geochimica et Cosmochimica Acta* 237 (2018): 339-356.

Lansdown, J., Quay, P., and King, S.: CH4 production via CO2 reduction in a temperate bog: A source of 13C-depIeted CH4, Geochim Cosmochim Ac, 56, 3493-3503, 1992.

Lansdown J. M. (1992) The carbon and hydrogen stable isotope composition of methane released from natural wetlands and ruminants. Ph.D. dissertation, Univ. of Washington.

Hornibrook, E. R., Longstaffe, F. J., and Fyfe, W. S.: Spatial distribution of microbial methane production pathways in temperate zone wetland soils: stable carbon and hydrogen isotope evidence, Geochim Cosmochim Ac, 61, 745-753, 1997.

Penning, H., Claus, P., Casper, P., & Conrad, R. (2006). Carbon isotope fractionation during acetoclastic methanogenesis by Methanosaeta concilii in culture and a lake sediment. *Applied and Environmental Microbiology*, *72*(8), 5648-5652.

Penning, H., S. C. Tyler, and R. Conrad. "Determination of isotope fractionation factors and
quantification of carbon flow by stable carbon isotope signatures in a methanogenic rice root
model system." *Geobiology* 4.2 (2006): 109-121.
Rice, A. L., Butenhoff, C. L., Teama, D. G., Röger, F. H., Khalil, M. A. K., and
Rasmussen, R. A.: Atmospheric methane isotopic record favors fossil sources flat in
1980s and 1990s with recent increase, Proceedings of the National Academy of Sciences,
113, 10791-10796, 2016.
Saunois, M., Stavert, A. R., Poulter, B., Bousquet, P., Canadell, J. G., Jackson, R. B., Raymond,
P. A., Dlugokencky, E. J., Houweling, S., and Patra, P. K.: The global methane budget 2000–
2017, Earth System Science Data, 12, 1561-1623, 2020.
Schwietzke, S., Sherwood, O. A., Bruhwiler, L. M., Miller, J. B., Etiope, G.,
Dlugokencky, E. J., Michel, S. E., Arling, V. A., Vaughn, B. H., and White, J. W.:
Upward revision of global fossil fuel methane emissions based on isotope database,
Nature, 538, 88-91, 2016.
Sherwood, O. A., Schwietzke, S., Arling, V. A., and Etiope, G.: Global inventory of gas
geochemistry data from fossil fuel, microbial and burning sources, version 2017, Earth
System Science Data, 9, 2017.
Valentine, D. L., Chidthaisong, A., Rice, A., Reeburgh, W. S., & Tyler, S. C. (2004). Carbon and
hydrogen isotope fractionation by moderately thermophilic methanogens. *Geochimica et*
*Cosmochimica Acta*, *68*(7), 1571-1590.
Wahlen, M.: Carbon dioxide, carbon monoxide and methane in the atmosphere:
abundance and isotopic composition, Stable isotopes in ecology and environmental
science, 93-113, 1994.
Waldron, S., Lansdown, J., Scott, E., Fallick, A., and Hall, A.: The global influence of
the hydrogen iostope composition of water on that of bacteriogenic methane from
shallow freshwater environments, Geochim Cosmochim Ac, 63, 2237-2245, 1999a.
Waldron, S., Hall, A. J., and Fallick, A. E.: Enigmatic stable isotope dynamics of deep
peat methane, Global Biogeochem Cy, 13, 93-100, 1999b.
Whiticar, M. J., Faber, E., and Schoell, M.: Biogenic methane formation in marine and
freshwater environments: $CO_2$ reduction *vs.* acetate fermentation—Isotope evidence,
Geochim Cosmochim Ac, 50, 693-709, 1986.
Whiticar, M. J.: Carbon and hydrogen isotope systematics of bacterial formation and
oxidation of methane, Chem Geol, 161, 291-314, 199

---

## Author Response (AR2)

**Douglas et al- Response to reviews- 2$^{nd}$ revision**

**We thank the two reviewers for reviewing the manuscript a 2$^{nd}$ time, for their overall positive assessment, and for their helpful suggestions for further clarification and improvement of the manuscript. Our specific responses to both reviews are below in bold type. We have also further revised the manuscript to clarify the text and avoid redundancy.**

**Reviewer 2**

Second review of "Global geographic variability in freshwater methane hydrogen isotope ratios and its implications for emissions source apportionment and microbial biogeochemistry" by Douglas et al.

The authors have incorporated the numerous suggestions that were raised from the different reviewers in considerable detail. The results are now discussed in a more balanced way regarding previous publications. It is not an easy paper to digest, as each section has its own in-depth discussion, but this reflects the complexity of the subject. Otherwise, the presentation quality is high in terms of structure and clarity of language. The core of the paper is the use of statistical tools to detect correlations. It is a very useful exercise even if it turns out that often no clear relations can be found. This is not always surprising given the rather limited number of datapoints and the many parameters that are involved. In my opinion, the discussion sometimes goes a bit too far in such cases, trying to find out which parameter would cause which effect. Such discussions could be reduced, which would also reduce redundancies between different sections. In some cases, possible signals are mentioned, but then it is stated in the next sentence that the effect is not significant. This could also be skipped, but I leave these points for consideration of the authors. Otherwise, I only have a few minor comments.

**We thank the reviewer for their positive assessment. We have edited the manuscript to simplify the results and discussion section where possible, notably in sections 3.3, 3.4, 3.5, and 3.6. We have largely omitted discussion of results that are not statistically significant.**

Specific comments:
Abstract L23 ff: Related to my general comment, would you really expect to find one dominant process that explains the residual variability? I think it is not really surprising that there is not a clear result, given the vast differences in the conditions in the various datasets. This could be left out.

**This statement was in response to previous studies (Waldron et al., 1999; Chanton et al. 2006), who had asserted that specific processes were dominant in controlling residual variability. We discuss this in section 3.4. However, we appreciate that the abstract could be simplified by omitting this, and we have done so, focusing instead on the general finding of a complex set of processes controlling residual variability.**

L58 ff: The methods represented by the references also include isotope ratio mass spectrometry.

**We have added to this sentence to acknowledge the complementary use of isotope ratio mass spectrometry**

L104: subscript 4

**We fixed this typo.**

L119: Remove: "The isotope notation used in this study is briefly introduced here." This is standard notation, not only used in this study

**This was a holdover from the original manuscript when we had some specific derived isotopic terminology introduced in this section. We have now omitted this sentence as suggested.**

Table 1 and Section 3.8: Although the authors did not agree with my comment on the small errors of the d13C signatures of the fossil fuel sources, I still think it is valid. I acknowledge that the estimates come from published literature, but I am still skeptical. Given that the reported signatures span a range of about 50 ‰, the fluxes are uncertain, emissions and signatures can change over time, etc., I wonder whether the flux weighted average signature is really known to ±0.5 ‰. I think this could be mentioned as additional uncertainty, in particular since there is a clear discrepancy between the bottom-up and top-down estimates for d13C.

**We acknowledge that uncertainty related to the relative fluxes and isotopic signatures of fossil fuel emissions from different regions and resource types (i.e. conventional vs unconventional reservoirs) could add additional uncertainty to the average value and 95% CI applied in our model. It would be challenging to quantify this uncertainty with available data, and doing so is beyond the scope of this study. But we have added statements about this in section 2.4 and 3.8.**

**Reviewer 3:**

The authors have made extensive and well-considered revisions to the manuscript. I support publication of this work in Biogeosciences.

**We thank Dr. Hornibrook for his review and positive assessment.**

Minor corrections
Line 18: "is similar observations from in incubation experiments"
> Sentence unclear. Was this meant to say?: "similar to observations from incubation experiments'

**This was a typo- we have fixed it.**

P25, line 44; Awkward sentence: "Comparing the broader categories of inland waters and wetlands with a we do find a significant"

**This was also a typo- we deleted a phrase that was confusing our meaning here.**